# U-BENCH: A COMPREHENSIVE UNDERSTANDING OF U-NET THROUGH 100-VARIANT BENCHMARKING

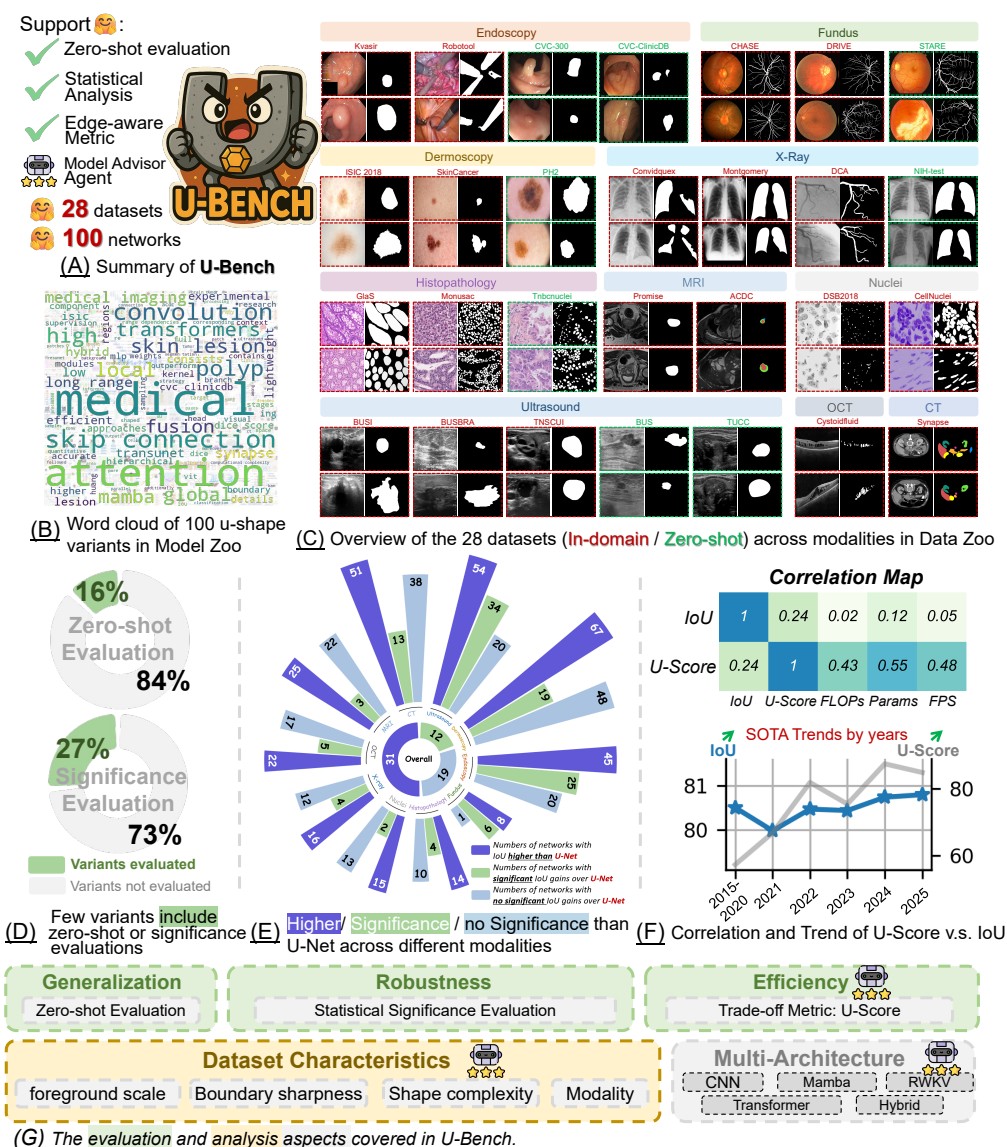

Figure 1: Overview of U-Bench. (A) **The summary of U-Bench**, which encompasses the most comprehensive large-scale evaluation of U-shaped architectures. (B) **Word cloud of 100 published U-shaped variants in U-Bench Model Zoo.** (C) **Examples of the 28 datasets in U-Bench Data Zoo.** The red / green box: in-domain / zero-shot split for evaluation. (D) **Literature analysis.** Among 100 recent works, 84% papers neglect zero-shot evaluation and 73% papers lack of statistical significance testing. (E) **Significance analysis.** Only a minority achieve statistically significant gains over U-Net. (F) **Overview of a new metric, U-score.** Top: IoU does not account for efficiency, while U-Score demonstrates a strong correlation with both segmentation performance and efficiency metrics. Bottom: while IoU shows a trend of saturation, U-Score highlights the yearly trends toward more efficient models. (G) **The evaluation and analysis aspects covered in U-Bench.**

## ABSTRACT

Over the past decade, U-Net has been the dominant architecture in medical image segmentation, leading to the development of thousands of U-shaped variants. Despite its widespread adoption, there is still no comprehensive benchmark to systematically evaluate their performance and utility, largely because of insufficient statistical validation and limited consideration of efficiency and generalization across diverse datasets. To bridge this gap, we present U-Bench, the first large-scale, statistically rigorous 2D benchmark that evaluates 100 U-Net variants across 28 datasets and 10 imaging modalities. Our contributions are threefold: (1) Comprehensive Evaluation: U-Bench evaluates models along three key dimensions: statistical robustness, zero-shot generalization, and computational efficiency. We introduce a novel metric, U-Score, which jointly captures the performance-efficiency trade-off, offering a deployment-oriented perspective on model progress. (2) Systematic Analysis and Model Selection Guidance: We summarize key findings from the large-scale evaluation and systematically analyze the impact of dataset characteristics and architectural paradigms on model performance. Based on these insights, we propose a model advisor agent to guide researchers in selecting the most suitable models for specific datasets and tasks. (3) Public Availability: We provide all code, models, protocols, and weights, enabling the community to reproduce our results and extend the benchmark with future methods. In summary, U-Bench not only exposes gaps in previous evaluations but also establishes a foundation for fair, reproducible, and practically relevant benchmarking in the next decade of U-Net-based segmentation models. The weights, datasets, and results will be released after the acceptance. Code is available at: `https://anonymous.4open.science/r/U-Bench`.

## 1 INTRODUCTION

Medical image segmentation is a critical and challenging task that can greatly enhance diagnostic efficiency by offering doctors objective and precise references for regions of interest (Zhou et al., 2017). Over the past decade, U-Net (Ronneberger et al., 2015) has become a cornerstone of medical image segmentation, thanks to its encoder-decoder structure with skip connections that effectively combine multi-scale features. Building on its promising segmentation results across diverse modalities, numerous U-shaped variants have been proposed to further improve performance, with lightweight designs (Valanarasu & Patel, 2022; Tang et al., 2024; Chen et al., 2024a; Valanarasu et al., 2021; Cao et al., 2022), attention mechanisms (Oktay et al., 2018; Tang et al., 2023), multi-scale feature fusion (Zhou et al., 2018; Huang et al., 2020), and more recently Mamba- (Liu et al., 2024a; Wu et al., 2025b), RWKV (Receptance Weighted Key Value)-based (Ye et al., 2025; Jiang et al., 2025), as well as hybrid architectures (Chen et al., 2021; Tang et al., 2025b; Dong et al., 2025; Tang et al., 2025a). Over the past decade, more than ten thousand U-Net variants have been proposed, and by 2025, nearly a thousand studies have employed U-shaped networks for medical image segmentation.

Among the vast number of U-Net variants, a central challenge remains unresolved: ***How to conduct a fair and comprehensive comparison across them?*** Although several benchmarks and surveys have attempted to organize this proliferation (Tab. 1), they mostly lack a large-scale, systematic evaluation. Critical aspects such as robustness of improvements, zero-shot generalization, and computational efficiency are often overlooked, and they also fail to provide complete and in-depth analyses of dataset-specific characteristics and model architectures. Despite reported gains in recent works, many studies report metrics without statistical validation (73% omit it, Fig. 1D), use incomplete baseline comparisons, or rely on limited dataset coverage. Moreover, efficiency, although vital for real-world clinical deployment (Vashist, 2017; Wenderott et al., 2024; Xu et al., 2025), is rarely considered. Compounding this issue, evaluations are typically confined to in-distribution settings (84% of work ignores zero-shot evaluation, Fig. 1D), even though clinical practice inevitably involves domain shifts across institutions and annotation protocols (Yan et al., 2019; Koch et al., 2024). These gaps leave the robustness and practicality of U-Net variants in real-world scenarios largely unverified (Niu et al., 2024).

Table 1: Comparisons between U-Bench and other medical image segmentation benchmarks. Details can be found in the Appendix B.

| Category | Item | U-Bench (ours) | TorchStone (Bassi et al., 2024) | nnWNet (Zhou et al., 2025) | MedSegBench (Kuş & Aydin, 2024b) | nnU-Net Revisited (Isensee et al., 2024) |
|---|---|---|---|---|---|---|
| **Models** | | **100** | 19 | 20 | 6 | 19 |
| **Datasets** | | **28** | 3 | 8 | 35 | 6 |
| **Modalities** | Ultrasound | ✓ | | | ✓ | |
| | Dermoscopy | ✓ | | ✓ | ✓ | |
| | Endoscopy | ✓ | | ✓ | ✓ | |
| | Fundus | ✓ | | ✓ | ✓ | |
| | X-Ray | ✓ | | | ✓ | |
| | Histopathology | ✓ | | | ✓ | |
| | CT | ✓ | ✓ | ✓ | ✓ | ✓ |
| | MRI | ✓ | | ✓ | ✓ | ✓ |
| | Nuclei | ✓ | | | ✓ | |
| | OCT | ✓ | | | ✓ | |
| **Evaluation** | Robustness | ✓ | ✓ | ✓ | | ✓ |
| | Generalization | ✓ | ✓ | | | |
| | Efficiency | ✓ | | | | |
| **Architecture Analysis** | CNN | ✓ | ✓ | ✓ | ✓ | ✓ |
| | Transformer | ✓ | ✓ | ✓ | | ✓ |
| | Hybrid | ✓ | ✓ | ✓ | ✓ | ✓ |
| | Mamba | ✓ | | | | ✓ |
| | RWKV | ✓ | | | | |
| **Dataset Analysis** | Scale | ✓ | | | | |
| | Boundary | ✓ | | | | |
| | Shape | ✓ | | | | |

To systematically and comprehensively evaluate U-shaped medical image segmentation models, we introduce **U-Bench**, the first large-scale, statistically rigorous, and efficiency-oriented 2D benchmark for U-Net and its variants. U-Bench is built upon three key aspects: **(1) Broad dataset and model coverage:** we implement **100** recent U-Net variants and evaluate them on **28** benchmark datasets covering **10** diverse imaging modalities (ultrasound, dermoscopy, endoscopy, fundus photography, histopathology, nuclear imaging, X-ray, MRI, CT, and OCT; Fig. 1A, C). **(2) Rigorous and comprehensive evaluation:** all models are implemented to calculate performance gains over the baseline U-Net with statistical significance, ensuring robust and fair comparisons (Fig. 1E). To capture clinical utility, we further assess zero-shot generalization across modalities. Additionally, to address practical considerations in real-world edge deployment, we introduce the U-Score, a statistically grounded, large-scale metric that jointly accounts for accuracy, parameter numbers, computational cost, and inference speed (Fig. 1F). **(3) Public availability and reproducibility:** U-Bench implements models using official code implementations, pre-trained weights, and deep supervision strategies (if available). At the same time, U-Bench is released with all code, models, and protocols, enabling the community to reproduce our results and extend the benchmark with future methods.

Building on this large-scale evaluation, we identify key findings that challenge common assumptions. Traditional metrics like IoU show signs of saturation, offering a limited discriminative power (Fig. 1F). Additionally, reported improvements are often inconsistent or statistically insignificant (Fig. 1E). At the same time, an increasing focus on storage and computational cost is reflected in the rising trajectory of U-Score (Fig. 1F). To explore these dynamics, we conduct a systematic analysis of U-Net variants, examining the influence of dataset and architectural factors on model performance across different modalities, architectures, and computational resource limitations (Fig. 1G). Building on these analyses, we introduce a model advisor agent that suggests suitable architectures based on dataset and task attributes, turning an actionable guidance for practitioners in clinical and research contexts.

Our contribution can be summarized as:

- We provide a comprehensive evaluation benchmark of 100 U-shaped variants across 28 datasets from 10 modalities with a rigorous assessment across statistical robustness, zero-shot generalization, and computational efficiency. To better capture the trade-off between accuracy and efficiency, we introduce U-Score, a novel metric grounded in large-scale statistical analysis that enables fair and holistic evaluation.

- We summarize the observations over large-scale evaluation: Most variants show performance gains, but few show in-domain statistical significance over the original U-Net. Zero-shot performances show significant and promising improvements. U-Score shows

an increasing trajectory, indicating the shift from purely pursuing accuracy to balancing accuracy with efficiency.

- We disentangle different aspects, including dataset characteristics and architectural designs, revealing their impact on performance and efficiency, and further build a model recommender that helps researchers identify well-suited architectures under diverse data and resource conditions.

- We open-source U-Bench and all the pretrained weights, providing a large-scale benchmark with comprehensive evaluation for medical image segmentation, to foster fair, robust, generalizable, and efficient research in the community.

## 2 U-BENCH CONSTRUCTION

### 2.1 PRELIMINARIES: U-SHAPED DESIGN

A U-shaped model generally comprises four components: hierarchical encoder, decoder, bottleneck, and skip-connection. Given an input image $x \in \mathbb{R}^{3 \times H \times W}$, Encoder($\cdot$) extracts multi-scale features $f_i$ by $N$ stages from up-bottom, denoted as $\{f_i\}_{i=1}^N$, $f_i \in \mathbb{R}^{C_i \times \frac{H}{2^{(i-1)}} \times \frac{W}{2^{(i-1)}}}$. Bottleneck($\cdot$) processes the last output feature, and Decoder($\cdot$) is composed of $N-1$ stages for upsampling decoder features $d_j$ from bottom-up, each stage comprises Skip-connection($\cdot$) for feature fusion. Final prediction $\tilde{x}$ is produced by the segmentation head after the top decoder stage. The differences across variants are illustrated in Fig. 2: Convolutional Neural Networks (CNNs) and related architectures (Attention, Mamba, RWKV) form the core building blocks, which can be organized in pure CNN / Attention, parallel, or sequential configurations for both encoding and decoding. Detailed categorization can be found in the Appendix B and D.

Figure 2: **Summary of U-shaped networks.** The network comprises an encoder, a bottleneck, and a decoder with skip-connection, each of which can integrate attention gates and multi-scale fusion. The blue and green blocks represent different core building blocks, which can be integrated sequentially, in parallel, or used individually.

### 2.2 DATASET AND MODEL ZOO

**Dataset Zoo.** As shown in Fig. 1(C), the U-Bench dataset zoo consists of 28 diverse publicly available 2D medical image segmentation datasets spanning a wide range of imaging modalities, including ultrasound, dermoscopy, endoscopy, fundus photography, histopathology, nuclear imaging, X-ray, MRI, CT, and OCT. We train on 20 datasets and evaluate zero-shot generalization on 8 additional ones. Following prior work (Chen et al., 2021; Tang et al., 2025b; Valanarasu & Patel, 2022; Jiang et al., 2025; Wang et al., 2022a), all datasets are resized to 256×256 and augmented by random rotation and flipping; for models with fixed input size, we keep their original resolution (typically 224×224). Official splits are used when available; otherwise, a 7/3 split is applied. All details on datasets and preprocessing are provided in the Appendix C.

**Model Zoo.** We curate a collection of 100 publicly available and widely adopted U-Net variants, covering CNN-, Transformer-, Mamba-, and RWKV-based architectures, as well as their hybrid designs (Fig. 1(B)). To ensure strict reproducibility and fair comparison, we follow the official implementations for all models, adopting their predefined settings, pretrained weights, and deep supervision strategies when available. All model details are provided in the Appendix D.

### 2.3 EVALUATION PROTOCOL

**Evaluation Metrics.** Following previous works (Luo et al., 2025; Jiang et al., 2025; Tang et al., 2025b; Valanarasu & Patel, 2022; Tang et al., 2024), we evaluate segmentation performance using Intersection over Union (IoU). We also report the commonly used performance metrics, including Dice coefficient, and boundary evaluation metrics HD95 and Boundary-F1 in Appendix. To evaluate the statistical significance of performance differences between models, we conduct paired sample

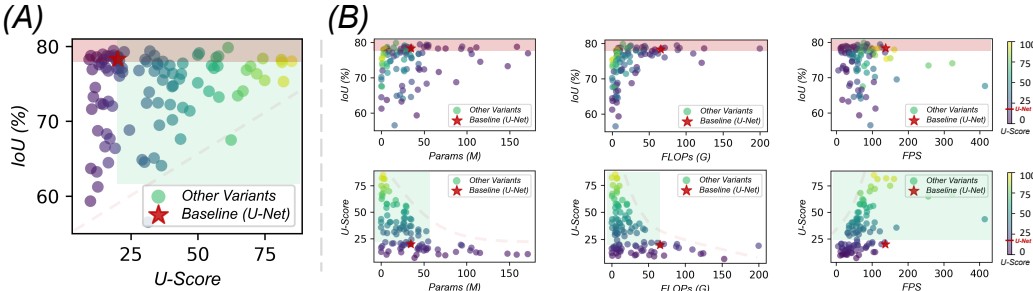

Figure 3: **Comparison between IoU and U-Score.** Red rectangle indicates the models perform better than U-Net in IoU, and green rectangle indicates the models perform better than U-Net in U-Score. (A) Across 100 variants, few methods show better IoU compared to baseline U-Net, while more than half of the methods show better U-Score. (B) The relationship between performance (IoU) and the increase in computational resources (FLOPs, parameters, FPS) is complex, whereas U-Score offers a clear distribution that effectively distinguishes favorable and unfavorable accuracy-efficiency trade-off.

$t$-tests, comparing each variant to the baseline U-Net. U-Bench also considers computational efficiency metrics, including Parameters (M), FLOPs (G), and FPS. All result details are provided in the Appendix D.

**Zero-shot Evaluation.** To evaluate the generalization capability of each model beyond the training distribution, we conduct zero-shot inference on unseen datasets within the same modality and task. Specifically, models are trained on source datasets and then directly evaluated on unseen datasets that share the same modality but differ in acquisition domain. Detailed dataset split can be found in Fig. 1(C). This approach aligns with clinical demands, where domain shifts frequently occur in real-world applications due to variations in devices, institutions, and patient populations.

**U-Score.** To assess real-world deployability, we propose U-Score, a unified metric that jointly accounts for accuracy and efficiency. For each model $i$, we compute segmentation accuracy evaluated by IoU $A_i$ parameter $P_i$, FLOPs $G_i$, and FPS $S_i$, which are percentile-normalized into $a_i, p_i, g_i, s_i \in [0,1]$ using the 10-th/90-th quantiles across the model zoo. Given a harmonic mean function $\mathcal{H}(\cdot)$ with equal weights for each input, an efficiency subscore $\text{Eff}_i = \mathcal{H}(p_i, g_i, s_i)$ is obtained, ensuring that no single factor dominates. The final U-Score is defined as $\text{U-Score}_i = \mathcal{H}(a_i, \text{Eff}_i)$ to incorporate segmentation accuracy and computational efficiency equally. This combination rewards models that achieve favorable accuracy-efficiency trade-offs and provides a deployment-oriented alternative to IoU. For example, although some models surpass U-Net in segmentation accuracy, they require substantially different levels of computational overhead, making direct comparisons with the baseline difficult. Additionally, we conduct extensive sensitivity analysis of U-Score covering the accuracy-efficiency trade-off, efficiency component weights, and normalization percentiles. More details of U-Score and sensitivity analysis can be found in Appendix E.

## 3 U-BENCH RESULTS & DISCUSSION

In this section, we present the results of the U-Bench benchmark across multiple dimensions, including accuracy, efficiency, and generalization. We organize the results as follows: In 3.1, we present and discuss retrospective analysis of the develop trends and statistical findings over 100 variants spanning different architectures and publication years. In 3.2, we disentangle influence factors into two aspects: dataset and architecture, and analyze how these factors impact model performance. In 3.3, we propose our ranking-based advisor agent, offering practical guidance for selecting optimal models based on dataset characteristics and resource constraints.

### 3.1 RETROSPECTIVE ANALYSIS OF THE PAST DECADE

**Finding 1: In-domain Top-1 performance has marginal gains in segmentation accuracy, while zero-shot improves more pronouncedly.** We analyzed 100 variants across different architectures

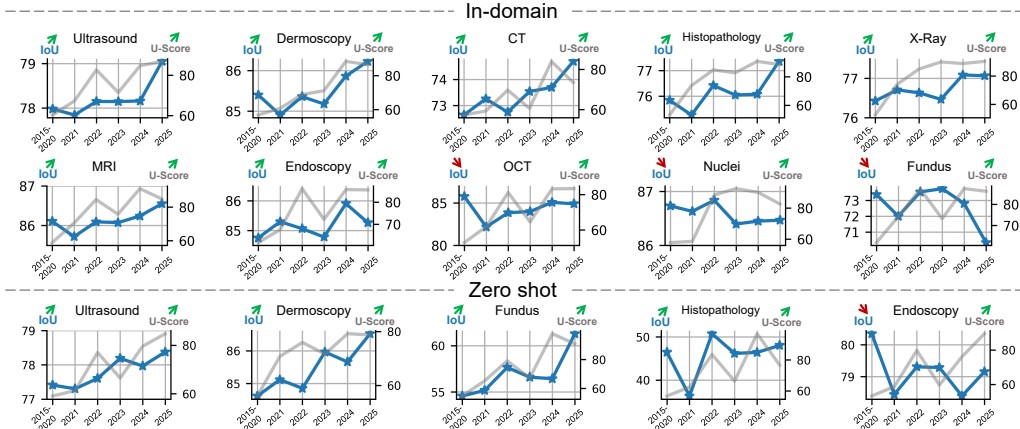

Figure 4: **Performance trends of SOTA models over the past decade.** The x-axis indicates publication year, with each point marking the yearly best result. The y-axes report two evaluation metrics: **IoU** (left axis) and **U-Score** (right axis). The trend's summary is shown as arrows at the top of the y-axis, with green ones highlighting improvements and red ones indicating stagnation. Source domain performance is show at the top, and zero-shot performance is shown at the bottom.

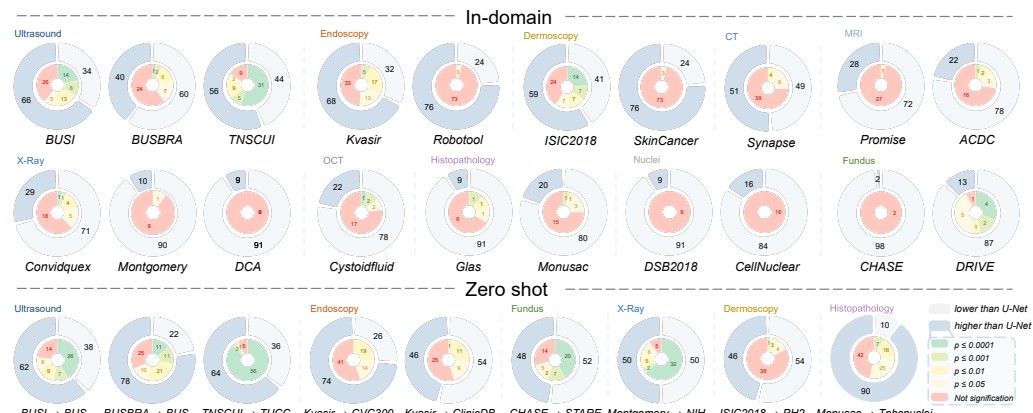

Figure 5: **Statistical significance analysis on the IoU against U-Net across 28 datasets across 10 modalities.** The outer blue pie represents the number of variants surpassing U-Net; the inner pie quantifies the statistical significance of the methods with improvements, annotated by non-significant to highly significant, with the number of works annotated in the middle. In general, in-domain improvements show limited statistical significance, while zero-shot performances show more significant improvements.

and publication years (the detailed list can be found in the Appendix B of Fig. 9), reporting the best-performing variant for each year, as shown in Fig. 4. Over the past decade, 70% of modalities have demonstrated steady progress of segmentation accuracy in both source and target domains, as reflected in IoU. However, IoU gains have been marginal (on average 1%-2%) and inconsistent. Some modalities (*i.e.* OCT, Nuclei, and Fundus) even show a sign of stagnation. In comparison, when considering zero-shot performances, the improvements have been more obvious (more than 3% on average) in 80% of the modalities.

**Finding 2: Although some in-domain improvements exist on average, few reach statistical significance, whereas the average zero-shot improvements remain consistently significant.** To rigorously distinguish modalities with genuine improvements from those with only numerical fluctuations, we perform $t$-tests between each variant and the U-Net baseline. The results are presented in Fig. 1(E) and Fig. 5. We observe that over 80% of variants fail to achieve statistically significant improvements. Even in the most heavily studied modalities, such as Ultrasound, Endoscopy, Dermoscopy, CT, and MRI, most gains are marginal and lack significance. Only a handful of datasets (e.g., BUSI, TNSCUI, Kvasir, ISIC2018, Convidquex) exhibit consistent clusters of superior vari-

Table 2: **Top-10 variants ranked by performance (IoU) and efficiency (U-Score) under in-domain and zero-shot settings.** Variants cover CNN, Transformer, Mamba, RWKV, and Hybrid architectures.

| In-domain (IoU) | | Zero-shot (IoU) | | In-domain (U-Score) | | Zero-shot (U-Score) | |
|---|---|---|---|---|---|---|---|
| Rank | Variants | Rank | Variants | Rank | Variants | Rank | Variants |
| 🥇 | RWKV-UNet | 🥇 | RWKV-UNet | 🥇 | LGMSNet | 🥇 | LGMSNet |
| 🥈 | nnUNet | 🥈 | G-CASCADE | 🥈 | MBSNet | 🥈 | LV-UNet |
| 🥉 | AURA-Net | 🥉 | Swin-umamba | 🥉 | CMUNeXt | 🥉 | U-KAN |
| #4 | UTANet | #4 | MADGNet | #4 | LV-UNet | #4 | Mobile U-ViT |
| #5 | ResEncUNet-L | #5 | CASCADE | #5 | Mobile U-ViT | #5 | RWKV-UNet |
| #6 | MADGNet | #6 | MFMSNet | #6 | Tinyunet | #6 | MBSNet |
| #7 | Swin-umamba | #7 | TransResUNet | #7 | U-RWKV | #7 | SwinUNETR |
| #8 | MFMSNet | #8 | DS-TransUNet | #8 | U-KAN | #8 | CMUNeXt |
| #9 | TransResUNet | #9 | CENet | #9 | DCSAU-Net | #9 | G-CASCADE |
| #10 | EViT-UNet | #10 | PraNet | #10 | RWKV-UNet | #10 | TA-Net |
| #25 | U-Net | #11 | ResEncUNet-L | #24 | ResEncUNet-L | #32 | ResEncUNet-L |
| | | #24 | nnUNet | #25 | nnUNet | #33 | nnUNet |
| | | #72 | U-Net | #64 | U-Net | #69 | U-Net |

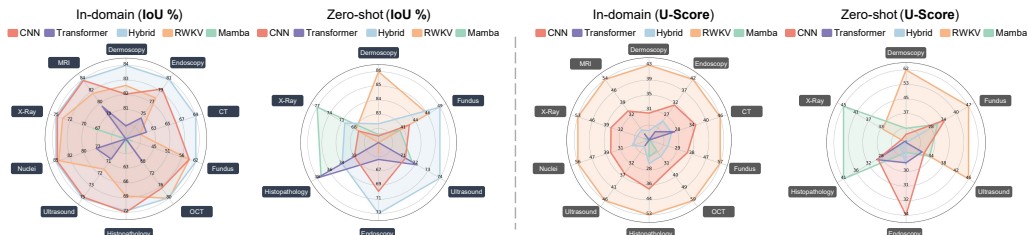

Figure 6: **Average performance of different architectures across modalities under in-domain and zero-shot settings.** Left: IoU-based comparison; Right: U-Score-based comparison of different architecture strengths in source-domain and zero-shot.

ants. In contrast, in experiments with zero-shot transfer, for variants that outperform U-Net, more than 50% of the variants are significant across 75% of the modalities.

**Possible explanations for findings 1 & 2:** We provide a possible explanation for these interesting observations. Considering in-domain evaluations, the improvements with statistical significance are typically associated with the lesion localization tasks which requires global semantic comparison. Specifically, lesions often exhibit significant differences from surrounding normal tissues, requiring a global context to model these distinctions (Zhou et al., 2017; Isensee et al., 2021). In recent years, with the growing adoption of long-range modeling techniques (e.g., attention-based Transformers, state-space models such as Mamba, and RNN-inspired hybrids like RWKV), architectural innovations have increasingly focused on capturing long-range dependencies, leading to more pronounced and steady improvements in these lesion segmentation tasks. On the other hand, long-range modeling techniques have been proven to be more generalizable (Jiang et al., 2024a; Harun et al., 2024; Hou et al., 2025; Gu et al., 2024), leading to improvements in zero-shot generalization ability. By contrast, modalities dominated by repetitive local patterns (e.g., Nuclei, Fundus) benefit far less from global modeling, exhibiting only marginal improvements. This underscores the complementary need for localized mechanisms to achieve precise boundary delineation.

**Finding 3: Increasing attention on efficiency.** We report the IoU and U-Score for all models with Parameter, FLOPs, and FPS, as shown in Fig. 3 and Fig 1(F). While IoU shows marginal improvements, suggesting a saturation point, illustrating that performance is no longer the key bottleneck in medical segmentation tasks, U-Score improvements are more pronounced, with an average increase of 33%. The increasing trend of U-Score reflects the growing emphasis and room for improvement on efficient models in the medical community, echoing the practical demand for clinical deployment beyond the lab research.

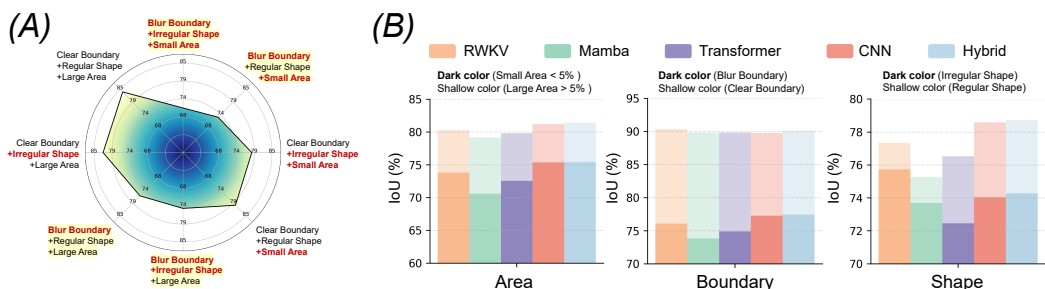

Figure 7: **Performance analysis under varying foreground properties.** (A) Foreground properties influence segmentation task difficulty. The yellow background indicates the challenge segmentation case. (B) Architectural influence on segmentation difficulty across diverse foreground properties. Dark / Shallow: hard / easy case.

## 3.2 INFLUENCING FACTOR ANALYSIS: ARCHITECTURES AND DATA CHARACTERISTICS

### 3.2.1 ARCHITECTURES

To analyze the performance regarding architectural choices, we divide 100 models into five families: CNN, Transformer, Mamba, RWKV, and Hybrid. The detailed descriptions are summarized in Appendix D. We present the top-10 variants across all datasets ranked by IoU and U-Score under in-domain and zero-shot settings, as shown in Tab. 2, and we calculate the average performance of each architecture family, as shown in Fig. 6.

Considering segmentation performance (IoU), Hybrid architectures achieve the highest accuracy by combining local priors with global attention. As shown in Tab. 2(Left), 5 of the top 10 models in both in-domain and zero-shot are hybrid, highlighting their high potential. On average, the hybrid family consistently delivers the best in-domain performance and competitive zero-shot generalization (Fig. 6(Left)), particularly excelling on lesion-centric tasks such as Ultrasound and Endoscopy. The newly proposed RWKV family ranks first in IoU for both in-domain and zero-shot evaluations, indicating promising potential despite limited prior research. In contrast, Mamba family shows weaker segmentation performance, which may be attributed to its architectural design, which, despite its strengths in certain tasks, might struggle with capturing fine-grained details or handling complex patterns in segmentation tasks.

Once computational demands are taken into account, as shown in Tab. 2(Right), the U-Score-based leaderboard is reshuffled, with the CNN family leading in performance, comprising 7 / 5 out of the top 10 models in in-domain / zero-shot settings, respectively. The newly proposed RWKV family achieves the best average in-domain results and competitive zero-shot performance (Fig. 6(Right)), further supporting its structural superiority and potential. In contrast, inefficient long-range modeling methods, including Transformer, and Hybrid architectures, face higher computational demands, leading to reduced performance when evaluated by U-Score. Although Mamba excels in efficiency, its inconsistent accuracy undermines the U-Score, offsetting its efficiency advantage.

### 3.2.2 DATA CHARACTERISTICS

We further investigate how performances vary with distinct foreground characteristics with three aspects: foreground scale, boundary sharpness, and shape complexity. The Appendix F.1 provides detailed definitions for the different scales of target area, edge, and shape regularity.

Figure 7(A) summarizes the characteristics of challenging cases: blurry boundaries are the dominant factor, with often causing substantial drops in segmentation performance, while small object size and irregular shapes further exacerbate the difficulty. When these foreground properties shift across datasets, different models exhibit varying performance patterns. As shown in Fig. 7(B), consistent with our earlier findings, hybrid architectures dominate both in easier and more challenging cases, proving that local and global fusion mechanism enables greater adaptability across diverse foreground properties, particularly for blurry boundaries. RWKV-based models show specific strength in capturing irregular but well-defined shapes, reflecting their ability to model long-range contours.

Nonetheless, boundary ambiguity, along with small and irregular targets, remains the central challenge; given its prevalence in medical images, uncertainty-aware designs are needed. Since architectural strengths are dataset-dependent, these observations highlight the importance of task-aware advising mechanisms that can match models to dataset properties.

### 3.3 MODEL ADVISOR AGENT

Based on our analysis, we introduce a ranking-based model advisor agent, designed to guide the community in selecting the most suitable models based on dataset characteristics and task requirements. This tool not only streamlines model selection but also helps users navigate the trade-offs between performance and efficiency, ensuring more informed, task-aware decisions.

The system overview is shown in Fig. 8. Our advisor agent system utilizes dataset-level characteristics (e.g., modality, boundary sharpness, shape complexity, and foreground scale) along with resource constraints (storage, computation, and speed) to predict the suitability of various U-shaped architectures. Rather than relying on a manual trial-and-error approach, our framework leverages XGBoost (Chen & Guestrin, 2016) as the recommended backbone and outputs candidate models and architectures that best satisfy the specified requirements. Crucially, the output is not a single "best" model but a prioritized list, offering more flexibility in choices to practitioners. Further details on the recommendation setup, dataset construction, implementation details, and evaluation metrics are provided in the Appendix F.2 and F.3.

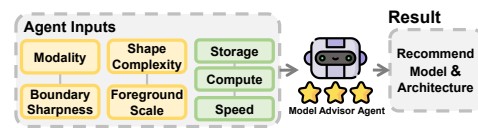

Figure 8: Our model advisor agent.

We design a set of experiments to validate the feasibility of automatic model suggestion in medical image segmentation. Our setup uses 18 in-domain datasets for training and holds out 2 datasets for validation. We use Normalized Discounted Cumulative Gain (NDCG), mean average precision (MAP) and Spearman correlation for evaluation (See Appendix F.3). As shown in Tab. 3, our experiments demonstrate that the proposed model advisor agent effectively recovers ranking orders that align with ground-truth IoU and U-Score rank in our benchmark. The results validate that our advisor agent system is able to prioritize suitable models across different task requirements, making it a reliable tool for model selection and deployment.

Table 3: NDCG, MAP, and Spearman correlation of our advisor agent.

| Ranking Metric | NDCG | | MAP | Spearman |
|---|---|---|---|---|
| | @5 | @20 | | |
| IoU | 0.75 | 0.76 | 0.24 | 0.36 |
| U-Score | 0.74 | 0.79 | 0.43 | 0.52 |

## 4 CONCLUSION

**Conclusion.** A key challenge in the field of medical image segmentation remains: How can we conduct a fair and comprehensive comparison across the numerous U-shaped variants? To address this, we introduce U-Bench, a framework that fills critical gaps in prior evaluations by offering a comprehensive, statistically rigorous, and efficiency-oriented approach. Our results challenge common assumptions in the field, revealing that while many variants show performance gains, few achieve statistical significance in-domain. In contrast, zero-shot generalization demonstrates substantial improvements, highlighting the potential for better model generalization across domains. Additionally, the newly proposed U-Score metric, which emphasizes efficiency alongside performance, signals a paradigm shift from models focused solely on accuracy to those that balance both performance and computational efficiency. Leveraging insights from our analysis of model architecture and dataset characteristics, we propose a ranking-based model recommender that transforms our large-scale evaluation into actionable guidance for selecting models tailored to specific tasks. By releasing U-Bench as an open-source platform, we provide the community with a robust, reproducible tool to advance research in medical image segmentation, enabling the development of models that are both accurate and computationally feasible for clinical deployment.

**Furture Work.** In future work, we aim to incorporate clinical perspectives in evaluating the trade-off between segmentation performance and computational efficiency. This could help guide the development of a clinical-driven metric to better align with real-world clinical needs.

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

## A    APPENDIX

In this appendix, we provide additional details and results to complement the main paper. The content is organized as follows:

**Appendix B**: Relate Work.

**Appendix C**: Details of U-Bench Data Zoo.

**Appendix D**: Details of U-Bench Model Zoo.

**Appendix E**: Details of U-Score.

**Appendix F**: Implementation and Evaluation Details.

**Appendix G**: Additional Results.

**Appendix J**: Reproducibility Checklist.

# B  RELATED WORK

In Appendix B, we present a broad view of the variations of U-shape networks, including the architecture of the network and existing medical segmentation benchmarks.

## B.1  MODEL ARCHITECTURE

As the core architecture for medical image segmentation, the U-Net has evolved into numerous variants in recent years, driven by advancements in feature representation capabilities, long-range dependency modeling techniques, and the trade-off between efficiency and accuracy. This section categorizes and organizes these U-Net variants based on their core paradigms and design motivations, systematically tracing their evolutionary path from foundational construction to integrated innovation. Fig. 9 summarizes the evolution of U-Net variants over time.

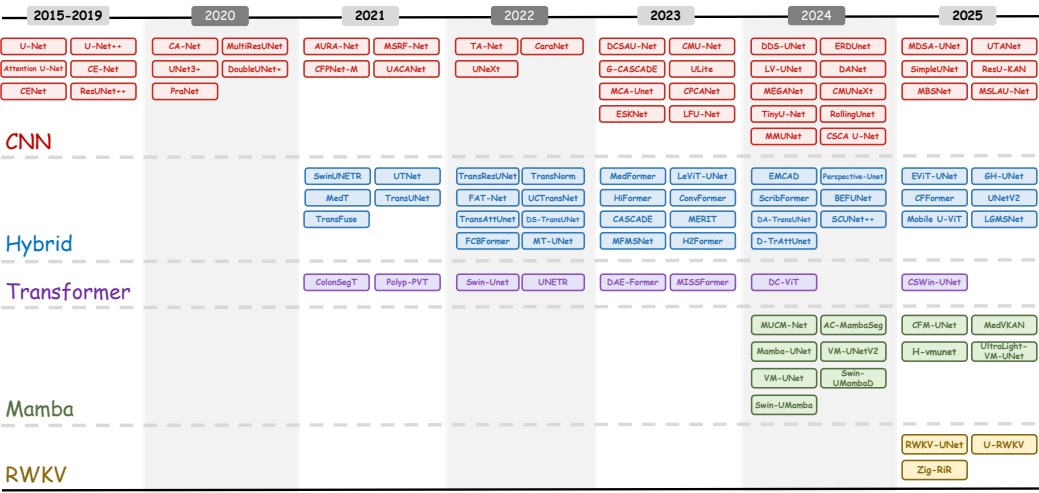

Figure 9: Time and architecture distribution of all evaluated models.

1. **U-shaped Networks Dominated by Convolutional Neural Networks (CNNs)** (2015-2021: Foundational Laying)

   U-shaped networks, using convolutional neural networks as their sole backbone, extract local features through convolution operations and utilize fixed skip connections to fuse multi-scale information, laying the foundation for the "encoder-decoder" paradigm in medical image segmentation. Their advantages lie in their ability to accurately capture local details (such as textures and edges), their relatively lightweight architecture, and their stable training process, providing a solid design baseline for subsequent complex variants. However, the local receptive field of convolutional operations limits the ability to model global semantic information, and fixed skip connections easily lead to a semantic gap between the encoder and decoder, limiting performance.

2. **Transformer-driven U-Networks** (2021-2023: Paradigm Shift)

   This variant introduces the Transformer architecture (including variants such as Vision Transformer (Dosovitskiy et al., 2020) and Swin Transformer (Liu et al., 2021)) to replace or enhance the traditional CNN backbone, leveraging the self-attention mechanism to effectively model long-range dependencies. However, the computational complexity of the self-attention mechanism grows quadratically with sequence length, resulting in low inference efficiency. Furthermore, this type of model is poorly adaptable to small-scale medical datasets and is prone to overfitting due to insufficient data, making it difficult to meet the stringent real-time requirements of clinical edge devices.

3. **U-Networks Based on State-Space Models (SSMs) and Recurrent Paradigms** (2023-2025: Efficiency-Oriented)

Recent research explores replacing the quadratic-cost self-attention with linear-time alternatives. One line leverages state-space models (e.g., Mamba (Gu & Dao, 2023)) that adopt selective state updates to capture long-range dependencies while achieving linear complexity, markedly improving inference efficiency and adaptability to small sample sizes. Another complementary line introduces RWKV (Peng et al., 2023), a recurrent-inspired model that combines Transformer-like expressiveness with RNN-style recurrence, enabling efficient sequential processing and stronger generalization across varying input lengths. Together, these paradigms alleviate the computational and data-dependency limitations of Transformers.

4. **Multi-Paradigm Fusion U-Networks (Hybrid Networks)** (2020-2025: Fusion and Innovation)

This phase aims to integrate the advantages of CNNs in local feature extraction, the global semantic modeling capabilities of Transformers. The goal is to achieve a balance between accuracy, efficiency, and generalization by fusing different architectures. This type of network variant can adapt to complex clinical scenarios such as multimodal imaging and cross-center data heterogeneity, significantly improving the practical value of segmentation results. However, the architectural design complexity increases significantly, and the coordination mechanisms between modules of different paradigms (such as the timing of feature interactions and weight distribution) still need further optimization.

The development of the four types of U-shaped network variants follows a technological evolutionary path of "local refinement → global correlation → efficiency considerations → multi-paradigm collaboration," reflecting the shift in clinical needs from static, single-scenario segmentation toward more efficient, generalized solutions adaptable across diverse conditions.

## B.2 MEDICAL SEGMENTATION BENCHMARKS

To fill the research gap in the evaluation of U-net systems, we comprehensively compare previous segmentation evaluation benchmarks with the U-Bench proposed in this paper, thereby clarifying the innovative positioning of U-Bench.

### B.2.1 RELATED WORK

Medical image segmentation has seen rapid progress, driven by deep learning architectures and large-scale datasets. However, the validity and reproducibility of many reported advances have been challenged due to inconsistent evaluation protocols, limited dataset diversity, and insufficient consideration of deployment constraints.

TorchStone (Bassi et al., 2024) addressed some of these limitations by introducing a large-scale collaborative benchmark for abdominal organ segmentation, leveraging diverse CT scans from multiple hospitals worldwide. While it emphasized out-of-distribution generalization and standardized evaluation, its scope was limited to a single anatomical region and modality, making it less suitable for assessing broader architectural capabilities. MedSegBench (Kuş & Aydin, 2024b) expanded coverage across modalities, incorporating 35 datasets from ultrasound, MRI, X-ray, and others. It provided standardized splits and evaluated multiple encoder-decoder variants, aiming to foster universal segmentation models. However, its focus remained on a smaller set of architectures and lacked comprehensive analysis of robustness, efficiency, and cross-paradigm comparisons. nnWNet (Zhou et al., 2025) proposed architectural modifications to integrate convolutions and transformers within a U-Net framework, addressing the need for continuous transmission of local and global features. Although it benchmarked on multiple 2D and 3D datasets, its evaluation was limited to a small number of models and lacked systematic efficiency analysis. nnU-Net Revisited (Isensee et al., 2024) critically examined recent architectural claims, showing that properly configured CNN-based U-Nets could still match or outperform newer transformer and Mamba-based models when trained with sufficient resources. This study highlighted the importance of rigorous baselines and computational reproducibility, yet it did not provide a multi-modal, multi-dataset framework for comparing a large number of variants. Collectively, these efforts underscore the need for a unified, statistically rigorous, and comprehensive benchmark that systematically evaluates a broad spectrum of U-Net variants across diverse modalities, datasets, and deployment metrics.

### B.2.2 TARGETED IMPROVEMENTS OF U-BENCH

As summarized in Tab. 1, existing medical image segmentation benchmarks suffer from limited modality coverage, insufficient evaluation diversity, narrow architectural scope, and lack of dataset-specific analysis-all of which hinder comprehensive assessment of model generalization. To address these gaps, U-Bench is designed with three targeted innovations, establishing a more comprehensive and clinically relevant evaluation framework while aligning with its core goals: evaluating 100 U-Net variants across 28 datasets and 10 modalities, introducing the performance-efficiency balanced U-Score, and enabling fair, reproducible benchmarking.

1. **Multimodality and Full Task Coverage**

   U-Bench encompasses 10 major medical imaging modalities (ultrasound, dermoscopy, endoscopy, fundus, histopathology, nuclei, X-Ray, MRI, CT, OCT) and integrates 28 datasets (sample sizes: 20-17,000). It covers tasks from macroscopic organ segmentation (e.g., lung CT, cardiac MRI) to microscopic structure segmentation (e.g., histopathological nuclei, retinal microvasculature), with standardized train/test splits. This design tests cross-modality adaptability of models, matching real-world clinical multimodal diagnostic workflows.

2. **Multi-Dimensional Evaluation System**

   Beyond traditional accuracy metrics (IoU, Dice), U-Bench introduces three critical evaluation dimensions and a unified U-Score to quantify clinical utility: *Computational Efficiency*: Standardized reporting of model parameters (M), inference FLOPs (G), and FPS to reflect deployability on resource-constrained devices. *Generalization Performance*: Zero-shot transfer tests on 8 unseen target datasets (distinct from 20 training source datasets) to assess robustness to domain shifts (e.g., cross-center ultrasound, unseen dermoscopic lesions). *Statistical Significance*: Paired t-tests between each variant and the original U-Net ($p < 0.05$ as significant) to validate reliable performance gains. *U-Score*: A comprehensive metric using quantile normalization and weighted harmonic mean to balance accuracy and efficiency, bridging academic performance and clinical deployment value.

3. **Large-Scale Reproducible Validation**

   U-Bench includes 100 publicly available U-Net variants, covering mainstream architectures from 2015 to 2025 (CNN, Transformer, Mamba, RWKV, hybrid designs). To ensure reproducibility, all models adopt official implementations, pre-trained weights (if available), and deep supervision strategies (if applicable).

## C DETAILS OF DATA ZOO

We summarize the dataset statistics used in this paper in Table 4. This table details the datasets used for experimental evaluation, covering 10 core imaging modalities, including ultrasound (e.g., BUSI), dermoscopy (e.g., ISIC2018), endoscopy (e.g., Kvasir-SEG), fundus (e.g., CHASE), histopathology (e.g., Glas), nuclear (e.g., DSB2018), X-ray (e.g., Montgomery), MRI (e.g., ACDC,), CT (e.g., Synapse), and OCT (e.g., Cystoidfluid). For each dataset, we provide key information such as the segmentation class (binary or multiclass), the number of samples, the year of publication, and a basic description. All datasets used are publicly available. Therefore, we provide access links in the relevant references and supplementary tables. The details are available in Tab. 4. A brief description of the dataset is as follows:

**BUSI.** The Breast Ultrasound Images (BUSI) dataset (Al-Dhabyani et al., 2020), collected from 600 female patients in 2018, contains 133 normal, 487 benign, and 210 malignant cases with corresponding ground truth labels. The data labels are obtained using ultrasound scans to examine breast cancer lesion areas.

**BUS.** The Breast UltraSound (BUS) public dataset (Zhang et al., 2022) includes 562 images (306 benign, 256 malignant) collected via five ultrasound devices, used for generalization experiments. The data labels are obtained using ultrasound scans to examine breast cancer lesion (or non-lesion) areas.

**BUSBRA.** The BUS-BRA dataset (Gómez-Flores et al., 2024) comprises 1875 anonymized images from 1064 patients (corresponding to 722 benign and 342 malignant cases) acquired via four ultra-

sound scanners. The data labels are obtained using ultrasound scans to examine breast cancer lesion (or non-lesion) areas.

**TNSCUI.** The Thyroid Nodule Segmentation and Classification in Ultrasound Images 2020 dataset[1] includes 3644 cases from the Chinese Artificial Intelligence Alliance for Thyroid and Breast Ultrasound. The data label is the thyroid nodule area obtained by thyroid ultrasound.

**TUCC.** The Thyroid Ultrasound (TUCC) dataset[2] collects data from 167 patients, including 192 biopsy-confirmed nodules. The data label is the thyroid nodule area obtained by thyroid ultrasound.

**ISIC2018.** The ISIC 2018 dataset (Codella et al., 2018) is a large-scale dermoscopy dataset for lesion segmentation, containing 2594 skin lesion images. The data label is the melanoma (or non-lesion) area of the skin disease obtained by dermoscopy imaging.

**PH2.** The PH$^2$ database (Mendonça et al., 2013) includes 200 dermoscopic images with manual segmentation and clinical diagnosis. The data label is the melanoma (or non-lesion) area of the skin disease obtained by dermoscopy imaging.

**SkinCancer.** The SkinCancer dataset (Kuş & Aydin, 2024a) contains 206 dermoscopic samples extracted from DermIS and DermQuest. The data label is the melanoma (or non-lesion) area of the skin disease obtained by dermoscopy imaging

**Covidquex.** The Covidquex dataset (Kuş & Aydin, 2024a) includes 2,913 chest X-ray images ($256 \times 256$ pixels) for binary segmentation. The dataset is labeled with COVID-infected areas on chest X-rays.

**Montgomery.** The Montgomery dataset (Jaeger et al., 2014) contains 138 chest X-rays (80 normal, 58 with tuberculosis). The data label is the tuberculosis lesion (or non-lesion) area on the lung X-ray.

**NIH-test.** The NIH-test dataset (Tang et al., 2019) is a manually annotated chest X-ray dataset with 100 lung masks. The data labels are lung segmentations from chest X-rays.

**DCA.** The DCA dataset (Kuş & Aydin, 2024a) contains 134 fundus images ($300 \times 300$ pixels). The data label is the blood vessel segmentation of the fundus image.

**Kvasir.** The Kvasir dataset (Jha et al., 2020b) contains 1000 gastrointestinal polyp images and corresponding ground truth. The data labels are pathological areas of gastrointestinal endoscopic imaging.

**CVC-300.** The CVC-300 dataset (Vázquez et al., 2017) comprises 60 colonoscopy polyp images ($500 \times 574$ pixels). The data labels are pathological areas of gastrointestinal endoscopic imaging.

**CVC-ClinicDB.** The CVC-ClinicDB dataset (Bernal et al., 2015) includes 612 images from 29 colonoscopy sequences. The data labels are pathological areas of gastrointestinal endoscopic imaging.

**Robotool.** The Robotool dataset (Kuş & Aydin, 2024a) consists of 500 images extracted from multiple surgical videos. The data label is the instrument area imaged by the endoscope.

**Promise.** The Promise dataset (Kuş & Aydin, 2024a) includes 1,473 prostate MRI samples ($512 \times 512$ pixels).

**ACDC.** The ACDC dataset (Bernard et al., 2018) contains 100 cardiac MRI scans. The data labels for left ventricle (LV), right ventricle (RV), and myocardium (MYO) in heart segmentation.

**CHASE.** The CHASE dataset (Fraz et al., 2012; Guo et al., 2021) includes 28 retinal images (one per eye from 14 children). The data label is the vascular area of the fundus image.

**Stare.** The Stare dataset (Hoover et al., 2000) includes 20 ocular fundus vessel images with manual annotations. The data label is the vascular area of the fundus image.

**DRIVE.** The DRIVE dataset (Staal et al., 2004) is collected from a Dutch diabetic retinopathy screening program. The data label is the vascular area of the fundus image.

---

[1]Available at: https://tn-scui2020.grand-challenge.org/Home.

[2]Available at: https://aimi.stanford.edu/datasets/thyroid-ultrasound-cine-clip.

Table 4: Dataset information summary, where 'O' in split type represents ourself-split, and 'S' represents splitting by data source

| Modal | Dataset | Category | Quantity | Year | Split type | Source |
|---|---|---|---|---|---|---|
| Ultrasound | BUSI | Binary | 0.5k~1k | 2020 | O | [link] |
| | BUS | Binary | 0.5k~1k | 2022 | O | [link] |
| | BUSBRA | Binary | 1k~2k | 2024 | O | [link] |
| | TNSCUI | Binary | 3k~4K | 2020 | O | [link] |
| | TUCC | Binary | 10k~20k | - | O | [link] |
| Dermoscopy | ISIC2018 | Binary | 2k~3k | 2018 | O | [link] |
| | PH$^2$ | Binary | <0.5k | 2013 | S | [link] |
| | SkinCancer | Binary | 206 | 2024 | S | [link] |
| X-Ray | Covidquex | Binary | 2k ~ 3k | 2021 | S | [link] |
| | Montgomery | Binary | <0.5k | 2014 | S | [link] |
| | NIH-test | Binary | <0.5k | 2019 | S | [link] |
| | DCA | Binary | <0.5k | 2019 | S | [link] |
| Endoscopy | Kvasir-SEG | Binary | 1k~2k | 2020 | S | [link] |
| | CVC-300 | Binary | <0.5k | 2017 | S | [link] |
| | CVC-ClinicDB | Binary | 0.5k~1k | 2015 | S | [link] |
| | Robotool | Binary | 0.5k~1k | 2021 | S | [link] |
| MRI | Promise | Binary | 1k~2k | 2024 | S | [link] |
| | ACDC | 4-Class | <0.5k | 2018 | S | [link] |
| Fundus | CHASE | Binary | <0.5k | 2012 | S | [link] |
| | Stare | Binary | <0.5k | 2000 | S | [link] |
| | DRIVE | Binary | <0.5k | - | S | [link] |
| CT | Synapse | 9-Class | 3k~4k | 2023 | S | [link] |
| OCT | Cystoidfluid | Binary | 1k~2k | 2024 | S | [link] |
| Nuclear | DSB2018 | Binary | 0.5k~1k | 2018 | S | [link] |
| | Cell | Binary | 0.5k~1k | 2018 | S | [link] |
| Histopathology | Monusac | Binary | <0.5k | 2016 | S | [link] |
| | Tnbcnuclei | Binary | <0.5k | 2018 | S | [link] |
| | Glas | Binary | <0.5k | 2015 | S | [link] |

**Cell.** The Cell dataset (Kuş & Aydin, 2024a) consists of 670 nuclei images with a resolution of 320×256 pixels. The data label is the cell nucleus segmentation area.

**Glas.** The Glas dataset (Sirinukunwattana et al., 2015) contains 165 H&E stained slide images for gland segmentation. The data label is the glandular lesion (or non-lesion) area of the Hematoxylin and Eosin image.

**Monusac.** The Monusac dataset (Kuş & Aydin, 2024a) includes 310 H&E stained digital tissue images. The data labels are the nucleus regions of H&E stained histology images.

**Tnbcnuclei.** The Tnbcnuclei dataset (Kuş & Aydin, 2024a) contains 50 pathological samples for binary segmentation. The data labels are the cell nucleus regions of Hematoxylin and Eosin stained histology images.

**Synapse.** The Synapse multi-organ dataset includes 30 abdominal CT scans with 8-organ segmentation. The data labels are 8 abdominal organs (aorta, gallbladder, left kidney, right kidney, liver, pancreas, spleen, stomach).

**Cystoidfluid.** The Cystoidfluid dataset (Kuş & Aydin, 2024a) contains 1,006 Optical Coherence Tomography images. The dataset is labeled the Cystoid Macular Edema (CME) region of the retina.

Table 5: Hybrid architecture model comparison.

| Model | Deep Supervision | Pre-training | Zero-shot | P-value | Params (M) | FLOPs (G) | FPS |
|---|:---:|:---:|:---:|:---:|---:|---:|---:|
| BEFUNet (Manzari et al., 2024) | | ✓ | | ✓ | 42.61 | 7.95 | 69.89 |
| CASCADE (Rahman & Marculescu, 2023a) | ✓ | ✓ | | | 35.27 | 8.15 | 57.91 |
| CFFormer (Li et al., 2025b) | | | ✓ | ✓ | 158.44 | 71.17 | 30.28 |
| DA-TransUNet (Sun et al., 2024) | | | | ✓ | 2.60 | 6.92 | 67.48 |
| DS-TransUNet (Lin et al., 2022) | ✓ | ✓ | ✓ | | 171.34 | 51.15 | 24.28 |
| D-TrAttUnet (Bougourzi et al., 2024) | ✓ | | | ✓ | 104.16 | 54.00 | 53.85 |
| EMCAD (Rahman et al., 2024) | ✓ | ✓ | | | 26.76 | 5.60 | 56.17 |
| EViT-UNet (Li et al., 2025d) | | ✓ | | | 54.79 | 8.36 | 16.73 |
| FAT-Net (Wu et al., 2022) | | ✓ | | | 29.62 | 42.80 | 76.01 |
| FCNFormer (Sanderson & Matuszewski, 2022) | | ✓ | ✓ | | 52.94 | 40.88 | 25.70 |
| GH-UNet (Wang et al., 2025b) | ✓ | | | ✓ | 12.81 | 21.58 | 14.61 |
| H2Former (He et al., 2023) | | | | | 33.63 | 32.25 | 55.26 |
| HiFormer (Heidari et al., 2023) | | ✓ | | | 34.14 | 17.75 | 68.12 |
| LeViT-UNet (Xu et al., 2023a) | | ✓ | | | 17.53 | 27.24 | 102.91 |
| LGMSNet (Dong et al., 2025) | | | ✓ | ✓ | 2.32 | 4.89 | 105.04 |
| MedFormer (Gao et al., 2023) | | | | | 28.07 | 21.79 | 59.85 |
| MedT (Valanarasu et al., 2021) | | ✓ | | | 1.37 | 2.41 | 5.15 |
| MERIT (Rahman & Marculescu, 2023c) | | ✓ | | ✓ | 147.68 | 33.28 | 18.69 |
| MFMSNet (Wu et al., 2023) | ✓ | ✓ | ✓ | | 31.56 | 10.08 | 13.44 |
| Mobile U-ViT (Tang et al., 2025b) | | | ✓ | | 6.21 | 10.43 | 96.80 |
| MT-UNet (Wang et al., 2022c) | | | | ✓ | 75.07 | 57.72 | 11.23 |
| Perspective-Unet (Hu et al., 2024) | | | | | 111.08 | 124.48 | 41.80 |
| ScribFormer (Li et al., 2024b) | ✓ | | | ✓ | 47.91 | 44.63 | 35.25 |
| SCUNet++ (Chen et al., 2024b) | | ✓ | | | 43.54 | 16.68 | 59.66 |
| SwinUNETR (Hatamizadeh et al., 2021) | | | | | 6.29 | 4.86 | 84.41 |
| TransAttUnet (Chen et al., 2022) | | ✓ | | | 22.65 | 88.78 | 99.76 |
| TransFuse (Zhang et al., 2021) | ✓ | ✓ | | | 26.17 | 11.53 | 59.97 |
| TransNorm (Azad et al., 2022) | | ✓ | | | 105.59 | 39.28 | 42.59 |
| TransResUNet (Tomar et al., 2022) | | ✓ | | | 27.07 | 24.06 | 85.84 |
| TransUNet (Chen et al., 2021) | | ✓ | | | 93.23 | 32.23 | 58.45 |
| UCTransNet (Wang et al., 2022a) | | | | ✓ | 66.24 | 43.06 | 35.12 |
| UNetV2 (Peng et al., 2025) | ✓ | | ✓ | ✓ | 25.13 | 5.40 | 60.33 |
| UTNet (Gao et al., 2021) | | | | | 14.41 | 20.49 | 76.67 |

**DSB2018.** The DSB2018 dataset (Hamilton, 2018) includes 670 Hematoxylin and Eosin (H&E)-stained nuclear images. The data label is the cell nucleus segmentation area.

# D  DETAILS OF MODEL ZOO

We conducted a comprehensive statistical analysis of the 100 models evaluated by U-bench, as shown in Tab. 6, 5 and 7.

Tab. 6 and Tab. 5 summarize the basic information of the single architecture and hybrid architecture respectively., quantifying critical metrics including deep supervision adoption, pre-training status, zero-shot capability, statistical significance (P-value), parameter count (Params), computational cost (FLOPs), and inference speed (FPS);

Tab. 7 further clarifies the training foundation of all evaluated models, documenting their publication year, venue, target dataset modality, and open-source repository links for reproducibility.

# E  DETAILS OF U-SCORE

Clinical deployment of segmentation models often requires operation under constrained resources. However, existing evaluations focus predominantly on segmentation performance, while failing to balance key computational factors such as model size, inference cost, and speed. This disconnect makes it difficult to assess real-world deployability. To bridge this gap, we introduce U-Score, a unified metric that quantifies the trade-off between performance and efficiency using quantile statistics under large-scale benchmark. Specifically, we report the 10th and 90th percentiles of IoU, Params, FLOPs, and FPS, as summarized in Tab. 8 and 9. The formulation is represented as follow.

Given model $i$, we compute IoU $A_i$ across datasets, parameter $P_i$ in millions, FLOPs $G_i$ in GLOPs, and runtime speed $S_i$ in FPS. We normalize each component using the 10th and 90th percentiles computed over the model zoo. Let $Q_{10}^M$ and $Q_{90}^M$ denote the 10th and 90th percentiles of metric $M$.

Table 6: Single-architecture model comparison.

| Architecture | Model | Deep Supervision | Pre-training | Zero-shot | P-value | Params (M) | FLOPs (G) | FPS |
|---|---|---|---|---|---|---|---|---|
| CNN | AtU-Net (Oktay et al., 2018) | | | | ✓ | 34.88 | 66.63 | 126.09 |
| | AURA-Net (Cohen & Uhlmann, 2021) | | | | | 52.84 | 25.15 | 121.63 |
| | CA-Net (Gu et al., 2020) | | | | ✓ | 2.79 | 5.99 | 31.71 |
| | CaraNet (Lou et al., 2022) | ✓ | ✓ | | | 44.59 | 11.50 | 26.82 |
| | CENet (Gu et al., 2019b) | | | | | 33.36 | 10.64 | 23.63 |
| | CE-Net (Gu et al., 2019a) | | ✓ | | | 29.00 | 8.90 | 103.22 |
| | CFPNet-M (Lou et al., 2021) | | | | | 0.76 | 3.47 | 73.13 |
| | CMU-Net (Tang et al., 2023) | | | | | 49.93 | 91.25 | 83.28 |
| | CMUNeXt (Tang et al., 2024) | | | | | 3.15 | 7.42 | 161.14 |
| | CPCANet (Huang et al., 2023a) | | | | | 43.39 | 13.36 | 16.23 |
| | CSCA U-Net (Shu et al., 2024) | ✓ | | | | 35.27 | 13.74 | 44.99 |
| | DANet (Pramanik et al., 2024) | ✓ | | | ✓ | 94.51 | 33.24 | 48.18 |
| | DCSAU-Net (Xu et al., 2023b) | | | | | 10.81 | 23.83 | 56.84 |
| | DDS-UNet (Ou et al., 2025) | | | | | 43.62 | 17.40 | 36.87 |
| | DoubleUNetPlus (Jha et al., 2020a) | ✓ | | ✓ | | 29.29 | 53.96 | 100.99 |
| | ERDUnet (Li et al., 2024a) | | ✓ | | | 10.21 | 10.29 | 43.18 |
| | ESKNet (Chen et al., 2023) | | | ✓ | ✓ | 26.71 | 45.28 | 75.38 |
| | G-CASCADE (Rahman & Marculescu, 2023b) | ✓ | ✓ | | | 26.63 | 5.54 | 62.77 |
| | LFU-Net (Deng et al., 2023) | | | | | 0.05 | 0.76 | 167.91 |
| | LV-UNet (Jiang et al., 2024b) | | | | | 0.92 | 0.21 | 139.30 |
| | MALUNet (Ruan et al., 2022) | | | | | 0.18 | 0.08 | 108.64 |
| | MBSNet (Ye et al., 2021) | | | | | 3.98 | 6.86 | 115.10 |
| | MCA-Unet (Amer & Ye, 2023) | ✓ | | | | 8.66 | 58.02 | 12.26 |
| | MDSA-UNet (Li et al., 2025c) | | | | | 6.58 | 5.65 | 77.36 |
| | MEGANet (Bui et al., 2024) | ✓ | | | | 29.27 | 11.71 | 59.62 |
| | MMUNet (Yuan et al., 2024b) | | | | | 17.73 | 24.04 | 46.93 |
| | MSLAU-Net (Lan et al., 2025) | | ✓ | | | 21.88 | 6.27 | 35.34 |
| | MSRF-Net (Srivastava et al., 2021) | ✓ | | ✓ | ✓ | 22.50 | 109.73 | 33.16 |
| | MultiResUNet (Ibtehaz & Rahman, 2020) | | | | | 7.25 | 18.76 | 84.31 |
| | PraNet (Fan et al., 2020) | ✓ | ✓ | ✓ | | 50.01 | 11.96 | 27.55 |
| | ResNet34UnetPlus (Zhou et al., 2018) | | | | ✓ | 26.90 | 37.63 | 84.54 |
| | ResU-KAN (Wang et al., 2025a) | | | | | 18.59 | 7.78 | 67.56 |
| | ResUNetPlusPlus (Jha et al., 2019) | | | | | 14.48 | 70.99 | 91.72 |
| | RollingUnet (Liu et al., 2024c) | | | | | 7.10 | 8.28 | 31.85 |
| | SimpleUNet (Yu et al., 2025) | | | | | 0.06 | 0.74 | 414.31 |
| | TA-Net (Wang et al., 2022b) | | | | | 29.57 | 9.32 | 94.43 |
| | TinyU-Net (Chen et al., 2024a) | | | | | 0.48 | 1.66 | 150.32 |
| | UACANet (Kim et al., 2021) | ✓ | ✓ | ✓ | ✓ | 67.11 | 31.55 | 27.79 |
| | U-KAN (Li et al., 2025a) | | | | | 9.38 | 6.89 | 93.47 |
| | ULite (Dinh et al., 2023) | | | | | 0.88 | 0.76 | 323.06 |
| | U-Net (Ronneberger et al., 2015) | | | | | 34.53 | 65.52 | 137.05 |
| | UNet3+ (Huang et al., 2020) | | | | | 26.97 | 199.74 | 50.70 |
| | UNeXt (Valanarasu & Patel, 2022) | | | | ✓ | 1.47 | 0.57 | 256.68 |
| | UTANet (Luo et al., 2025) | | | | ✓ | 45.03 | 87.59 | 85.63 |
| | nnUNet (Isensee et al., 2021) | ✓ | | | ✓ | 92.46 | 115.79 | 314.56 |
| | ResEncUNet-L (Isensee et al., 2024) | ✓ | | | ✓ | 393.06 | 435.17 | 124.54 |
| Mamba | AC-MambaSeg (Nguyen et al., 2024) | | | | | 7.42 | 6.27 | 35.64 |
| | CFM-UNet (Niu et al., 2025) | | | ✓ | ✓ | 52.96 | 6.17 | 38.08 |
| | H-vmunet (Wu et al., 2025a) | | | | | 6.44 | 0.74 | 16.30 |
| | Mamba-UNet (Wang et al., 2024) | | ✓ | | | 15.48 | 4.60 | 94.47 |
| | MedVKAN (Zhu et al., 2025) | | | | | 43.58 | 14.13 | 45.91 |
| | MUCM-Net (Yuan et al., 2024a) | | | | | 0.08 | 0.06 | 107.24 |
| | Swin-UMamba (Liu et al., 2024a) | | ✓ | | | 55.06 | 43.93 | 58.52 |
| | Swin-UMambaD (Liu et al., 2024b) | | ✓ | | | 21.74 | 6.20 | 65.48 |
| | UltraLight-VM-UNet (Wu et al., 2025b) | | ✓ | | | 0.04 | 0.06 | 82.45 |
| | VM-UNet (Ruan & Xiang, 2024) | | ✓ | | ✓ | 34.62 | 7.56 | 48.08 |
| | VM-UNetV2 (Zhang et al., 2024b) | | ✓ | | | 17.91 | 4.40 | 62.85 |
| RWKV | RWKV-UNet (Jiang et al., 2025) | | ✓ | | | 17.10 | 14.58 | 76.14 |
| | U-RWKV (Ye et al., 2025) | | | | | 2.82 | 6.90 | 107.52 |
| | Zig-RiR (Chen et al., 2025) | | | | ✓ | 24.25 | 3.31 | 35.59 |
| Transformer | DC-ViT (Zhang et al., 2024a) | | | ✓ | | 6.84 | 20.87 | 65.34 |
| | ColonSegT (Jha et al., 2021) | | | | | 5.01 | 62.16 | 135.43 |
| | ConvFormer (Lin et al., 2023) | | ✓ | | | 115.61 | 121.13 | 43.05 |
| | CSWin-UNet (Liu et al., 2025) | | ✓ | | | 23.57 | 6.14 | 33.05 |
| | DAE-Former (Azad et al., 2023) | | ✓ | | ✓ | 29.69 | 34.10 | 55.23 |
| | MISSFormer (Huang et al., 2023b) | | ✓ | | ✓ | 35.45 | 7.25 | 57.68 |
| | Polyp-PVT (Dong et al., 2021) | ✓ | ✓ | ✓ | ✓ | 25.11 | 5.30 | 67.64 |
| | SwinUnet (Cao et al., 2022) | | ✓ | | | 41.34 | 8.69 | 63.25 |
| | UNETR (Hatamizadeh et al., 2022) | | | | ✓ | 87.51 | 26.41 | 104.25 |

Table 7: Training modal information of all evaluation models.

| Architecture | Model | Year | Publication | Modality | Github |
|---|---|---|---|---|---|
| | AttU-Net | 2018 | MIDL | CT | [link] |
| | AURA-Net | 2021 | ISBI | Microscopy | [link] |
| | CA-Net | 2020 | TMI | Dermoscopy, MRI | [link] |
| | CaraNet | 2022 | SPIE Medical Imaging | Colonoscopy, MRI | [link] |
| | CENet | 2025 | MICCAI | Dermoscopy, CT, MRI | [link] |
| | CE-Net | 2019 | TMI | Fundus, CT, Microscopy, OCT | [link] |
| | CFPNet-M | 2021 | Medical Imaging | Thermography, Microscopy, Colonoscopy, Dermoscopy, Fundus | [link] |
| | CMU-Net | 2023 | ISBI | Ultrasound | [link] |
| | CMUNeXt | 2024 | ISBI | Ultrasound | [link] |
| | CPCANet | 2023 | CMI | MRI, Dermoscopy | [link] |
| | CSCA U-Net | 2024 | AIIM | Colonoscopy, Pathology, Ultrasound | [link] |
| | DANet | 2024 | Plos one | Ultrasound | [link] |
| | DCSAU-Net | 2023 | CBM | Colonoscopy, Microscopy, Dermoscopy | [link] |
| | DDS-UNet | 2025 | The Visual Computer | Ultrasound, Dermoscopy, Colonoscopy | [link] |
| | DoubleUNetPlus | 2020 | IEEE CBMS | Colonoscopy, Dermoscopy, Microscopy | [link] |
| | ERDUnet | 2024 | TCSVT | Microscopy, Dermoscopy, Colonoscopy, Pathology, MRI | [link] |
| | ESKNet | 2023 | CMPB | Ultrasound | [link] |
| | G-CASCADE | 2023 | WACV | CT, MRI, Dermoscopy, Colonoscopy | [link] |
| | LFU-Net | 2023 | CMI | CT, MRI | [link] |
| | LV-UNet | 2024 | BIBM | Dermoscopy, Ultrasound, Colonoscopy | [link] |
| | MALUNet | 2022 | BIBM | Dermoscopy | [link] |
| | MBSNet | 2021 | MSSP | Dermoscopy, Ultrasound, Colonoscopy | [link] |
| CNN | MCA-Unet | 2023 | CMPBU | CT | [link] |
| | MDSA-UNet | 2025 | JBHI | Ultrasound, CT, Dermoscopy | [link] |
| | MEGANet | 2024 | WACV | Colonoscopy | [link] |
| | MMUNet | 2024 | BSPC | Histological image | [link] |
| | MSLAU-Net | 2025 | arXiv (cs.CV) | CT, MRI, Colonoscopy | [link] |
| | MSRF-Net | 2021 | JBHI | Colonoscopy, Microscopy, Dermoscopy | [link] |
| | MultiResUNet | 2020 | Neural networks | Microscopy, Dermoscopy, Colonoscopy, MRI | [link] |
| | PraNet | 2020 | MICCAI | Colonoscopy | [link] |
| | ResNet34UnetPlus | 2018 | TMI | Microscopy, CT, MRI | [link] |
| | ResUNetPlusPlus | 2019 | ISM | Colonoscopy | [link] |
| | RollingUnet | 2024 | AAAI | Ultrasound, Histological image, Dermoscopy, Fundus | [link] |
| | SimpleUNet | 2025 | arXiv | Ultrasound, Dermoscopy, Colonoscopy | [link] |
| | TA-Net | 2022 | WACV | Histological image | [link] |
| | TinyU-Net | 2024 | MICCAI | Dermoscopy, CT | [link] |
| | UACANet | 2021 | ACM MM | Colonoscopy | [link] |
| | ULite | 2023 | APSIPA | Dermoscopy, Microscopy, Histological image | [link] |
| | U-Net | 2015 | MICCAI | Microscopy, Microscopy | [link] |
| | UNet3+ | 2020 | ICASSP | CT | [link] |
| | UNeXt | 2022 | MICCAI | Dermoscopy, Ultrasound | [link] |
| | UTANet | 2025 | AAAI | Histology Image, Microscopy, Abdominal CT, Dermoscopy | [link] |
| | ResU-KAN | 2025 | Applied Intelligence | Ultrasound, Histological, Colonoscopy | [link] |
| | U-KAN | 2025 | AAAI | Ultrasound, Histological image, Colonoscopy | [link] |
| | Mamba-UNet | 2024 | CoRR | MRI, CT | [link] |
| | MedVKAN | 2025 | arxiv | Microscopy, MRI, Ultrasound, CT | [link] |
| | Swin-UMambaD | 2024 | TMI | MRI, Endoscopy, Microscopy | [link] |
| | UltraLight-VM-UNet | 2025 | Patterns | Dermoscopy | [link] |
| | VM-UNet | 2024 | CoRR | Dermoscopy, CT | [link] |
| Mamba | VM-UNetV2 | 2024 | ISBRA | Dermoscopy, Colonoscopy | [link] |
| | AC-MambaSeg | 2024 | ICGTSD | Dermoscopy | [link] |
| | CFM-UNet | 2025 | Scientific Reports | CT, MRI, Colonoscopy, MRI | [link] |
| | MUCM-Net | 2024 | CoRR | Dermoscopy | [link] |
| | Swin-UMamba | 2024 | MICCAI | MRI, Endoscopy, Microscopy | [link] |
| | H-vmunet | 2024 | arxiv | Colonoscopy,Dermoscopy,CT,MRI | [link] |
| | Zig-RiR | 2025 | TMI | Dermoscopy, CT, MRI, Microscopy | [link] |
| RWKV | RWKV-UNet | 2025 | CoRR | CT, MRI, Ultrasound, Colonoscopy, Dermoscopy, Histological image | [link] |
| | U-RWKV | 2025 | MICCAI | Ultrasound, Colonoscopy, Dermoscopy, CT | [link] |
| | DC-ViT | 2024 | CVPR | Natural images | [link] |
| | ColonSegT | 2021 | IEEE ACCESS | Colonoscopy | [link] |
| | CSWin-UNet | 2025 | Information Fusion | CT, MRI, Dermoscopy | [link] |
| Transformer | DAE-Former | 2023 | IWPIM | CT, Dermoscopy | [link] |
| | MISSFormer | 2023 | TMI | CT, MRI | [link] |
| | Polyp-PVT | 2021 | arXiv | Colonoscopy | [link] |
| | SwinUnet | 2022 | ECCVW | CT, MRI | [link] |
| | UNETR | 2022 | WACV | CT, MRI | [link] |
| | BEFUNet | 2024 | arXiv | CT, Microscopy, Dermoscopy | [link] |
| | CASCADE | 2023 | WACV | CT, MRI, Colonoscopy | [link] |
| | CFFormer | 2025 | ESA | Ultrasound, Dermoscopy, Colonoscopy, CT, MRI | [link] |
| | ConvFormer | 2023 | MICCAI | Ultrasound, Dermoscopy, CT | [link] |
| | DA-TransUNet | 2024 | FBB | CT, Colonoscopy, X-ray, Dermoscopy, Endoscopy | [link] |
| | DS-TransUNet | 2022 | TIM | Colonoscopy, Dermoscopy, Histology, Microscopy | [link] |
| | D-TrAttUnet | 2024 | CBM | CT, Histology Image, Microscopy | [link] |
| | EMCAD | 2024 | CVPR | Colonoscopy, Dermoscopy, Ultrasound, CT, MRI | [link] |
| | EViT-UNet | 2025 | ISBI | CT, Histology Image, Microscopy | [link] |
| | FAT-Net | 2022 | MIA | Dermoscopy | [link] |
| | FCNFormer | 2022 | MICCAI | Colonoscopy | [link] |
| | GH-UNet | 2025 | Digital Medicine | Dermoscopy, Colonoscopy, Fundus, MRI, CT | [link] |
| | H2Former | 2023 | TMI | Fundus, Colonoscopy, Dermoscopy, MRI, CT | [link] |
| | HiFormer | 2023 | WACV | CT, Dermoscopy, Microscopy | [link] |
| | LeViT-UNet | 2023 | PRCV | CT, MRI | [link] |
| | LGMSNet | 2025 | ECAI | Ultrasound, Dermoscopy, Colonoscopy, CT | [link] |
| Hybrid | MADGNet | 2024 | CVPR | Ultrasound, Dermoscopy, Colonoscopy, Microscopy, CT | [link] |
| | MedFormer | 2023 | arXiv | MRI, CT | [link] |
| | MedT | 2021 | MICCAI | Ultrasound, Histology Image, Microscopy | [link] |
| | MERIT | 2023 | MIDL | CT, MRI | [link] |
| | MFMSNet | 2023 | UMB | Ultrasound | [link] |
| | Mobile U-ViT | 2025 | ACM MM | Ultrasound, Dermoscopy, Colonoscopy, CT | [link] |
| | MT-UNet | 2022 | ICASSP | CT, MRI | [link] |
| | Perspective-Unet | 2024 | MICCAI | CT, MRI | [link] |
| | ScribFormer | 2024 | TMI | MRI, CT | [link] |
| | SCUNet++ | 2024 | WACV | CT | [link] |
| | SwinUNETR | 2021 | MICCAI | MRI | [link] |
| | TransAttUnet | 2022 | TIM | Dermoscopy, X-ray, CT, Biological Image, Histology Image | [link] |
| | TransFuse | 2021 | MICCAI | Colonoscopy, Dermoscopy, X-ray, MRI | [link] |
| | TransNorm | 2022 | IEEE Access | CT, Dermoscopy, Microscopy | [link] |
| | TransResUNet | 2022 | CoRR | Colonoscopy | [link] |
| | TransUNet | 2021 | arXiv | Abdominal CT, MRI | [link] |
| | UCTransNet | 2022 | AAAI | Histology Image, Microscopy, CT | [link] |
| | UNetV2 | 2025 | ISBI | Dermoscopy, Colonoscopy | [link] |
| | UTNet | 2021 | MICCAI | MRI | [link] |

Table 8: In-domain per-dataset 10th and 90th percentiles of IoU, Params, FLOPs, and FPS.

| Modality | Dataset | IoU (%) | | Params (M) | | FLOPs (G) | | FPS | |
|---|---|---|---|---|---|---|---|---|---|
| | | $Q_{10}^A$ | $Q_{90}^A$ | $Q_{10}^P$ | $Q_{90}^P$ | $Q_{10}^G$ | $Q_{90}^G$ | $Q_{10}^S$ | $Q_{90}^S$ |
| Ultrasound | BUSI | 0.58 | 0.71 | 0.39 | 4.32 | 0.88 | 4.20 | 24.28 | 121.63 |
| | BUSBRA | 0.78 | 0.84 | 0.39 | 4.32 | 0.88 | 4.20 | 24.28 | 121.63 |
| | TNSCUI | 0.66 | 0.78 | 0.39 | 4.32 | 0.88 | 4.20 | 24.28 | 121.63 |
| Dermoscopy | ISIC2018 | 0.81 | 0.84 | 0.39 | 4.32 | 0.88 | 4.20 | 24.28 | 121.63 |
| | SkinCancer | 0.79 | 0.85 | 0.39 | 4.32 | 0.88 | 4.20 | 24.28 | 121.63 |
| Endoscopy | Kvasir | 0.75 | 0.84 | 0.39 | 4.32 | 0.88 | 4.20 | 24.28 | 121.63 |
| | Robotool | 0.69 | 0.85 | 0.39 | 4.32 | 0.88 | 4.20 | 24.28 | 121.63 |
| Fundus | CHASE | 0.47 | 0.81 | 0.39 | 4.32 | 0.88 | 4.20 | 24.28 | 121.63 |
| | DRIVE | 0.15 | 0.62 | 0.39 | 4.32 | 0.88 | 4.20 | 24.28 | 121.63 |
| Nuclei | DSB2018 | 0.85 | 0.88 | 0.39 | 4.32 | 0.88 | 4.20 | 24.28 | 121.63 |
| | CellNuclear | 0.78 | 0.84 | 0.39 | 4.32 | 0.88 | 4.20 | 24.28 | 121.63 |
| Histopathology | Glas | 0.63 | 0.83 | 0.39 | 4.32 | 0.88 | 4.20 | 24.28 | 121.63 |
| | Monusac | 0.53 | 0.67 | 0.39 | 4.32 | 0.88 | 4.20 | 24.28 | 121.63 |
| X-Ray | Covidquex | 0.63 | 0.70 | 0.39 | 4.32 | 0.88 | 4.20 | 24.28 | 121.63 |
| | Montgomery | 0.92 | 0.96 | 0.39 | 4.32 | 0.88 | 4.20 | 24.28 | 121.63 |
| | DCA | 0.51 | 0.63 | 0.39 | 4.32 | 0.88 | 4.20 | 24.28 | 121.63 |
| MRI | ACDC | 0.73 | 0.85 | 0.39 | 4.32 | 0.88 | 4.20 | 24.28 | 121.63 |
| | Promise | 0.78 | 0.87 | 0.39 | 4.32 | 0.88 | 4.20 | 24.28 | 121.63 |
| CT | Synapse | 0.55 | 0.72 | 0.39 | 4.32 | 0.88 | 4.20 | 24.28 | 121.63 |
| OCT | Cystoidfluid | 0.63 | 0.83 | 0.39 | 4.32 | 0.88 | 4.20 | 24.28 | 121.63 |

Table 9: Zero-shot per-dataset 10th and 90th percentiles of IoU, Params, FLOPs, and FPS.

| Source | Target | IoU (%) | | Params (M) | | FLOPs (G) | | FPS | |
|---|---|---|---|---|---|---|---|---|---|
| | | $Q_{10}^A$ | $Q_{90}^A$ | $Q_{10}^P$ | $Q_{90}^P$ | $Q_{10}^G$ | $Q_{90}^G$ | $Q_{10}^S$ | $Q_{90}^S$ |
| BUSI | BUS | 0.60 | 0.81 | 0.39 | 4.32 | 0.88 | 4.20 | 24.28 | 121.63 |
| BUSBRA | BUS | 0.78 | 0.85 | 0.39 | 4.32 | 0.88 | 4.20 | 24.28 | 121.63 |
| TNSCUI | TUCC | 0.56 | 0.64 | 0.39 | 4.32 | 0.88 | 4.20 | 24.28 | 121.63 |
| ISIC2018 | PH2 | 0.82 | 0.85 | 0.39 | 4.32 | 0.88 | 4.20 | 24.28 | 121.63 |
| Kvasir | CVC300 | 0.61 | 0.80 | 0.39 | 4.32 | 0.88 | 4.20 | 24.28 | 121.63 |
| Kvasir | CVC-ClinicDB | 0.60 | 0.75 | 0.39 | 4.32 | 0.88 | 4.20 | 24.28 | 121.63 |
| CHASE | STARE | 0.30 | 0.54 | 0.39 | 4.32 | 0.88 | 4.20 | 24.28 | 121.63 |
| Monusac | Tnbcnuclei | 0.25 | 0.44 | 0.39 | 4.32 | 0.88 | 4.20 | 24.28 | 121.63 |
| Montgomery | NIH-test | 0.58 | 0.82 | 0.39 | 4.32 | 0.88 | 4.20 | 24.28 | 121.63 |

The normalized scores are defined as:

$$a_i = \text{clip}\left(\frac{A_i - Q_{10}^A}{Q_{90}^A - Q_{10}^A}, \, 0, \, 1\right), \quad p_i = \text{clip}\left(\frac{\log Q_{90}^P - \log P_i}{\log Q_{90}^P - \log Q_{10}^P}, \, 0, \, 1\right),$$
$$g_i = \text{clip}\left(\frac{\log Q_{90}^G - \log G_i}{\log Q_{90}^G - \log Q_{10}^G}, \, 0, \, 1\right), \quad s_i = \text{clip}\left(\frac{S_i - Q_{10}^S}{Q_{90}^S - Q_{10}^S}, \, 0, \, 1\right). \tag{1}$$

Then, we compute an efficiency subscore via the weighted harmonic mean of $p_i$, $g_i$, and $s_i$. Since we regard storage, cost, and speed as equally important, we set $w_P = w_G = w_S = \frac{1}{3}$, leading to:

$$\text{Eff}_i = \frac{3}{\frac{1}{p_i} + \frac{1}{g_i} + \frac{1}{s_i}}. \tag{2}$$

Finally, we combine accuracy and efficiency via a harmonic mean. To balance the two factors equally, we set $\alpha = 0.5$, yielding:

$$\text{U-Score}_i = \frac{2}{\frac{1}{a_i} + \frac{1}{\text{Eff}_i}}. \tag{3}$$

Table 10: Robustness of U-Score rankings over 105 configurations (Five choices of $\alpha$, Seven efficiency weightings $w$, Three quantile pairs). Baseline: $\alpha = 0.5$, $(w_P, w_G, w_S) = (1, 1, 1)$, $q$-pair $= (0.10, 0.90)$.

| Metric | Mean | Std | Min | 25% | Median | 75% |
|---|---|---|---|---|---|---|
| Kendall-$\tau$ vs. base U-Score ranking | 0.77 | 0.08 | 0.55 | 0.71 | 0.77 | 0.83 |
| Spearman-$\rho$ vs. base U-Score ranking | 0.90 | 0.06 | 0.72 | 0.86 | 0.91 | 0.95 |
| Kendall-$\tau$ vs. IoU | 0.19 | 0.14 | -0.07 | 0.06 | 0.19 | 0.30 |
| Top-10 keep | 0.86 | 0.12 | 0.60 | 0.80 | 0.90 | 0.90 |

### E.1 ABLATION U-SCORE SENSITIVITY OF WEIGHTS AND PERCENTILES

We supplement the system with a comprehensive sensitivity analysis, performing a grid search over the following configurations: the accuracy–efficiency trade-off $\alpha \in \{0.25, 0.33, 0.5, 0.67, 0.75\}$, efficiency weights $(w_P, w_G, w_S) \in \{(1,1,1), (2,1,1), (1,2,1), (1,1,2), (3,1,1), (1,3,1), (1,1,3)\}$, and quantile pairs $(10/90, 5/95, 20/80)$. We evaluate ranking stability using Kendall-$\tau$, Spearman-$\rho$, and Top-$k$ monotonicity. For each configuration, we compute: (I) Kendall-$\tau$ and Spearman-$\rho$ between the resulting ranking and the baseline *U-Score* ranking (our default setting: $\alpha = 0.5$, $(w_P, w_G, w_S) = (1, 1, 1)$, $q$-pair $= (0.10, 0.90)$); (II) Kendall-$\tau$ against the IoU-only ranking; and (III) Top-$k$ set stability (the overlap of the Top-10 models with the baseline Top-$k$ under a deterministic tie-breaking scheme).

The results are summarized in Table 10. We observe that the rankings are highly stable: Kendall-$\tau$ versus the baseline has a mean of 0.77, a median of 0.77, and remains $\geq 0.55$ even in the worst configuration; Spearman-$\rho$ versus the baseline has a mean of 0.90 and a median of 0.91. The Top-10 set also exhibits strong robustness, with a mean overlap of 0.86, a median of 0.90, and never dropping below 0.60. Importantly, U-Score is distinct from pure accuracy: Kendall-$\tau$ relative to the IoU-only ranking is much lower, with a mean of 0.19 (median 0.19), demonstrating that U-Score genuinely captures the accuracy–efficiency trade-off. These results collectively underscore the robustness and stability of the proposed U-Score.

### E.2 ABLATION U-SCORE SENSITIVITY OF ACCURACY-EFFICIENCY ($\alpha$)

We further investigate the impact of the accuracy–efficiency trade-off parameter $\alpha$ on U-Score rankings, isolating $\alpha$ while keeping $(w_P, w_G, w_S) = (1, 1, 1)$ and $q$-pair $= (0.10, 0.90)$. Table 11 shows that changing $\alpha$ produces stable rankings, with Kendall-$\tau$ values consistently above 0.8 across all settings, the Top-10 overlap remains between 0.80 and 1.00. These results demonstrate that U-Score provides a robust benchmark, where users can adjust $\alpha$ to reflect deployment priorities without introducing significant ranking shifts.

In Table 12 and Table 13, we isolate the effect of the quantile pair $(q_{\text{pair}})$ and efficiency weights $(w_P, w_G, w_S)$. The results indicate that moderate changes in quantile ranges (e.g., $(0.05, 0.95)$ to $(0.20, 0.80)$) have minimal impact on U-Score rankings, with Kendall-$\tau$ values ranging from 0.76 to 1.00. The Top-10 overlap stays high, with most configurations showing stable rankings. This shows that U-Score remains reliable even under adjustments to the quantile clipping range. Similarly, varying the efficiency weights $(w_P, w_G, w_S)$ does not drastically alter the leaderboard. Kendall-$\tau$ values between 0.86 and 0.94 and Top-10 overlap values above 0.9. This flexibility allows users to bias U-Score towards their desired trade-offs (e.g., memory, computation, or latency).

Finally, we add a comprehensive Leave-One-Modality-Out (LOMO) evaluation using the suggested 'closest-dataset IoU heuristic' as a strong, label-free baseline. On IoU-only ranking (Table 14, left), this heuristic is extremely competitive because IoU is highly modality-specific, and transferring the ranking from the closest source dataset provides a near-oracle estimate. The advisor therefore does not surpass the heuristic in accuracy-only setting. However, when switching to the multi-objective U-Score (accuracy–efficiency trade-offs), the picture reverses: the advisor outperforms the heuristic in 7 out of 10 modalities, including large gains on CT (+0.056 NDCG@10) and Pathology (+0.027), demonstrating that handcrafted similarity based solely on IoU does not generalize to multi-objective optimization, while the advisor captures cross-modal architectural patterns that transfer better. Finally, our deployment-time evaluation reveals complementary strengths. The heuristic

Table 11: Sensitivity of U-Score to the accuracy–efficiency weight $\alpha$. We report Kendall-$\tau$ vs. the baseline U-Score ranking ($\alpha = 0.5$, $w_P, w_G, w_S = (1, 1, 1)$, $q$-pair $= (0.10, 0.90)$), Kendall-$\tau$ vs. IoU-only, and the median Top-10 keep.

| $\alpha$ | $\tau_{\text{base}}$ | $\tau_{\text{IoU}}$ | Top-10 keep |
|---|---|---|---|
| 0.25 | 0.82 | -0.01 | 0.90 |
| 0.33 | 0.89 | 0.06 | 1.00 |
| 0.50 | 1.00 | 0.17 | 1.00 |
| 0.67 | 0.92 | 0.25 | 0.90 |
| 0.75 | 0.87 | 0.31 | 0.80 |

Table 12: Sensitivity of U-Score to the quantile pair ($q_{q\text{-pair}}$). We report Kendall-$\tau$ vs. the baseline U-Score ranking ($\alpha = 0.5$, $w_P, w_G, w_S = (1, 1, 1)$, $q$-pair $= (0.10, 0.90)$), Kendall-$\tau$ vs. IoU-only, and the median Top-10 keep.

| $q_{\text{pair}}$ | $\tau_{\text{base}}$ | $\tau_{\text{IoU}}$ | Top-10 keep |
|---|---|---|---|
| (0.05, 0.95) | 0.90 | 0.14 | 1.00 |
| (0.10, 0.90) | 1.00 | 0.17 | 1.00 |
| (0.20, 0.80) | 0.76 | 0.31 | 0.80 |

Table 13: Sensitivity of U-Score to the efficiency weight $(w_P, w_G, w_S)$. We report Kendall-$\tau$ vs. the baseline U-Score ranking ($\alpha = 0.5$, $w_P, w_G, w_S = (1, 1, 1)$, $q$-pair $= (0.10, 0.90)$), Kendall-$\tau$ vs. IoU-only, and the median Top-10 keep.

| $(w_P, w_G, w_S)$ | $\tau_{\text{base}}$ | $\tau_{\text{IoU}}$ | Top-10 keep |
|---|---|---|---|
| (1, 1, 1) | 1.00 | 0.17 | 1.00 |
| (1, 1, 2) | 0.92 | 0.19 | 1.00 |
| (1, 1, 3) | 0.87 | 0.21 | 1.00 |
| (1, 2, 1) | 0.90 | 0.15 | 1.00 |
| (1, 3, 1) | 0.86 | 0.14 | 1.00 |
| (2, 1, 1) | 0.94 | 0.17 | 0.90 |
| (3, 1, 1) | 0.90 | 0.17 | 0.90 |

Table 14: Model Advisor vs. closest-dataset heuristic under Leave-One-Modality-Out (LOMO). A-NDCG and B-NDCG denote Advisor and Baseline NDCG@10, respectively. A-Trial and B-Trial report the number of trials needed to find a top-10% model. The heuristic is extremely strong for IoU-only ranking due to modality-specific performance, while the advisor generalizes better for the multi-objective U-Score and yields substantial trial reductions when no close source modality exists.

| Modality | IoU LOMO | | | | U-Score LOMO | | | |
|---|---|---|---|---|---|---|---|---|
| | A-NDCG | B-NDCG | A-Trial | B-Trial | A-NDCG | B-NDCG | A-Trial | B-Trial |
| Ultrasound | 0.676 | 0.946 | 3.67 | 1.00 | 0.845 | 0.841 | 1.00 | 2.00 |
| Dermoscopy | 0.671 | 0.917 | 4.00 | 1.00 | 0.831 | 0.813 | 10.0 | 2.00 |
| Endoscopy | 0.616 | 0.861 | 3.00 | 2.50 | 0.837 | 0.855 | 4.00 | 2.50 |
| Fundus | 0.783 | 0.949 | 3.00 | 1.00 | 0.834 | 0.875 | 7.00 | 1.50 |
| Histopathology | 0.728 | 0.905 | 4.67 | 1.00 | 0.871 | 0.844 | 5.67 | 1.33 |
| Nuclear | 0.759 | 0.924 | 4.00 | 1.00 | 0.864 | 0.853 | 7.00 | 3.00 |
| X-ray | 0.729 | 0.954 | 3.67 | 1.33 | 0.859 | 0.870 | 5.00 | 1.00 |
| MRI | 0.682 | 0.855 | 3.00 | 1.00 | 0.851 | 0.838 | 4.00 | 2.00 |
| CT | 0.715 | 0.757 | 2.00 | 1.00 | 0.858 | 0.802 | 1.00 | 5.00 |
| OCT | 0.752 | 0.946 | 5.00 | 1.00 | 0.857 | 0.839 | 5.00 | 2.00 |

excels when a close source modality exists, but collapses in hard cases (e.g., CT: 5 trials), whereas the advisor reliably identifies a top-10% model in just 1 trial, highlighting its substantial practical benefit in real model-selection scenarios.

# F IMPLEMENTATION AND EVALUATION DETAILS

## F.1 FOREGROUND CHARACTERIZATION METRICS

We employ three metrics to characterize dataset-level foreground properties: scale, boundary sharpness, and shape regularity.

**Foreground scale.** Foreground scale is measured as the ratio between the foreground area $A_f$ and the total image area $A_t$.

$$A = \frac{A_f}{A_t}.$$  (4)

We categorize samples as *small-scale* if $A < 0.05$ and *large-scale* otherwise.

**Shape complexity.** We quantify the sharpness of the segmented foreground boundaries using a composite score $S$ derived from two standard geometric descriptors: *circularity* and *solidity*. We categorize samples with $S < 0.5$ as *irregular*, and those with $S \geq 0.5$ as *regular*. The boundary sharpness score $S$ is defined as:

$$S = 0.5 \times \text{Circularity} + 0.5 \times \text{Solidity}.$$  (5)

Circularity measures how close the shape is to a perfect circle. It is defined as: Circularity $= \frac{4\pi A_f}{P^2}$, where $A_f$ is the foreground area and $P$ is the perimeter of the contour. Solidity evaluates the extent to which a shape fills its convex hull. It is given by: Solidity $= \frac{A_f}{A_c}$, where $A_f$ is the foreground area and $A_c$ is the area of its convex hull.

**Boundary Sharpness.** We assess boundary sharpness using two complementary measures: *boundary width* and *boundary contrast*. Given a binary mask $m$, we first construct a narrow boundary ring by applying morphological dilation and erosion. The boundary width is then computed as the ratio between the area of this ring and the contour perimeter: $w = \frac{\text{Area(Ring)}}{P+\epsilon}$, where $P$ denotes the sum of contour perimeters. A larger $w$ indicates blurrier boundaries, while a smaller $w$ corresponds to sharper edges. To evaluate intensity separation across the boundary, we form two narrow bands: one inside the mask and one outside, each of width $t$ pixels. Let $(\mu_{in}, \sigma_{in})$ and $(\mu_{out}, \sigma_{out})$ denote the mean and standard deviation of pixel intensities inside and outside the boundary band. The boundary contrast is defined as CNR $= \frac{|\mu_{in}-\mu_{out}|}{\sigma_{in}+\sigma_{out}+\epsilon}$. To obtain a unified measure of boundary clarity, we normalize both $w$ and CNR to $[0, 1]$ across the dataset. A composite blur score is then computed as:

$$B = \frac{w_{norm}}{w_{norm} + c_{norm} + \epsilon},$$  (6)

where $w_{norm}$ and $c_{norm}$ are the normalized boundary width and contrast, respectively. We categorize samples with $b < 0.6$ as *clear*, and those with $b \geq 0.6$ as *blur*.

## F.2 MODEL ADVISOR AGENT SETTINGS

We construct a comprehensive feature space that integrates both continuous and discretized descriptors from models and datasets. For model-level attributes, we discretize storage (parameter) into four scales (Tiny: 0-10M, Small: 10-50M, Medium: 50-200M, Large: >200M), computation cost (FLOPs) into three levels (Low: 0-10 GFLOPs, Medium: 10-100 GFLOPs, High: >100 GFLOPs), and inference speed (FPS) into three categories (Slow: <15 FPS, Medium: 15-60 FPS, Fast: >60 FPS). On the data characteristics side, we discretize foreground-related properties:foreground scale ($< 0.05$ vs. $\geq 0.05$, denoting small vs. large targets), shape complexity ($< 0.5$ vs. $\geq 0.5$, irregular vs. regular), and boundary sharpness ($< 0.6$ vs. $\geq 0.6$, clear vs. blurry). We train an XGBRanker with the rank:pairwise objective on 18 in-domain datasets, reserving 2 datasets (BUSI and Skin-Cancer) for testing. Scores are normalized into relevance values within each dataset, with higher relevance indicating better relative performance. Dataset-level grouping is used to enforce within-dataset ranking consistency during training. At evaluation, the ranker outputs predicted scores, which are converted into ranked lists for each dataset. Performance is assessed using NDCG@50/20 for ranking quality, MAP for precision under binary relevance, and correlation metrics (Spearman) to quantify alignment between predicted and ground-truth orderings. Finally, the agent exports the recommended models per test dataset, providing a practical reference list for downstream selection.

### F.2.1 DATA PREPROCESSING

**Experimental Data Split.** For all experiments on the unified dataset, we used the same train-test split. For data without a clear train-test split in the dataset source, we adopted a random split; for data with a known train-test split in the dataset source, we followed the original split. The division method of each dataset is shown in Tab. 4, where 'O' denotes our self-defined division and 'S' denotes the division consistent with the referenced data source.

### F.2.2 RETRAINING AND INFERENCE DETAILS

**Training and Inference Protocol.** The experiments are utilizing eight NVIDIA H20 GPUs. A total of 100 models are trained with 6000 hours. We provide a fully reproducible description in the Table 16. All FPS values in U-Score are measured as pure forward-pass latency (no dataloader, I/O, or post-processing), using batch size 1, on our compute cluster equipped with $8\times$ NVIDIA H20 GPUs and Intel Xeon Platinum 8558 CPUs ($2\times48$ cores, OMP/MKL threads fixed to 1). The implementation is based on Python 3.9 and PyTorch 2.4.0 (CUDA 12.4, cuDNN 9.1); no torch.compile, TensorRT, graph mode, or mixed-precision inference is used (FP32 only). Before timing, each model is warmed up for 10 iterations, followed by 30 timed iterations, using torch.cuda.Event with GPU synchronization. The model structure files are primarily obtained from the open-source code of the original models, with only minor modifications (e.g. input and output channels) to some input parameters to adapt to our framework.

Table 15: Hyperparameters in U-Bench

| Optimizer | Learning Rate | Epochs | Random Seed | Batch Size |
|---|---|---|---|---|
| SGD (Momentum=0.9, Weight Decay=0.0001) | 0.01 | 300 | 41 | 8 |

**Data augmentation.** Following prior works (Valanarasu & Patel, 2022; Tang et al., 2023; 2024; Chen et al., 2024a; Tang et al., 2025b; Ye et al., 2025; Jiang et al., 2025; Dong et al., 2025), we rescale all images to a resolution of $256 \times 256$ by default and apply standard data augmentation. For models that require a fixed input size (e.g., Swin Transformer variants designed for $224 \times 224$ inputs), we preserve their original settings without rescaling. For 3D datasets (e.g. Synapse and ACDC), we follow the same approach as previous methods such as TransUNet Chen et al. (2021), CASCADE Rahman & Marculescu (2023a), SwinUnet Cao et al. (2022), and Mobile U-ViT Tang et al. (2025b), where we process 3D data by slicing along the axial (transverse) plane (Z-axis). Augmentations include random $90°$ rotations, random horizontal and vertical flips, and normalization. To ensure fair comparisons, the same preprocessing pipeline is applied consistently across all experiments. Notably, for models that adopt deep supervision, we retain their original training strategy to enable accurate performance evaluation.

**Training Settings.** Following previous work (Dong et al., 2025; Tang et al., 2024; 2025b; Chen et al., 2021; Wang et al., 2024; Ye et al., 2025), we unify training settings across all models to ensure fair comparisons, as summarized in Tab. 15. Moreover, we use commonly adopted loss configurations (Dong et al., 2025; Tang et al., 2024; 2025b; Ye et al., 2025) to promote generalizable results and enable more equitable performance evaluation.

**Loss Function.** Specifically, for the ground truth $y$ and the predicted output $\hat{y}$, the loss function is defined as:

$$\mathcal{L} = 0.5 \times BCE(\hat{y}, y) + Dice(\hat{y}, y), \tag{7}$$

where BCE denotes the binary cross-entropy loss and Dice denotes the Dice loss. Note that for 3D data ACDC and Synapse, we follow CASCADE (Rahman & Marculescu, 2023a) weights $0.5 \times BCE(\hat{y}, y), 0.7 \times Dice(\hat{y}, y)$

### F.3 METRICS

**Intersection over Union (IoU)** IoU quantifies the overlap between two regions (predicted $A$ and ground-truth $B$) as:

$$\text{IoU}(\hat{Y}, Y) = \frac{|\hat{Y} \cap Y|}{|\hat{Y} \cup Y|}, \tag{8}$$

Table 16: Measurement Protocol.

| *Inference Hardware* | |
|---|---|
| GPU Model | 8 × NVIDIA H20 (FPS measured on a single GPU) |
| GPU Memory | 96 GB per GPU |
| CPU Model | Intel Xeon Platinum 8558 |
| CPU Cores / Threads | 48 cores × 2 sockets, 2 threads/core (192 threads total) |
| OMP Threads | `OMP_NUM_THREADS=1` |
| MKL Threads | `MKL_NUM_THREADS=1` |
| *Torch & Cuda Version* | |
| Python Version | Python 3.9.0 |
| PyTorch Version | PyTorch 2.4.0 |
| CUDA Version | CUDA 12.4 |
| cuDNN Version | cuDNN 9.1.0 |
| *Precision & Compute Settings* | |
| Precision Mode | FP32 only |
| Mixed Precision | AMP / FP16 / BF16 disabled |
| TF32 | Disabled (`allow_tf32=False`) |
| Acceleration Libraries | No TensorRT / no `torch.compile` / no CUDA Graphs |
| *Resolution Policy* | |
| CNN, ViT, RWKV, Hybrid Models | 256 × 256 |
| Swin-based variants | 224 × 224 |
| FLOPs Alignment | Same resolution as FPS measurement |
| *Timing Protocol* | |
| Batch Size | 1 |
| Warm-up Iterations | 10 |
| Timed Iterations | 30 |
| Timing Method | `torch.cuda.Event(enable_timing=True)` |
| Synchronization | Explicit `torch.cuda.synchronize()` per iteration |
| *Isolation & Clean Measurement* | |
| Dataloader | Disabled (no workers) |
| I/O Operations | Disabled |
| Post-Processing | Disabled |
| Model Mode | `model.eval()`, dropout disabled |
| GPU Usage Policy | Single-GPU measurement, no concurrent jobs |

where $|\hat{Y} \cap Y|$ is the area of intersection, and $|\hat{Y} \cup Y|$ is the area of union.

**U-Score.** To address the limitation that existing evaluations primarily focus on segmentation performance while failing to balance key computational factors such as model size, inference cost, and speed-making it difficult to assess practical deployment capabilities-we construct the U-Score based on quantile statistics. A detailed description is provided in Appendix E.

**Normalized Discounted Cumulative Gain (NDCG)**

NDCG evaluates the "usefulness" of a ranked list by accounting for two key factors: (1) the *relevance* of each item, and (2) the *position* of relevant items (penalizing lower-ranked relevant items via discounting). It is normalized to a range of $[0, 1]$ to enable cross-task comparisons.

First, the *Discounted Cumulative Gain (DCG)* is defined to measure the cumulative relevance of a ranked list up to position $k$ (denoted as DCG@$k$):

$$\text{DCG@}k = \sum_{i=1}^{k} \frac{\text{rel}_i}{\log_2(i+1)} \tag{9}$$

where:

- $k$: The cutoff position (e.g., $k = 10$ for NDCG@10, focusing on top-10 results).

- $\text{rel}_i$: The *relevance score* of the $i$-th item in the ranked list. For binary relevance (relevant/irrelevant).

- $\log_2(i+1)$: The discount factor, which reduces the contribution of items ranked later (since users are less likely to inspect lower positions).

To normalize DCG across different queries/tasks (where the maximum possible relevance varies), the *Ideal DCG (IDCG)*-the maximum possible DCG@k for a given set of items-is computed by

ranking all relevant items in descending order of $\text{rel}_i$:

$$\text{IDCG@}k = \sum_{i=1}^{\min(k,|R|)} \frac{\text{rel}'_i}{\log_2(i+1)} \tag{10}$$

where:

- $R$: The set of all relevant items for the query/task.
- $|R|$: The total number of relevant items.
- $\text{rel}'_i$: The $i$-th highest relevance score among all items in $R$ (i.e., the ideal ranking).

NDCG@k is defined as the ratio of DCG@k to IDCG@k. To avoid division by zero (when no relevant items exist, IDCG@$k = 0$), NDCG@k is set to 0 in this edge case:

$$\text{NDCG@}k = \begin{cases} 0 & \text{if IDCG@}k = 0, \\ \frac{\text{DCG@}k}{\text{IDCG@}k} & \text{otherwise.} \end{cases} \tag{11}$$

For experiments with multiple queries/tasks (e.g., a retrieval dataset with 1k queries), the *mean NDCG@k*-the average of NDCG@k across all queries-is reported. In Table 3, NDCG@k values for $k = 5$ and $k = 20$ are provided.

**Mean Average Precision (MAP)**

MAP quantifies the average precision of relevant items in a ranked list, aggregated across all queries/tasks. It is particularly useful for scenarios where "early relevant items" (high precision at top positions) are critical (e.g., information retrieval, recommendation systems).

First, *Average Precision (AP)* for a single query $q$ is defined as the average of the precision of the ranked list at the position of each relevant item:

$$\text{AP}(q) = \frac{1}{|R_q|} \sum_{r \in R_q} \text{Prec}(k_r) \tag{12}$$

where:

- $q$: A single query (or task instance) from the query set $Q$;
- $R_q$: The set of all relevant items for query $q$ (if $|R_q| = 0$, $\text{AP}(q) = 0$ by convention);
- $k_r$: The position of relevant item $r$ in the ranked list for $q$;
- $\text{Prec}(k_r)$: The precision at position $k_r$, defined as $\text{Prec}(k_r) = \frac{\text{numRel}(k_r)}{k_r}$, where $\text{numRel}(k_r)$ is the number of relevant items in the top-$k_r$ positions.

For a set of $|Q|$ queries, MAP is the average of AP scores across all queries:

$$\text{MAP} = \frac{1}{|Q|} \sum_{q \in Q} \text{AP}(q) \tag{13}$$

Similar to NDCG, MAP ranges from $[0, 1]$: a value of 1 indicates all relevant items are ranked first (perfect precision at every relevant position), while 0 indicates no relevant items are retrieved.

**Spearman's Rank Correlation Coefficient**

Spearman's rank correlation coefficient quantifies the *monotonic relationship* between two ranked variables. It is particularly useful for evaluating how well the order of items (e.g., predicted rankings by a model and ground-truth rankings) aligns, making it relevant for tasks where the consistency of relative ordering matters (e.g., comparing ranked recommendations or human judgments).

Formally, Spearman's rank correlation coefficient $\rho$ between two variables $X$ (e.g., model-generated ranks) and $Y$ (e.g., ground-truth ranks) (each with $n$ paired observations) is defined as:

$$\rho = 1 - \frac{6 \sum_{i=1}^{n} d_i^2}{n(n^2 - 1)} \tag{14}$$

where:

Table 17: Top-10 performing variants across each dataset on source domains. Baseline U-Net is highlighted (gray background), and statistical significance of p-value is highlighted: p<0.0001 , p<0.001 , p<0.05 , p≤0.05 , and P > 0.05 (Not significant) .

| Rank | Ultrasound BUSI | | BUSBRA | | TNSCUI | | Endoscopy Kvasir | | Robotool | | Dermoscopy ISIC2018 | | SkinCancer | |
|---|---|---|---|---|---|---|---|---|---|---|---|---|---|---|
| 1 | RWKV-UNet | 72.32 | RWKV-UNet | 84.76 | RWKV-UNet | 80.06 | Swin-umamba | 85.56 | MEGANet | 86.25 | RWKV-UNet | 84.97 | RWKV-UNet | 87.48 |
| 2 | PraNet | 71.63 | EViT-UNet | 84.36 | MEGANet | 79.01 | VMUnet | 84.90 | RWKV-UNet | 85.95 | CFFormer | 84.89 | DA-TransUNet | 87.22 |
| 3 | Mobile U-ViT | 71.59 | CaraNet | 84.31 | MADGNet | 78.91 | UACANet | 84.81 | AURA-Net | 85.78 | MSLAU-Net | 84.50 | MSLAU-Net | 86.80 |
| #4 | DA-TransUNet | 71.47 | MADGNet | 84.27 | TA-Net | 78.85 | CFFormer | 84.57 | TA-Net | 85.62 | Swin-umamba | 84.49 | PraNet | 86.38 |
| #5 | MEGANet | 71.47 | TA-Net | 84.17 | UACANet | 78.83 | EViT-UNet | 84.53 | EViT-UNet | 85.39 | PraNet | 84.42 | FCBFormer | 86.33 |
| #6 | TransResUNet | 71.27 | FAT-Net | 84.15 | EViT-UNet | 78.75 | FCBFormer | 84.50 | TransResUNet | 85.30 | TransResUNet | 84.38 | EMCAD | 86.20 |
| #7 | MADGNet | 71.24 | UACANet | 84.07 | CaraNet | 78.69 | PraNet | 84.39 | MADGNet | 85.21 | TA-Net | 84.34 | MCA-UNet | 86.17 |
| #8 | CFFormer | 70.91 | FCBFormer | 84.00 | Swin-umamba | 78.64 | CASCADE | 84.34 | CE-Net | 85.11 | CE-Net | 84.32 | CaraNet | 86.01 |
| #9 | ESKNet | 70.88 | MEGANet | 83.99 | FAT-Net | 78.57 | CE-Net | 84.32 | PraNet | 85.10 | CaraNet | 84.26 | TransNorm | 85.86 |
| #10 | CASCADE | 70.81 | AURA-Net | 83.95 | UTANet | 78.51 | MADGNet | 84.32 | UACANet | 84.65 | AURA-Net | 84.25 | AURA-Net | 85.56 |
| | U-Net (#68) | 65.58 | U-Net (#41) | 82.91 | U-Net (#58) | 75.99 | U-Net (#70) | 80.11 | U-Net (#23) | 81.24 | U-Net (#61) | 82.78 | U-Net (#77) | 80.94 |

| Rank | X-Ray Covidquex | | Montgomery | | DCA | | MRI Promise | | ACDC | | Fundus CHASE | | DRIVE | |
|---|---|---|---|---|---|---|---|---|---|---|---|---|---|---|
| 1 | AURA-Net | 70.85 | RWKV-UNet | 96.21 | DA-TransUNet | 64.90 | RWKV-UNet | 87.56 | CENet | 85.54 | CMU-Net | 84.33 | FCBFormer | 64.25 |
| 2 | RWKV-UNet | 70.75 | DA-TransUNet | 96.17 | UTANet | 64.23 | FCBFormer | 87.29 | Swin-umambaD | 85.45 | AttU-Net | 84.20 | MT-UNet | 63.21 |
| 3 | CaraNet | 70.61 | MEGANet | 96.12 | EViT-UNet | 63.81 | MADGNet | 87.26 | DoubleUNet | 85.33 | U-Net | 84.07 | ColonSegNet | 63.19 |
| #4 | EViT-UNet | 70.33 | TransAttUNet | 96.03 | MADGNet | 63.81 | EViT-UNet | 87.05 | RWKV-UNet | 85.20 | UNet3plus | 83.69 | UTNet | 63.17 |
| #5 | TA-Net | 70.20 | RollingUnet | 96.01 | MEGANet | 63.77 | Perspective-Unet | 87.03 | DDANet | 85.11 | Perspective-Unet | 82.86 | ESKNet | 63.15 |
| #6 | MEGANet | 70.19 | DDANet | 95.97 | ESKNet | 63.69 | MEGANet | 87.00 | AttU-Net | 85.01 | UCTransNet | 82.82 | CMU-Net | 62.85 |
| #7 | PraNet | 70.09 | MT-UNet | 95.90 | DDANet | 63.65 | TransResUNet | 86.95 | EViT-UNet | 84.91 | ESKNet | 82.69 | Swin-umamba | 62.75 |
| #8 | CE-Net | 70.04 | TransResUNet | 95.89 | RWKV-UNet | 63.61 | U-KAN | 86.89 | FCBFormer | 84.90 | ColonSegNet | 82.20 | UNet3plus | 62.54 |
| #9 | TransResUNet | 69.81 | Mobile U-ViT | 95.89 | UTNet | 63.54 | PraNet | 86.88 | G-CASCADE | 84.89 | MT-UNet | 82.00 | RollingUnet | 62.49 |
| #10 | MADGNet | 69.77 | UNet3plus | 95.88 | U-Net | 63.30 | CMU-Net | 86.87 | MSRFNet | 84.78 | Swin-umamba | 81.65 | D-TrAttUNet | 62.48 |
| | U-Net (#31) | 68.52 | U-Net (#11) | 95.87 | | | U-Net (#29) | 86.30 | U-Net (#23) | 84.32 | | | U-Net (#14) | 61.81 |

| Rank | Histopathology Glas | | Monusac | | Nuclear DSB2018 | | CellNuclear | | CT Synapse | | OCT Cystoidfluid | |
|---|---|---|---|---|---|---|---|---|---|---|---|---|
| 1 | EMCAD | 85.85 | MT-UNet | 69.27 | MT-UNet | 88.74 | MT-UNet | 84.93 | CENet | 74.70 | UNet3plus | 85.76 |
| 2 | RWKV-UNet | 85.75 | RWKV-UNet | 68.96 | DoubleUNet | 88.61 | TransAttUnet | 84.88 | Perspective-Unet | 73.69 | Swin-umamba | 85.06 |
| 3 | CASCADE | 85.17 | UTANet | 68.39 | TransAttUnet | 88.49 | AURA-Net | 84.87 | G-CASCADE | 73.54 | UTANet | 84.89 |
| #4 | MSLAU-Net | 84.38 | CA-Net | 68.39 | DCSAU-Net | 88.44 | CA-Net | 84.85 | CASCADE | 73.30 | MMUNet | 84.21 |
| #5 | UTANet | 84.22 | DDANet | 68.38 | UTNet | 88.39 | UTANet | 84.77 | AURA-Net | 73.25 | H2Former | 83.99 |
| #6 | DDANet | 83.78 | TransAttUnet | 68.25 | D-TrAttUnet | 88.27 | ColonSegNet | 84.70 | MEGANet | 73.18 | Perspective-Unet | 83.90 |
| #7 | MERIT | 83.77 | UTNet | 67.76 | ESKNet | 88.23 | DA-TransUNet | 84.63 | DS-TransUNet | 72.74 | FCBFormer | 83.84 |
| #8 | MBSNet | 83.57 | EViT-UNet | 67.21 | AURA-Net | 88.23 | RollingUnet | 84.62 | DoubleUNet | 72.63 | D-TrAttUNet | 83.74 |
| #9 | CENet | 83.47 | D-TrAttUNet | 66.96 | LGMSNet | 88.16 | RWKV-UNet | 84.56 | MSLAU-Net | 72.60 | MedFormer | 83.58 |
| #10 | U-Net | 83.30 | AttU-Net | 66.96 | DDANet | 88.16 | FCBFormer | 84.54 | RWKV-UNet | 72.56 | EViT-UNet | 83.54 |
| | | | U-Net (#21) | 66.44 | U-Net (#16) | 88.05 | U-Net (#17) | 84.33 | U-Net (#52) | 67.90 | U-Net (#23) | 82.39 |

Table 18: Top-10 performing variants across each dataset on target domains. Baseline U-Net is highlighted (gray background), and statistical significance of p-value is highlighted: p<0.0001 , p<0.001 , p<0.05 , p≤0.05 , and P > 0.05 (Not significant) .

| Rank | Ultrasound (Source → Target) BUSI → BUS | | BUSBRA → BUS | | TNSCUI → TUCC | | Rank | Endoscopy (Source → Target) Kvasir → CVC300 | | Kvasir → CVC-ClinicDB | |
|---|---|---|---|---|---|---|---|---|---|---|---|
| 1 | Swin-umamba | 82.91 | MEGANet | 86.62 | MSLAU-Net | 66.15 | 1 | PraNet | 83.31 | PraNet | 77.39 |
| 2 | EMCAD | 82.83 | DoubleUNet | 86.51 | MERIT | 66.00 | 2 | RWKV-UNet | 82.14 | DS-TransUNet | 77.38 |
| 3 | CENet | 82.70 | CENet | 86.29 | EViT-UNet | 65.83 | 3 | UACANet | 81.72 | CASCADE | 77.19 |
| #4 | G-CASCADE | 82.61 | FCBFormer | 86.10 | Polyp-PVT | 65.23 | #4 | MERIT | 81.39 | Swin-umambaD | 76.83 |
| #5 | DA-TransUNet | 82.32 | CASCADE | 85.96 | LGMSNet | 65.16 | #5 | MADGNet | 81.24 | EMCAD | 76.56 |
| #6 | PraNet | 82.13 | Polyp-PVT | 85.65 | G-CASCADE | 65.10 | #6 | TA-Net | 81.24 | TransResUNet | 76.42 |
| #7 | CASCADE | 82.11 | TransResUNet | 85.62 | CaraNet | 64.61 | #7 | EViT-UNet | 81.11 | MADGNet | 76.33 |
| #8 | TransNorm | 82.11 | ResNet34UnetPlus | 85.46 | H2Former | 64.48 | #8 | UTANet | 80.78 | CFFormer | 76.18 |
| #9 | MCA-UNet | 81.88 | MCA-UNet | 85.28 | MEGANet | 64.37 | #9 | DS-TransUNet | 80.58 | DoubleUNet | 75.72 |
| #10 | CaraNet | 81.32 | G-CASCADE | 85.27 | Swin-umamba | 64.00 | #10 | CASCADE | 80.57 | MEGANet | 75.64 |
| | U-Net (#63) | 72.44 | U-Net (# 79) | 81.37 | U-Net (# 65) | 60.50 | | U-Net (#75) | 70.33 | U-Net (#47) | 69.87 |

| Rank | Dermoscopy ISIC2018 → PH2 | | Rank | Fundus CHASE → DRIVE | | Rank | X-Ray Montgomery → NIH-test | | Rank | Histopathology Monusac → Tnbcnuclei | |
|---|---|---|---|---|---|---|---|---|---|---|---|
| 1 | MSLAU-Net | 86.52 | 1 | MSRFNet | 55.60 | 1 | MEGANet | 88.19 | 1 | TA-Net | 50.74 |
| 2 | RWKV-UNet | 86.00 | 2 | DS-TransUNet | 55.52 | 2 | TransResUNet | 87.69 | 2 | CENet | 48.03 |
| 3 | G-CASCADE | 85.96 | 3 | MBSNet | 54.66 | 3 | CaraNet | 86.22 | 3 | ResNet34UnetPlus | 46.44 |
| #4 | MERIT | 85.94 | #4 | RWKV-UNet | 54.27 | #4 | DA-TransUNet | 85.87 | #4 | EMCAD | 46.41 |
| #5 | MMUNet | 85.66 | #5 | CENet | 53.78 | #5 | PraNet | 84.58 | #5 | G-CASCADE | 46.18 |
| #6 | H2Former | 85.39 | #6 | CSCAUNet | 52.80 | #6 | MADGNet | 83.67 | #6 | UNetV2 | 45.32 |
| #7 | CMUNeXt | 85.32 | #7 | MCA-UNet | 52.69 | #7 | Swin-umambaD | 83.13 | #7 | CSWin-UNet | 45.18 |
| #8 | UACANet | 85.12 | #8 | EViT-UNet | 52.49 | #8 | TransUnet | 83.03 | #8 | DAEFormer | 45.16 |
| #9 | MADGNet | 85.08 | #9 | Tinyunet | 52.26 | #9 | TransNorm | 82.90 | #9 | DA-TransUNet | 44.53 |
| #10 | MCA-UNet | 85.08 | #10 | TransResUNet | 52.00 | #10 | RWKV-UNet | 82.41 | #10 | MedVKAN | 44.42 |
| | U-Net (#47) | 84.00 | | U-Net (#57) | 39.64 | | U-Net (#51) | 71.33 | | U-Net (#91) | 26.05 |

- $d_i$: The difference between the rank of $X_i$ and the rank of $Y_i$ (i.e., $d_i = \text{rank}(X_i) - \text{rank}(Y_i)$).

- $n$: The total number of paired observations.

# G  ADDITIONAL RESULTS

### G.1 PER-DATASET TOP-10 AND U-NET COMPARISON

We report the top-10 performing methods across each dataset, evaluated on both source and target domains. As shown in Tab. 17 and Tab. 18. For reference, the position of the vanilla U-Net is highlighted with a gray background, and we also compute the statistical significance of each variant relative to U-Net.

**Top10 performance on source domain.** On widely studied datasets and modalities-such as ultrasound, polyp segmentation, ISIC2018 (Dermoscopy), Synapse (CT), Drive (Fundus), ACDC (MRI), and Covidquex (X-ray)-most top-10 variants achieve significant improvements over U-Net. This trend is consistent with the increasing popularity of these datasets and 'novelty desion' for long-range dependency modeling, such as incorporating Transformers, Mamba, RWKV, and hybrid designs. In contrast, on other datasets and modalities the improvements remain marginal. For example, in Montgomery (X-ray lung segmentation), DCA, Chase (Fundus), nuclear segmentation, and Histopathology, the relative gains over U-Net are not significant. This suggests that progress in these modalities has been limited, because they rely on stable local patterns rather than long-range context. These observations highlight an important direction for future research: designing models that are modality-aware, particularly tailored for domains dominated by local and repetitive structures.

**Top10 performance on target domain.** On the target-domain datasets, nearly all top-10 methods achieve substantial improvements, highlighting the superior generalization ability of recent variants. These gains are primarily driven by two factors: the adoption of long-range dependency modeling and the increased model complexity. Together, these characteristics enhance the representational capacity and adaptability of the variants, which aligns with the prevailing trend toward more novel and increasingly complex model architectures.

In addition, we provide the visualization results of the top 5 models and U-net of the dataset for visualization analysis. The results are shown in Fig. 15 and 16.

### G.2 GPU-HOURS OF VARIANTS TO ACHIEVE OPTIMAL PERFORMANCE

We calculate the GPU hours required to achieve optimal performance for each variants and architecture family. The results are shown in Table 19 and Table 26. The Transformer family has the highest median GPU hours (1.22) and a large IQR (1.78), indicating high computational demands and variability. The Mamba family also requires significant resources (median = 1.19), but with more consistent performance (IQR = 0.47). The RWKV family is the most efficient, with the lowest median (0.63) and IQR (0.30), reflecting lightweight models with minimal variation. The CNN and Hybrid families show moderate GPU hours (0.92 and 0.90) and similar IQRs (0.67), offering a balance between performance and efficiency.

Table 19: GPU Hours for Achieving Optimal Performance by Model Family

| Architecture | Median | IQR |
|---|---|---|
| CNN | 0.92 | 0.67 |
| Hybrid | 0.90 | 0.67 |
| Transformer | 1.22 | 1.78 |
| RWKV | 0.63 | 0.30 |
| Mamba | 1.19 | 0.47 |

## H RELIABLE CROSS-ARCHITECTURE EVALUATION

We add a $\Delta$IoU column in Table 20 comparing our reproduced in-domain results with the original papers' reported metrics. The results show that our unified recipe reproduces prior performance faithfully: most $\Delta$IoU values lie within a small and symmetric range around zero (typically ±3–5%), and the few larger gaps correspond to architectures originally tailored to narrow modalities rather than evidence of training failure. Notably, several models even achieve higher IoU under our unified pipeline (e.g., RWKV-UNet, DA-TransUNet, U-KAN and DDANet), demonstrating that our implementation does not systematically disadvantage any architecture. To further validate the unified recipe, we conduct a controlled reproduction study on BUSI across 100 variants, using both (i) each

paper's official hyperparameters and (ii) our unified U-Bench recipe. The results are shown Fig. 10 The unified recipe yields higher median IoU and fewer collapsed low-tail cases, confirming that the U-Bench implementation is reliable as the official pipelines.

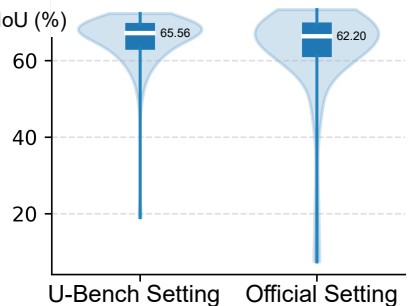

Figure 10: Comparison between official and U-Bench settings, showing higher median IoU and fewer low-tail failures with the unified recipe on BUSI.

# I  SCALING ANALYSIS OF DATASET SIZE WITH DIFFERENT VARIANTS FAMILIES

We present the results of a comprehensive scaling analysis examining the relationship between dataset size and model performance. We conduct experiments using dataset subsampling at 25%, 50%, 75%, and 100%, exploring both compute efficiency and data efficiency across different architectures. To ensure the analysis is representative, we select the top 3 variants from each model family (Mamba, CNN, Hybrid, RWKV, and Transformer) as representatives. The total and architecture-specific scaling results are shown in Fig. 11, where we observe consistent improvements in performance as dataset size increases, indicating predictable scaling behavior across architectures. Additionally, we examine performance across multiple modality families (shown in Fig. 12), where similar trends are observed. As dataset size increases, we consistently observe performance gains, as exemplified by the Ultrasound modality, where performance improves from 0.6928 at 25% to 0.7463 at 100%. These findings suggest the strong relationship between dataset scaling and model performance, reinforcing the predictability of scaling behavior across both architectures and modalities.

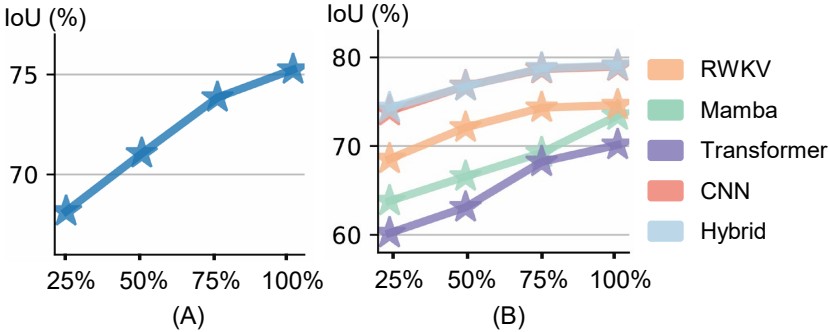

Figure 11: Overall Scaling Behavior (A) and Architecture-Specific Scaling Trends (B) with increasing dataset size (25%, 50%, 75%, and 100%) across different architectures (Mamba, CNN, Hybrid, RWKV, Transformer). The left figure shows the overall scaling trend, while the right figure displays the scaling behavior for each architecture family, highlighting the predictable improvement in performance as dataset scaling grows.

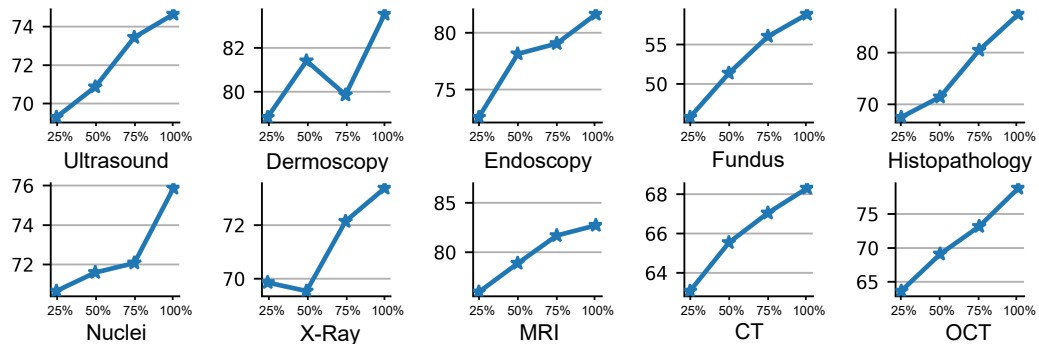

Figure 12: **Modality-Specific Scaling Trends.** The x-axis indicates increasing dataset size (25%, 50%, 75%, and 100%). The y-axes report IoU (%) performance.

## J INCORPORATE NEW DATASETS AND ALGORITHMS IN U-BENCH

We implement U-bench using the PyTorch framework. Figure 13 illustrates the comprehensive workflow of U-bench, a system tailored for medical image analysis. It features a versatile 2D/3D Dataloader that seamlessly accommodates multiple medical imaging modalities, including MRI, CT, X-Ray, Dermoscopy, and Fundus and so on. A rich assortment of models with diverse architectural designs—spanning CNN, Transformer, RWKV, Mamba, and Hybrid—are registered via a Model JSON configuration and then leveraged by the Trainer module for 2D/3D slice-wise training. The evaluation pipeline encompasses in-domain testing, zero-shot inference, statistical significance assessments and custom assessments of U-score. Finally, results are systematically logged and visualized using tools such as Weight & Biases (wandb), ensuring thorough tracking of metrics and checkpoints. We demonstrate how to integrate new datasets and algorithms through example pseudocode Figure 14.

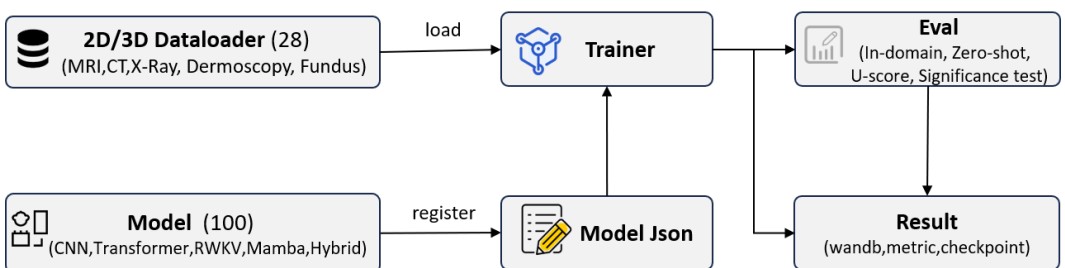

Figure 13: Overall workflow of U-bench. The Dataloader supports multiple medical imaging modalities (e.g., MRI, CT, X-Ray, Dermoscopy, Fundus). Models (with diverse architectures like CNN, Transformer, RWKV, Mamba, Hybrid) are registered and used by the Trainer for 2D/3D slice training. Evaluation covers In-domain/Zero-shot tasks, with results logged via tools like wandb.

### J.1 ADDING A NEW DATASET

If the existing `Dataset` classes cannot meet your processing requirements, you can implement your own dataset with the structure shown as in Figure 14 (a).

Additionally, you need to add your dataset name and loading method in the Dataloader file, as shown in the Figure 14 (b).

### J.2 ADDING A NEW ALGORITHM

1. First, define your model in the `models` directory, ensuring the first two parameters are `input_channel` and `num_classes` to adapt to our project (as shown in Figure 14 (c)).

2. Then, properly import your "modelname" in the `__init__.py` file under the `models` directory.

```
class CustomDataset(data.Dataset):                                          (a)
    def __init__(self):
        # TODO
        # 1. Initialize file path or list of file names.
        pass
    def __getitem__(self, index):
        # TODO
        # 1. Read one data from file (e.g. using numpy.fromfile, PIL.Image.open).
        # 2. Preprocess the data (e.g. torchvision.Transform).
        # 3. Return a data pair (e.g. image and label).

        pass
    def __len__(self):
        # You should change 0 to the total size of your dataset.
        return num
```

```
if "X" in args.base_dir:  # for XDataset xxxx;                              (b)
    db_train = XDataset(dataset_dir=args.base_dir, mode="train", transform=
                                          train_transform)
    db_val = XDataset(dataset_dir=args.base_dir, mode="test", transform=val_transform)
```

```
def modelname(input_channel=3, num_classes=2, base_channels=64):           (c)
    # Model implementation
    pass
```

```
{                                                                          (d)
    "modelname": "XXXX",
    "id": X,
    "state": "XXX",
    "deep_supervision": 0 or 1,
    "filename": "XXXX"
}
```

Figure 14: Pseudocode display. (a) Pseudocode for datasets; (b) Pseudocode for data loading; (c) Pseudocode for model definition and input parameters; (d) Example of model registration using a JSON file

3. Finally, register your model in the `model_id.json` file with the format shown in Figure 14 (d). Note that `modelname`, `id`, and `deep_supervision` are required fields, and `modelname` serves as the unique identifier for the model.

## K  DATASET AVAILABILITY AND ETHICAL COMPLIANCE

All datasets used in this benchmark, except for TNSCUI and TUCC, are publicly available. These datasets are listed in the manuscript along with appropriate citations and links for access. We have ensured that the use of each dataset complies with the respective ethical guidelines and regulations. Detailed information regarding the ethical considerations for each dataset can be found in the dataset documentation provided by the respective authors.

## L  EXPLANATION OF THE USE OF LLM

Large language models (LLMs) are employed solely to improve the clarity and readability of the manuscript.

Table 20: Reproduction-gap analysis across 100 U-shape segmentation models. For each dataset, we report our reproduced IoU and the corresponding ΔIoU relative to the original papers' reported metrics. Background denotes that the original paper adopts the same split ratio as our evaluation, while the specific partition may differ. A small and symmetric ΔIoU distribution indicates strong implementation fidelity and shows that the unified U-Bench recipe neither systematically benefits nor harms specific model families.

| Rank | Network | Ultrasound BUSI | Δ | TNSCUI | Δ | Dermoscopy ISIC2018 | Δ | Endoscopy Kvasir | Δ | Fundus CHASE | Δ | DRIVE | Δ | Histopathology DSB2018 | Δ | Glas | Δ | Nuclear Cell | Δ | X-Ray Montgomery | Δ | MRI ACDC | Δ | CT Synapse | Δ |
|---|---|---|---|---|---|---|---|---|---|---|---|---|---|---|---|---|---|---|---|---|---|---|---|---|---|
| #1 | RWKV-UNet | 72.32 | +2.95 | 80.06 | | 84.97 | | 84.53 | +0.61 | 70.85 | | 59.75 | | 88.10 | | 85.75 | -0.03 | 84.56 | | 96.21 | | 85.20 | -0.28 | 72.56 | +0.11 |
| #2 | UTANet | 69.00 | | 78.51 | | 83.84 | | 83.06 | | 79.81 | | 60.86 | | 87.70 | | 84.22 | -1.61 | 84.77 | | 95.76 | | 84.73 | | 69.73 | -9.21 |
| #3 | AURA-Net | 70.63 | | 78.32 | | 84.25 | | 83.53 | | 80.08 | | 58.22 | | 88.23 | | 82.77 | | 84.87 | | 95.67 | | 84.35 | | 73.25 | |
| #4 | Swin-umamba | 70.04 | | 78.64 | | 84.49 | | 85.56 | | 81.65 | | 62.75 | | 87.81 | | 79.06 | | 84.49 | | 95.56 | | 84.51 | | 71.69 | |
| #5 | TransResUNet | 71.27 | | 78.04 | | 84.38 | | 83.38 | +1.24 | 81.01 | | 58.85 | | 87.09 | | 82.35 | | 83.78 | | 95.89 | | 83.79 | | 69.65 | |
| #6 | FCBFormer | 68.24 | | 78.10 | | 83.60 | | 84.50 | -4.53 | 81.17 | | 64.25 | | 88.08 | | 79.81 | | 84.54 | | 95.05 | | 84.90 | | 72.06 | |
| #7 | MEGANet | 71.47 | | 79.01 | | 84.50 | | 82.60 | -2.01 | 73.27 | | 52.76 | | 87.33 | | 82.92 | | 84.37 | | 96.12 | | 83.99 | | 73.18 | |
| #8 | DA-TransUNet | 71.47 | | 77.95 | | 83.95 | +1.17 | 82.20 | +1.18 | 78.65 | | 58.05 | | 87.48 | | 82.89 | | 84.63 | | 96.17 | | 84.30 | | 70.15 | +3.76 |
| #9 | ESKNet | 70.88 | +0.68 | 78.29 | | 83.34 | | 82.11 | | 82.69 | | 63.15 | | 88.23 | | 82.48 | | 84.35 | | 95.65 | | 84.06 | | 70.78 | |
| #10 | CFFormer | 70.91 | -4.88 | 78.44 | | 84.89 | | 84.57 | -0.49 | 76.61 | | 60.05 | | 87.57 | | 82.17 | | 84.41 | | 95.87 | | 83.99 | | 70.89 | -0.99 |
| #11 | EViT-UNet | 70.50 | | 78.75 | | 84.03 | | 83.94 | | 74.06 | | 58.95 | | 87.32 | | 80.53 | -5.97 | 84.21 | | 95.27 | | 84.91 | | 67.17 | -0.72 |
| #12 | CMU-Net | 69.25 | -4.02 | 77.28 | | 82.74 | | 83.07 | | 84.33 | | 62.85 | | 87.87 | | 81.88 | | 84.13 | | 95.54 | | 84.24 | | 68.42 | |
| #13 | DDANet | 68.33 | | 77.53 | | 83.20 | | 83.06 | +5.06 | 78.94 | | 61.40 | | 88.16 | | 83.78 | | 84.17 | | 95.97 | | 85.11 | | 71.83 | |
| #14 | MADGNet | 71.24 | -3.96 | 78.91 | | 84.18 | | 84.32 | | 71.18 | | 54.35 | | 86.92 | | 83.03 | | 84.14 | | 95.69 | | 84.28 | | 72.20 | |
| #15 | Perspective-Unet | 67.00 | | 78.20 | | 82.78 | | 82.70 | | 82.86 | | 60.47 | | 87.29 | | 81.53 | | 82.99 | | 95.20 | | 83.80 | -2.31 | 73.69 | +0.34 |
| #16 | AttU-Net | 65.74 | | 76.30 | | 82.60 | | 79.23 | | 84.20 | | 62.27 | | 87.94 | | 83.06 | | 84.49 | | 95.73 | | 85.01 | | 71.19 | |
| #17 | UNet3+ | 65.06 | | 75.61 | | 82.87 | | 78.83 | | 83.69 | | 62.54 | | 87.99 | | 82.79 | | 84.19 | | 95.88 | | 84.27 | | 70.41 | |
| #18 | UTNet | 69.60 | | 77.42 | | 83.56 | | 80.92 | | 80.83 | | 63.17 | | 88.39 | | 80.42 | | 84.47 | | 95.73 | | 84.07 | | 69.16 | |
| #19 | Mobile U-ViT | 71.59 | -2.32 | 78.12 | -0.12 | 83.88 | +0.57 | 83.21 | -5.86 | 77.58 | | 60.69 | | 87.67 | | 79.05 | | 83.91 | | 95.89 | | 83.79 | | 70.84 | +4.32 |
| #20 | U-Net | 65.58 | | 75.99 | | 82.78 | | 80.11 | | 84.07 | | 61.81 | | 88.05 | | 83.30 | | 84.33 | | 95.87 | | 84.32 | | 67.90 | |
| #21 | MT-UNet | 67.01 | | 72.66 | | 82.84 | | 80.08 | | 82.00 | | 63.21 | | 88.74 | | 80.31 | | 84.93 | | 95.90 | | 83.68 | +1.15 | 69.81 | +5.08 |
| #22 | RollingUnet | 67.68 | -0.13 | 76.17 | | 83.07 | -1.07 | 81.50 | | 78.51 | +8.11 | 62.49 | | 88.14 | | 82.54 | -5.48 | 84.62 | | 96.01 | | 83.88 | | 70.12 | |
| #23 | UCTransNet | 68.01 | | 75.42 | | 83.10 | | 80.34 | | 82.82 | | 60.58 | | 87.72 | | 81.39 | -1.57 | 84.18 | | 95.70 | | 84.21 | | 69.66 | |
| #24 | MBSNet | 68.82 | | 76.78 | | 83.09 | | 81.44 | | 77.57 | | 61.61 | | 87.78 | | 83.57 | | 83.97 | | 95.79 | | 84.20 | | 69.66 | |
| #25 | TransUNet | 69.62 | | 77.61 | | 83.51 | | 80.93 | | 79.87 | | 59.76 | | 87.26 | | 79.33 | | 84.17 | | 95.72 | | 83.40 | +2.06 | 70.70 | +7.46 |
| #26 | TransAttUnet | 68.46 | | 77.65 | | 83.04 | -0.76 | 81.53 | | 75.63 | | 62.45 | | 88.49 | +3.51 | 81.04 | -0.09 | 84.88 | | 96.03 | -1.79 | 83.88 | | 70.11 | |
| #27 | LGMSNet | 69.45 | -3.95 | 77.71 | -0.32 | 83.68 | +0.22 | 81.77 | | 77.64 | | 60.05 | | 88.16 | | 81.92 | | 84.32 | | 95.56 | | 83.08 | | 69.35 | |
| #28 | CENet | 69.16 | | 77.34 | | 83.89 | | 84.32 | | 70.80 | | 58.38 | | 88.06 | | 83.47 | | 83.57 | | 94.46 | | 85.54 | +0.04 | 74.70 | +0.73 |
| #29 | H2Former | 69.80 | | 77.84 | | 83.31 | | 81.88 | -4.41 | 74.94 | | 57.93 | | 86.58 | | 82.16 | | 83.51 | | 95.82 | | 84.72 | -1.16 | 70.90 | +16.43 |
| #30 | CA-Net | 68.96 | | 77.34 | | 82.87 | | 80.58 | | 74.01 | | 61.97 | | 88.12 | | 81.63 | | 84.85 | | 95.62 | | 84.68 | | 70.14 | |
| #31 | CMUNeXt | 68.68 | -2.88 | 77.18 | -0.65 | 82.80 | | 80.21 | | 78.25 | | 60.99 | | 87.62 | | 82.12 | | 84.00 | | 95.54 | | 83.13 | | 68.05 | |
| #32 | D-TrAttUnet | 65.38 | | 75.21 | | 82.56 | | 80.78 | | 79.19 | | 62.48 | | 88.27 | | 79.38 | -6.58 | 84.15 | | 95.13 | | 83.05 | | 70.25 | |
| #33 | MedFormer | 67.08 | | 77.81 | | 83.09 | | 82.06 | | 79.90 | | 61.77 | | 87.64 | | 77.93 | | 83.61 | | 94.87 | | 83.30 | -2.12 | 67.21 | |
| #34 | FAT-Net | 70.17 | | 78.57 | | 83.91 | +1.89 | 83.39 | | 75.39 | | 50.01 | | 86.07 | | 82.81 | | 82.76 | | 95.43 | | 84.78 | | 64.45 | |
| #35 | MCA-UNet | 69.49 | | 78.26 | | 83.72 | | 82.25 | | 75.56 | | 58.90 | | 87.52 | | 79.85 | | 83.87 | | 95.22 | | 84.40 | | 58.69 | |
| #36 | MSLAU-Net | 68.69 | | 78.18 | | 83.56 | | 83.83 | | 70.05 | | 49.20 | | 85.98 | | 84.38 | | 82.70 | | 94.83 | | 84.39 | -1.02 | 72.60 | +1.39 |
| #37 | UNet++ | 66.44 | | 76.87 | | 82.59 | | 81.83 | | 79.72 | | 58.40 | | 87.11 | | 80.43 | | 83.56 | +10.64 | 95.30 | | 83.25 | | 67.43 | |
| #38 | TA-Net | 69.89 | | 78.85 | | 84.34 | | 83.33 | | 71.68 | | 50.29 | | 86.95 | | 81.60 | -1.04 | 83.60 | | 94.76 | | 82.78 | | 63.54 | |
| #39 | ResU-KAN | 67.38 | -0.36 | 77.59 | | 83.36 | | 82.70 | | 76.12 | | 57.84 | | 86.86 | | 79.44 | -8.55 | 83.93 | | 94.83 | | 81.95 | | 69.60 | |
| #40 | U-KAN | 67.10 | +3.72 | 77.15 | | 82.73 | | 82.11 | | 75.80 | | 57.71 | | 86.68 | | 79.75 | -7.89 | 83.55 | | 94.99 | | 82.44 | | 69.82 | |
| #41 | MSRFNet | 65.81 | | 77.30 | | 82.86 | -0.87 | 80.60 | -8.54 | 69.36 | | 57.25 | | 88.01 | +2.67 | 81.12 | | 83.82 | | 95.54 | | 83.56 | | 68.07 | |
| #42 | ColonSegNet | 62.77 | | 71.03 | | 82.06 | | 79.81 | +7.42 | 82.20 | | 63.19 | | 88.01 | | 80.56 | | 84.70 | | 95.65 | | 83.56 | | 68.07 | |
| #43 | GH-UNet | 66.20 | | 76.69 | | 82.55 | | 81.36 | -5.83 | 70.27 | | 57.92 | | 86.01 | | 81.91 | | 83.01 | | 95.01 | | 83.21 | -3.03 | 71.08 | +7.58 |
| #44 | CE-Net | 69.99 | | 78.21 | | 84.32 | | 82.11 | | 70.23 | | 49.23 | | 86.35 | | 81.20 | | 83.19 | | 95.13 | | 82.89 | | 64.67 | |
| #45 | ScribFormer | 66.37 | | 76.55 | | 83.13 | | 79.33 | | 74.34 | | 60.58 | | 87.93 | | 77.57 | | 84.13 | | 95.63 | | 83.52 | +3.67 | 70.96 | |
| #46 | DS-TransUNet | 63.91 | | 73.09 | | 83.30 | -1.93 | 84.02 | -1.88 | 72.97 | | 55.46 | | 87.82 | +1.70 | 81.46 | +3.01 | 84.37 | | 94.98 | | 82.65 | | 72.74 | |
| #47 | DCSAU-Net | 67.37 | | 78.00 | | 83.47 | -0.63 | 81.07 | | 70.08 | | 60.31 | | 88.44 | +3.44 | 77.00 | | 83.86 | | 95.14 | | 80.49 | | 67.80 | |
| #48 | DDS-UNet | 67.63 | | 76.19 | | 83.67 | -0.22 | 81.86 | +3.76 | 74.09 | | 54.06 | | 86.38 | | 79.88 | | 83.32 | | 95.14 | | 84.11 | | 67.21 | |
| #49 | TransNorm | 69.43 | | 77.72 | | 83.57 | +2.56 | 81.29 | | 76.98 | | 57.50 | | 86.53 | | 76.72 | | 83.66 | | 95.13 | | 81.96 | -3.57 | 66.17 | |
| #50 | MedVKAN | 66.71 | -4.19 | 77.26 | | 83.01 | | 81.93 | | 67.95 | | 57.50 | | 87.20 | | 78.18 | | 83.47 | | 95.53 | | 82.76 | | 72.63 | |
| #51 | DoubleUNet | 66.56 | | 74.95 | | 81.07 | -1.05 | 84.09 | | 78.51 | | 60.29 | | 88.61 | +4.54 | 77.13 | | 80.26 | | 95.53 | | 85.13 | | 72.63 | |
| #52 | AC-MambaSeg | 65.96 | | 76.17 | | 83.17 | | 81.48 | | 78.73 | | 54.28 | | 86.25 | | 75.39 | | 82.21 | | 94.33 | | 80.47 | | 65.96 | |
| #53 | MMUNet | 67.40 | | 77.60 | | 83.13 | | 82.79 | | 76.15 | | 58.40 | | 86.65 | | 73.61 | -12.88 | 83.09 | | 93.96 | | 80.32 | | 65.87 | |
| #54 | U-RWKV | 65.72 | -5.29 | 72.70 | | 82.68 | +5.29 | 78.28 | -1.30 | 77.13 | | 59.50 | | 87.98 | | 78.14 | | 84.00 | | 95.18 | | 81.13 | | 64.59 | |
| #55 | HiFormer | 67.28 | | 77.71 | | 83.96 | +0.45 | 83.36 | | 66.14 | | 47.23 | | 85.69 | | 79.74 | | 82.22 | | 95.09 | | 83.25 | | 69.18 | +1.54 |
| #56 | ResUNet++ | 62.36 | | 73.01 | | 82.30 | | 78.36 | -0.91 | 79.93 | | 61.55 | | 87.15 | | 76.16 | | 83.75 | | 94.86 | | 81.45 | | 62.47 | |
| #57 | CASCADE | 70.81 | | 77.49 | | 83.53 | | 84.34 | -3.42 | 59.33 | | 33.42 | | 86.20 | | 85.17 | | 81.76 | | 94.38 | | 84.43 | -0.12 | 73.30 | +2.83 |
| #58 | CSCAUNet | 67.19 | | 77.52 | | 82.79 | -3.12 | 82.25 | -2.35 | 70.20 | | 28.59 | | 86.87 | +0.62 | 81.58 | | 82.99 | | 95.13 | | 82.19 | | 71.09 | |
| #59 | Tinyunet | 61.96 | | 74.71 | | 82.12 | -3.37 | 77.22 | | 70.74 | | 59.07 | | 86.81 | | 79.47 | | 83.40 | | 95.11 | | 81.56 | | 67.22 | |
| #60 | LV-UNet | 66.35 | +1.46 | 75.39 | | 83.53 | | 81.88 | +1.19 | 64.48 | | 46.42 | | 86.56 | | 78.69 | | 82.37 | | 94.42 | | 80.41 | | 64.14 | |
| #61 | G-CASCADE | 69.81 | | 77.21 | | 83.52 | -3.01 | 83.56 | -4.34 | 55.59 | | 36.32 | | 86.90 | | 82.70 | | 81.43 | | 94.54 | | 84.89 | -0.69 | 73.54 | +0.32 |
| #62 | DC-UNet | 61.31 | | 73.10 | | 80.97 | | 77.01 | | 61.08 | | 58.03 | | 86.36 | | 81.13 | | 82.86 | | 95.13 | | 83.74 | | 66.64 | |
| #63 | ERDUnet | 66.72 | | 76.04 | | 82.46 | -1.65 | 76.04 | -8.61 | 65.86 | | 55.23 | | 87.43 | | 72.75 | -14.16 | 83.07 | | 93.56 | | 79.46 | | 63.86 | |
| #64 | ConvFormer | 67.45 | | 75.96 | | 82.86 | | 80.25 | | 62.90 | | 37.90 | | 84.32 | | 81.02 | | 81.68 | | 95.55 | | 83.28 | | 64.13 | |
| #65 | ULite | 63.97 | | 70.75 | | 82.26 | -1.37 | 76.05 | | 65.37 | | 58.10 | | 87.45 | +2.59 | 74.47 | -3.37 | 83.27 | | 94.25 | | 74.04 | | 61.42 | |
| #66 | SwinUNETR | 62.50 | | 70.31 | | 82.70 | | 80.02 | | 70.20 | | 61.53 | | 87.18 | | 70.26 | | 83.74 | | 94.25 | | 74.04 | | 62.33 | |
| #67 | SCUNet++ | 61.48 | | 73.90 | | 83.66 | | 82.91 | | 75.17 | | 34.54 | | 85.46 | | 79.68 | | 84.20 | | 94.41 | | 82.98 | | 62.33 | |
| #68 | UNeXt | 62.11 | -4.84 | 71.13 | | 82.41 | +0.71 | 75.64 | | 65.51 | | 46.52 | | 86.29 | | 76.30 | | 82.83 | | 94.76 | | 77.43 | | 60.17 | |
| #69 | MERIT | 69.24 | | 77.46 | | 83.85 | | 83.40 | | 48.57 | | 40.41 | | 82.49 | | 83.77 | | 81.39 | | 91.78 | | 57.79 | -27.94 | 68.78 | -4.98 |
| #70 | MDSA-UNet | 67.63 | | 76.44 | | 83.02 | -0.04 | 79.28 | | 53.47 | | 37.81 | | 85.63 | | 78.23 | | 81.63 | | 94.64 | | 78.88 | -4.69 | 64.59 | |
| #71 | CPCANet | 66.01 | | 72.75 | | 82.13 | | 81.49 | | 70.88 | | 20.77 | | 85.48 | | 69.88 | | 81.07 | | 93.73 | | 81.68 | -4.54 | 67.38 | |
| #72 | LeViT-UNet | 58.38 | | 66.10 | | 81.50 | | 75.36 | | 71.19 | | 44.36 | | 85.95 | | 73.78 | | 81.86 | | 94.80 | | 79.28 | -3.07 | 64.37 | -0.28 |
| #73 | DAEFormer | 63.39 | | 73.07 | | 82.17 | -2.11 | 80.99 | | 69.98 | | 28.50 | | 86.82 | | 71.62 | | 81.23 | | 92.72 | | 81.81 | | 65.07 | -5.33 |
| #74 | PraNet | 71.63 | | 78.69 | | 84.42 | | 84.39 | +0.39 | 36.43 | | 26.97 | | 78.92 | | 82.65 | | 77.50 | | 94.17 | | 84.18 | | 70.84 | |
| #75 | CaraNet | 70.61 | | 78.69 | | 84.26 | | 83.24 | -3.26 | 37.69 | | 26.30 | | 79.04 | | 83.09 | | 77.18 | | 94.44 | | 83.96 | | 68.97 | |
| #76 | TransFuse | 69.77 | | 77.79 | | 84.11 | | 83.01 | -3.99 | 49.62 | | 18.52 | | 85.15 | | 75.03 | | 81.48 | | 93.98 | | 80.97 | | 64.11 | |
| #77 | MedT | 60.85 | | 70.13 | | 82.12 | | 74.78 | | 63.13 | | 54.62 | | 87.18 | | 70.33 | +0.72 | 82.62 | | 93.31 | | 73.87 | | 58.81 | |
| #78 | EMCAD | 68.56 | +1.55 | 70.95 | | 84.01 | +0.60 | 84.20 | -2.28 | 58.61 | | 44.65 | | 86.97 | +0.51 | 79.86 | | 82.55 | | 94.69 | | 73.31 | -12.09 | 69.28 | |
| #79 | UCANet | 69.88 | | 78.83 | | 83.76 | | 84.81 | -1.09 | 36.30 | | 25.48 | | 78.16 | | 79.86 | | 76.54 | | 94.49 | | 82.52 | | 69.28 | |
| #80 | MultiResUNet | 66.58 | | 75.20 | | 82.47 | | 80.57 | | 30.58 | | 19.22 | | 84.69 | | 80.93 | | 82.38 | | 92.49 | | 81.98 | | 64.45 | |
| #81 | MUCM-Net | 61.25 | | 68.99 | | 81.99 | -1.41 | 74.14 | | 57.12 | | 41.63 | | 84.87 | | 73.10 | | 79.58 | | 93.63 | | 73.44 | | 54.66 | |
| #82 | LFU-Net | 57.58 | | 62.52 | | 80.51 | | 63.84 | | 59.19 | | 55.87 | | 86.71 | | 74.73 | | 81.55 | | 93.01 | | 72.68 | | 51.19 | |
| #83 | MissFormer | 62.20 | | 72.78 | | 81.49 | | 80.23 | | 60.91 | | 19.32 | | 85.73 | | 65.47 | | 82.55 | | 91.53 | | 77.04 | -6.21 | 55.53 | -5.53 |
| #84 | UNETR | 52.28 | | 48.34 | | 80.87 | | 72.01 | | 60.45 | | 61.35 | | 87.52 | | 61.49 | | 83.17 | | 91.61 | | 71.60 | | 54.86 | |
| #85 | CFM-UNet | 58.88 | | 72.51 | | 81.81 | | 76.53 | -4.59 | 47.18 | | 14.58 | | 84.85 | | 70.27 | | 80.60 | | 92.87 | | 69.65 | | 59.89 | |
| #86 | Zig-RiR | 54.17 | | 66.18 | | 81.75 | -5.14 | 74.98 | | 56.12 | | 35.55 | | 84.71 | | 68.28 | | 80.35 | | 92.84 | | 66.22 | -20.26 | 50.85 | -31.55 |
| #87 | SimpleUNet | 55.08 | | 62.93 | | 79.52 | -2.06 | 69.60 | -1.27 | 52.60 | | 54.68 | | 85.98 | | 77.27 | | 79.75 | | 94.08 | | 65.87 | | 62.85 | |
| #88 | BEFUnet | 65.22 | | 73.99 | | 82.91 | +1.03 | 79.22 | | 67.89 | | 58.15 | +1.00 | 87.03 | | 79.05 | | 83.22 | | 94.49 | | 82.65 | | 64.19 | |
| #89 | UNetV2 | 61.91 | | 65.36 | | 79.88 | | 76.88 | | 51.04 | | 15.83 | | 84.46 | | 60.78 | | 79.57 | | 90.36 | | 75.57 | | 57.19 | -10.13 |
| #90 | UNetV2 | 58.12 | | 70.15 | | 81.49 | -2.66 | 79.25 | -8.75 | 40.72 | | 15.17 | | 84.52 | | 67.10 | | 80.56 | | 91.43 | | 72.43 | | 40.58 | |
| #91 | Swin-umambaD | 64.94 | | 75.19 | | 81.93 | | 84.27 | | 50.01 | | 14.24 | | 82.55 | | 53.22 | | 34.65 | | 94.39 | | 85.45 | | 55.09 | |
| #92 | VMUNetV2 | 61.69 | | 74.85 | | 82.91 | +1.54 | 84.03 | -0.12 | 46.22 | | 13.02 | | 82.92 | | 59.07 | | 62.69 | | 90.97 | | 82.07 | | 65.87 | |
| #93 | VMUNet | 67.38 | | 75.31 | | 83.01 | +1.66 | 84.90 | | 53.54 | | 16.37 | | 85.88 | | 52.81 | | 9.39 | | 93.87 | | 83.91 | | 68.52 | +0.34 |
| #94 | UltraLight-VM-UNet | 57.58 | | 59.18 | | 80.45 | -0.11 | 63.17 | | 50.32 | | 14.64 | | 85.99 | | 60.37 | | 78.05 | | 89.39 | | 72.64 | | 50.34 | |
| #95 | Polyp-PVT | 70.54 | | 76.68 | | 83.33 | | 83.95 | -2.45 | 35.87 | | 1.28 | | 81.59 | | 51.22 | | 64.22 | | 91.57 | | 84.52 | | 54.32 | |
| #96 | H-vmunet | 61.83 | | 64.49 | | 81.66 | | 74.22 | | 46.58 | | 10.85 | | 85.58 | | 67.11 | | 77.57 | | 91.57 | | 70.28 | | 54.32 | |
| #97 | CSWin-UNet | 56.38 | | 69.71 | | 81.67 | -2.00 | 75.09 | | 27.08 | | 14.59 | | 85.60 | | 63.12 | | 79.19 | | 88.68 | | 73.93 | -10.34 | 56.83 | -11.40 |
| #98 | MALUNet | 59.10 | | 68.27 | | 80.99 | +0.74 | 72.56 | | 11.87 | | 15.08 | | 84.85 | | 61.99 | | 78.91 | | 90.18 | | 63.01 | | 51.51 | |
| #99 | SwinUnet | 49.81 | | 27.35 | | 81.18 | | 67.11 | | 49.19 | | 10.66 | | 85.52 | | 57.82 | | 64.08 | | 90.07 | | 70.62 | -11.20 | 50.08 | -15.39 |
| #100 | MambaUnet | 19.01 | | 50.64 | | 79.22 | | 84.28 | | 53.76 | | 11.99 | | 86.53 | | 52.52 | | 0.67 | | 56.49 | | 84.62 | +7.66 | 68.88 | +31.85 |

Table 21: Average performance of 100 u-shape medical image segmentation networks with IoU. Baseline U-Net is highlighted (gray background), and statistical significance of p-value is highlighted: p<0.0001 , p<0.001 , p<0.05 , p≤0.05 , and P > 0.05 (Not significant) .

| Rank | Network | Ultrasound | | | Dermoscopy | | Endoscopy | | Fundus | | Histopathology | | | Nuclear | Covidquex | X-Ray | | MRI | | CT | OCT | Avg |
|---|---|---|---|---|---|---|---|---|---|---|---|---|---|---|---|---|---|---|---|---|---|---|
| | | BUSI | BUSBRA | TNSCUI | ISIC2018 | SkinCancer | Kvasir | Robotool | CHASE | DRIVE | DSB2018 | Glas | Monusac | Cell | | Montgomery | DCA | ACDC | Promise | Synapse | Cystoidfluid | |
| 1 | RWKV-UNet | 72.32 | 84.76 | 80.06 | 84.97 | 87.48 | 84.53 | 85.95 | 70.85 | 59.75 | 88.10 | 85.75 | 68.96 | 84.56 | 70.75 | 96.21 | 63.61 | 85.20 | 87.56 | 72.56 | 72.56 | 79.84 |
| 2 | UTANet | 69.00 | 83.93 | 78.51 | 83.84 | 84.48 | 83.06 | 84.62 | 79.81 | 60.86 | 87.70 | 84.22 | 68.39 | 84.77 | 69.29 | 95.76 | 64.23 | 84.73 | 86.87 | 69.73 | 69.73 | 79.43 |
| 3 | AURA-Net | 70.63 | 83.95 | 78.32 | 84.25 | 85.56 | 83.53 | 85.78 | 80.08 | 58.22 | 88.23 | 82.77 | 66.56 | 84.87 | 70.85 | 95.67 | 63.25 | 84.35 | 86.66 | 73.25 | 73.25 | 79.37 |
| #4 | Swin-umamba | 70.04 | 83.48 | 78.64 | 84.49 | 83.92 | 85.56 | 84.04 | 81.65 | 62.75 | 87.81 | 79.06 | 64.12 | 84.49 | 69.48 | 95.56 | 62.73 | 84.51 | 86.29 | 71.69 | 71.69 | 79.27 |
| #5 | TransResUNet | 71.27 | 83.86 | 78.04 | 84.38 | 84.79 | 83.38 | 85.30 | 81.01 | 58.85 | 87.09 | 82.35 | 66.86 | 83.78 | 69.81 | 95.89 | 63.25 | 83.79 | 86.95 | 69.65 | 69.65 | 79.13 |
| #6 | FCBFormer | 68.24 | 84.00 | 78.10 | 83.60 | 86.33 | 84.50 | 77.58 | 81.17 | 64.25 | 88.08 | 79.81 | 66.29 | 84.54 | 67.60 | 95.05 | 61.70 | 84.90 | 87.29 | 72.06 | 72.06 | 78.95 |
| #7 | MEGANet | 71.47 | 83.99 | 79.01 | 84.50 | 85.23 | 84.29 | 86.25 | 73.27 | 52.76 | 87.33 | 82.92 | 65.86 | 84.37 | 70.19 | 96.12 | 63.77 | 83.99 | 87.00 | 73.18 | 73.18 | 78.85 |
| #8 | DA-TransUNet | 71.47 | 83.91 | 77.95 | 83.95 | 87.22 | 82.20 | 83.09 | 78.65 | 58.05 | 87.48 | 82.89 | 64.26 | 84.63 | 69.61 | 96.17 | 64.90 | 84.30 | 85.84 | 70.15 | 70.15 | 78.82 |
| #9 | ESKNet | 70.88 | 83.77 | 78.29 | 83.34 | 81.93 | 82.11 | 81.36 | 82.69 | 63.15 | 88.23 | 82.48 | 66.92 | 84.35 | 68.82 | 95.65 | 63.69 | 84.06 | 86.34 | 70.78 | 70.78 | 78.79 |
| #10 | CFFormer | 70.91 | 82.42 | 78.44 | 84.89 | 83.87 | 84.57 | 83.80 | 76.61 | 60.05 | 87.57 | 82.17 | 64.51 | 84.41 | 68.40 | 95.87 | 62.17 | 83.99 | 86.58 | 70.89 | 70.89 | 78.76 |
| #11 | EViT-UNet | 70.50 | 84.36 | 78.75 | 84.03 | 82.90 | 83.94 | 85.39 | 74.06 | 58.95 | 87.32 | 80.53 | 67.61 | 84.21 | 70.33 | 95.27 | 63.81 | 84.91 | 87.05 | 67.17 | 67.17 | 78.73 |
| #12 | CMU-Net | 69.25 | 83.39 | 77.28 | 82.74 | 81.90 | 83.07 | 80.25 | 84.33 | 62.85 | 87.87 | 81.88 | 66.48 | 84.13 | 68.28 | 95.54 | 62.64 | 84.24 | 86.87 | 68.42 | 68.42 | 78.70 |
| #13 | DDANet | 68.33 | 83.36 | 77.53 | 83.20 | 80.91 | 83.06 | 80.35 | 78.94 | 61.40 | 88.16 | 83.78 | 68.38 | 84.17 | 68.32 | 95.97 | 63.65 | 85.11 | 86.20 | 71.83 | 71.83 | 78.69 |
| #14 | MADGNet | 71.24 | 84.27 | 78.91 | 84.18 | 84.67 | 84.32 | 85.21 | 71.18 | 54.35 | 86.92 | 83.03 | 64.66 | 84.14 | 69.77 | 95.69 | 63.81 | 84.28 | 87.26 | 72.20 | 72.20 | 78.62 |
| #15 | Perspective-Unet | 67.00 | 83.53 | 78.20 | 82.78 | 84.29 | 82.70 | 79.77 | 82.86 | 60.47 | 87.29 | 81.53 | 64.66 | 82.99 | 67.54 | 95.20 | 62.35 | 83.80 | 87.03 | 71.69 | 71.69 | 78.58 |
| #16 | AttU-Net | 65.74 | 83.13 | 76.30 | 82.60 | 81.74 | 79.23 | 81.04 | 84.20 | 62.27 | 87.44 | 83.06 | 66.96 | 84.49 | 68.93 | 95.73 | 63.14 | 85.01 | 86.69 | 71.19 | 71.19 | 78.57 |
| #17 | UNet3+ | 65.06 | 82.92 | 75.61 | 82.80 | 81.85 | 78.83 | 81.13 | 83.69 | 62.54 | 87.99 | 82.79 | 66.36 | 84.19 | 68.69 | 95.88 | 63.27 | 84.27 | 86.68 | 70.41 | 70.41 | 78.54 |
| #18 | UTNet | 69.60 | 83.09 | 77.42 | 83.56 | 82.08 | 80.92 | 79.66 | 80.83 | 63.17 | 88.39 | 80.42 | 67.76 | 84.47 | 68.31 | 95.73 | 63.54 | 84.07 | 86.04 | 69.16 | 69.16 | 78.52 |
| #19 | Mobile U-ViT | 71.59 | 83.64 | 78.12 | 83.88 | 83.64 | 83.21 | 80.31 | 77.58 | 60.69 | 87.67 | 79.05 | 66.63 | 83.91 | 68.57 | 95.89 | 63.19 | 83.79 | 85.29 | 70.84 | 70.84 | 78.38 |
| #20 | U-Net | 65.58 | 82.91 | 75.99 | 82.78 | 80.94 | 80.11 | 81.24 | 84.07 | 61.81 | 88.05 | 83.30 | 66.44 | 84.33 | 68.52 | 95.87 | 63.30 | 84.32 | 86.30 | 67.90 | 67.90 | 78.31 |
| #21 | MT-UNet | 67.01 | 81.80 | 72.66 | 82.84 | 84.41 | 80.08 | 80.82 | 82.00 | 63.21 | 88.74 | 80.31 | 69.27 | 84.93 | 67.50 | 95.90 | 63.23 | 83.68 | 85.63 | 69.81 | 69.81 | 78.26 |
| #22 | RollingUnet | 67.68 | 83.44 | 76.17 | 83.07 | 81.00 | 81.50 | 80.34 | 78.51 | 62.49 | 88.14 | 82.54 | 66.52 | 84.62 | 68.83 | 96.01 | 63.09 | 83.88 | 85.87 | 70.12 | 70.12 | 78.25 |
| #23 | UCTransNet | 68.01 | 82.51 | 75.42 | 83.10 | 82.77 | 80.34 | 81.23 | 82.82 | 60.38 | 87.72 | 81.39 | 66.59 | 84.18 | 68.34 | 95.70 | 62.25 | 84.21 | 86.01 | 69.62 | 69.62 | 78.24 |
| #24 | MBSNet | 68.82 | 83.60 | 76.78 | 83.09 | 82.86 | 81.44 | 78.25 | 77.57 | 61.61 | 87.78 | 83.57 | 66.81 | 83.97 | 68.54 | 95.79 | 62.95 | 84.20 | 85.98 | 69.66 | 69.66 | 78.23 |
| #25 | TransUNet | 69.62 | 82.84 | 77.61 | 83.51 | 84.85 | 80.93 | 78.30 | 79.87 | 59.76 | 87.26 | 79.33 | 65.21 | 84.17 | 68.95 | 95.72 | 62.85 | 83.40 | 86.01 | 70.70 | 70.70 | 78.15 |
| #26 | TransAttUnet | 68.46 | 83.28 | 77.65 | 83.04 | 79.11 | 81.53 | 80.01 | 75.63 | 62.45 | 88.49 | 81.04 | 68.25 | 84.88 | 69.34 | 96.03 | 63.04 | 83.88 | 85.54 | 70.11 | 70.11 | 78.13 |
| #27 | LGMSNet | 69.45 | 83.22 | 77.71 | 83.68 | 85.22 | 81.77 | 78.45 | 77.64 | 60.05 | 88.16 | 81.92 | 64.88 | 84.32 | 68.14 | 95.56 | 62.01 | 83.08 | 86.02 | 69.35 | 69.35 | 78.02 |
| #28 | CENet | 69.16 | 82.83 | 77.34 | 83.89 | 84.13 | 84.32 | 80.07 | 70.80 | 58.38 | 88.06 | 83.47 | 64.12 | 83.57 | 67.65 | 94.46 | 61.63 | 85.54 | 83.62 | 74.70 | 74.70 | 78.00 |
| #29 | H2Former | 69.80 | 83.41 | 77.84 | 83.31 | 83.02 | 81.88 | 78.40 | 74.94 | 57.93 | 86.58 | 82.16 | 65.58 | 83.51 | 67.45 | 95.82 | 62.18 | 84.72 | 86.35 | 70.90 | 70.90 | 77.99 |
| #30 | CA-Net | 68.96 | 82.87 | 77.34 | 82.87 | 82.48 | 80.58 | 77.61 | 74.01 | 61.97 | 88.12 | 81.63 | 68.39 | 84.85 | 68.59 | 95.62 | 63.01 | 84.68 | 85.36 | 70.14 | 70.14 | 77.95 |
| #31 | CMUNeXt | 68.68 | 83.63 | 77.18 | 82.80 | 85.39 | 80.21 | 77.83 | 78.25 | 60.99 | 87.62 | 82.12 | 65.35 | 84.00 | 68.08 | 95.54 | 62.59 | 83.13 | 85.57 | 68.05 | 68.05 | 77.88 |
| #32 | D-TrAttUnet | 65.38 | 82.03 | 75.21 | 82.56 | 83.82 | 80.78 | 78.93 | 79.19 | 62.48 | 88.27 | 79.38 | 66.96 | 84.15 | 67.07 | 95.13 | 62.67 | 83.05 | 85.87 | 70.25 | 70.25 | 77.85 |
| #33 | MedFormer | 67.08 | 82.43 | 77.81 | 83.09 | 82.16 | 82.06 | 77.61 | 79.90 | 61.77 | 87.64 | 77.93 | 66.52 | 83.61 | 67.29 | 94.87 | 62.43 | 83.30 | 86.53 | 67.21 | 67.21 | 77.74 |
| #34 | FAT-Net | 70.17 | 84.15 | 78.57 | 83.91 | 84.11 | 83.39 | 84.34 | 75.39 | 50.01 | 86.07 | 82.81 | 64.82 | 82.76 | 69.48 | 95.43 | 63.15 | 84.78 | 86.06 | 64.45 | 64.45 | 77.74 |
| #35 | MCA-UNet | 69.49 | 83.78 | 78.26 | 83.72 | 86.17 | 82.25 | 79.97 | 75.56 | 58.90 | 87.52 | 79.85 | 65.09 | 83.87 | 67.63 | 95.22 | 62.13 | 84.40 | 85.84 | 69.69 | 69.69 | 77.57 |
| #36 | MSLAU-Net | 68.69 | 82.69 | 78.18 | 83.56 | 86.80 | 83.83 | 82.93 | 70.05 | 49.20 | 85.98 | 84.38 | 62.82 | 82.70 | 68.30 | 94.83 | 61.94 | 84.39 | 84.84 | 72.60 | 72.60 | 77.55 |
| #37 | U-Net++ | 66.44 | 82.10 | 76.87 | 82.59 | 83.16 | 81.83 | 79.08 | 79.72 | 58.40 | 87.11 | 80.43 | 64.63 | 83.56 | 66.74 | 95.30 | 61.41 | 83.25 | 86.40 | 67.43 | 67.43 | 77.45 |
| #38 | TA-Net | 69.89 | 84.17 | 78.85 | 84.34 | 83.32 | 83.33 | 85.62 | 71.68 | 50.29 | 86.95 | 81.60 | 64.34 | 83.60 | 70.20 | 94.76 | 62.51 | 82.78 | 86.39 | 63.54 | 63.54 | 77.45 |
| #39 | ResU-KAN | 67.38 | 83.01 | 77.39 | 83.36 | 82.12 | 82.70 | 77.54 | 76.12 | 57.84 | 86.86 | 79.44 | 64.16 | 83.89 | 67.75 | 94.83 | 62.05 | 81.95 | 86.35 | 69.60 | 69.60 | 77.38 |
| #40 | U-KAN | 67.10 | 82.81 | 77.15 | 82.73 | 84.94 | 82.11 | 77.77 | 75.80 | 57.71 | 86.68 | 79.75 | 63.29 | 83.55 | 66.93 | 94.99 | 61.34 | 82.44 | 86.89 | 69.82 | 69.82 | 77.34 |
| #41 | MSRFNet | 65.81 | 83.16 | 77.30 | 82.86 | 80.87 | 80.60 | 78.32 | 69.36 | 57.27 | 88.01 | 81.12 | 66.29 | 83.82 | 67.64 | 95.54 | 63.25 | 84.78 | 85.81 | 70.16 | 70.16 | 77.14 |
| #42 | ColonSegNet | 62.77 | 78.47 | 71.03 | 82.06 | 81.92 | 79.81 | 77.87 | 82.20 | 63.19 | 87.81 | 84.70 | 66.67 | 84.00 | 66.47 | 95.18 | 62.82 | 81.13 | 83.99 | 66.07 | 66.07 | 77.11 |
| #43 | GH-UNet | 66.20 | 82.86 | 76.69 | 82.55 | 84.98 | 81.36 | 77.58 | 70.27 | 57.92 | 86.01 | 81.91 | 63.09 | 83.01 | 67.16 | 95.01 | 62.17 | 83.21 | 86.19 | 71.08 | 71.08 | 77.09 |
| #44 | CE-Net | 69.99 | 83.74 | 78.21 | 84.32 | 83.64 | 82.11 | 85.11 | 70.23 | 49.23 | 86.35 | 81.20 | 62.25 | 83.19 | 70.04 | 95.13 | 62.14 | 82.89 | 85.96 | 64.67 | 64.67 | 77.08 |
| #45 | ScribFormer | 66.37 | 82.02 | 76.55 | 83.13 | 80.01 | 79.33 | 75.38 | 74.34 | 60.58 | 87.93 | 77.57 | 66.84 | 84.13 | 67.98 | 95.63 | 62.26 | 83.52 | 84.89 | 70.96 | 70.96 | 77.02 |
| #46 | DS-TransUNet | 63.91 | 81.29 | 73.09 | 83.30 | 83.74 | 84.02 | 82.80 | 72.97 | 55.46 | 87.82 | 81.46 | 64.05 | 83.10 | 66.44 | 94.37 | 60.83 | 84.68 | 83.10 | 72.74 | 72.74 | 76.91 |
| #47 | DCSAU-Net | 67.37 | 83.21 | 78.00 | 83.47 | 81.47 | 81.07 | 77.78 | 70.08 | 60.31 | 88.44 | 77.00 | 64.48 | 83.86 | 68.51 | 94.98 | 62.40 | 85.42 | 86.33 | 66.33 | 66.33 | 76.83 |
| #48 | DDS-UNet | 67.63 | 82.74 | 76.19 | 83.67 | 83.58 | 81.86 | 78.22 | 74.09 | 54.06 | 86.38 | 79.88 | 62.80 | 83.32 | 62.45 | 95.14 | 61.94 | 80.49 | 85.78 | 67.80 | 67.80 | 76.75 |
| #49 | TransNorm | 69.43 | 83.14 | 77.72 | 83.57 | 85.86 | 81.29 | 79.36 | 76.98 | 57.25 | 86.53 | 76.72 | 63.39 | 83.66 | 67.96 | 95.13 | 61.95 | 74.11 | 86.37 | 61.95 | 61.95 | 76.72 |
| #50 | MedVKAN | 66.71 | 83.92 | 77.26 | 83.01 | 82.77 | 81.93 | 78.33 | 67.95 | 57.50 | 83.78 | 78.18 | 65.57 | 83.47 | 68.43 | 95.00 | 61.78 | 83.96 | 86.06 | 66.17 | 66.17 | 76.70 |
| #51 | DoubleUNet | 66.56 | 81.79 | 74.95 | 83.07 | 79.15 | 84.09 | 75.65 | 78.51 | 60.29 | 88.61 | 77.13 | 65.59 | 80.26 | 67.40 | 95.53 | 60.54 | 85.33 | 81.01 | 72.63 | 72.63 | 76.54 |
| #52 | AC-MambaSeg | 65.96 | 82.05 | 76.17 | 83.17 | 83.31 | 81.48 | 75.51 | 78.73 | 54.28 | 86.25 | 75.39 | 62.73 | 82.21 | 68.10 | 94.33 | 60.34 | 80.47 | 86.32 | 69.96 | 69.96 | 76.34 |
| #53 | MMUNet | 67.40 | 83.15 | 77.60 | 83.13 | 80.89 | 82.79 | 76.90 | 76.15 | 58.40 | 86.65 | 73.61 | 61.44 | 83.09 | 66.84 | 93.96 | 60.19 | 80.32 | 84.96 | 65.87 | 65.87 | 76.34 |
| #54 | U-RWKV | 65.72 | 81.95 | 72.70 | 82.68 | 80.76 | 78.28 | 75.33 | 77.13 | 59.50 | 87.98 | 78.14 | 65.00 | 84.00 | 66.47 | 95.18 | 62.82 | 81.13 | 85.64 | 64.59 | 64.59 | 76.34 |
| #55 | HiFormer | 67.28 | 83.53 | 77.71 | 83.96 | 84.90 | 83.36 | 82.95 | 66.14 | 47.23 | 85.69 | 79.74 | 62.18 | 83.22 | 68.38 | 95.09 | 60.02 | 83.25 | 85.99 | 69.18 | 69.18 | 76.32 |
| #56 | ResUNet++ | 62.36 | 80.43 | 73.01 | 82.30 | 82.58 | 78.36 | 75.13 | 79.93 | 61.55 | 87.15 | 76.16 | 63.59 | 83.75 | 66.50 | 94.86 | 59.50 | 81.45 | 86.40 | 62.47 | 62.47 | 76.04 |
| #57 | CASCADE | 70.81 | 82.42 | 77.49 | 83.53 | 85.19 | 84.34 | 80.35 | 59.33 | 33.42 | 86.20 | 85.17 | 58.27 | 81.76 | 68.22 | 94.38 | 61.34 | 84.43 | 86.42 | 73.30 | 73.30 | 75.65 |
| #58 | CSCAUNet | 67.19 | 82.79 | 77.52 | 82.79 | 82.24 | 82.25 | 78.05 | 70.20 | 28.59 | 86.87 | 81.58 | 62.25 | 83.81 | 66.01 | 94.86 | 54.13 | 82.19 | 86.24 | 71.09 | 71.09 | 75.51 |
| #59 | Tinyunet | 61.96 | 82.01 | 74.71 | 82.12 | 81.44 | 77.22 | 75.09 | 70.74 | 59.07 | 86.81 | 79.47 | 60.71 | 83.40 | 66.80 | 95.11 | 59.83 | 81.56 | 84.17 | 67.22 | 67.22 | 75.46 |
| #60 | LV-UNet | 66.35 | 82.08 | 75.39 | 83.53 | 83.55 | 81.88 | 80.21 | 64.48 | 46.42 | 86.56 | 78.69 | 62.87 | 82.37 | 68.61 | 94.42 | 61.21 | 80.41 | 84.82 | 64.14 | 64.14 | 75.31 |
| #61 | G-CASCADE | 69.81 | 82.21 | 77.21 | 83.52 | 84.46 | 83.56 | 78.54 | 55.59 | 36.32 | 86.90 | 82.70 | 60.80 | 81.43 | 67.94 | 94.54 | 59.94 | 84.89 | 83.98 | 73.54 | 73.54 | 75.30 |
| #62 | DC-UNet | 61.31 | 80.30 | 73.10 | 80.97 | 78.43 | 77.01 | 76.19 | 68.10 | 58.03 | 86.36 | 81.13 | 64.79 | 82.86 | 68.79 | 95.13 | 57.07 | 83.74 | 84.25 | 60.55 | 60.55 | 74.74 |
| #63 | ERDUnet | 66.72 | 82.01 | 76.04 | 82.48 | 79.44 | 76.04 | 73.28 | 65.86 | 55.23 | 87.43 | 72.75 | 64.59 | 83.07 | 66.42 | 93.56 | 61.13 | 79.46 | 84.38 | 63.86 | 63.86 | 74.64 |
| #64 | ConvFormer | 67.45 | 82.41 | 75.96 | 82.86 | 82.15 | 80.25 | 79.03 | 62.90 | 37.90 | 84.32 | 81.02 | 58.43 | 81.68 | 69.05 | 95.55 | 59.80 | 83.28 | 85.62 | 64.13 | 64.13 | 74.51 |
| #65 | ULite | 63.97 | 80.70 | 70.75 | 82.26 | 82.79 | 76.05 | 73.45 | 65.37 | 58.10 | 87.45 | 74.47 | 61.12 | 83.27 | 65.05 | 94.31 | 58.95 | 75.68 | 83.29 | 63.12 | 63.12 | 74.04 |
| #66 | SwinUNETR | 62.50 | 78.55 | 70.31 | 82.70 | 81.35 | 80.02 | 71.71 | 70.20 | 61.53 | 87.18 | 70.36 | 61.59 | 83.74 | 64.15 | 91.16 | 70.49 | 83.96 | 86.17 | 67.33 | 73.89 | 74.01 |
| #67 | SCUNet++ | 65.48 | 80.70 | 73.90 | 81.66 | 82.49 | 82.91 | 72.35 | 75.17 | 34.54 | 85.46 | 79.68 | 59.11 | 81.20 | 66.55 | 94.41 | 60.15 | 82.98 | 80.10 | 67.33 | 67.33 | 73.72 |
| #68 | UNeXt | 62.11 | 80.42 | 71.13 | 82.41 | 81.00 | 75.64 | 75.64 | 65.51 | 46.52 | 86.29 | 76.30 | 60.97 | 82.83 | 65.74 | 94.76 | 59.27 | 77.43 | 84.83 | 60.17 | 60.17 | 73.41 |
| #69 | MERIT | 69.24 | 83.18 | 77.46 | 83.85 | 84.32 | 83.40 | 83.91 | 48.57 | 40.41 | 82.49 | 83.77 | 52.13 | 81.86 | 67.52 | 91.78 | 57.42 | 75.79 | 85.24 | 68.78 | 68.78 | 72.77 |
| #70 | MDSA-UNet | 67.63 | 81.63 | 76.44 | 83.02 | 82.19 | 79.28 | 77.47 | 53.47 | 37.81 | 85.63 | 78.23 | 58.66 | 81.63 | 66.92 | 94.64 | 57.73 | 78.88 | 84.68 | 65.07 | 65.07 | 72.66 |
| #71 | CPCANet | 66.01 | 81.01 | 72.75 | 82.13 | 83.38 | 81.49 | 73.42 | 70.88 | 20.77 | 85.48 | 68.68 | 69.88 | 81.07 | 64.93 | 93.73 | 57.00 | 81.86 | 81.88 | 63.30 | 63.30 | 72.66 |
| #72 | LeViT-UNet | 58.38 | 73.72 | 66.10 | 81.77 | 70.93 | 80.99 | 73.48 | 69.50 | 14.29 | 86.82 | 71.62 | 58.11 | 81.23 | 65.21 | 91.33 | 67.64 | 79.28 | 82.16 | 64.37 | 64.37 | 72.43 |
| #73 | DAEFormer | 63.39 | 79.42 | 73.07 | 82.17 | 80.93 | 80.09 | 73.48 | 36.43 | 26.97 | 78.92 | 82.65 | 52.12 | 77.50 | 70.09 | 94.17 | 50.33 | 84.18 | 86.88 | 70.84 | 70.84 | 72.37 |
| #74 | PraNet | 71.63 | 83.90 | 78.40 | 84.42 | 86.38 | 84.39 | 85.10 | 36.43 | 26.97 | 79.04 | 83.09 | 52.01 | 77.50 | 70.09 | 94.41 | 50.33 | 84.18 | 86.88 | 70.84 | 70.84 | 72.37 |
| #75 | CaraNet | 70.61 | 84.31 | 78.69 | 84.26 | 86.01 | 83.24 | 84.53 | 37.69 | 26.30 | 79.04 | 83.09 | 52.01 | 77.18 | 70.61 | 94.44 | 51.74 | 83.96 | 86.66 | 68.97 | 68.97 | 72.26 |
| #76 | TransFuse | 69.77 | 83.21 | 77.79 | 84.11 | 82.86 | 83.01 | 82.69 | 49.62 | 18.52 | 85.15 | 75.03 | 56.85 | 81.08 | 69.30 | 93.08 | 50.91 | 80.97 | 86.19 | 64.94 | 64.94 | 72.23 |
| #77 | MedT | 60.85 | 79.80 | 70.13 | 83.12 | 76.65 | 74.78 | 70.27 | 63.13 | 54.02 | 87.18 | 70.33 | 59.70 | 82.62 | 65.25 | 93.31 | 60.11 | 73.87 | 81.40 | 58.55 | 58.55 | 72.02 |
| #78 | EMCAD | 68.56 | 82.37 | 70.95 | 84.01 | 86.20 | 84.20 | 77.52 | 58.61 | 44.65 | 86.97 | 85.85 | 62.40 | 82.55 | 68.54 | 94.69 | 62.49 | 73.31 | 85.61 | 0.00 | 77.23 | 71.84 |
| #79 | UACANet | 69.88 | 84.07 | 78.83 | 83.76 | 83.88 | 84.81 | 84.65 | 36.30 | 25.48 | 78.16 | 79.86 | 51.53 | 76.54 | 69.67 | 94.49 | 49.19 | 82.52 | 86.43 | 69.28 | 69.28 | 71.62 |
| #80 | MultiResUNet | 65.58 | 82.71 | 75.20 | 82.47 | 82.21 | 80.57 | 77.38 | 30.58 | 19.22 | 84.69 | 80.93 | 62.97 | 82.38 | 68.28 | 92.49 | 52.49 | 81.98 | 85.24 | 64.45 | 64.45 | 71.39 |
| #81 | MUCM-Net | 61.25 | 79.47 | 68.59 | 81.99 | 82.57 | 74.14 | 66.91 | 57.12 | 41.63 | 84.87 | 73.10 | 58.08 | 79.58 | 64.00 | 93.63 | 58.29 | 73.44 | 80.99 | 71.47 | 71.47 | 70.19 |
| #82 | LFU-Net | 57.58 | 77.72 | 62.52 | 80.51 | 75.02 | 63.84 | 69.14 | 59.19 | 55.87 | 86.71 | 74.73 | 56.03 | 81.55 | 64.25 | 93.29 | 52.49 | 81.98 | 78.17 | 51.19 | 72.90 | 69.48 |
| #83 | MissFormer | 62.20 | 79.26 | 72.78 | 81.89 | 76.54 | 80.27 | 73.36 | 60.91 | 19.32 | 85.73 | 65.47 | 54.45 | 62.55 | 63.38 | 91.53 | 57.05 | 77.04 | 78.66 | 63.91 | 73.03 | 68.83 |
| #84 | UNETR | 52.28 | 71.40 | 48.34 | 80.87 | 80.45 | 72.01 | 62.20 | 65.47 | 61.05 | 87.52 | 61.49 | 58.35 | 83.17 | 58.54 | 91.61 | 58.15 | 71.60 | 81.25 | 54.86 | 74.12 | 68.74 |
| #85 | CFM-UNet | 58.88 | 78.27 | 72.51 | 81.41 | 86.04 | 76.53 | 69.41 | 47.18 | 14.58 | 84.85 | 70.27 | 55.18 | 86.00 | 60.00 | 92.84 | 53.96 | 66.22 | 84.07 | 50.85 | 74.49 | 67.67 |
| #86 | Zig-RiR | 54.17 | 75.18 | 66.18 | 81.75 | 79.06 | 74.98 | 71.85 | 56.12 | 35.55 | 81.71 | 68.28 | 46.33 | 80.35 | 63.39 | 94.85 | 59.95 | 54.66 | 71.47 | 48.79 | 77.76 | 67.63 |
| #87 | SimpleUNet | 55.08 | 75.77 | 62.93 | 79.52 | 77.94 | 69.60 | 73.54 | 62.60 | 54.68 | 85.98 | 77.27 | 62.98 | 79.75 | 65.58 | 94.08 | 55.08 | 80.70 | 76.57 | 62.85 | 0.03 | 67.63 |
| #88 | CFPNet-M | 65.22 | 81.46 | 75.09 | 82.91 | 81.64 | 79.22 | 77.20 | 67.89 | 58.15 | 87.03 | 79.05 | 63.39 | 84.46 | 60.78 | 94.49 | 60.65 | 82.65 | 0.00 | 64.19 | 0.00 | 67.50 |
| #89 | BEFUnet | 61.91 | 77.66 | 65.36 | 79.88 | 82.59 | 76.88 | 69.54 | 54.65 | 15.83 | 84.46 | 60.78 | 49.14 | 79.57 | 60.60 | 90.36 | 53.78 | 75.57 | 77.30 | 57.19 | 62.79 | 66.61 |
| #90 | UNetV2 | 58.12 | 78.20 | 70.15 | 81.49 | 81.37 | 79.25 | 79.70 | 40.72 | 15.17 | 84.52 | 67.10 | 54.72 | 80.56 | 62.63 | 91.43 | 50.68 | 73.87 | 81.40 | 40.58 | 67.34 | 66.39 |
| #91 | Swin-umambaD | 64.94 | 80.15 | 75.19 | 81.93 | 70.71 | 84.27 | 75.72 | 50.01 | 14.24 | 87.55 | 51.22 | 55.18 | 84.65 | 66.46 | 94.39 | 59.19 | 85.45 | 47.58 | 55.09 | 67.86 | 66.30 |
| #92 | VMUNetV2 | 61.69 | 80.54 | 74.85 | 82.91 | 78.35 | 84.03 | 61.46 | 46.22 | 13.02 | 82.92 | 79.07 | 51.24 | 62.69 | 66.01 | 90.97 | 47.05 | 82.07 | 79.86 | 65.87 | 6.44 | 64.86 |
| #93 | VMUNet | 67.38 | 45.08 | 75.31 | 83.01 | 83.08 | 84.90 | 75.42 | 45.94 | 11.85 | 83.88 | 52.81 | 64.00 | 9.39 | 66.93 | 93.87 | 53.84 | 80.97 | 47.79 | 68.52 | 68.86 | 64.15 |
| #94 | UltraLight-VM-UNet | 57.58 | 72.38 | 59.18 | 80.45 | 80.96 | 63.17 | 60.58 | 50.32 | 14.64 | 85.99 | 60.37 | 53.99 | 78.05 | 60.11 | 89.39 | 50.84 | 72.64 | 77.57 | 50.81 | 68.76 | 61.49 |
| #95 | Polyp-PVT | 70.54 | 82.16 | 76.68 | 83.33 | 84.71 | 83.95 | 80.64 | 35.87 | 1.28 | 81.59 | 51.22 | 46.40 | 44.22 | 67.43 | 94.24 | 0.06 | 84.52 | 83.80 | 68.56 | 60.33 | 64.08 |
| #96 | H-vmunet | 61.83 | 77.23 | 66.49 | 81.66 | 79.14 | 74.22 | 62.31 | 46.58 | 10.85 | 85.58 | 67.11 | 53.34 | 77.57 | 61.19 | 91.57 | 4.64 | 70.28 | 82.58 | 54.32 | 65.49 | 63.43 |
| #97 | CSWin-UNet | 56.38 | 78.28 | 69.71 | 81.67 | 81.18 | 75.09 | 70.03 | 27.08 | 14.59 | 85.60 | 63.12 | 50.60 | 79.19 | 52.82 | 88.68 | 7.24 | 73.93 | 78.19 | 56.83 | 64.89 | 63.25 |
| #98 | MALUNet | 59.10 | 74.33 | 68.27 | 80.99 | 77.25 | 72.56 | 56.44 | 11.87 | 15.08 | 84.83 | 61.99 | 54.83 | 78.91 | 59.74 | 90.18 | 51.78 | 63.01 | 47.01 | 51.51 | 64.54 | 61.24 |
| #99 | SwinUnet | 49.81 | 69.94 | 27.35 | 81.18 | 79.64 | 67.11 | 67.26 | 49.19 | 10.66 | 85.52 | 57.82 | 49.81 | 64.08 | 60.35 | 90.07 | 45.24 | 70.62 | 47.52 | 50.08 | 63.49 | 59.34 |
| #100 | MambaUnet | 19.01 | 80.34 | 50.64 | 79.22 | 42.31 | 84.28 | 60.88 | 53.76 | 11.99 | 86.53 | 52.52 | 64.15 | 0.67 | 65.20 | 56.49 | 59.51 | 84.62 | 47.62 | 68.88 | 61.96 | 56.53 |

Table 22: Average performance of 100 u-shape medical image segmentation networks with U-Score. Baseline U-Net is highlighted (gray background), and statistical significance (calculated by IoU) of p-value is highlighted: p<0.0001 , p<0.001 , p<0.05 , p≤0.05 , and P > 0.05 (Not significant) .

| Rank | Network | Ultrasound BUSI | BUSBRA | TNSCUI | Dermoscopy ISIC2018 | SkinCancer | Endoscopy Kvasir | Robotool | Fundus CHASE | DRIVE | Histopathology DSB2018 | Glas | Monusac | Nuclear Cell | X-Ray Covidquex | Montgomery | DCA | MRI ACDC | Promise | CT Synapse | OCT Cystoidfluid | Avg |
|---|---|---|---|---|---|---|---|---|---|---|---|---|---|---|---|---|---|---|---|---|---|---|
| 1 | LGMSNet | 86.60 | 85.79 | 88.20 | 83.21 | 89.72 | 77.54 | 69.33 | 86.33 | 88.65 | 90.80 | 88.31 | 84.05 | 89.78 | 80.45 | 87.70 | 86.53 | 84.55 | 86.55 | 83.36 | 82.59 | 84.49 |
| 2 | LV-UNet | 78.85 | 82.62 | 85.36 | 87.93 | 83.00 | 84.97 | 83.02 | 68.19 | 79.63 | 71.97 | 87.64 | 82.50 | 82.15 | 91.70 | 79.68 | 90.90 | 78.00 | 86.67 | 69.73 | 84.68 | 81.96 |
| 3 | CMUNeXt | 81.96 | 86.88 | 84.44 | 64.46 | 88.85 | 65.67 | 65.44 | 85.35 | 87.62 | 82.70 | 86.90 | 83.98 | 85.96 | 78.45 | 85.68 | 86.61 | 83.05 | 82.44 | 77.97 | 82.98 | 81.37 |
| 4 | MBSNet | 81.21 | 85.22 | 81.57 | 70.00 | 67.66 | 73.03 | 66.41 | 83.10 | 86.64 | 83.32 | 87.32 | 87.02 | 84.37 | 80.48 | 86.60 | 86.31 | 85.46 | 83.19 | 81.24 | 83.02 | 81.16 |
| 5 | Tinyunet | 45.28 | 81.90 | 81.53 | 47.59 | 54.63 | 37.62 | 54.37 | 82.24 | 96.30 | 77.54 | 90.03 | 70.35 | 91.56 | 73.91 | 90.27 | 83.80 | 84.65 | 81.65 | 83.25 | 90.12 | 74.93 |
| 6 | Mobile U-ViT | 77.81 | 76.35 | 77.00 | 74.58 | 67.76 | 73.98 | 67.44 | 74.46 | 76.66 | 73.54 | 70.83 | 77.17 | 75.13 | 72.49 | 77.81 | 77.60 | 75.20 | 71.62 | 75.33 | 72.18 | 74.25 |
| 7 | U-RWKV | 67.91 | 72.99 | 62.42 | 60.99 | 39.91 | 48.14 | 51.99 | 82.99 | 85.32 | 86.03 | 76.72 | 84.80 | 84.96 | 63.88 | 80.96 | 86.37 | 73.79 | 81.88 | 65.36 | 82.54 | 72.00 |
| 8 | U-KAN | 66.86 | 71.29 | 73.84 | 56.88 | 75.10 | 69.02 | 58.72 | 72.17 | 74.12 | 61.24 | 71.74 | 67.77 | 72.92 | 61.75 | 70.32 | 72.14 | 70.77 | 77.30 | 72.83 | 76.59 | 69.67 |
| 9 | DCSAU-Net | 66.85 | 72.47 | 74.96 | 68.11 | 46.94 | 63.17 | 58.03 | 64.49 | 74.72 | 76.06 | 65.18 | 70.21 | 73.28 | 70.64 | 69.12 | 73.92 | 70.36 | 70.67 | 63.66 | 70.13 | 68.15 |
| 10 | ULite | 62.66 | 65.19 | 52.52 | 52.91 | 74.11 | 18.95 | 41.97 | 70.39 | 95.19 | 89.40 | 72.50 | 72.85 | 90.39 | 50.16 | 77.77 | 78.81 | 39.76 | 74.12 | 64.52 | 92.51 | 66.83 |
| 11 | UNeXt | 46.75 | 61.09 | 55.94 | 58.30 | 46.96 | 11.45 | 58.13 | 70.75 | 79.78 | 65.60 | 79.51 | 71.93 | 86.46 | 60.55 | 85.18 | 80.66 | 56.34 | 86.75 | 46.86 | 88.44 | 64.87 |
| 12 | CFPNet-M | 63.23 | 66.18 | 72.40 | 64.41 | 52.24 | 55.62 | 60.18 | 66.76 | 80.90 | 71.09 | 76.17 | 72.31 | 77.07 | 65.61 | 70.20 | 75.70 | 77.34 | 9.36 | 61.88 | 9.36 | 62.40 |
| 13 | RWKV-UNet | 61.72 | 61.72 | 61.72 | 61.72 | 61.72 | 61.72 | 61.72 | 54.56 | 60.59 | 61.51 | 61.72 | 61.72 | 61.72 | 61.72 | 61.72 | 61.72 | 61.72 | 61.72 | 61.72 | 61.14 | 61.27 |
| 14 | MDSA-UNet | 68.43 | 62.18 | 71.46 | 62.01 | 48.81 | 52.72 | 57.40 | 30.24 | 54.31 | 41.12 | 68.49 | 48.22 | 60.75 | 61.48 | 66.64 | 58.63 | 56.33 | 68.16 | 59.17 | 59.39 | 57.84 |
| 15 | DDANet | 56.79 | 58.95 | 59.28 | 52.96 | 35.00 | 58.05 | 54.34 | 59.52 | 60.36 | 60.77 | 60.77 | 60.77 | 59.88 | 56.60 | 60.77 | 60.77 | 60.77 | 59.28 | 60.37 | 58.60 | 57.73 |
| 16 | ResU-KAN | 56.11 | 59.24 | 60.59 | 55.84 | 46.81 | 58.61 | 49.07 | 59.25 | 60.38 | 53.31 | 58.34 | 57.81 | 60.63 | 55.76 | 56.75 | 60.35 | 56.99 | 61.21 | 59.16 | 62.21 | 57.42 |
| 17 | UTNet | 58.08 | 57.02 | 58.05 | 55.41 | 44.94 | 50.90 | 52.14 | 59.49 | 59.67 | 59.67 | 57.04 | 59.67 | 59.60 | 55.61 | 59.08 | 59.67 | 58.58 | 57.85 | 56.11 | 58.52 | 56.85 |
| 18 | CE-Net | 58.93 | 59.33 | 59.54 | 59.90 | 53.74 | 54.81 | 59.90 | 52.61 | 53.68 | 46.56 | 58.08 | 51.78 | 56.06 | 59.90 | 56.39 | 58.13 | 56.75 | 57.87 | 48.69 | 57.72 | 56.01 |
| 19 | TA-Net | 57.20 | 58.27 | 58.27 | 58.27 | 51.03 | 56.36 | 58.27 | 52.49 | 52.95 | 50.96 | 56.94 | 54.56 | 55.90 | 58.27 | 52.86 | 57.18 | 55.11 | 57.30 | 45.15 | 55.82 | 55.16 |
| 20 | SwinUNETR | 44.82 | 23.25 | 43.28 | 57.72 | 47.03 | 59.57 | 24.69 | 67.66 | 79.66 | 70.15 | 46.66 | 63.75 | 76.58 | 31.48 | 64.59 | 74.15 | 19.26 | 66.82 | 48.36 | 73.38 | 54.14 |
| 21 | TransResUNet | 51.29 | 51.13 | 50.84 | 51.29 | 49.95 | 49.89 | 51.29 | 51.23 | 50.59 | 46.42 | 50.80 | 51.24 | 49.84 | 51.29 | 51.29 | 51.27 | 50.16 | 51.29 | 49.12 | 50.47 | 50.52 |
| 22 | MEGANet | 50.68 | 50.68 | 50.68 | 50.68 | 50.35 | 50.68 | 50.68 | 47.14 | 47.58 | 47.22 | 50.17 | 49.62 | 50.44 | 50.68 | 50.68 | 50.64 | 49.80 | 50.68 | 50.68 | 49.38 | 49.98 |
| 23 | G-CASCADE | 55.18 | 50.86 | 54.54 | 52.17 | 53.69 | 54.97 | 47.41 | 31.28 | 41.82 | 49.10 | 55.99 | 44.42 | 46.18 | 51.48 | 49.99 | 51.00 | 56.29 | 49.38 | 56.29 | 47.19 | 49.96 |
| 24 | MultiResUNet | 54.42 | 62.15 | 59.75 | 46.64 | 50.21 | 54.76 | 51.51 | 9.10 | 14.90 | 9.10 | 64.61 | 59.08 | 58.68 | 62.00 | 30.04 | 20.35 | 61.06 | 62.39 | 52.87 | 56.12 | 46.99 |
| 25 | EMCAD | 50.79 | 49.06 | 36.93 | 53.27 | 53.54 | 53.41 | 43.37 | 53.65 | 64.34 | 47.44 | 53.54 | 47.25 | 48.53 | 50.90 | 48.62 | 52.59 | 9.01 | 51.23 | 8.80 | 47.80 | 44.37 |
| 26 | MedFormer | 43.08 | 43.96 | 46.61 | 41.60 | 37.91 | 43.92 | 39.29 | 46.80 | 47.06 | 45.49 | 43.63 | 46.88 | 45.66 | 42.06 | 44.06 | 46.42 | 45.73 | 46.77 | 43.11 | 47.22 | 44.36 |
| 27 | HiFormer | 43.32 | 46.40 | 46.45 | 46.30 | 46.25 | 45.95 | 45.83 | 39.93 | 42.43 | 31.62 | 45.03 | 41.87 | 42.37 | 44.76 | 44.80 | 43.50 | 45.61 | 45.94 | 44.93 | 43.05 | 43.82 |
| 28 | SimpleUNet | 9.52 | 9.52 | 9.52 | 9.52 | 9.52 | 9.52 | 42.70 | 63.30 | 91.07 | 57.53 | 82.95 | 83.42 | 48.75 | 58.36 | 73.58 | 50.40 | 79.71 | 9.52 | 63.09 | 9.52 | 43.53 |
| 29 | CASCADE | 47.32 | 44.00 | 46.37 | 44.44 | 46.96 | 47.32 | 43.32 | 35.67 | 34.42 | 37.14 | 47.32 | 33.45 | 49.84 | 44.54 | 42.02 | 45.33 | 46.99 | 46.72 | 47.32 | 42.12 | 43.09 |
| 30 | DC-UNet | 28.87 | 39.02 | 43.88 | 8.79 | 8.79 | 26.53 | 40.05 | 45.90 | 51.91 | 42.40 | 58.83 | 50.81 | 49.32 | 51.35 | 50.51 | 42.02 | 52.03 | 47.84 | 47.38 | 35.90 | 40.76 |
| 31 | MMUNet | 41.80 | 43.76 | 44.42 | 40.31 | 29.06 | 43.28 | 36.64 | 43.51 | 44.24 | 39.00 | 37.52 | 39.20 | 42.76 | 39.10 | 38.25 | 42.02 | 39.91 | 42.47 | 39.99 | 45.18 | 40.62 |
| 32 | MUCM-Net | 37.64 | 44.13 | 29.61 | 41.94 | 69.92 | 9.50 | 9.50 | 46.09 | 70.37 | 17.77 | 65.37 | 50.99 | 45.25 | 9.50 | 63.43 | 73.16 | 11.65 | 48.34 | 9.50 | 56.28 | 40.50 |
| 33 | ERDUnet | 45.10 | 45.34 | 47.42 | 37.87 | 10.27 | 15.87 | 29.01 | 42.01 | 48.23 | 47.47 | 39.68 | 47.92 | 47.36 | 41.26 | 38.96 | 47.88 | 42.08 | 45.80 | 40.87 | 47.07 | 40.37 |
| 34 | TransFuse | 51.82 | 51.09 | 52.06 | 52.55 | 44.95 | 50.69 | 50.90 | 14.12 | 12.08 | 23.48 | 44.83 | 20.09 | 42.19 | 52.07 | 35.46 | 8.78 | 47.12 | 51.70 | 44.40 | 32.95 | 39.67 |
| 35 | U-Net++ | 37.94 | 38.71 | 40.79 | 34.03 | 37.76 | 39.08 | 37.55 | 41.69 | 41.27 | 38.83 | 40.76 | 40.36 | 40.76 | 36.48 | 40.76 | 40.56 | 40.84 | 41.58 | 38.97 | 41.71 | 39.52 |
| 36 | AC-MambaSeg | 36.31 | 37.59 | 39.01 | 37.05 | 37.17 | 37.52 | 31.33 | 40.25 | 39.21 | 33.41 | 36.23 | 37.47 | 37.19 | 38.56 | 36.69 | 38.44 | 36.72 | 40.32 | 39.68 | 40.75 | 37.54 |
| 37 | H2Former | 37.89 | 37.76 | 38.03 | 35.65 | 34.46 | 35.98 | 33.92 | 36.87 | 37.66 | 33.53 | 38.07 | 37.63 | 37.25 | 35.23 | 38.34 | 37.72 | 36.99 | 37.96 | 37.83 | 38.42 | 36.93 |
| 38 | CSCAUNet | 37.08 | 38.27 | 39.32 | 34.08 | 33.37 | 37.82 | 34.70 | 36.57 | 26.63 | 36.08 | 39.33 | 36.18 | 37.90 | 38.48 | 38.37 | 37.09 | 37.91 | 39.36 | 39.43 | 39.32 | 36.87 |
| 39 | Polyp-PVT | 59.01 | 52.95 | 56.26 | 52.98 | 57.09 | 58.47 | 53.52 | 8.94 | 8.94 | 8.94 | 8.94 | 8.94 | 8.94 | 51.84 | 50.03 | 8.94 | 58.75 | 50.92 | 54.83 | 8.94 | 36.41 |
| 40 | LFU-Net | 9.52 | 9.52 | 9.52 | 9.52 | 9.52 | 9.52 | 9.50 | 46.00 | 9.50 | 75.41 | 73.55 | 33.05 | 73.33 | 36.16 | 50.12 | 65.53 | 9.52 | 9.52 | 9.52 | 64.18 | 35.63 |
| 41 | LeViT-UNet | 8.87 | 8.87 | 8.87 | 16.62 | 40.89 | 8.87 | 41.26 | 50.54 | 47.97 | 39.30 | 43.31 | 40.44 | 48.14 | 15.43 | 51.52 | 45.56 | 45.61 | 42.28 | 45.71 | 52.94 | 35.15 |
| 42 | MSLAU-Net | 35.37 | 35.00 | 36.43 | 34.93 | 36.58 | 36.23 | 35.76 | 33.66 | 34.15 | 28.80 | 36.58 | 33.90 | 34.40 | 35.01 | 34.56 | 35.79 | 36.36 | 34.66 | 36.54 | 36.19 | 35.05 |
| 43 | FAT-Net | 35.87 | 36.12 | 36.12 | 35.50 | 34.54 | 35.43 | 35.99 | 34.85 | 33.94 | 29.13 | 36.03 | 34.96 | 34.09 | 35.93 | 35.39 | 36.05 | 36.12 | 35.46 | 31.52 | 35.20 | 34.91 |
| 44 | ESKNet | 34.67 | 34.51 | 34.59 | 32.47 | 26.68 | 32.90 | 33.10 | 34.67 | 34.67 | 34.67 | 34.48 | 34.67 | 34.54 | 33.84 | 34.64 | 34.29 | 34.28 | 34.14 | 31.96 | 33.84 | 33.81 |
| 45 | MedVKAN | 32.82 | 35.55 | 34.88 | 31.92 | 31.82 | 33.51 | 31.60 | 32.04 | 34.84 | 33.47 | 33.62 | 34.88 | 34.51 | 34.24 | 34.02 | 34.71 | 33.74 | 35.51 | 32.49 | 34.31 | 33.72 |
| 46 | Swin-umambaD | 47.87 | 41.34 | 54.28 | 31.67 | 8.96 | 40.57 | 29.28 | 16.16 | 8.96 | 57.03 | 8.96 | 38.97 | 8.96 | 47.96 | 52.11 | 52.56 | 60.39 | 8.96 | 39.10 | 8.96 | 33.62 |
| 47 | AURA-Net | 32.24 | 32.24 | 32.19 | 32.24 | 32.24 | 31.78 | 32.24 | 32.07 | 31.74 | 32.24 | 32.16 | 32.10 | 32.24 | 32.24 | 31.99 | 32.23 | 32.05 | 32.11 | 32.24 | 31.44 | 32.10 |
| 48 | SCUNet++ | 24.80 | 32.31 | 34.87 | 19.30 | 33.52 | 37.75 | 21.52 | 37.51 | 30.51 | 25.12 | 37.54 | 30.89 | 33.28 | 33.68 | 35.47 | 36.62 | 37.76 | 36.28 | 34.79 | 31.83 | 31.83 |
| 49 | MambaUnet | 9.21 | 49.43 | 9.21 | 9.21 | 9.21 | 73.88 | 9.21 | 30.68 | 9.21 | 56.94 | 9.21 | 67.51 | 9.21 | 44.29 | 9.21 | 63.62 | 73.55 | 9.21 | 67.96 | 9.21 | 31.46 |
| 50 | CA-Net | 31.32 | 31.09 | 31.59 | 28.58 | 28.25 | 28.97 | 28.24 | 30.82 | 32.06 | 32.08 | 31.71 | 32.11 | 32.11 | 31.19 | 31.79 | 32.00 | 32.07 | 31.07 | 31.43 | 31.09 | 30.98 |
| 51 | DAEFormer | 30.49 | 26.03 | 33.73 | 27.99 | 26.64 | 35.25 | 25.48 | 35.76 | 26.17 | 35.15 | 30.94 | 28.73 | 33.38 | 28.97 | 25.47 | 33.89 | 36.77 | 27.30 | 34.41 | 34.61 | 30.36 |
| 52 | RollingUnet | 29.90 | 30.96 | 30.27 | 28.73 | 23.17 | 29.38 | 29.56 | 30.97 | 31.34 | 31.38 | 31.24 | 31.23 | 31.38 | 30.72 | 31.38 | 31.31 | 30.99 | 30.75 | 30.72 | 30.80 | 30.31 |
| 53 | VMUNet | 40.60 | 8.49 | 40.65 | 38.45 | 38.91 | 43.80 | 34.87 | 35.68 | 8.49 | 32.09 | 8.49 | 41.33 | 8.49 | 38.42 | 36.78 | 25.66 | 43.08 | 8.49 | 41.36 | 27.48 | 30.08 |
| 54 | VMUNetV2 | 34.11 | 46.07 | 55.66 | 51.36 | 9.02 | 62.66 | 9.02 | 9.02 | 9.02 | 9.02 | 58.56 | 9.02 | 9.02 | 46.74 | 9.02 | 9.02 | 57.95 | 27.31 | 53.51 | 9.02 | 29.21 |
| 55 | DDS-UNet | 28.01 | 28.37 | 28.38 | 28.50 | 27.72 | 27.88 | 26.52 | 28.27 | 28.45 | 25.77 | 28.54 | 27.58 | 28.50 | 27.45 | 28.48 | 28.83 | 27.16 | 28.74 | 27.95 | 29.15 | 28.01 |
| 56 | MissFormer | 31.14 | 27.98 | 39.07 | 26.82 | 8.61 | 39.96 | 10.76 | 35.52 | 13.88 | 32.19 | 17.05 | 12.93 | 8.61 | 14.53 | 8.61 | 38.24 | 33.38 | 12.69 | 38.96 | 37.71 | 24.43 |
| 57 | Swin-umamba | 22.51 | 22.44 | 22.64 | 22.64 | 21.89 | 22.64 | 22.54 | 22.64 | 22.64 | 22.39 | 22.01 | 22.00 | 22.64 | 22.57 | 22.44 | 22.52 | 22.58 | 22.46 | 22.56 | 22.64 | 22.47 |
| 58 | DoubleUNet | 23.37 | 23.30 | 23.60 | 7.39 | 7.39 | 24.74 | 21.05 | 24.55 | 24.66 | 24.81 | 23.56 | 24.48 | 20.84 | 23.40 | 24.54 | 23.96 | 24.81 | 19.74 | 24.81 | 22.88 | 21.89 |
| 59 | ColonSegNet | 19.46 | 13.13 | 19.77 | 18.34 | 20.63 | 21.21 | 21.55 | 23.57 | 23.57 | 23.47 | 23.18 | 23.31 | 23.57 | 21.64 | 23.42 | 23.40 | 23.27 | 22.27 | 22.74 | 23.57 | 21.75 |
| 60 | ResULNet++ | 18.15 | 19.59 | 20.45 | 18.82 | 20.53 | 18.60 | 18.86 | 22.27 | 22.31 | 21.46 | 21.11 | 21.58 | 22.07 | 20.63 | 21.61 | 21.31 | 21.45 | 22.22 | 19.56 | 22.33 | 20.75 |
| 61 | TransAttUnet | 21.27 | 21.49 | 21.60 | 20.40 | 7.09 | 20.80 | 20.78 | 21.32 | 21.76 | 21.50 | 21.76 | 21.76 | 21.76 | 21.64 | 21.76 | 21.72 | 21.57 | 21.35 | 21.44 | 21.42 | 20.71 |
| 62 | GH-UNet | 20.09 | 20.84 | 20.92 | 18.90 | 21.15 | 20.29 | 19.51 | 20.30 | 21.06 | 18.48 | 21.16 | 20.45 | 20.71 | 20.09 | 20.75 | 21.08 | 20.97 | 21.11 | 21.15 | 21.21 | 20.51 |
| 63 | MADGNet | 20.48 | 20.48 | 20.48 | 20.48 | 20.21 | 20.48 | 20.48 | 19.66 | 20.06 | 19.46 | 20.48 | 20.07 | 20.37 | 20.46 | 20.39 | 20.48 | 20.39 | 20.48 | 20.05 | 20.36 | 20.31 |
| 64 | AttU-Net | 19.28 | 20.29 | 20.14 | 18.49 | 18.07 | 18.30 | 19.97 | 20.60 | 20.59 | 20.59 | 20.65 | 20.57 | 20.79 | 20.35 | 20.53 | 20.57 | 20.60 | 20.55 | 19.97 | 20.45 | 20.07 |
| 65 | U-Net | 19.25 | 20.24 | 20.11 | 18.94 | 16.59 | 19.01 | 20.05 | 20.65 | 20.62 | 20.59 | 20.65 | 20.57 | 20.79 | 20.23 | 20.65 | 20.65 | 20.56 | 20.50 | 19.97 | 20.33 | 20.05 |
| 66 | CENet | 19.96 | 19.80 | 20.03 | 20.02 | 19.74 | 20.23 | 19.39 | 19.39 | 20.04 | 20.18 | 20.23 | 19.71 | 19.92 | 19.43 | 19.28 | 19.92 | 20.23 | 19.08 | 20.23 | 20.11 | 19.85 |
| 67 | ScribFormer | 19.66 | 19.85 | 20.35 | 19.66 | 13.08 | 18.50 | 17.88 | 20.23 | 20.66 | 20.62 | 19.96 | 20.50 | 20.13 | 20.08 | 20.62 | 20.56 | 19.48 | 19.51 | 19.54 | 19.58 | 19.40 |
| 68 | TransUnet | 18.15 | 19.32 | 19.58 | 19.18 | 19.54 | 18.66 | 18.43 | 19.65 | 19.61 | 19.12 | 19.28 | 19.45 | 19.63 | 19.56 | 19.65 | 19.65 | 19.48 | 19.51 | 19.54 | 19.58 | 19.40 |
| 69 | UNet3+ | 18.17 | 19.29 | 19.07 | 18.14 | 17.47 | 17.20 | 19.10 | 19.65 | 19.65 | 19.57 | 19.63 | 19.57 | 19.57 | 19.34 | 19.65 | 19.65 | 19.57 | 19.61 | 19.44 | 19.65 | 19.15 |
| 70 | UNetV2 | 18.33 | 14.01 | 34.44 | 16.29 | 37.92 | 42.15 | 13.31 | 8.87 | 8.87 | 8.87 | 26.46 | 16.15 | 41.48 | 8.87 | 8.87 | 8.87 | 8.87 | 33.98 | 8.87 | 25.95 | 19.10 |
| 71 | DA-TransUNet | 18.84 | 18.83 | 18.76 | 18.70 | 18.84 | 18.33 | 18.64 | 18.70 | 18.66 | 18.45 | 18.82 | 18.42 | 18.84 | 18.82 | 18.84 | 18.84 | 18.77 | 18.60 | 18.31 | 18.60 | 18.69 |
| 72 | UTANet | 18.33 | 18.78 | 18.59 | 18.37 | 18.30 | 18.33 | 18.59 | 18.52 | 18.53 | 18.37 | 18.49 | 18.59 | 18.49 | 18.54 | 18.59 | 18.58 | 18.31 | 18.59 | 18.50 |
| 73 | MedT | 16.28 | 19.47 | 18.84 | 19.00 | 7.32 | 7.32 | 8.00 | 21.22 | 23.44 | 23.00 | 19.79 | 21.13 | 22.99 | 19.84 | 20.38 | 23.05 | 11.40 | 19.94 | 17.06 | 22.26 | 18.09 |
| 74 | CMU-Net | 17.30 | 17.35 | 17.33 | 16.19 | 15.79 | 17.27 | 16.90 | 17.50 | 17.50 | 17.37 | 17.40 | 17.44 | 17.41 | 17.12 | 17.37 | 17.41 | 17.43 | 17.50 | 17.08 | 17.42 | 17.20 |
| 75 | Zig-RiR | 8.20 | 8.20 | 8.20 | 20.73 | 8.20 | 8.70 | 18.82 | 24.92 | 29.79 | 8.20 | 24.18 | 8.20 | 25.65 | 21.79 | 8.20 | 34.05 | 8.20 | 32.21 | 17.16 |
| 76 | UltraLight-VM-UNet | 9.42 | 9.42 | 9.42 | 9.42 | 43.98 | 9.42 | 9.42 | 19.12 | 9.42 | 54.15 | 9.42 | 28.61 | 14.28 | 9.42 | 9.42 | 9.42 | 9.42 | 9.42 | 39.94 | 16.60 |
| 77 | TransNorm | 17.18 | 17.12 | 17.24 | 16.96 | 17.34 | 16.64 | 16.58 | 17.13 | 17.16 | 16.21 | 16.66 | 16.83 | 17.14 | 16.86 | 17.03 | 17.16 | 10.51 | 17.25 | 15.41 | 17.23 | 16.58 |
| 78 | CPCANet | 17.62 | 17.21 | 17.16 | 15.48 | 17.84 | 17.88 | 14.77 | 17.90 | 11.06 | 14.80 | 15.74 | 15.77 | 17.08 | 15.51 | 17.04 | 16.98 | 18.03 | 17.48 | 17.97 | 18.11 | 16.57 |
| 79 | EViT-UNet | 16.17 | 16.18 | 16.18 | 16.11 | 17.33 | 16.57 | 16.13 | 16.18 | 15.85 | 16.08 | 15.80 | 15.99 | 16.16 | 16.18 | 15.94 | 17.05 | 7.53 | 24.54 | 19.95 | 21.37 | 15.99 |
| 80 | CFM-UNet | 7.53 | 11.76 | 23.36 | 17.63 | 18.89 | 15.45 | 7.53 | 7.53 | 11.51 | 21.35 | 14.33 | 22.65 | 22.11 | 20.23 | 17.05 | 7.53 | 24.54 | 19.95 | 15.67 |
| 81 | MSRFNet | 15.31 | 15.93 | 15.98 | 15.15 | 13.42 | 15.29 | 15.24 | 15.47 | 15.96 | 16.06 | 15.97 | 16.05 | 15.98 | 15.60 | 16.00 | 16.11 | 16.11 | 15.94 | 15.94 | 15.92 | 15.67 |
| 82 | BEFUnet | 28.54 | 8.49 | 8.49 | 8.49 | 37.47 | 23.09 | 8.49 | 17.70 | 8.49 | 8.49 | 8.49 | 8.49 | 28.87 | 8.49 | 8.49 | 25.45 | 25.84 | 8.49 | 18.20 | 8.49 | 15.35 |
| 83 | UCTransNet | 14.85 | 14.83 | 14.78 | 14.54 | 14.39 | 14.34 | 14.83 | 15.18 | 14.89 | 15.01 | 15.05 | 15.13 | 14.89 | 15.11 | 15.05 | 15.10 | 15.03 | 14.96 | 15.09 | 14.92 |
| 84 | CaraNet | 18.49 | 18.50 | 18.50 | 18.50 | 18.50 | 18.28 | 18.49 | 6.71 | 14.25 | 6.71 | 18.50 | 6.71 | 6.71 | 18.50 | 17.68 | 8.06 | 18.37 | 18.45 | 18.11 | 6.71 | 14.74 |
| 85 | FCBFormer | 14.36 | 14.61 | 14.58 | 14.37 | 14.61 | 14.61 | 13.75 | 14.61 | 14.61 | 14.60 | 14.41 | 14.56 | 14.61 | 14.17 | 14.37 | 14.46 | 14.61 | 14.61 | 14.60 | 14.61 | 14.49 |
| 86 | D-TrAttUnet | 13.53 | 13.82 | 13.89 | 13.35 | 13.93 | 13.65 | 13.67 | 14.19 | 14.25 | 14.25 | 14.03 | 14.25 | 14.20 | 13.67 | 14.04 | 14.09 | 14.12 | 14.13 | 14.25 | 13.97 |
| 87 | PraNet | 17.33 | 17.33 | 17.33 | 17.33 | 17.33 | 17.33 | 13.14 | 6.55 | 13.77 | 6.55 | 17.30 | 6.55 | 6.55 | 17.33 | 16.40 | 6.55 | 17.26 | 17.33 | 17.21 | 6.55 | 13.86 |
| 88 | MCA-UNet | 13.30 | 13.36 | 13.37 | 13.23 | 13.39 | 13.14 | 13.00 | 13.22 | 13.31 | 13.20 | 13.22 | 13.25 | 13.30 | 13.02 | 13.23 | 13.30 | 10.82 | 13.37 | 13.13 |
| 89 | MALUNet | 11.18 | 9.50 | 25.67 | 9.50 | 9.50 | 9.50 | 9.50 | 16.69 | 9.50 | 19.76 | 33.35 | 9.50 | 9.50 | 13.00 | 9.50 | 9.50 | 9.50 | 9.92 | 12.65 |
| 90 | UNETR | 7.08 | 7.08 | 7.08 | 7.08 | 15.58 | 7.08 | 7.08 | 19.80 | 21.51 | 21.11 | 7.08 | 18.22 | 21.05 | 7.08 | 7.08 | 20.05 | 7.08 | 18.04 | 7.08 | 19.72 | 12.64 |
| 91 | CSWin-UNet | 7.91 | 12.87 | 22.18 | 17.55 | 24.16 | 7.91 | 7.91 | 7.91 | 7.91 | 23.13 | 7.91 | 7.91 | 23.03 | 7.91 | 13.31 | 7.91 | 14.18 | 10.23 | 12.29 |
| 92 | Perspective-Unet | 12.04 | 12.30 | 12.34 | 11.72 | 12.20 | 12.20 | 12.01 | 12.36 | 12.32 | 12.12 | 12.29 | 12.21 | 12.15 | 12.03 | 12.22 | 12.29 | 12.29 | 12.36 | 12.36 | 12.36 | 12.26 |
| 93 | ConvFormer | 12.19 | 12.21 | 12.25 | 11.88 | 11.69 | 11.87 | 12.02 | 11.64 | 11.67 | 5.70 | 12.36 | 11.28 | 11.95 | 12.38 | 12.39 | 12.16 | 12.34 | 12.32 | 11.81 | 12.04 | 11.71 |
| 94 | MERIT | 12.46 | 12.45 | 12.49 | 12.46 | 12.40 | 12.47 | 12.47 | 6.20 | 11.91 | 12.36 | 12.21 | 5.73 | 11.88 | 5.73 | 12.38 | 12.37 | 11.71 | 10.99 |
| 95 | DS-TransUNet | 10.13 | 10.39 | 10.35 | 10.56 | 10.60 | 10.78 | 10.72 | 10.62 | 10.70 | 10.74 | 10.75 | 10.64 | 10.66 | 10.51 | 10.49 | 10.70 | 10.37 | 10.80 | 10.64 | 10.59 |
| 96 | UACANet | 12.60 | 12.65 | 12.65 | 12.52 | 12.40 | 12.65 | 12.65 | 5.74 | 10.33 | 5.74 | 12.49 | 5.74 | 5.74 | 12.65 | 12.28 | 5.74 | 12.47 | 12.60 | 12.49 | 5.74 | 10.39 |
| 97 | CFFormer | 10.13 | 9.97 | 10.13 | 10.13 | 9.97 | 10.13 | 10.10 | 10.05 | 10.10 | 10.03 | 10.11 | 10.02 | 10.12 | 10.02 | 10.13 | 10.08 | 10.10 | 10.11 | 10.09 | 10.12 | 10.08 |
| 98 | H-vmunet | 18.06 | 7.26 | 7.26 | 14.53 | 7.26 | 7.26 | 7.26 | 7.26 | 7.26 | 18.26 | 15.94 | 7.26 | 7.26 | 7.26 | 7.26 | 7.26 | 21.04 | 7.26 | 11.43 | 10.05 |
| 99 | SwinUnet | 8.48 | 8.48 | 8.48 | 8.48 | 13.72 | 8.48 | 8.48 | 11.91 | 8.48 | 27.87 | 8.48 | 8.48 | 8.48 | 8.48 | 8.48 | 8.48 | 8.48 | 8.48 | 9.89 |
| 100 | MT-UNet | 9.34 | 9.30 | 9.12 | 9.17 | 9.44 | 9.16 | 9.38 | 9.52 | 9.52 | 9.52 | 9.45 | 9.52 | 9.52 | 9.32 | 9.52 | 9.52 | 9.48 | 9.45 | 9.45 | 9.48 | 9.41 |

Table 23: Average performance of 100 u-shape medical image segmentation networks with Dice. Baseline U-Net is highlighted (gray background).

| Rank | Network | Ultrasound | | | Dermoscopy | | Endoscopy | | Fundus | | Histopathology | | | Nuclear | X-Ray | | | MRI | | CT | OCT | Avg |
|---|---|---|---|---|---|---|---|---|---|---|---|---|---|---|---|---|---|---|---|---|---|---|
| | | BUSI | BUSBRA | TNSCUI | ISIC2018 | SkinCancer | Kvasir | Robotool | CHASE | DRIVE | DSB2018 | Glas | Monusac | Cell | Covidquex | Montgomery | DCA | ACDC | Promise | Synapse | Cystoidfluid | |
| #1 | RWKV-UNet | 83.94 | 91.75 | 88.92 | 91.87 | 93.32 | 91.62 | 92.44 | 82.94 | 74.80 | 93.68 | 92.33 | 81.63 | 91.64 | 82.87 | 98.07 | 77.76 | 92.01 | 93.37 | 84.10 | 90.59 | 88.79 |
| #2 | UTANet | 81.65 | 91.26 | 87.96 | 91.21 | 91.59 | 90.75 | 91.67 | 88.77 | 75.67 | 93.45 | 91.44 | 81.23 | 91.76 | 81.86 | 97.83 | 78.22 | 91.74 | 92.97 | 82.17 | 91.83 | 88.54 |
| #3 | AURA-Net | 82.78 | 91.27 | 87.84 | 91.45 | 92.22 | 91.03 | 92.35 | 88.94 | 73.60 | 93.75 | 90.57 | 79.92 | 91.82 | 82.94 | 97.79 | 77.49 | 91.51 | 92.85 | 84.56 | 89.31 | 88.50 |
| #4 | Swin-umamba | 82.38 | 91.00 | 88.04 | 91.59 | 91.25 | 92.22 | 91.33 | 88.90 | 77.11 | 93.51 | 88.30 | 78.14 | 91.59 | 81.99 | 97.73 | 77.10 | 91.60 | 92.64 | 83.51 | 91.93 | 88.44 |
| #5 | TransResUNet | 83.23 | 91.22 | 87.66 | 91.53 | 91.77 | 90.94 | 92.06 | 89.51 | 74.09 | 93.10 | 90.32 | 80.14 | 91.17 | 82.22 | 97.90 | 77.49 | 91.18 | 93.02 | 82.11 | 90.24 | 88.35 |
| #6 | FCBFormer | 81.12 | 91.31 | 87.70 | 91.07 | 92.67 | 91.60 | 92.06 | 89.61 | 78.23 | 93.66 | 88.77 | 79.73 | 91.62 | 80.67 | 97.46 | 76.31 | 91.83 | 93.21 | 83.76 | 91.21 | 88.24 |
| #7 | MEGANet | 83.36 | 91.30 | 88.28 | 91.60 | 92.03 | 91.47 | 92.62 | 84.57 | 69.07 | 93.24 | 90.66 | 79.42 | 91.52 | 82.49 | 98.02 | 77.88 | 91.30 | 93.05 | 84.51 | 89.82 | 88.18 |
| #8 | DA-TransUNet | 83.36 | 91.25 | 87.61 | 91.28 | 93.17 | 90.23 | 90.76 | 88.05 | 73.45 | 93.32 | 90.64 | 78.25 | 91.67 | 82.09 | 98.05 | 78.72 | 91.48 | 92.38 | 82.45 | 88.73 | 88.16 |
| #9 | ESKNet | 82.96 | 91.17 | 87.83 | 90.91 | 90.07 | 90.18 | 89.72 | 90.53 | 77.41 | 93.75 | 90.40 | 80.18 | 91.51 | 81.53 | 97.78 | 77.82 | 91.34 | 92.67 | 82.89 | 88.92 | 88.13 |
| #10 | CFFormer | 82.98 | 90.36 | 87.92 | 91.82 | 91.23 | 91.64 | 91.18 | 86.75 | 75.04 | 93.37 | 90.21 | 78.42 | 91.54 | 81.24 | 97.89 | 76.68 | 91.30 | 92.81 | 82.97 | 90.77 | 88.12 |
| #11 | EViT-UNet | 82.70 | 91.52 | 88.11 | 91.32 | 90.65 | 91.27 | 92.12 | 85.10 | 74.18 | 93.23 | 89.21 | 80.68 | 91.43 | 82.58 | 97.58 | 77.91 | 91.84 | 93.08 | 80.36 | 91.03 | 88.10 |
| #12 | CMU-Net | 81.83 | 90.94 | 87.19 | 90.55 | 90.05 | 90.75 | 89.04 | 91.50 | 77.19 | 93.55 | 90.04 | 79.86 | 91.38 | 81.15 | 97.72 | 77.03 | 91.45 | 92.97 | 81.25 | 90.41 | 88.08 |
| #13 | DDANet | 81.18 | 90.92 | 87.34 | 90.83 | 90.45 | 90.74 | 89.11 | 88.23 | 76.09 | 93.71 | 91.18 | 81.22 | 91.40 | 81.18 | 97.95 | 77.79 | 91.95 | 92.59 | 83.60 | 89.64 | 88.08 |
| #14 | MADGNet | 83.21 | 91.47 | 88.21 | 91.41 | 91.70 | 91.49 | 92.01 | 83.16 | 70.42 | 93.00 | 90.73 | 78.54 | 91.39 | 82.19 | 97.80 | 77.91 | 91.47 | 93.20 | 83.85 | 90.32 | 88.03 |
| #15 | Perspective-Unet | 80.24 | 91.03 | 87.77 | 90.58 | 91.47 | 90.53 | 88.75 | 90.63 | 75.36 | 93.21 | 89.83 | 78.54 | 90.71 | 80.62 | 97.54 | 76.81 | 91.19 | 93.06 | 84.85 | 91.24 | 88.00 |
| #16 | AttU-Net | 79.33 | 90.79 | 86.56 | 90.47 | 89.95 | 88.41 | 89.53 | 91.42 | 76.75 | 93.58 | 90.75 | 80.21 | 91.59 | 81.61 | 97.82 | 77.40 | 91.90 | 92.87 | 83.17 | 90.16 | 88.00 |
| #17 | UNet3+ | 78.83 | 90.66 | 86.11 | 90.59 | 90.02 | 88.16 | 89.58 | 91.12 | 76.95 | 93.61 | 90.59 | 79.78 | 91.41 | 81.44 | 97.90 | 77.51 | 91.47 | 92.86 | 82.64 | 92.34 | 87.98 |
| #18 | UTNet | 82.08 | 90.76 | 87.27 | 91.05 | 90.16 | 89.45 | 88.68 | 89.40 | 77.43 | 93.84 | 89.15 | 80.78 | 91.58 | 81.17 | 97.82 | 77.71 | 91.34 | 92.50 | 81.77 | 90.21 | 87.97 |
| #19 | Mobile U-ViT | 83.44 | 91.09 | 87.72 | 91.23 | 91.09 | 90.83 | 89.08 | 87.38 | 75.54 | 93.43 | 88.30 | 79.98 | 91.25 | 81.35 | 97.90 | 77.44 | 91.18 | 92.06 | 82.93 | 88.93 | 87.88 |
| #20 | U-Net | 79.21 | 90.65 | 86.36 | 90.58 | 89.46 | 88.96 | 89.65 | 91.34 | 76.40 | 93.65 | 90.89 | 79.84 | 91.50 | 81.32 | 97.89 | 77.53 | 91.49 | 92.65 | 80.88 | 90.35 | 87.83 |
| #21 | MT-UNet | 80.25 | 89.99 | 84.17 | 90.61 | 91.54 | 88.94 | 89.39 | 90.11 | 77.46 | 94.03 | 89.08 | 81.84 | 91.85 | 80.60 | 97.91 | 77.47 | 91.11 | 92.26 | 82.22 | 89.78 | 87.81 |
| #22 | RollingUnet | 80.73 | 90.97 | 86.48 | 90.75 | 89.51 | 89.81 | 89.10 | 87.96 | 76.92 | 93.70 | 90.43 | 79.90 | 91.67 | 81.54 | 97.97 | 77.37 | 91.23 | 92.40 | 82.44 | 89.66 | 87.80 |
| #23 | UCTransNet | 80.96 | 90.42 | 85.99 | 90.77 | 90.57 | 89.10 | 89.65 | 90.60 | 75.30 | 93.46 | 89.74 | 79.94 | 91.41 | 81.19 | 97.80 | 76.73 | 91.43 | 92.48 | 82.09 | 90.28 | 87.79 |
| #24 | MBSNet | 81.53 | 91.07 | 86.86 | 90.77 | 90.63 | 89.77 | 87.80 | 87.37 | 76.25 | 93.49 | 91.05 | 80.10 | 91.29 | 81.33 | 97.85 | 77.27 | 91.42 | 92.46 | 82.12 | 89.68 | 87.78 |
| #25 | TransUnet | 82.09 | 90.61 | 87.40 | 91.01 | 91.80 | 89.46 | 87.83 | 88.81 | 74.81 | 93.20 | 88.47 | 78.96 | 91.41 | 81.44 | 97.81 | 77.18 | 90.95 | 92.48 | 82.84 | 90.14 | 87.73 |
| #26 | TransAttUnet | 81.28 | 90.88 | 87.42 | 90.73 | 88.34 | 89.83 | 88.90 | 86.12 | 76.88 | 93.89 | 89.53 | 81.13 | 91.82 | 81.90 | 97.98 | 77.33 | 91.23 | 92.21 | 82.43 | 89.40 | 87.72 |
| #27 | LGMSNet | 81.97 | 90.84 | 87.46 | 91.11 | 92.02 | 89.97 | 87.92 | 87.41 | 75.04 | 93.71 | 90.06 | 78.70 | 91.49 | 81.05 | 97.73 | 76.55 | 90.76 | 92.48 | 81.90 | 88.78 | 87.65 |
| #28 | CENet | 81.77 | 90.61 | 87.22 | 91.24 | 91.38 | 91.49 | 88.93 | 82.90 | 73.73 | 93.65 | 90.99 | 78.14 | 91.05 | 80.71 | 97.15 | 76.26 | 92.20 | 91.08 | 85.52 | 90.26 | 87.64 |
| #29 | H2Former | 82.21 | 90.95 | 87.54 | 90.90 | 90.72 | 90.04 | 87.89 | 85.68 | 73.36 | 92.81 | 90.21 | 79.21 | 91.01 | 80.56 | 97.87 | 76.68 | 91.73 | 92.68 | 82.97 | 91.30 | 87.63 |
| #30 | CA-Net | 81.63 | 90.63 | 87.22 | 90.64 | 90.40 | 89.25 | 87.39 | 85.06 | 76.52 | 93.69 | 89.88 | 81.22 | 91.81 | 81.37 | 97.72 | 77.31 | 91.70 | 92.10 | 82.45 | 88.90 | 87.61 |
| #31 | CMUNeXt | 81.43 | 91.08 | 87.12 | 90.59 | 92.12 | 89.02 | 87.54 | 87.80 | 75.77 | 93.40 | 90.19 | 79.04 | 91.30 | 81.01 | 97.72 | 76.99 | 90.79 | 92.23 | 80.99 | 89.29 | 87.57 |
| #32 | D-TrAttUnet | 79.06 | 90.13 | 85.85 | 90.44 | 91.20 | 89.37 | 88.22 | 88.39 | 76.91 | 93.77 | 88.51 | 80.21 | 91.39 | 80.29 | 97.50 | 77.05 | 90.74 | 92.40 | 82.52 | 91.15 | 87.54 |
| #33 | MedFormer | 80.30 | 90.37 | 87.52 | 90.76 | 90.21 | 90.15 | 87.39 | 88.83 | 76.37 | 93.41 | 87.59 | 79.89 | 91.07 | 80.45 | 97.37 | 76.87 | 90.89 | 92.78 | 80.39 | 91.06 | 87.48 |
| #34 | FAT-Net | 82.47 | 91.39 | 88.00 | 91.25 | 91.37 | 90.95 | 91.50 | 85.97 | 66.68 | 92.52 | 90.60 | 78.66 | 90.57 | 81.99 | 97.66 | 77.41 | 91.76 | 92.51 | 78.38 | 89.43 | 87.47 |
| #35 | MCA-UNet | 82.00 | 91.17 | 87.80 | 91.14 | 92.57 | 90.26 | 88.87 | 86.08 | 74.13 | 93.34 | 88.80 | 78.86 | 91.23 | 80.69 | 97.55 | 76.64 | 91.54 | 92.38 | 73.96 | 90.75 | 87.37 |
| #36 | MSLAU-Net | 81.44 | 90.53 | 87.75 | 91.05 | 92.93 | 91.20 | 90.67 | 82.38 | 65.95 | 92.46 | 91.53 | 77.16 | 90.53 | 81.17 | 97.35 | 76.50 | 91.53 | 91.80 | 84.12 | 90.27 | 87.35 |
| #37 | UNet++ | 79.84 | 90.17 | 86.93 | 90.47 | 90.80 | 90.01 | 88.32 | 88.71 | 73.74 | 93.11 | 89.15 | 78.51 | 91.05 | 80.05 | 97.59 | 76.09 | 90.86 | 92.70 | 80.54 | 90.46 | 87.29 |
| #38 | TA-Net | 82.27 | 91.41 | 88.17 | 91.51 | 90.90 | 90.91 | 92.25 | 83.50 | 66.93 | 93.02 | 89.87 | 78.30 | 91.07 | 82.49 | 97.31 | 76.93 | 90.58 | 92.70 | 77.71 | 89.37 | 87.29 |
| #39 | ResU-KAN | 80.51 | 90.72 | 87.25 | 90.92 | 90.18 | 90.53 | 87.35 | 86.44 | 73.29 | 92.97 | 88.54 | 78.17 | 91.24 | 80.78 | 97.35 | 76.58 | 90.08 | 92.68 | 82.07 | 90.82 | 87.25 |
| #40 | U-KAN | 80.31 | 90.60 | 87.10 | 90.55 | 91.86 | 90.17 | 87.50 | 86.23 | 73.19 | 92.86 | 88.73 | 77.52 | 91.04 | 80.19 | 97.43 | 76.04 | 90.38 | 92.98 | 82.23 | 90.67 | 87.22 |
| #41 | MSRFNet | 79.38 | 90.81 | 87.19 | 90.62 | 89.43 | 89.26 | 87.84 | 83.91 | 72.83 | 93.62 | 89.57 | 79.73 | 91.20 | 80.70 | 97.22 | 77.49 | 91.76 | 92.36 | 82.46 | 89.34 | 87.09 |
| #42 | ColonSegNet | 77.12 | 87.94 | 83.06 | 90.14 | 90.06 | 88.77 | 87.56 | 90.23 | 77.44 | 93.62 | 89.23 | 79.33 | 91.72 | 80.00 | 97.78 | 76.97 | 91.05 | 91.29 | 81.00 | 90.95 | 87.08 |
| #43 | GH-UNet | 79.67 | 90.63 | 86.80 | 90.44 | 91.88 | 89.72 | 87.38 | 82.54 | 73.35 | 92.48 | 90.06 | 77.37 | 90.72 | 80.35 | 97.44 | 76.67 | 90.84 | 92.58 | 83.10 | 90.47 | 87.07 |
| #44 | CE-Net | 82.35 | 91.15 | 87.77 | 91.49 | 90.98 | 90.18 | 91.95 | 82.51 | 65.98 | 92.68 | 89.62 | 76.74 | 90.83 | 82.38 | 97.51 | 76.65 | 90.64 | 92.37 | 82.84 | 89.61 | 87.06 |
| #45 | ScribFormer | 79.79 | 90.12 | 86.72 | 90.79 | 88.90 | 88.47 | 85.96 | 85.28 | 75.46 | 93.58 | 87.37 | 80.12 | 91.38 | 80.94 | 97.77 | 76.74 | 91.02 | 91.83 | 83.02 | 89.48 | 87.02 |
| #46 | DS-TransUNet | 77.98 | 89.68 | 84.45 | 90.89 | 91.15 | 91.32 | 90.59 | 84.37 | 71.35 | 93.52 | 89.78 | 78.09 | 90.77 | 79.83 | 97.10 | 75.64 | 91.71 | 90.77 | 84.22 | 88.33 | 86.95 |
| #47 | DCSAU-Net | 80.50 | 90.84 | 87.64 | 90.99 | 89.79 | 89.55 | 87.50 | 82.41 | 75.24 | 93.87 | 87.00 | 78.40 | 91.22 | 81.31 | 97.42 | 76.85 | 90.50 | 92.13 | 79.75 | 88.75 | 86.90 |
| #48 | DDS-UNet | 80.69 | 90.56 | 86.49 | 91.11 | 91.06 | 90.03 | 87.78 | 85.11 | 70.18 | 92.69 | 88.82 | 77.15 | 90.90 | 80.56 | 97.51 | 76.50 | 89.19 | 92.34 | 80.81 | 90.44 | 86.86 |
| #49 | TransNorm | 81.96 | 90.79 | 87.46 | 91.05 | 92.40 | 89.68 | 88.49 | 86.99 | 72.81 | 92.78 | 86.82 | 77.60 | 91.10 | 80.92 | 97.50 | 76.51 | 85.13 | 92.69 | 76.50 | 90.17 | 86.83 |
| #50 | MedVKAN | 80.03 | 91.26 | 87.17 | 90.71 | 90.63 | 90.07 | 87.85 | 80.92 | 73.02 | 93.16 | 87.75 | 79.21 | 90.99 | 81.26 | 97.44 | 76.38 | 90.08 | 92.94 | 79.64 | 88.88 | 86.82 |
| #51 | DoubleUNet | 79.92 | 89.98 | 85.68 | 90.54 | 88.36 | 91.36 | 86.14 | 87.96 | 75.23 | 93.96 | 87.09 | 79.22 | 89.05 | 80.53 | 97.71 | 75.42 | 92.09 | 89.51 | 84.15 | 85.94 | 86.73 |
| #52 | AC-MambaSeg | 79.49 | 90.14 | 86.47 | 90.81 | 90.90 | 89.80 | 86.05 | 88.10 | 70.36 | 92.62 | 85.97 | 77.10 | 90.24 | 81.02 | 97.08 | 75.27 | 89.18 | 92.66 | 82.32 | 90.77 | 86.68 |
| #53 | MMUNet | 80.53 | 90.80 | 87.39 | 90.79 | 89.44 | 90.59 | 86.94 | 86.46 | 73.74 | 92.85 | 84.80 | 76.12 | 90.77 | 80.12 | 96.89 | 75.15 | 89.08 | 91.87 | 79.42 | 91.43 | 86.61 |
| #54 | U-RWKV | 79.32 | 90.08 | 84.19 | 90.52 | 89.35 | 87.82 | 85.93 | 87.09 | 74.61 | 93.61 | 87.73 | 79.45 | 91.30 | 79.86 | 97.53 | 77.17 | 89.58 | 92.26 | 78.48 | 89.42 | 86.58 |
| #55 | HiFormer | 80.44 | 91.02 | 87.46 | 91.28 | 91.83 | 90.93 | 90.68 | 79.62 | 64.16 | 92.30 | 88.73 | 76.68 | 90.25 | 81.22 | 97.48 | 75.01 | 90.86 | 92.47 | 81.78 | 87.42 | 86.57 |
| #56 | ResUNet++ | 76.82 | 89.15 | 84.40 | 90.29 | 90.46 | 87.87 | 85.80 | 88.85 | 76.20 | 93.13 | 86.47 | 77.74 | 91.16 | 80.02 | 97.36 | 74.61 | 89.78 | 92.70 | 76.90 | 90.81 | 86.39 |
| #57 | CASCADE | 82.91 | 90.36 | 87.32 | 91.03 | 92.01 | 91.51 | 89.11 | 74.47 | 50.09 | 92.59 | 91.09 | 73.64 | 89.07 | 81.11 | 97.11 | 76.04 | 91.56 | 92.72 | 84.59 | 86.74 | 86.14 |
| #58 | CSCAUNet | 80.37 | 90.59 | 87.33 | 90.58 | 90.25 | 90.26 | 87.67 | 82.49 | 44.47 | 92.98 | 89.86 | 76.73 | 90.70 | 81.34 | 97.50 | 74.85 | 90.23 | 92.61 | 83.10 | 90.03 | 86.05 |
| #59 | Tinyunet | 76.51 | 90.12 | 85.53 | 90.18 | 89.77 | 87.14 | 85.77 | 82.86 | 74.27 | 92.94 | 88.56 | 75.55 | 90.95 | 80.09 | 97.49 | 74.87 | 89.84 | 91.41 | 80.40 | 88.78 | 86.02 |
| #60 | LV-UNet | 79.77 | 90.16 | 85.97 | 91.03 | 91.04 | 90.04 | 89.02 | 78.40 | 63.40 | 92.79 | 88.08 | 77.20 | 90.34 | 81.38 | 97.13 | 75.94 | 89.14 | 91.79 | 78.15 | 87.71 | 85.91 |
| #61 | G-CASCADE | 82.22 | 90.24 | 87.14 | 91.02 | 91.57 | 91.05 | 87.98 | 71.45 | 53.29 | 92.99 | 90.53 | 75.24 | 89.77 | 80.91 | 97.20 | 74.96 | 91.83 | 91.29 | 84.75 | 85.91 | 85.80 |
| #62 | DC-UNet | 76.01 | 89.08 | 84.46 | 89.49 | 87.91 | 87.01 | 86.48 | 81.02 | 73.44 | 92.68 | 89.58 | 78.63 | 90.63 | 81.51 | 97.50 | 72.66 | 91.15 | 91.45 | 79.98 | 82.73 | 85.54 |
| #63 | ERDUnet | 80.04 | 90.11 | 86.39 | 90.40 | 88.54 | 86.39 | 84.58 | 79.42 | 71.16 | 93.29 | 84.22 | 78.49 | 90.75 | 79.82 | 96.67 | 75.87 | 88.55 | 91.53 | 77.95 | 88.27 | 85.48 |
| #64 | ConvFormer | 80.56 | 90.36 | 86.34 | 90.63 | 90.20 | 89.04 | 88.29 | 77.23 | 54.97 | 91.49 | 89.52 | 73.76 | 89.92 | 81.69 | 97.73 | 74.85 | 90.87 | 92.25 | 78.15 | 86.60 | 85.39 |
| #65 | ULite | 78.02 | 89.32 | 82.87 | 90.27 | 90.58 | 86.39 | 84.69 | 79.06 | 73.50 | 93.30 | 85.37 | 75.87 | 90.87 | 78.82 | 97.07 | 74.18 | 86.16 | 90.88 | 77.39 | 89.27 | 85.08 |
| #66 | SwinUNETR | 76.92 | 87.98 | 82.57 | 90.53 | 89.72 | 88.90 | 83.52 | 82.49 | 76.18 | 93.15 | 82.54 | 76.23 | 91.15 | 78.16 | 97.04 | 75.90 | 85.09 | 91.28 | 76.10 | 88.65 | 85.07 |
| #67 | SCUNet++ | 76.15 | 89.32 | 84.99 | 89.90 | 90.41 | 90.66 | 83.96 | 85.82 | 51.35 | 92.93 | 86.69 | 74.30 | 89.63 | 79.12 | 97.12 | 75.11 | 90.70 | 88.98 | 80.48 | 86.18 | 84.99 |
| #68 | UNeXt | 76.63 | 89.15 | 83.13 | 90.36 | 89.50 | 86.13 | 86.13 | 79.16 | 63.50 | 92.64 | 86.55 | 75.75 | 90.61 | 79.33 | 97.31 | 74.43 | 87.28 | 91.79 | 75.13 | 88.44 | 84.67 |
| #69 | MERIT | 81.82 | 90.81 | 87.30 | 91.21 | 91.49 | 90.95 | 91.25 | 65.38 | 57.66 | 90.41 | 91.17 | 76.63 | 89.74 | 80.61 | 95.72 | 72.95 | 73.25 | 92.03 | 81.50 | 84.24 | 84.57 |
| #70 | MDSA-UNet | 80.69 | 89.89 | 86.65 | 90.72 | 89.86 | 88.44 | 87.31 | 69.68 | 54.87 | 92.26 | 87.78 | 73.95 | 89.89 | 80.18 | 97.25 | 73.20 | 88.19 | 91.70 | 78.49 | 85.55 | 84.55 |
| #71 | CPCANet | 79.52 | 89.51 | 84.23 | 90.19 | 90.94 | 89.80 | 84.67 | 82.96 | 54.30 | 92.17 | 82.27 | 73.39 | 89.54 | 78.74 | 96.77 | 72.61 | 89.92 | 90.80 | 80.51 | 88.19 | 84.16 |
| #72 | LeViT-UNet | 73.72 | 84.87 | 79.59 | 90.81 | 89.97 | 85.95 | 86.39 | 83.17 | 61.45 | 92.44 | 84.25 | 74.22 | 90.03 | 77.59 | 97.33 | 73.13 | 88.44 | 90.21 | 78.32 | 88.71 | 84.04 |
| #73 | DAEFormer | 77.59 | 88.53 | 84.44 | 90.21 | 89.46 | 89.49 | 84.71 | 82.34 | 44.36 | 92.94 | 83.46 | 73.50 | 89.64 | 78.94 | 96.22 | 73.28 | 89.99 | 89.43 | 78.84 | 86.01 | 84.01 |
| #74 | PraNet | 80.47 | 91.25 | 87.89 | 91.46 | 92.69 | 91.54 | 91.95 | 53.41 | 42.49 | 88.22 | 90.50 | 68.53 | 87.32 | 82.42 | 97.00 | 66.96 | 91.41 | 92.98 | 82.93 | 76.57 | 83.97 |
| #75 | CaraNet | 82.78 | 91.49 | 88.07 | 91.46 | 92.48 | 90.85 | 91.62 | 54.75 | 41.65 | 88.29 | 90.76 | 68.43 | 87.12 | 82.77 | 97.14 | 68.19 | 91.28 | 92.85 | 81.64 | 76.46 | 83.90 |
| #76 | TransFuse | 82.19 | 90.83 | 87.51 | 91.37 | 90.63 | 90.72 | 90.52 | 66.32 | 31.26 | 91.98 | 85.73 | 72.49 | 89.55 | 81.87 | 96.42 | 67.47 | 89.48 | 92.58 | 78.74 | 82.04 | 83.88 |
| #77 | MedT | 75.66 | 88.76 | 82.44 | 90.18 | 86.78 | 85.57 | 82.54 | 77.40 | 70.65 | 93.15 | 82.58 | 74.77 | 90.48 | 78.97 | 96.54 | 75.08 | 84.97 | 89.74 | 74.07 | 86.07 | 83.74 |
| #78 | EMCAD | 81.35 | 90.33 | 83.01 | 91.31 | 92.59 | 91.42 | 87.33 | 73.90 | 61.74 | 93.03 | 92.39 | 76.85 | 90.44 | 81.34 | 97.27 | 76.92 | 84.60 | 92.25 | 0.00 | 87.16 | 83.61 |
| #79 | UACANet | 82.27 | 91.34 | 88.16 | 91.16 | 91.23 | 91.78 | 91.69 | 53.27 | 40.62 | 87.74 | 88.80 | 68.01 | 86.71 | 82.13 | 97.17 | 65.94 | 90.42 | 92.72 | 81.86 | 77.38 | 83.46 |
| #80 | MultiResUNet | 79.21 | 90.53 | 85.84 | 90.40 | 90.24 | 89.24 | 87.25 | 46.83 | 32.24 | 91.71 | 89.46 | 77.28 | 90.34 | 81.15 | 96.10 | 68.84 | 90.10 | 92.03 | 78.38 | 86.37 | 83.31 |
| #81 | MUCM-Net | 75.97 | 88.56 | 81.37 | 90.10 | 90.45 | 85.15 | 80.17 | 72.71 | 58.78 | 91.82 | 84.46 | 73.48 | 88.63 | 76.61 | 96.71 | 73.65 | 84.68 | 89.50 | 70.69 | 83.36 | 82.49 |
| #82 | LFU-Net | 73.08 | 87.47 | 76.94 | 89.20 | 85.72 | 77.93 | 81.76 | 74.36 | 71.69 | 92.88 | 85.54 | 71.87 | 89.84 | 78.24 | 96.38 | 72.56 | 84.18 | 87.75 | 67.72 | 84.33 | 81.99 |
| #83 | MissFormer | 76.69 | 88.43 | 84.24 | 90.04 | 86.71 | 89.03 | 82.63 | 75.71 | 32.38 | 92.32 | 79.13 | 70.50 | 76.96 | 77.59 | 95.58 | 72.66 | 87.03 | 88.18 | 77.98 | 84.42 | 81.54 |
| #84 | UNETR | 68.66 | 83.31 | 65.17 | 89.43 | 89.16 | 83.73 | 76.70 | 79.13 | 75.82 | 93.34 | 76.15 | 73.70 | 90.81 | 73.85 | 95.62 | 73.54 | 83.45 | 89.66 | 70.85 | 85.13 | 81.47 |
| #85 | CFM-UNet | 74.12 | 87.81 | 84.06 | 89.99 | 89.28 | 86.70 | 81.94 | 64.11 | 25.45 | 91.81 | 82.54 | 71.12 | 89.26 | 79.18 | 96.30 | 69.55 | 82.11 | 91.12 | 74.91 | 82.80 | 81.07 |
| #86 | Zig-RiR | 70.27 | 85.83 | 79.65 | 89.96 | 88.31 | 85.70 | 83.62 | 71.89 | 52.45 | 89.93 | 81.15 | 63.32 | 89.11 | 75.00 | 96.29 | 69.71 | 79.68 | 91.35 | 67.42 | 85.38 | 80.72 |
| #87 | SimpleUNet | 71.03 | 86.21 | 77.25 | 88.59 | 87.60 | 82.08 | 84.75 | 77.00 | 50.70 | 92.46 | 87.18 | 77.28 | 88.74 | 72.67 | 92.61 | 71.03 | 89.32 | 86.73 | 77.19 | 0.06 | 80.69 |
| #88 | CFPNet-M | 78.95 | 89.79 | 85.77 | 90.66 | 89.89 | 88.41 | 87.14 | 30.87 | 73.54 | 93.07 | 88.30 | 77.36 | 90.84 | 80.15 | 97.16 | 75.50 | 90.50 | 0.00 | 78.19 | 0.00 | 80.60 |
| #89 | BEFUnet | 76.48 | 87.43 | 79.05 | 88.81 | 90.47 | 86.93 | 82.04 | 67.58 | 27.33 | 91.57 | 75.61 | 65.90 | 88.62 | 75.47 | 94.94 | 69.94 | 86.08 | 87.20 | 72.77 | 77.14 | 79.96 |
| #90 | UNetV2 | 73.51 | 87.77 | 82.46 | 89.80 | 89.73 | 88.43 | 82.78 | 57.87 | 26.35 | 91.61 | 80.31 | 70.73 | 89.24 | 77.02 | 95.52 | 67.27 | 84.01 | 89.36 | 57.73 | 80.49 | 79.80 |
| #91 | Swin-umambaD | 78.74 | 88.98 | 85.84 | 90.07 | 82.84 | 91.47 | 84.24 | 66.68 | 24.92 | 93.36 | 65.74 | 73.54 | 51.46 | 79.85 | 97.11 | 74.36 | 92.16 | 64.48 | 74.02 | 90.85 | 78.85 |
| #92 | VMUNetV2 | 76.30 | 89.22 | 85.61 | 90.66 | 87.86 | 91.32 | 76.13 | 63.22 | 23.04 | 90.66 | 88.31 | 68.31 | 67.76 | 79.52 | 95.27 | 63.99 | 90.15 | 88.80 | 79.43 | 12.10 | 76.69 |
| #93 | VMUNet | 80.51 | 62.14 | 85.92 | 90.72 | 90.76 | 91.83 | 86.63 | 77.71 | 28.14 | 92.40 | 69.12 | 78.05 | 17.17 | 80.21 | 96.84 | 69.99 | 91.25 | 64.67 | 81.32 | 81.56 | 78.45 |
| #94 | UltraLight-VM-UNet | 73.08 | 83.97 | 74.35 | 89.17 | 89.48 | 77.43 | 75.45 | 66.95 | 25.54 | 92.47 | 75.28 | 71.54 | 37.67 | 75.08 | 94.40 | 67.41 | 84.15 | 87.37 | 67.38 | 81.49 | 78.40 |
| #95 | Polyp-PVT | 82.73 | 90.21 | 86.80 | 90.91 | 91.72 | 91.28 | 89.28 | 52.80 | 2.52 | 89.86 | 67.74 | 63.39 | 61.33 | 80.55 | 97.03 | 0.13 | 91.61 | 91.19 | 91.35 | 75.26 | 78.11 |
| #96 | H-vmunet | 76.42 | 87.15 | 79.88 | 89.91 | 88.36 | 85.20 | 76.78 | 63.55 | 19.58 | 92.23 | 80.32 | 69.57 | 87.37 | 75.93 | 95.60 | 8.86 | 82.55 | 90.46 | 70.40 | 79.15 | 77.82 |
| #97 | CSWin-UNet | 72.11 | 87.81 | 82.15 | 89.91 | 89.61 | 85.77 | 82.37 | 42.62 | 25.47 | 92.24 | 77.39 | 67.20 | 88.39 | 77.09 | 94.00 | 13.50 | 85.01 | 87.76 | 72.48 | 88.71 | 77.49 |
| #98 | MALUNet | 74.70 | 85.27 | 81.14 | 89.49 | 87.16 | 84.10 | 72.16 | 21.23 | 26.21 | 91.80 | 76.53 | 70.83 | 88.21 | 74.80 | 94.84 | 68.23 | 77.31 | 64.55 | 68.00 | 78.45 | 75.96 |
| #99 | SwinUnet | 66.50 | 82.31 | 42.96 | 89.61 | 88.66 | 80.32 | 80.43 | 65.94 | 19.26 | 92.20 | 73.27 | 66.50 | 78.11 | 75.27 | 94.78 | 62.30 | 92.78 | 64.42 | 66.73 | 77.67 | 74.48 |
| #100 | MambaUnet | 31.94 | 89.10 | 67.23 | 88.40 | 59.46 | 91.47 | 75.69 | 69.93 | 21.41 | 92.78 | 68.87 | 78.16 | 1.32 | 78.93 | 72.20 | 74.62 | 91.67 | 64.52 | 81.58 | 76.51 | 72.23 |

Table 24: Average performance of 100 u-shape medical image segmentation networks with Boundary-F1. Baseline U-Net is highlighted (gray background).

| Rank | Network | Ultrasound | | | Dermoscopy | | Endoscopy | | Fundus | | Histopathology | | | Nuclear | X-Ray | | | MRI | | CT | OCT | Avg |
|---|---|---|---|---|---|---|---|---|---|---|---|---|---|---|---|---|---|---|---|---|---|---|
| | | BUSI | BUSBRA | TNSCUI | ISIC2018 | SkinCancer | Kvasir | Robotool | CHASE | DRIVE | DSB2018 | Glas | Monusac | Cell | Covidquex | Montgomery | DCA | ACDC | Promise | Synapse | Cystoidfluid | |
| #1 | RWKV-UNet | 83.64 | 94.50 | 91.40 | 93.77 | 95.85 | 92.39 | 95.28 | 95.70 | 92.10 | 98.74 | 94.59 | 91.17 | 96.96 | 84.23 | 99.49 | 91.33 | 98.67 | 96.99 | 78.34 | 97.39 | 93.13 |
| #2 | AURA-Net | 81.98 | 93.79 | 90.07 | 93.19 | 94.85 | 91.39 | 95.01 | 96.98 | 90.97 | 98.62 | 92.77 | 88.86 | 97.22 | 83.96 | 99.20 | 91.06 | 98.64 | 96.50 | 77.97 | 96.05 | 92.45 |
| #3 | UTANet | 80.43 | 93.69 | 90.03 | 92.96 | 94.13 | 91.43 | 94.55 | 96.75 | 92.30 | 98.50 | 93.78 | 90.29 | 97.02 | 82.91 | 99.22 | 91.91 | 98.34 | 96.48 | 75.86 | 97.45 | 92.40 |
| #4 | MADGNet | 82.87 | 94.23 | 90.53 | 93.28 | 93.93 | 92.77 | 94.76 | 95.18 | 89.19 | 98.51 | 92.99 | 88.55 | 96.89 | 83.07 | 99.25 | 91.83 | 98.47 | 96.77 | 77.74 | 96.93 | 92.39 |
| #5 | MEGANet | 83.12 | 94.04 | 90.59 | 93.47 | 94.16 | 92.12 | 95.47 | 95.66 | 87.33 | 98.52 | 92.94 | 89.05 | 96.94 | 83.56 | 99.49 | 91.77 | 98.48 | 96.65 | 77.64 | 95.99 | 92.35 |
| #6 | Swin-umamba | 81.72 | 93.71 | 90.35 | 93.51 | 94.30 | 93.71 | 94.27 | 96.99 | 92.60 | 98.49 | 90.74 | 87.15 | 96.86 | 83.43 | 99.16 | 90.22 | 98.34 | 96.09 | 78.11 | 97.21 | 92.35 |
| #7 | TransResUNet | 82.48 | 93.84 | 89.87 | 93.35 | 94.09 | 91.25 | 94.94 | 97.28 | 92.17 | 98.37 | 92.74 | 89.21 | 96.58 | 83.26 | 99.40 | 91.69 | 98.34 | 96.46 | 74.40 | 95.91 | 92.28 |
| #8 | DA-TransUNet | 83.15 | 94.03 | 89.70 | 92.87 | 95.70 | 90.72 | 93.15 | 96.80 | 92.06 | 98.49 | 92.75 | 88.09 | 97.08 | 83.35 | 99.44 | 92.59 | 98.61 | 95.97 | 73.86 | 95.63 | 92.20 |
| #9 | EViT-UNet | 81.78 | 94.26 | 90.48 | 93.14 | 93.09 | 92.69 | 95.00 | 95.98 | 91.69 | 98.53 | 91.50 | 90.66 | 96.90 | 83.43 | 99.09 | 92.06 | 98.48 | 96.55 | 71.54 | 97.10 | 92.20 |
| #10 | ESKNet | 82.52 | 93.59 | 89.78 | 92.51 | 92.59 | 90.43 | 92.36 | 97.49 | 92.86 | 98.53 | 92.53 | 88.91 | 96.74 | 82.38 | 99.22 | 91.11 | 98.27 | 96.15 | 76.38 | 96.52 | 92.04 |
| #11 | DDANet | 79.42 | 93.50 | 89.38 | 92.53 | 92.18 | 91.77 | 91.59 | 96.96 | 92.41 | 98.56 | 93.38 | 90.06 | 96.80 | 81.79 | 99.29 | 91.46 | 98.50 | 96.04 | 77.97 | 96.75 | 92.02 |
| #12 | FCBFormer | 80.11 | 93.87 | 89.94 | 92.79 | 95.55 | 92.54 | 89.78 | 97.23 | 93.49 | 98.59 | 91.17 | 88.51 | 96.77 | 81.55 | 98.92 | 88.88 | 98.62 | 96.61 | 77.13 | 97.22 | 91.96 |
| #13 | Mobile U-ViT | 83.03 | 93.65 | 90.02 | 93.01 | 93.66 | 91.59 | 91.51 | 96.62 | 92.39 | 98.48 | 90.10 | 89.51 | 96.60 | 82.32 | 99.30 | 90.98 | 98.13 | 95.54 | 76.29 | 96.30 | 91.96 |
| #14 | FAT-Net | 81.64 | 94.08 | 90.61 | 93.49 | 94.01 | 91.96 | 94.62 | 95.63 | 87.26 | 98.68 | 93.32 | 89.93 | 97.31 | 83.46 | 99.31 | 92.17 | 98.70 | 96.35 | 69.34 | 97.27 | 91.96 |
| #15 | CENet | 81.21 | 93.14 | 89.55 | 92.85 | 93.79 | 92.56 | 91.40 | 95.53 | 90.95 | 98.70 | 93.32 | 87.93 | 96.43 | 81.76 | 98.65 | 89.63 | 98.65 | 94.69 | 79.87 | 97.33 | 91.90 |
| #16 | CFFormer | 83.31 | 92.84 | 90.26 | 93.71 | 94.13 | 92.41 | 94.12 | 96.33 | 91.78 | 98.57 | 92.77 | 88.60 | 97.09 | 82.55 | 99.40 | 90.81 | 97.98 | 96.31 | 77.88 | 96.66 | 91.88 |
| #17 | Perspective-Unet | 78.46 | 93.61 | 89.75 | 92.16 | 93.92 | 90.99 | 91.28 | 97.62 | 91.67 | 98.47 | 92.12 | 87.77 | 96.35 | 81.47 | 99.02 | 90.17 | 98.32 | 96.51 | 78.93 | 97.19 | 91.79 |
| #18 | MSLAU-Net | 80.51 | 93.01 | 90.06 | 92.77 | 95.59 | 91.84 | 93.51 | 94.50 | 86.46 | 98.30 | 93.75 | 88.12 | 96.41 | 82.06 | 98.96 | 90.68 | 98.50 | 95.30 | 78.09 | 97.18 | 91.78 |
| #19 | H2Former | 81.76 | 93.65 | 89.71 | 92.62 | 93.51 | 90.40 | 90.48 | 95.97 | 91.47 | 98.37 | 92.43 | 88.95 | 96.69 | 81.39 | 99.31 | 90.68 | 98.61 | 96.16 | 75.98 | 96.89 | 91.75 |
| #20 | CMU-Net | 81.08 | 93.59 | 89.07 | 92.12 | 92.77 | 91.42 | 91.87 | 97.86 | 93.00 | 98.37 | 92.16 | 88.29 | 96.47 | 81.92 | 99.12 | 90.60 | 98.43 | 96.31 | 74.06 | 96.18 | 91.73 |
| #21 | MBSNet | 80.37 | 93.74 | 88.84 | 92.53 | 93.31 | 90.12 | 90.21 | 96.61 | 92.34 | 98.44 | 93.26 | 89.09 | 96.43 | 82.15 | 99.30 | 90.11 | 98.23 | 95.78 | 75.50 | 96.79 | 91.71 |
| #22 | UTNet | 81.35 | 93.24 | 89.32 | 92.79 | 92.76 | 89.55 | 91.22 | 97.20 | 93.06 | 98.59 | 91.33 | 89.55 | 96.88 | 82.03 | 99.24 | 91.11 | 98.06 | 96.04 | 74.14 | 96.68 | 91.71 |
| #23 | RollingUnet | 78.94 | 93.56 | 88.22 | 92.55 | 92.09 | 90.29 | 91.83 | 96.97 | 92.58 | 98.46 | 92.52 | 88.97 | 96.94 | 82.63 | 99.32 | 90.87 | 97.95 | 95.80 | 75.80 | 96.37 | 91.63 |
| #24 | AttU-Net | 76.87 | 93.27 | 88.31 | 92.21 | 92.24 | 88.58 | 92.37 | 97.89 | 92.71 | 98.46 | 93.00 | 88.68 | 96.82 | 82.53 | 99.24 | 91.08 | 98.47 | 96.30 | 77.08 | 96.52 | 91.63 |
| #25 | TransAttUnet | 80.35 | 93.46 | 89.62 | 92.36 | 90.29 | 90.27 | 91.52 | 96.76 | 92.61 | 98.62 | 91.60 | 88.88 | 97.03 | 81.16 | 99.35 | 90.52 | 97.80 | 95.58 | 75.26 | 95.92 | 91.60 |
| #26 | TA-Net | 81.45 | 94.27 | 90.59 | 93.32 | 93.35 | 91.91 | 95.14 | 95.00 | 87.84 | 98.57 | 92.41 | 89.22 | 96.64 | 83.68 | 98.84 | 91.05 | 97.98 | 96.36 | 67.67 | 96.29 | 91.58 |
| #27 | TransUnet | 80.89 | 92.86 | 89.41 | 92.76 | 94.52 | 89.68 | 90.37 | 96.75 | 91.05 | 98.47 | 90.49 | 88.78 | 96.80 | 82.76 | 99.25 | 90.27 | 98.17 | 95.94 | 75.50 | 96.81 | 91.58 |
| #28 | LGMSNet | 80.89 | 93.42 | 89.59 | 92.87 | 94.72 | 90.51 | 90.33 | 96.62 | 91.41 | 98.66 | 92.25 | 88.48 | 96.79 | 81.59 | 99.18 | 90.10 | 97.82 | 96.00 | 73.47 | 96.23 | 91.55 |
| #29 | HiFormer | 78.83 | 94.06 | 90.25 | 93.40 | 94.78 | 92.11 | 93.82 | 94.13 | 87.04 | 98.59 | 91.36 | 88.88 | 96.75 | 82.75 | 99.08 | 89.82 | 98.51 | 96.54 | 73.67 | 96.30 | 91.53 |
| #30 | UCTransNet | 79.43 | 92.95 | 87.73 | 92.37 | 93.27 | 89.45 | 92.43 | 97.51 | 92.00 | 98.36 | 92.04 | 89.48 | 96.92 | 82.17 | 99.25 | 90.36 | 98.12 | 95.92 | 74.59 | 96.33 | 91.53 |
| #31 | UNet3+ | 76.43 | 93.34 | 87.75 | 92.39 | 92.64 | 87.82 | 92.42 | 97.74 | 92.48 | 98.43 | 92.94 | 88.57 | 96.68 | 82.62 | 99.32 | 90.97 | 98.07 | 96.28 | 76.21 | 96.87 | 91.50 |
| #32 | CA-Net | 80.74 | 92.95 | 89.06 | 92.14 | 92.04 | 89.69 | 89.81 | 96.38 | 92.39 | 98.58 | 91.55 | 89.93 | 96.94 | 82.69 | 99.13 | 90.13 | 97.68 | 95.42 | 75.39 | 96.18 | 91.44 |
| #33 | MT-UNet | 78.67 | 92.34 | 85.51 | 92.34 | 94.28 | 89.68 | 92.37 | 97.29 | 93.23 | 98.66 | 91.26 | 90.36 | 96.89 | 81.88 | 99.26 | 90.81 | 97.80 | 95.58 | 74.95 | 95.70 | 91.42 |
| #34 | CE-Net | 82.07 | 93.94 | 90.00 | 93.40 | 93.79 | 90.89 | 94.90 | 94.67 | 85.45 | 98.38 | 92.26 | 87.77 | 96.52 | 83.67 | 99.09 | 90.55 | 98.03 | 96.00 | 68.29 | 96.27 | 91.40 |
| #35 | U-Net | 76.51 | 93.04 | 88.09 | 92.27 | 91.65 | 89.25 | 92.36 | 97.79 | 92.32 | 98.51 | 93.09 | 88.30 | 96.70 | 82.02 | 99.30 | 91.51 | 98.14 | 96.03 | 73.66 | 96.73 | 91.36 |
| #36 | D-TrAttUnet | 77.19 | 92.80 | 87.43 | 91.97 | 93.81 | 90.21 | 90.95 | 96.98 | 92.49 | 98.59 | 90.69 | 89.25 | 96.73 | 81.37 | 99.00 | 90.77 | 97.59 | 95.88 | 76.41 | 96.71 | 91.34 |
| #37 | ResU-KAN | 79.04 | 93.37 | 89.50 | 92.57 | 93.87 | 91.29 | 90.13 | 96.23 | 90.51 | 98.35 | 90.73 | 87.63 | 96.70 | 81.95 | 98.93 | 90.20 | 97.72 | 96.09 | 76.06 | 96.30 | 91.30 |
| #38 | GH-UNet | 77.93 | 93.18 | 88.51 | 92.07 | 94.52 | 90.05 | 90.23 | 95.42 | 89.95 | 98.17 | 92.23 | 86.84 | 96.44 | 81.54 | 99.03 | 90.54 | 98.23 | 96.15 | 76.50 | 95.93 | 91.17 |
| #39 | U-KAN | 79.33 | 93.09 | 89.01 | 92.11 | 94.43 | 90.67 | 89.94 | 96.16 | 90.51 | 98.17 | 91.04 | 86.84 | 96.63 | 81.01 | 99.01 | 89.53 | 97.87 | 96.50 | 74.93 | 96.45 | 91.16 |
| #40 | DS-TransUNet | 76.71 | 91.94 | 86.30 | 92.44 | 94.03 | 92.11 | 93.54 | 95.63 | 88.94 | 98.72 | 91.93 | 88.49 | 96.66 | 81.50 | 98.86 | 89.91 | 97.84 | 96.32 | 71.93 | 96.52 | 91.12 |
| #41 | MedFormer | 79.40 | 92.71 | 89.66 | 92.34 | 92.99 | 90.99 | 89.78 | 97.00 | 91.85 | 98.36 | 89.55 | 88.55 | 96.34 | 81.50 | 98.86 | 89.91 | 97.84 | 96.32 | 71.93 | 96.50 | 91.12 |
| #42 | MSRFNet | 78.06 | 93.21 | 89.24 | 92.13 | 91.65 | 89.54 | 90.34 | 95.74 | 88.96 | 98.59 | 91.78 | 88.52 | 96.38 | 81.55 | 99.11 | 90.73 | 97.70 | 95.77 | 76.05 | 96.50 | 91.08 |
| #43 | DCSAU-Net | 79.40 | 93.32 | 89.94 | 92.72 | 92.52 | 90.10 | 90.38 | 95.69 | 91.40 | 98.66 | 89.32 | 88.08 | 96.73 | 82.45 | 98.96 | 90.23 | 97.81 | 95.63 | 71.54 | 96.08 | 91.05 |
| #44 | MCA-UNet | 81.18 | 93.77 | 89.80 | 92.68 | 95.39 | 90.42 | 91.21 | 96.12 | 91.14 | 98.44 | 90.81 | 88.32 | 96.86 | 81.77 | 99.15 | 90.07 | 98.03 | 95.87 | 72.96 | 96.41 | 91.02 |
| #45 | DDS-UNet | 79.17 | 93.21 | 88.51 | 92.93 | 93.72 | 90.53 | 90.33 | 95.54 | 88.94 | 98.16 | 91.15 | 87.53 | 96.55 | 81.44 | 99.17 | 90.28 | 97.24 | 95.97 | 72.99 | 96.74 | 91.01 |
| #46 | MedVKAN | 78.44 | 93.89 | 89.07 | 92.41 | 93.28 | 90.25 | 90.60 | 94.92 | 90.88 | 98.40 | 89.73 | 88.13 | 96.37 | 82.05 | 99.02 | 90.20 | 97.10 | 96.48 | 71.41 | 96.21 | 90.94 |
| #47 | ScribFormer | 78.11 | 92.52 | 88.72 | 92.53 | 89.52 | 87.93 | 88.12 | 96.27 | 91.56 | 98.46 | 89.80 | 88.97 | 96.76 | 81.52 | 99.18 | 89.93 | 97.98 | 95.29 | 75.66 | 96.36 | 90.81 |
| #48 | CASCADE | 82.27 | 92.86 | 89.72 | 92.75 | 94.60 | 92.06 | 91.64 | 92.14 | 71.71 | 98.48 | 94.17 | 85.55 | 96.25 | 82.21 | 98.88 | 89.88 | 98.32 | 96.41 | 79.00 | 96.17 | 90.75 |
| #49 | G-CASCADE | 81.24 | 92.95 | 89.61 | 92.62 | 93.96 | 91.84 | 90.67 | 91.57 | 75.85 | 98.65 | 92.93 | 86.88 | 96.26 | 82.22 | 98.91 | 88.96 | 98.69 | 95.03 | 78.70 | 96.31 | 90.69 |
| #50 | ColonSegNet | 76.23 | 90.68 | 84.92 | 91.55 | 92.55 | 89.51 | 90.30 | 97.29 | 93.19 | 98.41 | 91.55 | 87.83 | 96.49 | 81.12 | 99.17 | 90.39 | 97.14 | 94.56 | 72.97 | 96.55 | 90.64 |
| #51 | TransNorm | 80.71 | 93.39 | 89.62 | 92.79 | 95.22 | 89.77 | 91.33 | 96.39 | 89.62 | 98.41 | 89.05 | 87.91 | 96.60 | 81.98 | 99.05 | 90.20 | 93.94 | 96.28 | 64.21 | 96.39 | 90.64 |
| #52 | MMUNet | 79.60 | 93.41 | 89.42 | 92.48 | 92.13 | 91.16 | 89.09 | 95.92 | 90.64 | 98.25 | 87.40 | 86.04 | 96.31 | 81.00 | 98.56 | 88.38 | 96.86 | 95.26 | 70.25 | 95.85 | 90.40 |
| #53 | U-RWKV | 78.17 | 92.57 | 85.64 | 92.02 | 91.53 | 88.88 | 88.76 | 96.20 | 91.07 | 98.42 | 90.12 | 88.62 | 96.49 | 80.89 | 98.97 | 90.81 | 96.59 | 95.84 | 69.63 | 95.92 | 90.36 |
| #54 | LV-UNet | 78.73 | 92.76 | 87.90 | 92.80 | 93.83 | 90.57 | 91.84 | 93.88 | 81.45 | 98.31 | 90.55 | 88.13 | 96.17 | 82.20 | 98.85 | 89.44 | 97.13 | 95.43 | 68.18 | 95.87 | 90.20 |
| #55 | DC-UNet | 73.74 | 91.44 | 85.90 | 90.84 | 90.79 | 87.07 | 89.04 | 96.10 | 91.64 | 98.18 | 92.03 | 87.35 | 96.12 | 82.46 | 99.01 | 87.54 | 97.94 | 94.91 | 71.05 | 94.90 | 89.90 |
| #56 | MERIT | 80.71 | 93.66 | 89.97 | 92.96 | 94.10 | 91.63 | 94.11 | 88.71 | 82.25 | 97.61 | 93.51 | 88.70 | 96.25 | 81.93 | 97.84 | 88.45 | 79.64 | 95.71 | 74.20 | 95.48 | 89.87 |
| #57 | Tinyunet | 74.11 | 92.65 | 87.13 | 91.86 | 92.63 | 87.33 | 88.02 | 95.21 | 90.61 | 98.16 | 91.03 | 84.56 | 96.17 | 80.82 | 99.07 | 88.86 | 96.86 | 94.76 | 71.34 | 95.84 | 89.85 |
| #58 | ConvFormer | 79.20 | 93.16 | 88.59 | 92.15 | 92.62 | 89.69 | 90.88 | 91.90 | 78.07 | 98.04 | 91.53 | 86.13 | 96.30 | 83.15 | 99.24 | 89.05 | 98.18 | 95.59 | 68.29 | 94.79 | 89.83 |
| #59 | CSCAUNet | 79.44 | 93.11 | 89.53 | 92.20 | 92.85 | 91.08 | 90.59 | 95.10 | 59.54 | 98.46 | 92.15 | 86.69 | 96.36 | 82.70 | 99.11 | 88.62 | 98.05 | 96.20 | 76.57 | 96.32 | 89.73 |
| #60 | CMUNeXt | 43.91 | 93.82 | 89.08 | 92.34 | 94.74 | 89.56 | 89.78 | 96.68 | 91.87 | 98.41 | 92.61 | 88.04 | 96.52 | 82.02 | 99.20 | 89.48 | 98.05 | 95.50 | 74.05 | 96.30 | 89.65 |
| #61 | ERDUnet | 78.88 | 92.59 | 88.55 | 92.07 | 91.07 | 87.45 | 87.25 | 94.32 | 88.27 | 98.50 | 86.48 | 88.59 | 96.38 | 80.71 | 98.39 | 89.33 | 96.04 | 94.99 | 67.11 | 95.90 | 89.64 |
| #62 | UNet++ | 73.97 | 92.15 | 85.42 | 92.00 | 93.44 | 90.16 | 89.77 | 94.53 | 89.47 | 98.45 | 91.25 | 86.07 | 95.96 | 79.38 | 99.16 | 88.75 | 96.47 | 91.13 | 67.21 | 93.96 | 89.44 |
| #63 | SCUNet++ | 74.42 | 92.12 | 87.34 | 91.73 | 93.98 | 92.06 | 87.32 | 94.74 | 73.64 | 98.37 | 91.42 | 86.36 | 96.35 | 82.02 | 99.22 | 89.48 | 98.05 | 93.07 | 72.52 | 95.58 | 89.43 |
| #64 | MDSA-UNet | 79.54 | 92.26 | 89.03 | 92.36 | 92.48 | 88.71 | 90.05 | 90.31 | 78.31 | 98.19 | 90.13 | 85.87 | 96.14 | 81.05 | 98.93 | 87.51 | 96.70 | 95.35 | 69.30 | 95.07 | 89.36 |
| #65 | SwinUNETR | 75.70 | 90.31 | 84.43 | 92.29 | 92.98 | 90.00 | 86.50 | 95.32 | 92.09 | 98.22 | 86.42 | 86.04 | 96.45 | 79.28 | 98.63 | 89.55 | 93.94 | 95.03 | 65.33 | 95.54 | 89.20 |
| #66 | ULite | 76.58 | 91.71 | 84.18 | 91.84 | 93.55 | 86.98 | 87.32 | 94.12 | 91.12 | 98.41 | 88.15 | 85.73 | 96.31 | 79.72 | 98.76 | 88.22 | 94.29 | 96.50 | 67.04 | 95.44 | 88.74 |
| #67 | UNeXt | 74.51 | 91.46 | 84.50 | 92.12 | 92.14 | 85.51 | 88.64 | 92.88 | 84.96 | 98.28 | 85.28 | 88.28 | 96.17 | 80.16 | 98.95 | 88.20 | 95.80 | 95.46 | 63.04 | 96.54 | 88.74 |
| #68 | ResUNet++ | 71.13 | 89.25 | 84.54 | 91.82 | 92.45 | 88.26 | 87.44 | 92.85 | 91.65 | 98.15 | 89.46 | 84.48 | 96.35 | 78.77 | 98.73 | 88.89 | 97.20 | 92.56 | 66.64 | 92.22 | 88.65 |
| #69 | PraNet | 82.74 | 94.12 | 90.38 | 93.50 | 95.24 | 92.67 | 94.88 | 75.13 | 66.64 | 96.57 | 92.85 | 81.03 | 95.21 | 83.86 | 98.83 | 80.79 | 98.75 | 96.60 | 76.42 | 89.32 | 88.48 |
| #70 | AC-MambaSeg | 75.35 | 92.08 | 87.17 | 91.94 | 92.96 | 88.81 | 88.74 | 91.50 | 85.02 | 98.22 | 87.43 | 84.88 | 95.80 | 79.99 | 98.66 | 86.30 | 95.62 | 88.91 | 68.78 | 92.27 | 88.47 |
| #71 | CaraNet | 82.08 | 94.40 | 90.45 | 93.39 | 94.95 | 91.59 | 94.46 | 76.50 | 59.81 | 96.58 | 93.06 | 81.05 | 95.92 | 84.24 | 99.01 | 82.74 | 98.64 | 94.61 | 74.64 | 89.60 | 88.43 |
| #72 | DAEFormer | 76.84 | 90.91 | 86.10 | 91.71 | 92.74 | 90.57 | 91.51 | 93.89 | 67.98 | 98.42 | 86.64 | 84.68 | 95.92 | 80.07 | 98.28 | 87.74 | 97.19 | 93.21 | 69.50 | 95.10 | 88.22 |
| #73 | LeViT-UNet | 71.86 | 86.60 | 80.48 | 91.22 | 92.57 | 86.14 | 89.08 | 94.02 | 82.20 | 98.22 | 86.71 | 84.77 | 96.10 | 78.10 | 98.96 | 87.37 | 96.37 | 93.67 | 69.53 | 95.15 | 87.96 |
| #74 | CPCANet | 78.67 | 91.83 | 85.59 | 91.70 | 93.82 | 90.15 | 87.35 | 93.23 | 56.96 | 98.22 | 84.76 | 84.05 | 95.76 | 79.29 | 98.66 | 86.10 | 97.33 | 94.50 | 72.63 | 96.30 | 87.88 |
| #75 | MedT | 74.60 | 91.35 | 84.04 | 91.70 | 88.46 | 86.56 | 85.04 | 93.32 | 88.36 | 98.45 | 86.53 | 85.02 | 96.17 | 79.95 | 98.32 | 88.58 | 95.52 | 93.36 | 59.98 | 94.71 | 87.86 |
| #76 | UACANet | 81.47 | 94.16 | 90.75 | 93.07 | 93.68 | 92.82 | 94.62 | 74.86 | 58.04 | 96.35 | 93.28 | 80.77 | 94.76 | 83.59 | 98.94 | 80.24 | 97.57 | 96.37 | 68.36 | 89.96 | 87.58 |
| #77 | TransFuse | 80.60 | 93.61 | 89.50 | 93.33 | 93.03 | 91.91 | 93.46 | 86.46 | 41.33 | 98.30 | 87.99 | 85.40 | 96.16 | 82.67 | 98.55 | 81.44 | 98.09 | 96.27 | 68.36 | 94.25 | 87.54 |
| #78 | EMCAD | 81.53 | 92.75 | 84.38 | 93.05 | 95.30 | 92.03 | 89.62 | 92.51 | 85.00 | 98.57 | 94.57 | 88.17 | 96.84 | 82.55 | 98.93 | 90.86 | 94.11 | 95.95 | 0.00 | 96.16 | 87.14 |
| #79 | MUCM-Net | 75.02 | 90.95 | 82.61 | 91.55 | 93.15 | 85.13 | 82.43 | 90.26 | 80.34 | 98.27 | 84.59 | 85.28 | 93.26 | 77.64 | 98.49 | 87.21 | 92.81 | 93.32 | 51.47 | 93.24 | 86.45 |
| #80 | MissFormer | 75.45 | 91.09 | 86.40 | 91.70 | 90.03 | 89.63 | 85.65 | 92.43 | 57.52 | 98.46 | 83.10 | 83.87 | 97.14 | 78.32 | 98.03 | 88.16 | 95.67 | 92.70 | 68.03 | 95.48 | 86.44 |
| #81 | MultiResUNet | 76.93 | 92.98 | 87.29 | 92.15 | 92.91 | 89.12 | 89.85 | 96.95 | 49.28 | 97.40 | 91.65 | 86.47 | 96.23 | 82.25 | 98.20 | 80.56 | 97.56 | 95.55 | 69.78 | 95.48 | 86.43 |
| #82 | LFU-Net | 71.50 | 89.87 | 77.89 | 90.34 | 93.18 | 77.37 | 76.45 | 77.77 | 84.57 | 71.60 | 88.90 | 90.36 | 88.59 | 82.67 | 96.54 | 88.67 | 92.28 | 95.51 | 50.93 | 94.57 | 85.95 |
| #83 | UNETR | 66.10 | 85.30 | 64.74 | 91.29 | 92.57 | 85.06 | 80.25 | 93.89 | 91.54 | 98.33 | 91.98 | 84.01 | 96.33 | 74.63 | 97.70 | 87.95 | 99.60 | 93.27 | 55.33 | 93.07 | 85.30 |
| #84 | DoubleUNet | 75.84 | 91.52 | 86.74 | 88.49 | 90.64 | 91.89 | 88.57 | 93.41 | 89.49 | 98.85 | 89.16 | 86.95 | 93.83 | 79.25 | 98.92 | 87.79 | 99.83 | 2.59 | 78.53 | 90.57 | 85.09 |
| #85 | Zig-RiR | 68.03 | 87.84 | 80.27 | 91.56 | 90.83 | 85.47 | 86.16 | 90.07 | 77.78 | 97.27 | 84.02 | 75.58 | 95.88 | 75.69 | 98.26 | 84.80 | 88.67 | 95.15 | 53.31 | 93.83 | 85.02 |
| #86 | CFM-UNet | 71.96 | 90.19 | 85.33 | 91.46 | 92.15 | 87.11 | 83.95 | 86.33 | 36.96 | 97.80 | 85.38 | 84.07 | 95.49 | 79.99 | 98.24 | 84.11 | 91.19 | 94.90 | 63.14 | 94.19 | 84.70 |
| #87 | BEFUnet | 76.32 | 90.28 | 80.46 | 90.52 | 93.38 | 85.10 | 80.11 | 90.01 | 40.56 | 98.08 | 89.91 | 86.97 | 92.71 | 80.48 | 98.55 | 95.95 | 96.12 | 89.62 | 66.11 | 0.36 | 82.74 |
| #88 | SimpleUNet | 68.77 | 88.70 | 78.21 | 90.17 | 90.17 | 82.33 | 87.66 | 93.98 | 89.03 | 98.08 | 89.11 | 80.11 | 95.76 | 79.63 | 98.40 | 86.21 | 94.87 | 93.12 | 66.11 | 93.42 | 82.04 |
| #89 | UNetV2 | 71.75 | 90.24 | 84.40 | 91.34 | 92.61 | 88.80 | 85.26 | 79.63 | 40.56 | 97.98 | 83.17 | 83.09 | 95.68 | 78.08 | 97.63 | 81.40 | 74.87 | 93.12 | 37.74 | 93.42 | 82.04 |
| #90 | VMUNet | 80.43 | 61.42 | 88.05 | 92.22 | 93.76 | 93.12 | 88.59 | 93.03 | 45.73 | 98.22 | 72.61 | 86.21 | 39.56 | 81.18 | 98.70 | 84.52 | 98.08 | 66.48 | 73.83 | 92.85 | 81.44 |
| #91 | Swin-umambaD | 77.14 | 91.17 | 87.84 | 91.38 | 84.26 | 92.18 | 86.69 | 86.40 | 40.02 | 98.58 | 68.90 | 85.79 | 56.68 | 80.65 | 98.92 | 88.57 | 98.79 | 96.08 | 55.71 | 92.19 | 81.44 |
| #92 | CFPNet-M | 77.75 | 92.45 | 87.81 | 92.41 | 92.93 | 88.68 | 90.08 | 94.99 | 90.64 | 98.44 | 91.26 | 87.67 | 96.44 | 81.24 | 98.92 | 89.74 | 97.59 | 0.00 | 68.54 | 0.95 | 80.93 |
| #93 | VMUNetV2 | 75.26 | 92.02 | 87.79 | 92.28 | 90.76 | 92.47 | 77.20 | 84.49 | 38.12 | 97.28 | 90.94 | 79.72 | 95.88 | 80.84 | 97.38 | 79.00 | 97.73 | 62.00 | 70.65 | 15.91 | 80.92 |
| #94 | UltraLight-VM-UNet | 69.19 | 84.47 | 73.63 | 89.57 | 91.82 | 88.80 | 81.68 | 77.06 | 77.14 | 83.87 | 31.82 | 83.67 | 93.59 | 73.50 | 98.38 | 78.39 | 91.14 | 81.18 | 44.46 | 87.67 | 78.97 |
| #95 | H-vmunet | 75.30 | 89.50 | 81.21 | 91.40 | 91.03 | 85.54 | 79.19 | 82.60 | 28.14 | 98.11 | 83.06 | 82.04 | 93.59 | 76.51 | 97.79 | 10.12 | 92.07 | 94.39 | 55.03 | 92.30 | 78.94 |
| #96 | CSWin-UNet | 70.25 | 90.31 | 83.26 | 91.24 | 92.29 | 85.73 | 85.10 | 60.73 | 53.09 | 97.92 | 80.56 | 79.77 | 92.98 | 77.71 | 96.55 | 17.59 | 93.03 | 91.18 | 58.03 | 89.81 | 78.51 |
| #97 | SwinUnet | 65.31 | 84.35 | 42.48 | 91.73 | 92.38 | 80.35 | 83.12 | 88.35 | 44.17 | 98.27 | 77.57 | 79.65 | 88.88 | 77.07 | 97.40 | 78.30 | 92.13 | 66.51 | 50.99 | 91.16 | 78.51 |
| #98 | Polyp-PVT | 82.42 | 92.89 | 89.22 | 92.56 | 94.48 | 91.69 | 92.27 | 74.25 | 3.34 | 97.27 | 68.90 | 76.39 | 67.69 | 81.82 | 98.78 | 0.34 | 98.63 | 95.03 | 73.11 | 89.09 | 78.01 |
| #99 | MALUNet | 72.41 | 87.22 | 82.48 | 90.89 | 89.96 | 83.96 | 74.23 | 28.82 | 37.96 | 97.90 | 79.53 | 82.81 | 95.58 | 75.40 | 97.12 | 82.07 | 84.98 | 66.05 | 42.84 | 90.20 | 77.12 |
| #100 | MambaUnet | 30.36 | 91.58 | 68.31 | 89.92 | 59.72 | 92.52 | 78.36 | 88.42 | 36.79 | 98.40 | 70.58 | 88.51 | 6.88 | 79.96 | 74.30 | 88.68 | 98.60 | 65.97 | 74.46 | 89.75 | 73.60 |

Table 25: Average performance of 100 u-shape medical image segmentation networks with HD95. Baseline U-Net is highlighted (gray background).

| Rank | Network | Ultrasound | | | Dermoscopy | | Endoscopy | | Fundus | | Histopathology | | | Nuclear | X-Ray | | | MRI | | CT | OCT | Avg |
|---|---|---|---|---|---|---|---|---|---|---|---|---|---|---|---|---|---|---|---|---|---|---|
| | | BUSI | BUSBRA | TNSCUI | ISIC2018 | SkinCancer | Kvasir | Robotool | CHASE | DRIVE | DSB2018 | Glas | Monusac | Cell | Covidquex | Montgomery | DCA | ACDC | Promise | Synapse | Cystoidfluid | |
| #1 | FAT-Net | 16.89 | 3.87 | 5.90 | 4.11 | 4.63 | 9.88 | 7.02 | 1.60 | 5.52 | 1.08 | 8.04 | 7.51 | 3.03 | 22.36 | 0.04 | 5.04 | 1.53 | 1.19 | 17.71 | 2.04 | 6.45 |
| #2 | Swin-umamba | 18.98 | 4.11 | 6.79 | 4.05 | 2.79 | 7.22 | 5.15 | 0.96 | 3.16 | 1.20 | 9.78 | 9.28 | 4.22 | 24.75 | 0.28 | 7.02 | 2.15 | 1.30 | 17.32 | 2.32 | 6.64 |
| #3 | RWKV-UNet | 16.56 | 3.62 | 6.21 | 3.68 | 2.53 | 10.51 | 6.53 | 1.34 | 3.64 | 1.14 | 6.15 | 5.33 | 3.92 | 24.68 | 0.00 | 7.15 | 1.16 | 0.96 | 26.85 | 2.29 | 6.71 |
| #4 | AURA-Net | 18.72 | 4.78 | 7.33 | 4.29 | 3.48 | 9.31 | 7.14 | 0.90 | 4.28 | 1.25 | 7.23 | 8.80 | 3.52 | 23.96 | 0.10 | 7.36 | 1.11 | 1.09 | 17.21 | 4.48 | 6.82 |
| #5 | CE-Net | 19.30 | 3.81 | 7.42 | 3.94 | 4.35 | 10.17 | 5.68 | 1.96 | 5.28 | 1.21 | 7.77 | 9.97 | 3.92 | 23.58 | 0.11 | 5.99 | 1.97 | 1.20 | 15.64 | 3.94 | 6.86 |
| #6 | TA-Net | 19.30 | 4.16 | 6.25 | 3.92 | 5.86 | 7.82 | 5.94 | 1.86 | 5.38 | 1.13 | 8.45 | 6.43 | 4.49 | 24.30 | 0.78 | 5.90 | 2.43 | 1.10 | 19.28 | 2.79 | 6.88 |
| #7 | MEGANet | 17.41 | 4.02 | 6.17 | 3.76 | 4.66 | 10.64 | 2.46 | 1.34 | 6.80 | 1.20 | 8.11 | 8.77 | 3.55 | 24.03 | 0.00 | 6.21 | 1.60 | 1.10 | 23.60 | 3.94 | 6.97 |
| #8 | HiFormer | 24.12 | 3.79 | 6.22 | 3.61 | 3.00 | 8.34 | 9.52 | 2.27 | 5.69 | 1.18 | 7.73 | 7.84 | 3.64 | 22.02 | 0.09 | 7.11 | 1.16 | 1.55 | 17.96 | 5.73 | 7.13 |
| #9 | TransResUNet | 21.50 | 3.84 | 7.47 | 4.51 | 5.34 | 9.81 | 8.30 | 0.70 | 3.42 | 1.18 | 8.35 | 7.67 | 4.07 | 25.10 | 0.05 | 6.11 | 2.35 | 1.26 | 19.97 | 3.37 | 7.22 |
| #10 | DA-TransUNet | 17.18 | 3.94 | 6.99 | 4.50 | 2.72 | 11.15 | 13.67 | 1.13 | 3.60 | 1.18 | 6.99 | 10.01 | 3.47 | 24.07 | 0.04 | 4.87 | 1.10 | 1.38 | 24.59 | 3.17 | 7.29 |
| #11 | EMCAD | 19.50 | 5.19 | 11.39 | 4.44 | 2.27 | 10.42 | 27.31 | 3.32 | 6.54 | 1.11 | 6.55 | 8.80 | 3.08 | 24.70 | 0.45 | 6.45 | 2.12 | 1.32 | 0.00 | 2.74 | 7.39 |
| #12 | MSLAU-Net | 22.67 | 4.95 | 6.78 | 4.91 | 2.17 | 11.24 | 9.78 | 2.23 | 6.71 | 1.24 | 7.02 | 8.74 | 3.64 | 29.43 | 0.48 | 6.68 | 1.73 | 1.56 | 13.54 | 2.70 | 7.41 |
| #13 | UTANet | 21.77 | 4.04 | 8.36 | 5.12 | 3.49 | 10.10 | 7.64 | 0.82 | 3.43 | 1.62 | 7.34 | 8.05 | 3.36 | 25.25 | 0.09 | 5.73 | 1.76 | 1.27 | 30.90 | 3.47 | 7.68 |
| #14 | MADGNet | 18.07 | 3.76 | 6.63 | 4.51 | 11.36 | 9.78 | 8.85 | 1.71 | 5.33 | 1.19 | 8.57 | 12.15 | 3.44 | 25.54 | 0.07 | 5.97 | 1.80 | 1.09 | 20.98 | 3.01 | 7.69 |
| #15 | CFFormer | 14.59 | 5.79 | 6.67 | 3.74 | 3.19 | 9.17 | 4.98 | 1.42 | 3.57 | 1.14 | 6.99 | 8.27 | 2.88 | 25.73 | 0.00 | 7.14 | 1.21 | 1.24 | 43.98 | 2.53 | 7.71 |
| #16 | CENet | 19.95 | 5.11 | 7.05 | 4.58 | 5.01 | 9.09 | 18.48 | 1.46 | 4.19 | 1.06 | 7.23 | 9.45 | 4.07 | 24.73 | 0.84 | 8.55 | 1.64 | 1.80 | 18.54 | 2.19 | 7.75 |
| #17 | DS-TransUNet | 25.10 | 5.79 | 9.03 | 5.04 | 2.95 | 9.92 | 5.10 | 1.48 | 5.73 | 1.10 | 9.91 | 8.16 | 4.12 | 27.57 | 0.46 | 8.24 | 2.49 | 1.74 | 18.37 | 3.31 | 7.78 |
| #18 | MERIT | 18.99 | 3.92 | 6.79 | 4.39 | 3.30 | 9.68 | 7.44 | 4.71 | 7.56 | 1.64 | 7.19 | 6.65 | 4.02 | 27.05 | 0.46 | 7.99 | 6.12 | 1.43 | 23.04 | 3.28 | 7.78 |
| #19 | EViT-UNet | 19.65 | 3.71 | 6.61 | 4.34 | 5.12 | 9.37 | 6.72 | 1.36 | 3.71 | 1.20 | 9.56 | 7.64 | 3.32 | 23.37 | 0.44 | 5.91 | 1.88 | 1.17 | 38.69 | 2.78 | 7.83 |
| #20 | FCBFormer | 21.47 | 4.27 | 6.73 | 4.95 | 2.44 | 9.63 | 18.10 | 0.81 | 2.65 | 0.63 | 10.80 | 9.01 | 4.00 | 26.60 | 0.37 | 8.53 | 1.13 | 1.14 | 23.23 | 2.88 | 7.97 |
| #21 | TransAttUnet | 20.35 | 4.77 | 8.36 | 5.11 | 5.98 | 13.31 | 10.18 | 1.14 | 3.16 | 1.16 | 7.07 | 8.09 | 3.27 | 25.09 | 0.07 | 7.48 | 2.65 | 1.46 | 24.85 | 5.99 | 7.98 |
| #22 | Mobile U-ViT | 17.78 | 4.30 | 6.94 | 4.55 | 4.98 | 10.16 | 13.83 | 1.33 | 3.35 | 1.40 | 13.00 | 8.14 | 3.81 | 26.15 | 0.04 | 6.88 | 1.61 | 1.54 | 27.24 | 3.99 | 8.05 |
| #23 | H2Former | 20.26 | 4.27 | 7.46 | 5.11 | 4.14 | 10.53 | 19.20 | 1.49 | 4.06 | 1.19 | 7.84 | 7.49 | 3.83 | 27.94 | 0.04 | 7.04 | 1.27 | 1.27 | 23.01 | 3.74 | 8.06 |
| #24 | TransNorm | 19.25 | 4.55 | 7.14 | 4.84 | 3.10 | 11.84 | 15.84 | 1.45 | 5.12 | 1.32 | 11.87 | 10.01 | 4.34 | 26.09 | 0.19 | 7.29 | 1.89 | 1.23 | 20.03 | 3.94 | 8.07 |
| #25 | CASCADE | 18.48 | 5.03 | 6.40 | 4.85 | 3.47 | 9.46 | 13.27 | 3.71 | 12.58 | 1.21 | 5.48 | 11.38 | 4.47 | 26.53 | 0.42 | 7.89 | 1.98 | 1.18 | 21.91 | 3.14 | 8.14 |
| #26 | G-CASCADE | 18.37 | 4.79 | 6.70 | 4.93 | 3.99 | 10.05 | 18.24 | 3.51 | 10.27 | 1.12 | 8.46 | 10.37 | 3.65 | 25.10 | 0.55 | 8.15 | 1.20 | 1.62 | 20.31 | 2.59 | 8.20 |
| #27 | DDANet | 21.76 | 5.42 | 8.48 | 5.45 | 5.82 | 8.76 | 17.33 | 1.11 | 3.18 | 1.20 | 7.35 | 7.54 | 3.16 | 26.78 | 0.08 | 7.42 | 1.40 | 1.48 | 26.41 | 4.39 | 8.23 |
| #28 | TransUnet | 21.72 | 5.47 | 7.56 | 5.08 | 3.32 | 14.61 | 16.66 | 1.00 | 4.42 | 1.33 | 12.14 | 8.51 | 4.01 | 24.43 | 0.05 | 8.35 | 2.18 | 1.46 | 19.80 | 2.79 | 8.24 |
| #29 | ResU-KAN | 21.19 | 4.29 | 7.63 | 5.10 | 5.04 | 12.07 | 21.24 | 1.60 | 4.67 | 1.31 | 11.66 | 7.64 | 3.92 | 27.50 | 0.23 | 8.53 | 1.81 | 1.29 | 16.46 | 2.86 | 8.30 |
| #30 | MT-UNet | 21.26 | 6.44 | 11.43 | 4.95 | 3.95 | 12.44 | 9.92 | 0.76 | 2.82 | 1.11 | 11.19 | 7.25 | 2.98 | 25.99 | 0.04 | 7.72 | 2.29 | 1.46 | 26.06 | 6.63 | 8.33 |
| #31 | ESKNet | 17.25 | 4.96 | 8.20 | 5.24 | 5.37 | 11.40 | 22.93 | 0.62 | 3.16 | 1.16 | 8.47 | 9.89 | 2.99 | 26.95 | 0.11 | 7.50 | 1.29 | 1.46 | 23.52 | 4.33 | 8.34 |
| #32 | UACANet | 18.12 | 3.64 | 6.03 | 4.17 | 4.98 | 9.69 | 3.11 | 11.37 | 23.06 | 1.95 | 10.71 | 7.53 | 3.63 | 22.87 | 0.30 | 9.83 | 1.55 | 1.18 | 18.11 | 5.53 | 8.37 |
| #33 | LGMSNet | 20.06 | 5.03 | 7.43 | 4.66 | 3.06 | 11.15 | 17.35 | 1.33 | 3.85 | 1.19 | 9.58 | 8.50 | 4.15 | 28.55 | 0.26 | 8.09 | 1.84 | 1.36 | 26.02 | 4.11 | 8.38 |
| #34 | Perspective-Unet | 28.51 | 4.33 | 8.29 | 5.68 | 4.32 | 10.36 | 15.57 | 0.67 | 4.24 | 1.21 | 10.70 | 9.25 | 4.36 | 27.48 | 0.35 | 8.84 | 1.46 | 1.21 | 18.95 | 2.48 | 8.41 |
| #35 | MBSNet | 20.82 | 4.14 | 9.30 | 4.74 | 4.78 | 10.85 | 25.20 | 1.41 | 3.40 | 1.27 | 6.53 | 8.20 | 3.60 | 28.06 | 0.04 | 6.50 | 2.10 | 1.42 | 21.95 | 4.89 | 8.46 |
| #36 | CaraNet | 18.97 | 3.90 | 6.25 | 4.11 | 2.38 | 11.05 | 7.32 | 10.53 | 22.53 | 1.74 | 5.63 | 9.07 | 4.04 | 22.50 | 0.07 | 8.07 | 1.11 | 1.14 | 23.78 | 5.88 | 8.51 |
| #37 | PraNet | 16.21 | 3.60 | 6.44 | 3.99 | 2.26 | 9.43 | 3.67 | 1.61 | 21.85 | 1.38 | 9.30 | 9.04 | 3.49 | 23.17 | 0.13 | 10.82 | 1.10 | 1.09 | 24.49 | 7.18 | 8.51 |
| #38 | CMU-Net | 17.84 | 4.30 | 8.41 | 5.63 | 5.30 | 10.55 | 19.96 | 0.57 | 3.02 | 1.30 | 10.67 | 8.48 | 4.12 | 27.62 | 0.33 | 7.93 | 1.36 | 1.32 | 25.41 | 6.62 | 8.54 |
| #39 | DCSAU-Net | 20.16 | 4.39 | 6.79 | 4.80 | 5.00 | 12.83 | 16.32 | 1.38 | 3.79 | 1.12 | 13.88 | 8.53 | 3.34 | 25.93 | 0.11 | 7.59 | 2.10 | 1.45 | 25.74 | 5.49 | 8.54 |
| #40 | GH-UNet | 25.15 | 4.24 | 9.73 | 5.25 | 3.02 | 11.71 | 20.76 | 1.53 | 4.97 | 1.32 | 8.37 | 10.07 | 4.10 | 29.10 | 0.13 | 7.14 | 1.28 | 1.23 | 19.59 | 5.52 | 8.71 |
| #41 | UTNet | 20.51 | 4.67 | 7.74 | 4.73 | 5.74 | 12.48 | 20.29 | 0.74 | 2.99 | 1.13 | 10.99 | 7.78 | 3.79 | 26.22 | 0.15 | 6.82 | 1.86 | 1.31 | 28.57 | 6.18 | 8.73 |
| #42 | RollingUnet | 30.04 | 4.19 | 11.15 | 5.17 | 5.76 | 11.08 | 18.23 | 1.11 | 3.27 | 1.29 | 7.89 | 9.28 | 2.93 | 28.02 | 0.04 | 7.67 | 1.69 | 1.49 | 24.80 | 2.59 | 8.88 |
| #43 | UCTransNet | 24.75 | 5.11 | 12.06 | 5.21 | 4.86 | 14.43 | 14.61 | 0.73 | 3.56 | 1.23 | 7.88 | 8.56 | 3.50 | 25.04 | 0.12 | 7.20 | 2.19 | 1.33 | 34.25 | 3.31 | 9.00 |
| #44 | DDS-UNet | 19.44 | 4.52 | 8.96 | 4.60 | 4.63 | 11.96 | 25.46 | 1.84 | 5.36 | 2.05 | 11.33 | 9.53 | 4.24 | 26.94 | 0.07 | 7.44 | 2.39 | 1.34 | 25.29 | 2.73 | 9.01 |
| #45 | U-KAN | 22.11 | 5.04 | 10.57 | 5.25 | 2.83 | 11.20 | 19.98 | 1.60 | 4.70 | 1.36 | 12.78 | 10.00 | 3.14 | 30.79 | 0.16 | 8.85 | 1.39 | 1.17 | 24.18 | 4.33 | 9.02 |
| #46 | MedFormer | 22.86 | 5.35 | 7.46 | 5.62 | 4.93 | 11.65 | 22.12 | 0.96 | 3.78 | 1.50 | 13.91 | 8.28 | 4.41 | 28.17 | 0.43 | 7.93 | 1.97 | 1.27 | 22.94 | 5.57 | 9.06 |
| #47 | AC-MambaSeg | 22.32 | 5.00 | 8.42 | 5.04 | 3.92 | 12.54 | 21.27 | 4.28 | 6.53 | 1.39 | 8.78 | 9.00 | 3.93 | 25.60 | 0.26 | 9.96 | 1.62 | 5.57 | 20.00 | 6.11 | 9.08 |
| #48 | MCA-UNet | 22.75 | 4.30 | 7.52 | 4.91 | 3.62 | 10.97 | 18.10 | 1.50 | 4.05 | 1.31 | 12.15 | 9.06 | 3.48 | 28.41 | 0.33 | 8.16 | 2.57 | 1.48 | 33.84 | 3.72 | 9.11 |
| #49 | MMUNet | 20.19 | 4.41 | 7.35 | 4.96 | 5.85 | 9.95 | 15.36 | 1.60 | 4.21 | 1.30 | 14.03 | 9.86 | 3.97 | 28.54 | 0.74 | 8.87 | 2.80 | 1.59 | 33.82 | 3.07 | 9.12 |
| #50 | LV-UNet | 24.22 | 6.08 | 9.65 | 4.50 | 4.30 | 11.56 | 13.24 | 2.44 | 10.78 | 1.24 | 10.29 | 8.54 | 3.95 | 26.97 | 0.13 | 8.03 | 1.69 | 1.50 | 30.67 | 2.73 | 9.13 |
| #51 | ConvFormer | 21.53 | 4.67 | 8.51 | 5.49 | 6.01 | 13.22 | 15.41 | 3.70 | 8.90 | 1.41 | 12.29 | 8.16 | 4.00 | 23.41 | 0.30 | 7.69 | 1.86 | 1.56 | 29.57 | 4.33 | 9.13 |
| #52 | D-TrAttUnet | 31.11 | 4.86 | 13.58 | 5.41 | 4.59 | 14.16 | 17.74 | 1.08 | 3.43 | 1.17 | 12.06 | 8.96 | 3.89 | 26.60 | 0.12 | 7.24 | 2.67 | 1.37 | 20.13 | 2.65 | 9.14 |
| #53 | CSCAUNet | 20.42 | 4.82 | 7.17 | 5.33 | 4.28 | 11.30 | 17.62 | 1.86 | 30.93 | 1.18 | 8.86 | 10.56 | 3.66 | 25.98 | 0.15 | 8.13 | 1.30 | 1.25 | 17.09 | 2.28 | 9.21 |
| #54 | CA-Net | 21.62 | 5.06 | 8.04 | 5.17 | 4.95 | 13.10 | 19.04 | 1.24 | 3.44 | 1.26 | 12.58 | 8.22 | 3.87 | 26.02 | 0.25 | 8.01 | 2.51 | 1.64 | 35.39 | 3.36 | 9.24 |
| #55 | SCUNet++ | 31.97 | 5.94 | 7.54 | 5.52 | 3.42 | 8.26 | 18.83 | 2.17 | 11.95 | 1.34 | 10.27 | 9.24 | 3.74 | 24.64 | 0.42 | 8.29 | 1.90 | 2.34 | 27.02 | 3.72 | 9.43 |
| #56 | AttU-Net | 42.34 | 4.36 | 12.61 | 5.67 | 5.35 | 15.40 | 13.76 | 0.62 | 3.20 | 1.19 | 8.14 | 9.19 | 3.25 | 25.65 | 0.12 | 7.08 | 1.31 | 1.30 | 26.35 | 4.07 | 9.55 |
| #57 | MSRFNet | 28.17 | 6.15 | 9.47 | 5.67 | 6.81 | 11.70 | 15.93 | 1.38 | 6.17 | 1.24 | 11.42 | 8.27 | 4.70 | 29.30 | 0.37 | 7.35 | 2.15 | 1.42 | 30.44 | 3.73 | 9.59 |
| #58 | U-Net | 38.06 | 7.49 | 11.79 | 5.55 | 5.29 | 15.67 | 14.27 | 0.65 | 3.42 | 1.27 | 7.34 | 7.93 | 4.05 | 25.46 | 0.07 | 6.86 | 2.07 | 1.38 | 29.69 | 4.27 | 9.63 |
| #59 | U-RWKV | 25.21 | 5.10 | 13.29 | 5.59 | 5.68 | 13.14 | 18.64 | 1.45 | 3.98 | 1.26 | 10.54 | 8.05 | 3.40 | 28.47 | 0.57 | 7.18 | 1.62 | 1.34 | 36.51 | 4.34 | 9.77 |
| #60 | MDSA-UNet | 23.73 | 6.92 | 8.16 | 5.13 | 5.56 | 12.23 | 15.33 | 4.25 | 8.86 | 1.33 | 12.87 | 9.25 | 4.50 | 28.55 | 0.11 | 8.16 | 2.26 | 1.60 | 33.21 | 3.90 | 9.79 |
| #61 | UNet3+ | 38.48 | 5.04 | 13.28 | 5.37 | 5.30 | 19.82 | 15.65 | 0.70 | 3.42 | 1.44 | 7.61 | 9.65 | 3.65 | 27.75 | 0.04 | 8.01 | 2.41 | 1.59 | 23.28 | 5.26 | 9.89 |
| #62 | ERDUnet | 18.61 | 5.11 | 7.93 | 5.51 | 5.45 | 13.80 | 24.72 | 2.21 | 5.77 | 1.15 | 15.38 | 7.72 | 4.07 | 28.39 | 0.64 | 7.40 | 2.20 | 1.62 | 37.29 | 3.65 | 9.93 |
| #63 | UNet++ | 33.07 | 6.10 | 14.07 | 5.16 | 4.01 | 11.10 | 20.51 | 2.06 | 5.01 | 1.31 | 10.22 | 8.97 | 4.39 | 30.02 | 0.07 | 8.15 | 2.04 | 8.71 | 19.62 | 4.55 | 9.96 |
| #64 | MissFormer | 22.63 | 6.32 | 8.32 | 5.35 | 4.62 | 9.72 | 22.36 | 3.47 | 14.29 | 1.21 | 15.18 | 12.81 | 6.96 | 28.23 | 0.67 | 6.27 | 2.59 | 2.45 | 23.97 | 3.80 | 10.06 |
| #65 | SwinUNETR | 29.15 | 7.11 | 12.56 | 5.26 | 3.81 | 10.71 | 24.42 | 1.74 | 3.43 | 1.69 | 14.91 | 8.84 | 4.24 | 31.13 | 0.54 | 7.51 | 2.20 | 1.55 | 27.44 | 5.20 | 10.17 |
| #66 | DAEFormer | 27.44 | 8.08 | 9.74 | 6.03 | 3.30 | 11.20 | 16.32 | 2.62 | 10.98 | 1.16 | 15.14 | 11.02 | 4.33 | 28.98 | 0.59 | 8.47 | 2.88 | 2.35 | 30.74 | 3.46 | 10.24 |
| #67 | ColonSegNet | 28.71 | 8.82 | 13.17 | 6.90 | 5.28 | 13.31 | 20.14 | 0.77 | 2.82 | 1.25 | 10.20 | 10.52 | 4.20 | 28.03 | 0.15 | 7.72 | 2.60 | 1.83 | 36.12 | 3.76 | 10.28 |
| #68 | MedVKAN | 29.92 | 4.33 | 10.91 | 5.44 | 4.82 | 17.36 | 20.60 | 1.92 | 4.20 | 1.37 | 16.05 | 8.76 | 4.35 | 27.39 | 0.12 | 7.95 | 2.98 | 1.18 | 36.11 | 4.30 | 10.50 |
| #69 | CPCANet | 25.51 | 5.88 | 11.94 | 5.86 | 4.01 | 10.95 | 23.11 | 3.40 | 15.72 | 1.27 | 16.98 | 10.30 | 4.08 | 31.32 | 0.24 | 10.57 | 2.49 | 1.77 | 28.31 | 3.40 | 10.85 |
| #70 | ULite | 30.38 | 6.70 | 13.38 | 5.21 | 3.36 | 18.99 | 20.89 | 2.33 | 3.66 | 1.37 | 13.36 | 10.21 | 4.50 | 29.69 | 0.48 | 8.83 | 3.31 | 1.86 | 32.76 | 6.18 | 10.87 |
| #71 | ScribFormer | 36.50 | 5.41 | 9.33 | 5.41 | 21.91 | 18.28 | 28.84 | 1.24 | 3.75 | 1.18 | 11.70 | 8.50 | 3.53 | 29.23 | 0.13 | 8.36 | 2.10 | 1.57 | 21.76 | 4.42 | 10.91 |
| #72 | CMUNeXt | 68.02 | 4.02 | 8.28 | 5.25 | 2.97 | 13.82 | 16.57 | 1.25 | 3.60 | 1.22 | 7.45 | 9.65 | 3.78 | 27.99 | 0.04 | 7.87 | 1.44 | 1.48 | 32.49 | 2.68 | 10.99 |
| #73 | Tinyunet | 43.96 | 5.73 | 14.57 | 5.53 | 4.21 | 17.08 | 16.25 | 1.72 | 4.15 | 1.33 | 9.85 | 13.27 | 4.17 | 28.45 | 0.05 | 9.09 | 3.50 | 1.75 | 30.15 | 5.72 | 11.03 |
| #74 | MedT | 27.06 | 6.79 | 11.45 | 5.68 | 7.76 | 15.91 | 24.98 | 2.91 | 5.52 | 1.22 | 14.38 | 9.87 | 4.32 | 28.02 | 0.82 | 8.38 | 2.72 | 2.22 | 35.53 | 6.44 | 11.10 |
| #75 | ResUNet++ | 43.65 | 10.23 | 16.86 | 5.41 | 4.88 | 12.89 | 21.54 | 3.02 | 3.51 | 1.98 | 10.83 | 9.63 | 4.11 | 28.62 | 0.13 | 7.49 | 1.32 | 2.86 | 34.44 | 5.90 | 11.46 |
| #76 | DC-UNet | 36.89 | 10.25 | 14.92 | 8.61 | 6.75 | 19.13 | 20.44 | 1.27 | 3.38 | 1.40 | 9.80 | 11.01 | 4.43 | 27.83 | 0.17 | 9.75 | 2.21 | 1.73 | 41.35 | 4.49 | 11.79 |
| #77 | LeViT-UNet | 36.94 | 14.19 | 22.28 | 6.32 | 4.90 | 18.82 | 18.30 | 2.81 | 8.31 | 1.42 | 16.07 | 10.81 | 4.41 | 33.18 | 0.45 | 9.97 | 2.74 | 2.79 | 26.21 | 4.25 | 12.26 |
| #78 | MUCM-Net | 27.64 | 7.47 | 13.01 | 6.19 | 5.99 | 18.46 | 26.90 | 5.24 | 8.47 | 1.97 | 15.25 | 8.50 | 11.93 | 34.50 | 0.70 | 9.89 | 2.74 | 2.27 | 52.20 | 6.77 | 13.30 |
| #79 | TransFuse | 23.31 | 5.24 | 8.59 | 4.11 | 4.62 | 10.07 | 7.85 | 9.71 | 71.79 | 1.37 | 11.24 | 9.55 | 4.33 | 27.99 | 0.07 | 19.61 | 1.18 | 1.15 | 39.85 | 4.70 | 13.32 |
| #80 | MultiResUNet | 33.16 | 5.57 | 13.42 | 5.19 | 4.89 | 15.13 | 16.34 | 34.22 | 42.97 | 1.60 | 8.33 | 9.99 | 4.05 | 27.30 | 0.11 | 20.20 | 1.93 | 1.70 | 28.34 | 4.17 | 13.46 |
| #81 | Swin-umambaD | 24.75 | 6.67 | 7.71 | 6.20 | 9.98 | 10.38 | 18.56 | 7.52 | 37.31 | 1.23 | 38.73 | 8.46 | 22.81 | 27.81 | 0.09 | 7.68 | 1.16 | 14.13 | 20.14 | 4.83 | 13.81 |
| #82 | Zig-RiR | 41.00 | 11.45 | 19.28 | 5.93 | 5.22 | 19.66 | 25.02 | 3.84 | 7.32 | 1.74 | 19.05 | 15.87 | 3.97 | 36.23 | 0.81 | 8.32 | 4.01 | 1.52 | 42.64 | 6.56 | 13.97 |
| #83 | BEFUnet | 22.37 | 6.62 | 11.49 | 6.25 | 2.74 | 14.89 | 23.80 | 4.37 | 66.83 | 1.34 | 24.56 | 10.85 | 3.82 | 30.78 | 0.65 | 9.99 | 3.21 | 6.98 | 25.86 | 8.91 | 14.32 |
| #84 | UNetV2 | 33.95 | 6.69 | 11.07 | 5.93 | 3.63 | 12.34 | 23.30 | 9.66 | 38.93 | 1.92 | 16.91 | 11.88 | 4.59 | 30.49 | 0.68 | 18.59 | 4.76 | 2.19 | 47.61 | 4.52 | 14.48 |
| #85 | VMUNet | 20.64 | 42.88 | 8.02 | 5.02 | 2.99 | 7.43 | 17.98 | 3.19 | 34.95 | 3.51 | 36.55 | 7.53 | 21.70 | 28.25 | 0.16 | 12.75 | 1.96 | 13.54 | 22.59 | 5.57 | 14.86 |
| #86 | UNETR | 38.76 | 12.61 | 26.22 | 6.00 | 3.53 | 18.24 | 31.84 | 2.38 | 3.95 | 1.62 | 21.71 | 11.78 | 4.11 | 35.13 | 0.85 | 9.18 | 4.42 | 2.96 | 58.95 | 7.80 | 15.14 |
| #87 | CFM-UNet | 30.41 | 8.25 | 12.37 | 6.08 | 3.89 | 15.08 | 30.62 | 5.88 | 79.85 | 1.64 | 14.60 | 7.11 | 5.05 | 30.54 | 0.57 | 10.88 | 2.54 | 1.66 | 31.85 | 4.29 | 15.16 |
| #88 | SwinUnet | 40.87 | 12.86 | 37.52 | 5.49 | 3.56 | 19.58 | 22.69 | 4.49 | 12.02 | 1.49 | 21.89 | 15.51 | 5.62 | 29.42 | 0.88 | 18.10 | 2.21 | 12.35 | 31.55 | 7.19 | 15.27 |
| #89 | VMUNetV2 | 29.09 | 5.98 | 7.95 | 5.47 | 5.03 | 9.16 | 42.38 | 7.28 | 27.01 | 4.27 | 8.99 | 14.26 | 9.35 | 27.25 | 1.00 | 18.27 | 1.95 | 2.39 | 23.34 | 58.01 | 15.42 |
| #90 | LFU-Net | 37.84 | 10.74 | 23.27 | 8.83 | 8.08 | 36.33 | 23.44 | 3.52 | 5.18 | 1.20 | 13.75 | 12.38 | 4.53 | 30.18 | 0.72 | 10.52 | 5.32 | 2.89 | 73.23 | 4.40 | 15.82 |
| #91 | UNeXt | 33.94 | 6.90 | 16.11 | 5.65 | 5.45 | 24.93 | 20.46 | 3.27 | 7.12 | 1.26 | 14.82 | 10.11 | 70.74 | 45.19 | 0.10 | 8.56 | 2.24 | 1.51 | 38.40 | 4.83 | 16.08 |
| #92 | MambaUnet | 65.69 | 6.42 | 25.66 | 7.76 | 27.29 | 9.15 | 44.50 | 6.75 | 31.76 | 1.23 | 37.83 | 8.06 | 32.47 | 29.55 | 21.84 | 8.63 | 1.16 | 14.07 | 24.36 | 8.30 | 20.62 |
| #93 | MALUNet | 28.45 | 10.48 | 12.20 | 6.52 | 6.46 | 16.91 | 46.28 | 70.61 | 61.67 | 1.70 | 25.06 | 11.97 | 4.43 | 35.96 | 1.02 | 14.63 | 6.30 | 14.09 | 32.92 | 9.30 | 20.85 |
| #94 | UltraLight-VM-UNet | 31.40 | 13.76 | 17.19 | 8.37 | 4.51 | 26.82 | 43.39 | 8.56 | 85.88 | 1.34 | 10.79 | 9.78 | 5.97 | 33.66 | 2.49 | 16.33 | 3.51 | 9.32 | 62.43 | 12.85 | 20.87 |
| #95 | SimpleUNet | 45.21 | 11.66 | 21.57 | 9.22 | 7.47 | 28.55 | 21.67 | 2.16 | 3.92 | 1.44 | 11.90 | 12.84 | 6.22 | 27.20 | 0.85 | 10.57 | 5.00 | 4.55 | 47.08 | 152.80 | 21.60 |
| #96 | H-vmunet | 26.61 | 8.07 | 13.43 | 5.97 | 4.86 | 17.70 | 38.09 | 10.96 | 60.71 | 1.56 | 15.68 | 11.36 | 4.86 | 31.97 | 0.70 | 150.71 | 3.61 | 1.80 | 29.50 | 6.96 | 23.10 |
| #97 | CSWin-UNet | 35.50 | 7.45 | 11.41 | 6.23 | 3.93 | 14.48 | 22.70 | 26.22 | 68.11 | 1.57 | 22.31 | 16.20 | 15.33 | 33.82 | 1.50 | 136.12 | 3.25 | 3.02 | 26.76 | 15.87 | 23.59 |
| #98 | DoubleUNet | 28.44 | 6.16 | 11.67 | 8.93 | 5.89 | 9.57 | 21.03 | 3.20 | 5.01 | 0.59 | 14.20 | 8.47 | 8.51 | 31.10 | 0.04 | 9.10 | 1.10 | 294.89 | 23.41 | 9.72 | 25.05 |
| #99 | Polyp-PVT | 17.55 | 5.27 | 6.67 | 4.85 | 3.42 | 11.33 | 9.42 | 11.02 | 171.10 | 1.80 | 38.73 | 13.03 | 36.12 | 26.07 | 0.19 | 198.40 | 1.79 | 1.55 | 21.17 | 4.88 | 29.22 |
| #100 | CFPNet-M | 23.57 | 5.24 | 8.66 | 5.04 | 4.00 | 14.95 | 16.03 | 1.73 | 4.20 | 1.18 | 10.23 | 9.17 | 4.33 | 26.73 | 0.14 | 8.15 | 1.72 | 362.04 | 33.21 | 362.04 | 45.12 |

Table 26: GPU-Hours of variants to Achieve Optimal Performance on Different Datasets.

| Network | Ultrasound | | | Dermoscopy | | Endoscopy | | Fundus | | Histopathology | | | Nuclear | X-Ray | | | MRI | | CT | OCT |
|---|---|---|---|---|---|---|---|---|---|---|---|---|---|---|---|---|---|---|---|---|
| | BUSI | BUSBRA | TNSCUI | ISIC2018 | SkinCancer | Kvasir | Robotool | CHASE | DRIVE | DSB2018 | Glas | Monusac | Cell | Covidquex | Montgomery | DCA | ACDC | Promise | Synapse | Cystoidfluid |
| AC-MambaSeg | 1.27 | 4.96 | 5.65 | 4.00 | 0.23 | 2.06 | 1.38 | 1.35 | 0.16 | 0.42 | 0.25 | 0.45 | 1.55 | 3.95 | 0.98 | 0.53 | 3.30 | 3.37 | 11.64 | 3.60 |
| AURA-Net | 0.13 | 0.47 | 1.05 | 1.15 | 0.10 | 1.22 | 0.15 | 0.25 | 0.05 | 0.13 | 0.06 | 0.11 | 0.19 | 0.34 | 0.40 | 0.02 | 1.23 | 0.80 | 2.13 | 0.59 |
| AttU-Net | 0.40 | 1.57 | 2.21 | 0.85 | 0.06 | 1.43 | 0.43 | 0.41 | 0.03 | 0.15 | 0.11 | 0.16 | 0.43 | 0.93 | 0.31 | 0.05 | 1.32 | 0.79 | 1.16 | 1.34 |
| BEFUnet | 0.30 | 0.70 | 1.59 | 1.10 | 0.15 | 0.91 | 0.33 | 0.28 | 0.04 | 0.11 | 0.11 | 0.17 | 0.43 | 0.69 | 0.86 | 0.12 | 0.46 | 0.97 | 2.45 | 0.72 |
| CASCADE | 0.15 | 0.62 | 1.29 | 0.84 | 0.04 | 1.00 | 0.37 | 0.37 | 0.09 | 0.11 | 0.12 | 0.13 | 0.39 | 1.17 | 0.89 | 0.10 | 0.55 | 0.99 | 2.00 | 0.93 |
| CA-Net | 0.40 | 1.18 | 2.74 | 1.90 | 0.08 | 1.28 | 0.39 | 0.31 | 0.06 | 0.33 | 0.11 | 0.24 | 0.52 | 1.57 | 0.87 | 0.12 | 1.59 | 1.11 | 2.74 | 0.76 |
| CENet | 0.49 | 2.47 | 5.30 | 3.96 | 0.43 | 1.35 | 1.21 | 1.01 | 0.12 | 0.89 | 0.38 | 0.30 | 1.13 | 5.32 | 0.91 | 0.29 | 3.57 | 4.07 | 9.23 | 2.83 |
| CE-Net | 0.24 | 0.83 | 2.54 | 2.69 | 0.05 | 0.58 | 0.53 | 0.75 | 0.13 | 0.30 | 0.18 | 0.34 | 0.86 | 1.89 | 0.72 | 0.14 | 3.19 | 0.98 | 5.32 | 2.00 |
| CFFormer | 0.08 | 0.52 | 2.00 | 1.79 | 0.20 | 1.34 | 0.35 | 0.47 | 0.06 | 0.20 | 0.12 | 0.14 | 0.52 | 0.41 | 0.76 | 0.05 | 1.62 | 1.45 | 1.48 | 1.24 |
| CFM-UNet | 0.48 | 1.79 | 3.21 | 1.57 | 0.27 | 1.54 | 0.52 | 0.45 | 0.06 | 0.80 | 0.16 | 0.34 | 0.73 | 1.93 | 0.89 | 0.18 | 2.32 | 1.55 | 4.03 | 1.05 |
| CFPNet-M | 0.46 | 0.61 | 1.69 | 1.20 | 0.08 | 1.26 | 0.44 | 0.39 | 0.09 | 0.29 | 0.11 | 0.30 | 0.26 | 1.25 | 0.78 | 0.12 | 1.67 | 1.19 | 2.87 | 0.49 |
| CMUNeXt | 0.12 | 0.30 | 1.13 | 0.48 | 0.08 | 1.18 | 0.23 | 0.21 | 0.05 | 0.17 | 0.05 | 0.10 | 0.26 | 0.39 | 0.42 | 0.04 | 0.32 | 0.58 | 1.15 | 0.50 |
| CMU-Net | 0.40 | 1.98 | 4.55 | 0.81 | 0.11 | 3.01 | 0.76 | 0.93 | 0.07 | 0.39 | 0.07 | 0.18 | 0.83 | 4.90 | 0.84 | 0.22 | 4.21 | 1.87 | 7.48 | 2.74 |
| CPCANet | 0.65 | 1.90 | 3.05 | 2.36 | 0.17 | 2.46 | 0.86 | 0.73 | 0.12 | 0.12 | 0.22 | 0.28 | 0.51 | 1.10 | 0.71 | 0.19 | 3.12 | 1.64 | 3.92 | 1.01 |
| CSCAUNet | 0.30 | 0.71 | 0.93 | 1.35 | 0.06 | 1.01 | 0.38 | 0.32 | 0.08 | 0.22 | 0.14 | 0.23 | 0.49 | 0.88 | 0.86 | 0.15 | 0.99 | 0.85 | 3.01 | 0.76 |
| CSWin-UNet | 0.40 | 1.45 | 2.73 | 1.85 | 0.19 | 1.34 | 0.36 | 0.05 | 0.08 | 0.75 | 0.14 | 0.23 | 0.56 | 2.11 | 0.93 | 0.06 | 1.11 | 1.15 | 2.23 | 0.87 |
| CaraNet | 0.76 | 2.14 | 3.26 | 2.42 | 0.22 | 1.19 | 0.35 | 0.55 | 0.09 | 0.58 | 0.15 | 0.32 | 0.69 | 1.87 | 0.72 | 0.20 | 2.51 | 0.94 | 5.05 | 1.43 |
| ColonSegNet | 0.47 | 0.59 | 2.10 | 1.76 | 0.17 | 1.26 | 0.47 | 0.43 | 0.06 | 0.16 | 0.07 | 0.16 | 0.44 | 1.17 | 0.57 | 0.07 | 1.12 | 1.21 | 2.95 | 1.16 |
| ConvFormer | 0.91 | 4.65 | 3.33 | 4.24 | 0.48 | 3.09 | 1.19 | 1.22 | 0.16 | 2.21 | 0.40 | 0.46 | 1.24 | 7.00 | 0.82 | 0.36 | 6.55 | 4.79 | 11.14 | 3.37 |
| DAEFormer | 1.29 | 3.28 | 9.74 | 5.54 | 0.47 | 3.18 | 1.12 | 1.09 | 0.13 | 0.61 | 0.34 | 0.78 | 1.98 | 4.23 | 1.01 | 0.36 | 1.91 | 4.37 | 9.72 | 3.01 |
| DA-TransUNet | 0.26 | 1.07 | 3.29 | 1.96 | 0.08 | 1.49 | 0.27 | 0.52 | 0.09 | 0.23 | 0.07 | 0.24 | 0.69 | 2.67 | 0.50 | 0.08 | 2.81 | 0.72 | 4.79 | 1.47 |
| DCSAU-Net | 0.37 | 1.19 | 2.52 | 1.43 | 0.16 | 1.07 | 0.49 | 0.42 | 0.08 | 0.72 | 0.14 | 0.28 | 0.60 | 1.25 | 0.91 | 0.13 | 1.94 | 1.38 | 3.35 | 1.06 |
| DC-UNet | 0.60 | 1.32 | 3.61 | 1.41 | 0.29 | 1.40 | 0.37 | 0.49 | 0.10 | 0.78 | 0.11 | 0.20 | 0.59 | 1.96 | 0.67 | 0.14 | 1.60 | 1.09 | 4.19 | 0.76 |
| DDANet | 0.40 | 0.78 | 2.22 | 1.24 | 0.07 | 1.29 | 0.37 | 0.39 | 0.08 | 0.17 | 0.09 | 0.17 | 0.35 | 1.02 | 0.40 | 0.08 | 1.27 | 0.93 | 2.22 | 0.99 |
| DDS-UNet | 0.47 | 1.62 | 3.05 | 4.59 | 0.33 | 2.80 | 1.08 | 0.83 | 0.12 | 0.28 | 0.20 | 0.47 | 0.70 | 1.64 | 0.54 | 0.19 | 2.81 | 1.78 | 2.45 | 2.34 |
| DS-TransUNet | 0.59 | 2.37 | 6.76 | 3.68 | 0.23 | 2.43 | 1.08 | 0.81 | 0.11 | 0.58 | 0.14 | 0.52 | 1.13 | 2.93 | 0.80 | 0.26 | 4.44 | 2.88 | 9.51 | 2.02 |
| D-TrAttUnet | 0.65 | 2.80 | 4.90 | 1.13 | 0.23 | 1.82 | 0.64 | 0.67 | 0.07 | 0.35 | 0.18 | 0.27 | 0.38 | 3.62 | 0.56 | 0.08 | 3.64 | 2.35 | 6.27 | 1.80 |
| DoubleUNet | 0.76 | 1.76 | 1.99 | 2.89 | 0.17 | 1.60 | 0.55 | 0.51 | 0.13 | 0.14 | 0.19 | 0.35 | 0.63 | 1.60 | 1.09 | 0.24 | 0.68 | 2.47 | 2.47 | 1.18 |
| EMCAD | 0.11 | 0.45 | 1.26 | 0.39 | 0.13 | 0.66 | 0.30 | 0.36 | 0.10 | 0.21 | 0.13 | 0.24 | 0.38 | 1.04 | 0.69 | 0.16 | 0.55 | 0.88 | 2.99 | 0.93 |
| ERDUnet | 0.31 | 0.83 | 1.83 | 1.52 | 0.09 | 1.00 | 0.34 | 0.32 | 0.07 | 0.45 | 0.09 | 0.19 | 0.50 | 1.04 | 0.78 | 0.12 | 1.62 | 1.25 | 2.82 | 0.72 |
| ESKNet | 0.78 | 3.97 | 6.14 | 3.95 | 0.28 | 2.77 | 0.87 | 1.08 | 0.08 | 0.59 | 0.15 | 0.36 | 0.90 | 2.24 | 0.38 | 0.07 | 3.32 | 3.57 | 2.80 | 2.86 |
| EViT-UNet | 0.86 | 1.80 | 4.61 | 1.63 | 0.41 | 1.04 | 0.78 | 0.73 | 0.11 | 0.22 | 0.23 | 0.49 | 0.75 | 1.24 | 0.78 | 0.20 | 4.18 | 1.58 | 7.04 | 1.95 |
| FAT-Net | 0.14 | 0.30 | 0.89 | 1.31 | 0.04 | 0.58 | 0.07 | 0.25 | 0.04 | 0.13 | 0.03 | 0.08 | 0.19 | 0.69 | 0.21 | 0.04 | 0.91 | 0.49 | 2.38 | 0.65 |
| FCBFormer | 1.04 | 2.81 | 3.86 | 2.90 | 0.21 | 1.80 | 0.49 | 0.88 | 0.06 | 0.11 | 0.06 | 0.19 | 0.88 | 1.88 | 0.39 | 0.05 | 1.71 | 1.67 | 8.78 | 2.71 |
| GH-UNet | 0.95 | 3.08 | 6.25 | 4.40 | 0.30 | 4.03 | 0.54 | 1.23 | 0.15 | 0.59 | 0.39 | 0.56 | 1.81 | 1.55 | 0.68 | 0.17 | 3.77 | 3.42 | 9.85 | 3.52 |
| G-CASCADE | 0.27 | 1.81 | 2.66 | 0.50 | 0.22 | 0.45 | 0.54 | 0.45 | 0.10 | 0.46 | 0.17 | 0.28 | 0.71 | 1.24 | 0.88 | 0.18 | 2.11 | 1.44 | 3.91 | 1.12 |
| H2Former | 0.36 | 1.50 | 2.52 | 1.37 | 0.11 | 1.53 | 0.44 | 0.52 | 0.08 | 0.37 | 0.10 | 0.22 | 0.38 | 0.79 | 0.44 | 0.10 | 2.60 | 1.34 | 4.47 | 1.34 |
| H-vmunet | 0.93 | 1.62 | 2.63 | 1.84 | 0.49 | 2.49 | 0.62 | 0.84 | 0.07 | 0.50 | 0.26 | 0.43 | 1.44 | 1.97 | 0.90 | 0.02 | 1.72 | 2.75 | 6.49 | 2.02 |
| HiFormer | 0.12 | 0.25 | 0.65 | 0.23 | 0.13 | 1.02 | 0.27 | 0.28 | 0.08 | 0.11 | 0.05 | 0.15 | 0.28 | 0.59 | 0.33 | 0.07 | 0.82 | 0.57 | 2.08 | 0.67 |
| LFU-Net | 0.52 | 1.88 | 3.25 | 2.27 | 0.20 | 1.25 | 0.24 | 0.34 | 0.07 | 0.62 | 0.12 | 0.24 | 0.60 | 2.12 | 0.90 | 0.15 | 1.39 | 1.06 | 2.45 | 0.91 |
| LGMSNet | 0.26 | 0.53 | 1.23 | 0.61 | 0.04 | 1.28 | 0.22 | 0.25 | 0.03 | 0.12 | 0.06 | 0.06 | 0.11 | 0.49 | 0.31 | 0.03 | 1.15 | 0.64 | 0.67 | 0.59 |
| LV-UNet | 0.14 | 0.33 | 0.87 | 0.14 | 0.08 | 1.06 | 0.13 | 0.15 | 0.09 | 0.35 | 0.04 | 0.08 | 0.15 | 0.55 | 0.89 | 0.07 | 0.59 | 0.42 | 0.38 | 0.32 |
| LeViT-UNet | 0.19 | 0.40 | 1.79 | 1.07 | 0.12 | 1.20 | 0.28 | 0.27 | 0.06 | 0.14 | 0.04 | 0.07 | 0.33 | 0.44 | 0.47 | 0.04 | 0.42 | 0.50 | 2.16 | 0.64 |
| MALUNet | 0.22 | 0.39 | 1.33 | 0.44 | 0.07 | 0.92 | 0.13 | 0.01 | 0.04 | 0.16 | 0.04 | 0.13 | 0.30 | 0.81 | 0.66 | 0.07 | 0.50 | 0.69 | 0.90 | 0.45 |
| MBSNet | 0.24 | 0.51 | 1.22 | 0.87 | 0.07 | 1.24 | 0.25 | 0.26 | 0.05 | 0.10 | 0.05 | 0.08 | 0.23 | 0.36 | 0.31 | 0.03 | 0.78 | 0.61 | 1.39 | 0.63 |
| MCA-UNet | 0.75 | 3.40 | 5.05 | 2.69 | 0.22 | 3.17 | 0.87 | 1.09 | 0.13 | 0.44 | 0.15 | 0.49 | 0.90 | 1.25 | 0.42 | 0.09 | 1.95 | 1.86 | 10.50 | 3.13 |
| MDSA-UNet | 0.72 | 1.67 | 3.72 | 2.52 | 0.31 | 1.75 | 0.48 | 0.57 | 0.11 | 1.09 | 0.18 | 0.45 | 0.87 | 2.38 | 0.94 | 0.18 | 2.87 | 1.48 | 4.89 | 1.50 |
| MEGANet | 0.10 | 0.34 | 1.16 | 1.61 | 0.12 | 1.21 | 0.19 | 0.27 | 0.09 | 0.07 | 0.10 | 0.11 | 0.24 | 0.38 | 0.57 | 0.08 | 0.43 | 0.37 | 1.69 | 0.66 |
| MERIT | 0.65 | 1.77 | 7.08 | 0.83 | 0.22 | 0.47 | 0.55 | 0.30 | 0.08 | 0.03 | 0.19 | 0.18 | 0.39 | 1.42 | 0.34 | 0.15 | 1.37 | 1.32 | 7.47 | 1.14 |
| MADGNet | 0.15 | 1.44 | 3.33 | 3.87 | 0.21 | 2.72 | 0.48 | 0.92 | 0.12 | 0.90 | 0.19 | 0.39 | 0.98 | 2.05 | 0.53 | 0.18 | 1.60 | 1.13 | 5.91 | 2.51 |
| MMUNet | 0.36 | 1.32 | 2.46 | 1.32 | 0.07 | 1.27 | 0.57 | 0.64 | 0.10 | 0.20 | 0.11 | 0.20 | 1.09 | 1.22 | 0.39 | 0.06 | 1.06 | 1.31 | 5.35 | 1.66 |
| MSLAU-Net | 0.23 | 0.46 | 1.34 | 1.85 | 0.10 | 1.24 | 0.40 | 0.45 | 0.08 | 0.15 | 0.12 | 0.28 | 0.56 | 0.74 | 0.15 | 0.06 | 1.44 | 0.57 | 3.65 | 1.11 |
| MSRFNet | 2.23 | 6.89 | 9.72 | 7.91 | 0.69 | 3.76 | 1.61 | 1.30 | 0.15 | 1.31 | 0.44 | 1.00 | 2.42 | 7.90 | 0.94 | 0.51 | 5.37 | 5.30 | 9.41 | 3.39 |
| MT-UNet | 1.76 | 2.67 | 6.79 | 3.97 | 0.39 | 3.81 | 1.34 | 1.20 | 0.14 | 0.97 | 0.29 | 0.70 | 1.74 | 6.62 | 0.68 | 0.18 | 2.08 | 2.90 | 10.77 | 3.52 |
| MUCM-Net | 0.08 | 0.30 | 1.08 | 0.60 | 0.02 | 1.03 | 0.12 | 0.16 | 0.05 | 0.08 | 0.05 | 0.08 | 0.11 | 0.27 | 0.69 | 0.03 | 0.42 | 0.48 | 1.20 | 0.37 |
| MambaUnet | 0.24 | 2.34 | 5.22 | 3.67 | 0.12 | 1.50 | 0.73 | 0.53 | 0.07 | 0.26 | 0.00 | 0.32 | 0.00 | 2.69 | 0.72 | 0.15 | 1.06 | 1.61 | 4.88 | 1.43 |
| MedFormer | 0.33 | 0.60 | 2.00 | 1.22 | 0.15 | 0.86 | 0.30 | 0.46 | 0.06 | 0.09 | 0.14 | 0.20 | 0.32 | 1.04 | 0.41 | 0.06 | 1.43 | 0.62 | 3.69 | 1.18 |
| MedT | 2.28 | 13.57 | 24.42 | 14.98 | 0.99 | 7.18 | 3.21 | 2.42 | 0.25 | 4.82 | 0.45 | 1.52 | 4.64 | 16.15 | 1.10 | 0.85 | 8.42 | 9.65 | 14.42 | 5.88 |
| MedVKAN | 0.26 | 0.91 | 3.67 | 2.26 | 0.20 | 0.99 | 0.52 | 0.47 | 0.10 | 0.74 | 0.16 | 0.18 | 0.71 | 1.81 | 0.71 | 0.16 | 2.29 | 1.47 | 2.70 | 1.12 |
| MissFormer | 0.31 | 1.10 | 2.39 | 1.64 | 0.16 | 1.17 | 0.41 | 0.38 | 0.06 | 0.15 | 0.13 | 0.16 | 0.05 | 1.59 | 0.90 | 0.14 | 1.31 | 1.46 | 3.59 | 0.99 |
| Mobile U-ViT | 0.49 | 0.88 | 1.59 | 1.65 | 0.06 | 1.22 | 0.14 | 0.36 | 0.06 | 0.20 | 0.07 | 0.21 | 0.20 | 0.73 | 0.37 | 0.05 | 0.56 | 0.49 | 3.11 | 0.94 |
| MultiResUNet | 0.24 | 0.48 | 1.83 | 0.84 | 0.06 | 0.99 | 0.38 | 0.20 | 0.03 | 0.15 | 0.03 | 0.16 | 0.16 | 0.73 | 0.37 | 0.05 | 0.56 | 0.49 | 2.01 | 0.91 |
| Perspective-Net | 2.05 | 4.88 | 8.89 | 7.78 | 0.48 | 5.11 | 1.66 | 1.61 | 0.17 | 1.12 | 0.49 | 1.05 | 2.19 | 9.13 | 0.71 | 0.37 | 8.15 | 3.61 | 14.38 | 4.47 |
| Polyp-PVT | 0.38 | 1.24 | 1.38 | 0.54 | 0.13 | 1.11 | 0.43 | 0.39 | 0.09 | 0.50 | 0.00 | 0.26 | 0.53 | 1.01 | 0.89 | 0.07 | 1.49 | 1.47 | 2.71 | 1.02 |
| PraNet | 0.23 | 0.41 | 2.65 | 3.21 | 0.09 | 1.51 | 0.26 | 0.48 | 0.08 | 0.61 | 0.15 | 0.23 | 0.92 | 2.02 | 0.74 | 0.15 | 2.44 | 1.52 | 4.76 | 1.39 |
| RWKV-UNet | 0.36 | 0.61 | 1.72 | 0.48 | 0.13 | 0.50 | 0.39 | 0.52 | 0.10 | 0.38 | 0.11 | 0.31 | 0.80 | 2.10 | 0.78 | 0.18 | 2.55 | 1.57 | 4.40 | 1.32 |
| UNet++ | 0.28 | 0.48 | 0.41 | 0.41 | 0.07 | 0.92 | 0.25 | 0.31 | 0.08 | 0.14 | 0.10 | 0.15 | 0.26 | 0.92 | 0.49 | 0.07 | 0.90 | 0.84 | 2.36 | 0.76 |
| ResUNet++ | 1.05 | 3.91 | 3.62 | 5.10 | 0.44 | 3.33 | 1.16 | 1.13 | 0.13 | 0.42 | 0.12 | 0.62 | 1.99 | 2.37 | 0.75 | 0.21 | 5.50 | 3.00 | 6.73 | 3.19 |
| ResU-KAN | 0.39 | 0.66 | 2.73 | 3.33 | 0.15 | 2.13 | 0.48 | 0.86 | 0.11 | 0.22 | 0.10 | 0.26 | 0.53 | 1.66 | 0.98 | 0.07 | 2.77 | 1.02 | 7.02 | 2.20 |
| RollingUnet | 0.62 | 0.74 | 2.86 | 1.72 | 0.11 | 1.40 | 0.48 | 0.50 | 0.10 | 0.31 | 0.15 | 0.20 | 0.50 | 1.01 | 0.56 | 0.04 | 2.08 | 0.83 | 2.51 | 1.27 |
| SCUNet++ | 0.48 | 0.88 | 3.90 | 2.00 | 0.26 | 3.75 | 0.88 | 0.32 | 0.09 | 0.14 | 0.11 | 0.50 | 0.53 | 1.18 | 2.22 | 0.14 | 1.42 | 2.35 | 2.56 | 0.76 |
| ScribFormer | 0.50 | 1.06 | 2.87 | 3.21 | 0.15 | 1.41 | 0.59 | 0.58 | 0.08 | 0.28 | 0.14 | 0.28 | 0.55 | 1.59 | 0.83 | 0.16 | 2.38 | 1.62 | 4.20 | 1.59 |
| SimpleUNet | 0.08 | 0.15 | 0.28 | 0.23 | 0.03 | 0.88 | 0.11 | 0.12 | 0.04 | 0.35 | 0.04 | 0.07 | 0.10 | 0.27 | 0.71 | 0.04 | 0.31 | 0.21 | 1.05 | 0.00 |
| SwinUNETR | 0.24 | 0.53 | 0.82 | 0.90 | 0.05 | 0.97 | 0.25 | 0.24 | 0.08 | 0.09 | 0.10 | 0.15 | 0.38 | 0.49 | 0.60 | 0.05 | 0.32 | 0.39 | 1.68 | 0.55 |
| SwinUnet | 0.47 | 1.54 | 2.63 | 1.76 | 0.14 | 1.03 | 0.36 | 0.30 | 0.06 | 0.52 | 0.12 | 0.14 | 0.51 | 1.84 | 0.95 | 0.14 | 1.38 | 0.54 | 2.46 | 0.74 |
| Swin-umamba | 0.32 | 1.08 | 1.80 | 0.82 | 0.05 | 1.61 | 0.18 | 0.68 | 0.05 | 0.19 | 0.11 | 0.12 | 0.28 | 2.52 | 0.72 | 0.06 | 0.94 | 1.29 | 6.07 | 1.84 |
| Swin-umambaD | 1.04 | 3.29 | 4.55 | 3.94 | 0.38 | 1.37 | 0.88 | 0.62 | 0.08 | 0.73 | 0.13 | 0.44 | 1.12 | 2.78 | 1.00 | 0.23 | 2.33 | 0.98 | 1.90 | 1.71 |
| TA-Net | 0.15 | 0.58 | 2.51 | 4.87 | 0.07 | 0.76 | 0.19 | 0.73 | 0.12 | 0.34 | 0.26 | 0.33 | 1.04 | 2.55 | 0.62 | 0.14 | 3.23 | 1.01 | 5.39 | 2.02 |
| Tinyunet | 0.22 | 0.40 | 2.05 | 1.39 | 0.10 | 1.03 | 0.26 | 0.31 | 0.07 | 0.08 | 0.08 | 0.16 | 0.33 | 0.76 | 0.70 | 0.10 | 1.33 | 0.42 | 2.33 | 0.74 |
| TransAttUnet | 0.91 | 1.36 | 3.14 | 2.42 | 0.13 | 1.46 | 0.77 | 0.58 | 0.08 | 0.69 | 0.20 | 0.29 | 0.69 | 2.41 | 0.91 | 0.23 | 2.28 | 1.86 | 4.11 | 1.56 |
| TransFuse | 0.22 | 1.20 | 1.28 | 1.72 | 0.13 | 0.57 | 0.36 | 0.24 | 0.07 | 0.48 | 0.12 | 0.21 | 0.38 | 0.54 | 0.84 | 0.11 | 0.44 | 0.68 | 0.77 | 0.58 |
| TransNorm | 0.38 | 1.04 | 3.90 | 2.49 | 0.07 | 1.43 | 0.46 | 0.59 | 0.08 | 0.08 | 0.14 | 0.48 | 0.34 | 2.31 | 0.39 | 0.10 | 3.02 | 1.43 | 3.89 | 1.04 |
| TransResUNet | 0.29 | 1.73 | 2.27 | 2.45 | 0.27 | 1.66 | 0.34 | 0.55 | 0.07 | 0.75 | 0.05 | 0.14 | 0.34 | 0.85 | 0.89 | 0.17 | 2.61 | 1.21 | 4.46 | 1.45 |
| TransUnet | 0.19 | 0.82 | 2.00 | 2.46 | 0.03 | 1.23 | 0.31 | 0.48 | 0.06 | 0.18 | 0.07 | 0.13 | 0.50 | 0.93 | 0.25 | 0.05 | 0.81 | 0.99 | 4.14 | 1.19 |
| UACANet | 0.53 | 2.40 | 3.12 | 2.26 | 0.29 | 1.52 | 0.37 | 0.42 | 0.08 | 0.75 | 0.10 | 0.28 | 0.73 | 1.80 | 0.79 | 0.16 | 2.38 | 1.25 | 2.81 | 1.17 |
| UCTransNet | 0.84 | 1.33 | 2.33 | 3.73 | 0.14 | 2.07 | 0.62 | 0.74 | 0.07 | 0.22 | 0.06 | 0.15 | 0.42 | 1.97 | 0.61 | 0.11 | 1.28 | 1.95 | 2.20 | 2.02 |
| ULite | 0.04 | 0.16 | 0.37 | 0.23 | 0.03 | 0.85 | 0.12 | 0.13 | 0.07 | 0.22 | 0.06 | 0.10 | 0.15 | 0.18 | 0.81 | 0.08 | 0.43 | 0.30 | 0.75 | 0.26 |
| UNETR | 0.56 | 1.70 | 2.81 | 1.42 | 0.21 | 1.24 | 0.43 | 0.33 | 0.07 | 0.35 | 0.10 | 0.12 | 0.62 | 1.29 | 0.74 | 0.15 | 1.15 | 1.31 | 2.01 | 0.95 |
| UNeXt | 0.10 | 0.25 | 0.31 | 0.26 | 0.07 | 0.95 | 0.09 | 0.15 | 0.07 | 0.18 | 0.05 | 0.08 | 0.03 | 0.22 | 0.72 | 0.03 | 0.13 | 0.25 | 1.06 | 0.25 |
| UNet3+ | 1.64 | 6.17 | 10.78 | 8.36 | 0.84 | 6.79 | 2.21 | 2.01 | 0.13 | 0.62 | 0.48 | 0.65 | 1.39 | 2.40 | 1.02 | 0.32 | 5.95 | 4.44 | 15.11 | 5.92 |
| UNetV2 | 0.31 | 1.49 | 2.41 | 0.93 | 0.19 | 1.02 | 0.40 | 0.34 | 0.05 | 0.70 | 0.13 | 0.35 | 0.54 | 1.33 | 0.85 | 0.11 | 1.67 | 1.03 | 2.97 | 0.85 |
| UTANet | 0.30 | 0.86 | 1.72 | 1.43 | 0.11 | 0.68 | 0.28 | 0.39 | 0.08 | 0.12 | 0.08 | 0.22 | 0.53 | 1.36 | 0.61 | 0.09 | 1.25 | 0.84 | 1.12 | 1.00 |
| UTNet | 0.27 | 0.57 | 2.14 | 1.26 | 0.14 | 0.78 | 0.34 | 0.42 | 0.07 | 0.22 | 0.09 | 0.22 | 0.24 | 1.12 | 0.43 | 0.08 | 0.68 | 1.05 | 3.51 | 1.09 |
| U-KAN | 0.17 | 0.33 | 1.61 | 0.98 | 0.15 | 0.83 | 0.19 | 0.37 | 0.07 | 0.10 | 0.04 | 0.12 | 0.31 | 0.63 | 0.18 | 0.06 | 1.82 | 0.46 | 3.15 | 0.97 |
| U-Net | 0.51 | 0.65 | 1.97 | 0.94 | 0.24 | 1.01 | 0.45 | 0.42 | 0.03 | 0.14 | 0.06 | 0.13 | 0.29 | 0.85 | 0.33 | 0.10 | 1.81 | 1.34 | 3.16 | 1.21 |
| U-RWKV | 0.14 | 0.46 | 1.11 | 0.76 | 0.07 | 1.12 | 0.20 | 0.25 | 0.02 | 0.15 | 0.03 | 0.10 | 0.24 | 0.28 | 0.24 | 0.07 | 0.68 | 0.37 | 0.58 | 0.51 |
| UltraLight-VM-UNet | 0.22 | 0.86 | 1.60 | 0.86 | 0.09 | 0.96 | 0.22 | 0.19 | 0.06 | 0.28 | 0.09 | 0.10 | 0.32 | 0.99 | 0.86 | 0.09 | 0.51 | 0.57 | 0.91 | 0.44 |
| VMUNet | 0.72 | 1.96 | 7.12 | 4.47 | 0.39 | 0.76 | 0.96 | 0.75 | 0.12 | 0.73 | 0.05 | 0.57 | 0.44 | 1.06 | 0.98 | 0.32 | 2.19 | 0.05 | 6.62 | 1.95 |
| VMUNetV2 | 0.65 | 1.97 | 3.43 | 2.12 | 0.25 | 1.04 | 0.56 | 0.39 | 0.06 | 0.65 | 0.15 | 0.30 | 0.58 | 1.36 | 0.88 | 0.17 | 1.40 | 1.50 | 3.63 | 0.05 |
| Zig-RiR | 0.34 | 0.65 | 1.70 | 0.53 | 0.14 | 1.33 | 0.21 | 0.41 | 0.05 | 0.19 | 0.11 | 0.08 | 0.36 | 0.49 | 0.41 | 0.15 | 2.12 | 1.04 | 1.33 | 1.05 |

Table 27: Average zeroshot performance of 100 u-shape medical image segmentation networks with IoU. Source → Target. Baseline U-Net is highlighted (gray background), and statistical significance of p-value is highlighted: $p<0.0001$ , $p<0.001$ , $p<0.05$ , $p\leq0.05$ , and $P>0.05$ (Not significant) .

| Rank | Network | Ultrasound BUSI → BUS | BUSBRA → BUS | TNSCUI → TUCC | Endoscopy Kvasir → CVC300 | Kvasir → CVC-ClinicDB | Dermoscopy ISIC2018 → PH2 | Fundus CHASE → Stare | X-Ray Montgomery → NIH-test | Histopathology Monusac → Tnbcnuclei | Avg |
|---|---|---|---|---|---|---|---|---|---|---|---|
| #1 | RWKV-UNet | 80.73 | 84.73 | 63.42 | 82.14 | 75.50 | 86.00 | 61.29 | 82.41 | 38.96 | 70.94 |
| #2 | DS-TransUNet | 78.70 | 84.44 | 61.95 | 80.58 | 77.38 | 84.05 | 57.65 | 76.21 | 44.20 | 70.07 |
| #3 | TransResUNet | 81.27 | 85.62 | 62.36 | 79.13 | 76.42 | 84.20 | 50.77 | 87.69 | 35.19 | 69.47 |
| #4 | CENet | 82.70 | 86.29 | 62.15 | 78.57 | 73.14 | 84.10 | 53.32 | 71.06 | 48.03 | 69.31 |
| #5 | MADGNet | 79.78 | 84.47 | 61.27 | 81.24 | 76.33 | 85.08 | 48.94 | 83.67 | 43.98 | 69.08 |
| #6 | G-CASCADE | 82.61 | 85.27 | 65.10 | 77.89 | 74.77 | 85.96 | 48.80 | 78.05 | 46.18 | 68.95 |
| #7 | DA-TransUNet | 82.32 | 84.15 | 61.81 | 71.24 | 70.64 | 82.81 | 56.44 | 85.87 | 44.53 | 68.65 |
| #8 | TA-Net | 81.02 | 84.38 | 62.89 | 81.24 | 72.03 | 84.15 | 45.67 | 77.53 | 50.74 | 68.58 |
| #9 | MEGANet | 81.24 | 86.62 | 64.37 | 78.67 | 75.64 | 84.99 | 45.75 | 88.19 | 38.86 | 68.39 |
| #10 | EMCAD | 82.83 | 84.95 | 61.20 | 79.93 | 76.56 | 83.63 | 49.19 | 73.37 | 46.41 | 68.24 |
| #11 | CFFormer | 81.12 | 79.53 | 62.34 | 78.56 | 76.18 | 85.04 | 52.38 | 80.49 | 33.32 | 68.09 |
| #12 | CASCADE | 82.11 | 85.96 | 63.48 | 80.57 | 77.19 | 84.63 | 46.97 | 77.07 | 37.76 | 67.91 |
| #13 | Swin-umamba | 82.91 | 85.02 | 64.00 | 79.96 | 73.57 | 84.62 | 54.52 | 81.96 | 38.39 | 67.86 |
| #14 | AC-MambaSeg | 79.84 | 82.31 | 63.01 | 72.82 | 70.73 | 84.34 | 52.78 | 78.58 | 40.53 | 67.61 |
| #15 | MCA-UNet | 81.88 | 85.28 | 63.78 | 72.00 | 69.49 | 85.08 | 56.59 | 76.85 | 32.40 | 67.60 |
| #16 | MSLAU-Net | 80.96 | 84.90 | 66.15 | 78.24 | 71.49 | 86.52 | 46.51 | 78.69 | 31.46 | 67.20 |
| #17 | AURA-Net | 78.40 | 84.67 | 61.52 | 79.18 | 73.50 | 83.91 | 51.22 | 81.54 | 31.25 | 67.10 |
| #18 | CSCAUNet | 78.37 | 84.65 | 63.40 | 77.50 | 70.97 | 84.54 | 48.61 | 73.47 | 36.58 | 67.09 |
| #19 | VMUNet | 81.18 | 82.96 | 62.31 | 76.54 | 74.15 | 84.27 | 52.96 | 77.96 | 44.40 | 66.79 |
| #20 | FAT-Net | 77.91 | 84.86 | 60.23 | 74.22 | 71.64 | 84.22 | 51.38 | 82.19 | 29.64 | 66.60 |
| #21 | FCBFormer | 79.14 | 86.10 | 63.52 | 74.82 | 73.16 | 83.85 | 57.20 | 74.55 | 42.88 | 66.46 |
| #22 | Perspective-Unet | 76.17 | 84.69 | 61.76 | 77.38 | 72.13 | 83.57 | 45.92 | 77.21 | 36.12 | 66.33 |
| #23 | Swin-umambaD | 80.68 | 84.39 | 62.69 | 78.38 | 76.83 | 84.39 | 39.76 | 83.13 | 38.59 | 66.33 |
| #24 | LGMSNet | 78.45 | 83.34 | 65.16 | 74.57 | 75.01 | 85.00 | 46.58 | 76.02 | 30.87 | 66.06 |
| #25 | CE-Net | 80.34 | 83.77 | 62.49 | 76.99 | 70.87 | 84.02 | 43.52 | 80.83 | 32.30 | 65.94 |
| #26 | LV-UNet | 74.19 | 82.72 | 63.61 | 78.86 | 69.42 | 84.99 | 49.02 | 78.82 | 28.67 | 65.83 |
| #27 | GH-UNet | 77.25 | 83.26 | 62.49 | 77.22 | 70.59 | 85.01 | 45.35 | 64.56 | 43.52 | 65.48 |
| #28 | UTANet | 77.05 | 84.62 | 61.59 | 80.78 | 70.99 | 84.59 | 45.20 | 75.38 | 23.55 | 65.29 |
| #29 | U-KAN | 77.25 | 83.97 | 62.65 | 72.87 | 69.77 | 83.29 | 46.24 | 69.99 | 38.33 | 65.28 |
| #30 | MMUNet | 76.53 | 81.53 | 61.69 | 74.35 | 68.61 | 85.66 | 48.19 | 70.73 | 37.11 | 65.19 |
| #31 | MBSNet | 76.37 | 83.08 | 62.45 | 78.22 | 72.68 | 84.65 | 45.63 | 64.98 | 28.80 | 65.15 |
| #32 | CA-Net | 75.84 | 83.01 | 62.74 | 73.61 | 67.49 | 82.44 | 49.38 | 63.74 | 39.98 | 65.01 |
| #33 | DDS-UNet | 77.36 | 83.49 | 63.09 | 75.43 | 71.24 | 82.18 | 45.01 | 74.29 | 38.82 | 65.01 |
| #34 | HiFormer | 72.21 | 84.20 | 63.94 | 78.91 | 74.53 | 82.75 | 50.25 | 81.65 | 32.41 | 64.95 |
| #35 | ResU-KAN | 72.73 | 83.11 | 62.79 | 71.83 | 72.61 | 84.42 | 47.60 | 73.55 | 31.51 | 64.82 |
| #36 | PraNet | 82.13 | 84.19 | 63.59 | 83.31 | 77.39 | 84.30 | 24.25 | 84.58 | 38.53 | 64.81 |
| #37 | SwinUNETR | 74.70 | 84.18 | 59.91 | 63.58 | 68.44 | 83.73 | 57.55 | 74.66 | 32.98 | 64.70 |
| #38 | H2Former | 78.22 | 83.81 | 64.48 | 74.03 | 70.50 | 85.39 | 47.19 | 63.56 | 28.62 | 64.68 |
| #39 | EViT-UNet | 79.07 | 84.95 | 65.83 | 81.11 | 71.49 | 83.79 | 54.16 | 43.53 | 30.14 | 64.66 |
| #40 | TransFuse | 80.18 | 83.93 | 61.74 | 79.39 | 72.54 | 84.84 | 35.95 | 82.58 | 32.58 | 64.64 |
| #41 | CaraNet | 81.32 | 84.87 | 64.61 | 79.12 | 73.86 | 84.85 | 26.94 | 86.22 | 33.32 | 64.39 |
| #42 | DAEFormer | 72.80 | 84.02 | 63.72 | 68.10 | 66.69 | 82.81 | 52.57 | 65.70 | 45.16 | 64.36 |
| #43 | VMUNetV2 | 72.72 | 84.35 | 63.70 | 78.80 | 74.66 | 84.65 | 36.68 | 73.96 | 36.84 | 64.13 |
| #44 | MissFormer | 72.82 | 84.76 | 61.39 | 71.99 | 68.27 | 83.38 | 52.27 | 70.49 | 40.61 | 63.99 |
| #45 | Mobile U-ViT | 77.80 | 84.42 | 60.34 | 74.94 | 69.72 | 83.87 | 46.86 | 72.77 | 31.85 | 63.92 |
| #46 | UTNet | 77.64 | 83.78 | 60.04 | 72.66 | 68.39 | 84.14 | 45.38 | 68.83 | 32.14 | 63.91 |
| #47 | DCSAU-Net | 70.88 | 84.08 | 61.95 | 74.78 | 68.86 | 82.77 | 43.94 | 62.76 | 41.48 | 63.81 |
| #48 | CPCANet | 79.14 | 84.65 | 60.52 | 76.92 | 68.33 | 84.30 | 45.44 | 62.21 | 37.67 | 63.68 |
| #49 | ESKNet | 76.99 | 84.39 | 63.07 | 77.23 | 73.85 | 83.81 | 40.79 | 71.68 | 36.36 | 63.64 |
| #50 | D-TrAttUnet | 71.01 | 79.89 | 59.21 | 74.85 | 69.36 | 84.46 | 47.43 | 71.20 | 32.90 | 63.62 |
| #51 | U-Net++ | 76.68 | 85.46 | 58.14 | 71.45 | 66.04 | 84.62 | 43.62 | 61.52 | 46.44 | 63.56 |
| #52 | UCTransNet | 68.11 | 80.41 | 60.58 | 75.06 | 68.57 | 83.54 | 45.90 | 73.57 | 34.93 | 63.51 |
| #53 | TransAttUnet | 74.80 | 83.73 | 61.24 | 73.98 | 71.26 | 83.87 | 50.12 | 55.80 | 28.16 | 63.47 |
| #54 | CMU-Net | 75.72 | 83.52 | 61.57 | 75.09 | 70.11 | 83.15 | 46.09 | 73.47 | 38.16 | 63.47 |
| #55 | SCUNet++ | 73.77 | 82.73 | 59.35 | 75.67 | 67.82 | 82.74 | 47.08 | 68.66 | 35.38 | 63.43 |
| #56 | UNet3+ | 70.13 | 83.70 | 60.40 | 71.98 | 69.52 | 83.27 | 49.76 | 73.74 | 29.88 | 63.33 |
| #57 | RollingUnet | 72.58 | 83.70 | 62.34 | 77.36 | 69.26 | 82.97 | 46.98 | 64.60 | 22.75 | 63.08 |
| #58 | MSRFNet | 69.01 | 81.90 | 58.35 | 72.59 | 73.48 | 83.15 | 55.17 | 58.97 | 21.74 | 62.99 |
| #59 | UACANet | 80.81 | 84.82 | 62.47 | 81.72 | 74.99 | 85.12 | 25.10 | 81.88 | 26.84 | 62.84 |
| #60 | Polyp-PVT | 81.02 | 85.65 | 65.23 | 73.87 | 75.17 | 83.90 | 28.02 | 77.00 | 32.42 | 62.80 |
| #61 | MERIT | 80.46 | 83.26 | 66.00 | 81.39 | 74.77 | 85.94 | 38.06 | 73.80 | 43.43 | 62.80 |
| #62 | TransNorm | 82.11 | 84.33 | 63.81 | 69.78 | 68.51 | 83.03 | 53.31 | 82.90 | 39.03 | 62.72 |
| #63 | DDANet | 71.75 | 80.45 | 60.11 | 78.78 | 70.72 | 83.43 | 47.81 | 61.04 | 22.99 | 62.65 |
| #64 | ERDUnet | 76.56 | 84.30 | 62.23 | 72.47 | 57.91 | 83.98 | 52.97 | 60.27 | 34.34 | 62.39 |
| #65 | MedFormer | 77.54 | 85.05 | 63.57 | 73.91 | 69.55 | 84.18 | 43.98 | 52.97 | 31.26 | 62.37 |
| #66 | U-Net | 72.44 | 81.37 | 60.50 | 70.33 | 69.87 | 84.00 | 46.77 | 71.33 | 26.05 | 62.23 |
| #67 | TransUnet | 80.27 | 83.90 | 60.54 | 72.24 | 68.20 | 83.39 | 52.10 | 83.03 | 34.36 | 61.87 |
| #68 | MedVKAN | 71.36 | 82.32 | 59.69 | 67.30 | 62.66 | 85.03 | 48.48 | 55.99 | 44.42 | 61.79 |
| #69 | Tinyunet | 67.40 | 78.61 | 59.50 | 70.89 | 59.72 | 82.99 | 44.68 | 67.16 | 34.12 | 61.63 |
| #70 | CFPNet-M | 71.42 | 82.93 | 60.77 | 71.50 | 66.16 | 84.69 | 48.80 | 59.24 | 36.32 | 61.63 |
| #71 | AttU-Net | 73.47 | 81.25 | 58.99 | 73.40 | 67.76 | 80.58 | 48.93 | 71.44 | 20.90 | 61.43 |
| #72 | DoubleUNet | 73.47 | 86.51 | 58.88 | 79.92 | 75.72 | 80.70 | 54.56 | 77.45 | 26.88 | 61.37 |
| #73 | MDSA-UNet | 71.81 | 82.57 | 61.04 | 69.72 | 65.42 | 84.23 | 37.16 | 62.69 | 41.23 | 60.94 |
| #74 | UNeXt | 70.12 | 82.21 | 58.82 | 63.13 | 61.57 | 82.39 | 44.89 | 66.99 | 32.78 | 60.78 |
| #75 | MT-UNet | 67.76 | 81.09 | 57.52 | 67.90 | 65.41 | 84.35 | 49.51 | 60.96 | 26.84 | 60.71 |
| #76 | UNetV2 | 68.88 | 82.84 | 61.87 | 69.68 | 67.73 | 82.38 | 31.55 | 68.47 | 45.32 | 60.24 |
| #77 | ULite | 69.48 | 82.31 | 56.63 | 62.46 | 59.87 | 83.99 | 52.23 | 57.63 | 32.26 | 60.23 |
| #78 | CMUNeXt | 19.06 | 82.31 | 62.22 | 76.45 | 67.38 | 85.32 | 49.66 | 67.20 | 43.51 | 59.53 |
| #79 | ResUNet++ | 59.29 | 79.77 | 60.51 | 68.06 | 67.79 | 81.01 | 44.02 | 57.08 | 30.00 | 59.45 |
| #80 | MUCM-Net | 70.99 | 82.13 | 58.58 | 54.35 | 50.60 | 83.24 | 45.83 | 76.29 | 32.00 | 59.37 |
| #81 | H-vmunet | 74.93 | 83.30 | 56.18 | 59.46 | 63.00 | 83.55 | 33.00 | 75.52 | 37.65 | 59.28 |
| #82 | U-RWKV | 66.30 | 81.90 | 61.57 | 58.94 | 59.56 | 84.87 | 44.96 | 59.35 | 26.51 | 59.16 |
| #83 | BEFUnet | 65.27 | 77.20 | 57.28 | 71.27 | 64.23 | 79.48 | 41.29 | 62.93 | 30.47 | 58.64 |
| #84 | ConvFormer | 71.15 | 82.91 | 62.96 | 76.65 | 67.22 | 83.44 | 30.23 | 56.70 | 36.17 | 58.36 |
| #85 | MedT | 64.12 | 82.59 | 61.63 | 57.05 | 56.94 | 84.14 | 49.33 | 53.30 | 31.27 | 58.27 |
| #86 | CSWin-UNet | 64.89 | 83.11 | 57.60 | 71.46 | 63.25 | 83.27 | 20.29 | 67.40 | 45.18 | 57.88 |
| #87 | ScribFormer | 60.36 | 82.41 | 60.43 | 74.42 | 66.66 | 82.68 | 40.84 | 65.48 | 13.77 | 57.61 |
| #88 | DC-UNet | 60.45 | 79.51 | 59.78 | 66.80 | 63.24 | 80.95 | 46.60 | 48.64 | 21.43 | 57.37 |
| #89 | UNETR | 59.78 | 76.24 | 46.17 | 61.36 | 55.46 | 81.63 | 56.52 | 62.18 | 32.47 | 57.02 |
| #90 | MambaUnet | 37.61 | 83.92 | 46.43 | 77.11 | 75.05 | 81.96 | 48.04 | 34.32 | | 56.55 |
| #91 | Zig-RiR | 57.05 | 81.08 | 53.31 | 65.19 | 61.02 | 84.54 | 33.02 | 68.03 | 27.10 | 56.25 |
| #92 | LeViT-UNet | 58.33 | 70.27 | 53.69 | 60.61 | 61.89 | 81.11 | 32.46 | 66.76 | 38.53 | 55.53 |
| #93 | CFM-UNet | 63.88 | 81.77 | 59.27 | 68.37 | 59.91 | 84.09 | 28.23 | 70.06 | 32.33 | 55.14 |
| #94 | SwinUnet | 60.42 | 78.35 | 37.59 | 45.60 | 48.58 | 83.05 | 39.79 | 65.60 | 43.81 | 54.06 |
| #95 | ColonSegNet | 51.95 | 68.99 | 50.15 | 69.58 | 67.46 | 82.00 | 43.98 | 58.15 | 22.87 | 53.73 |
| #96 | MultiResUNet | 69.39 | 77.67 | 58.39 | 71.18 | 65.60 | 82.43 | 19.31 | 60.99 | 24.38 | 53.42 |
| #97 | UltraLight-VM-UNet | 64.99 | 77.60 | 54.56 | 33.04 | 43.42 | 80.01 | 38.61 | 66.75 | 36.21 | 53.38 |
| #98 | SimpleUNet | 50.77 | 72.58 | 51.67 | 56.36 | 50.97 | 82.21 | 40.12 | 68.72 | 30.43 | 52.90 |
| #99 | LFU-Net | 51.14 | 75.95 | 52.15 | 52.06 | 48.23 | 82.00 | 42.61 | 62.63 | 30.99 | 52.81 |
| #100 | MALUNet | 71.04 | 81.40 | 58.31 | 58.57 | 55.02 | 81.67 | 3.42 | 72.04 | 38.51 | 52.48 |

Table 28: Average zeroshot performance of 100 u-shape medical image segmentation networks with U-Score. Source → Target. Baseline U-Net is highlighted (gray background), and statistical significance of p-value is highlighted: p<0.0001 , p<0.001 , p<0.05 , p≤0.05 , and P > 0.05 (Not significant) .

| Rank | Network | Ultrasound | | | Endoscopy | | Dermoscopy | Fundus | X-Ray | Histopathology | Avg |
|---|---|---|---|---|---|---|---|---|---|---|---|
| | | BUSI → BUS | BUSBRA → BUS | TNSCUI → TUCC | Kvasir → CVC300 | Kvasir → CVC-ClinicDB | ISIC2018 → PH2 | CHASE → Stare | Montgomery → NIH-test | Monusac → Tnbcnuclei | |
| 1 | LV-UNet | 79.62 | 79.18 | 97.84 | 96.96 | 76.12 | 99.23 | 92.89 | 92.65 | 25.00 | 82.67 |
| 2 | LGMSNet | 84.79 | 78.74 | 90.80 | 76.60 | 89.51 | 90.21 | 80.85 | 78.81 | 39.90 | 78.41 |
| 3 | MBSNet | 77.02 | 74.05 | 79.08 | 83.56 | 79.75 | 82.13 | 87.32 | 43.20 | 25.09 | 70.02 |
| #4 | CMUNeXt | 9.41 | 68.19 | 78.86 | 80.54 | 60.39 | 88.85 | 72.94 | 52.42 | 86.90 | 67.85 |
| #5 | U-KAN | 70.78 | 72.11 | 71.77 | 62.44 | 63.35 | 50.87 | 73.45 | 55.98 | 64.91 | 64.96 |
| #6 | Mobile U-ViT | 72.22 | 74.93 | 58.24 | 68.02 | 63.51 | 62.62 | 55.00 | 62.65 | 42.25 | 62.51 |
| #7 | SwinUNETR | 67.82 | 75.92 | 55.97 | 20.27 | 60.58 | 61.65 | 74.63 | 68.20 | 48.32 | 61.36 |
| #8 | RWKV-UNet | 61.22 | 60.87 | 60.37 | 61.72 | 61.72 | 61.72 | 61.72 | 61.72 | 54.63 | 60.74 |
| #9 | CFPNet-M | 61.28 | 70.75 | 65.26 | 62.37 | 52.32 | 79.91 | 52.40 | 12.08 | 63.30 | 59.42 |
| #10 | DCSAU-Net | 55.27 | 71.60 | 67.23 | 66.31 | 59.60 | 36.03 | 70.20 | 31.18 | 70.95 | 58.63 |
| #11 | TA-Net | 58.07 | 56.54 | 55.73 | 58.27 | 53.85 | 52.01 | 54.42 | 54.56 | 58.27 | 55.13 |
| #12 | G-CASCADE | 56.29 | 56.29 | 54.38 | 55.53 | 56.29 | 51.75 | 53.26 | 56.29 | 56.29 | 54.82 |
| #13 | VMUNetV2 | 51.87 | 61.86 | 62.58 | 61.93 | 62.12 | 60.43 | 45.65 | 54.54 | 51.76 | 54.32 |
| #14 | MUCM-Net | 66.09 | 70.89 | 46.67 | 9.50 | 9.50 | 57.21 | 72.47 | 84.26 | 48.26 | 54.06 |
| #15 | Tinyunet | 50.36 | 9.52 | 59.89 | 67.78 | 9.52 | 49.23 | 100.00 | 55.94 | 61.09 | 53.68 |
| #16 | ResU-KAN | 51.36 | 55.48 | 59.20 | 50.24 | 58.34 | 57.85 | 59.55 | 53.38 | 35.93 | 53.67 |
| #17 | Swin-umambaD | 59.87 | 58.55 | 57.07 | 58.75 | 60.39 | 44.17 | 43.86 | 60.39 | 52.99 | 53.55 |
| #18 | MDSA-UNet | 58.39 | 62.92 | 62.60 | 52.32 | 45.82 | 67.55 | 47.04 | 31.00 | 71.26 | 53.44 |
| #19 | CE-Net | 59.07 | 56.04 | 54.75 | 56.67 | 53.28 | 52.07 | 54.55 | 58.87 | 37.90 | 53.05 |
| #20 | UNeXt | 63.62 | 73.33 | 50.73 | 17.38 | 20.95 | 22.82 | 86.58 | 55.21 | 53.67 | 51.85 |
| #21 | EMCAD | 53.54 | 53.39 | 46.72 | 53.52 | 53.54 | 43.46 | 48.97 | 46.56 | 53.54 | 50.30 |
| #22 | UTNet | 56.14 | 55.88 | 45.86 | 50.03 | 47.93 | 53.00 | 55.60 | 44.01 | 37.29 | 49.59 |
| #23 | TransResUNet | 51.29 | 51.29 | 48.14 | 50.68 | 51.29 | 46.72 | 51.29 | 51.29 | 40.76 | 49.19 |
| #24 | MEGANet | 50.66 | 50.68 | 50.68 | 49.74 | 50.68 | 50.49 | 42.87 | 50.68 | 45.68 | 48.62 |
| #25 | Polyp-PVT | 58.92 | 59.13 | 59.13 | 51.69 | 58.76 | 50.25 | 18.09 | 54.79 | 37.99 | 45.77 |
| #26 | U-RWKV | 41.67 | 63.35 | 73.57 | 9.40 | 9.40 | 85.61 | 81.67 | 12.90 | 9.40 | 45.45 |
| #27 | TransFuse | 52.09 | 50.27 | 47.84 | 52.41 | 49.82 | 51.90 | 36.28 | 52.62 | 35.76 | 45.42 |
| #28 | CASCADE | 47.32 | 47.32 | 46.62 | 47.32 | 47.32 | 45.65 | 43.16 | 44.54 | 41.72 | 45.36 |
| #29 | DDANet | 48.46 | 36.58 | 46.87 | 59.54 | 53.69 | 45.34 | 59.11 | 21.08 | 8.97 | 43.40 |
| #30 | ULite | 60.72 | 9.81 | 11.01 | 11.15 | 9.52 | 79.09 | 94.81 | 9.52 | 50.56 | 43.18 |
| #31 | MissFormer | 40.85 | 46.97 | 42.48 | 40.26 | 39.50 | 36.98 | 34.67 | 39.01 | 44.66 | 41.08 |
| #32 | MambaUnet | 9.21 | 68.85 | 9.21 | 69.26 | 73.10 | 9.21 | 57.45 | 9.21 | 50.96 | 40.81 |
| #33 | MMUNet | 42.37 | 36.22 | 41.36 | 41.16 | 38.45 | 45.18 | 43.44 | 37.73 | 39.33 | 40.70 |
| #34 | HiFormer | 39.96 | 45.65 | 47.16 | 46.51 | 46.45 | 27.56 | 25.16 | 46.92 | 32.64 | 40.30 |
| #35 | VMUNet | 43.76 | 8.49 | 41.40 | 41.74 | 42.93 | 40.77 | 43.54 | 41.90 | 43.80 | 39.16 |
| #36 | AC-MambaSeg | 40.27 | 35.88 | 39.76 | 36.26 | 37.55 | 38.54 | 40.64 | 39.49 | 38.76 | 38.75 |
| #37 | ERDUnet | 46.97 | 48.94 | 47.06 | 43.09 | 8.71 | 44.47 | 42.03 | 16.31 | 38.61 | 38.59 |
| #38 | MedFormer | 44.92 | 47.19 | 46.67 | 42.40 | 41.31 | 43.24 | 42.03 | 8.61 | 29.61 | 38.55 |
| #39 | CSCAUNet | 38.73 | 39.51 | 39.38 | 38.79 | 37.00 | 38.48 | 39.97 | 36.01 | 34.81 | 38.02 |
| #40 | MALUNet | 66.47 | 61.83 | 42.64 | 9.50 | 9.50 | 9.50 | 9.50 | 72.61 | 79.25 | 37.03 |
| #41 | UNetV2 | 40.05 | 49.71 | 51.24 | 41.89 | 44.35 | 19.05 | 8.87 | 41.64 | 56.52 | 36.22 |
| #42 | U-Net++ | 39.73 | 42.08 | 25.92 | 35.74 | 31.99 | 40.71 | 37.85 | 19.88 | 42.08 | 35.15 |
| #43 | MSLAU-Net | 36.48 | 36.46 | 36.58 | 36.58 | 34.44 | 36.58 | 35.30 | 35.50 | 25.44 | 34.61 |
| #44 | DAEFormer | 34.51 | 37.81 | 38.89 | 29.09 | 31.36 | 25.45 | 35.60 | 27.95 | 39.11 | 33.83 |
| #45 | H2Former | 37.20 | 36.36 | 38.42 | 35.25 | 35.28 | 38.42 | 38.17 | 23.97 | 17.63 | 33.66 |
| #46 | FAT-Net | 34.91 | 35.46 | 30.98 | 33.42 | 34.12 | 33.85 | 35.59 | 36.12 | 20.71 | 33.09 |
| #47 | SCUNet++ | 35.20 | 35.43 | 30.44 | 36.89 | 32.95 | 24.36 | 35.07 | 31.51 | 32.84 | 33.05 |
| #48 | UltraLight-VM-UNet | 34.82 | 34.06 | 33.93 | 9.42 | 9.42 | 31.02 | 65.59 | 51.07 | 66.00 | 30.98 |
| #49 | ESKNet | 33.19 | 34.06 | 33.93 | 33.66 | 33.98 | 31.02 | 20.38 | 30.69 | 30.54 | 30.92 |
| #50 | AURA-Net | 31.44 | 31.96 | 30.05 | 32.01 | 31.50 | 29.45 | 30.93 | 32.10 | 22.91 | 30.38 |
| #51 | MedVKAN | 30.65 | 31.74 | 29.26 | 25.79 | 20.02 | 35.54 | 32.04 | 8.12 | 35.56 | 28.23 |
| #52 | CA-Net | 30.40 | 30.03 | 31.19 | 29.64 | 27.55 | 16.65 | 32.10 | 21.63 | 30.55 | 28.05 |
| #53 | RollingUnet | 28.23 | 30.21 | 30.15 | 30.60 | 28.46 | 23.58 | 30.69 | 22.48 | 7.88 | 26.15 |
| #54 | DDS-UNet | 28.39 | 28.11 | 28.82 | 28.02 | 27.83 | 9.00 | 26.41 | 27.44 | 27.57 | 25.83 |
| #55 | DC-UNet | 8.79 | 23.53 | 40.76 | 32.42 | 27.62 | 8.79 | 50.20 | 8.79 | 8.79 | 25.67 |
| #56 | BEFUnet | 25.62 | 8.49 | 18.93 | 36.88 | 28.37 | 8.49 | 35.89 | 24.46 | 25.98 | 24.72 |
| #57 | LFU-Net | 9.52 | 9.52 | 9.52 | 9.52 | 9.52 | 9.52 | 42.39 | 33.83 | 42.44 | 24.15 |
| #58 | SimpleUNet | 9.52 | 9.52 | 9.52 | 9.52 | 9.52 | 9.52 | 13.21 | 17.47 | 62.24 | 23.40 |
| #59 | MultiResUNet | 14.92 | 9.10 | 36.43 | 51.70 | 42.92 | 22.06 | 9.10 | 21.57 | 9.10 | 22.60 |
| #60 | Swin-umamba | 22.64 | 22.64 | 22.64 | 22.64 | 22.29 | 22.23 | 19.13 | 22.62 | 21.46 | 22.09 |
| #61 | CSWin-UNet | 20.18 | 29.87 | 18.57 | 28.02 | 20.46 | 26.01 | 7.91 | 25.61 | 31.76 | 21.63 |
| #62 | SwinUnet | 8.48 | 8.48 | 8.48 | 8.48 | 8.48 | 30.92 | 36.49 | 30.06 | 43.42 | 21.50 |
| #63 | GH-UNet | 20.77 | 20.52 | 20.79 | 20.92 | 20.32 | 21.28 | 20.69 | 16.76 | 21.19 | 20.32 |
| #64 | MADGNet | 20.32 | 20.23 | 19.44 | 20.48 | 20.48 | 19.36 | 19.97 | 20.48 | 20.43 | 20.23 |
| #65 | CENet | 20.23 | 20.23 | 19.63 | 20.07 | 19.88 | 19.36 | 20.23 | 18.67 | 20.23 | 19.87 |
| #66 | LeViT-UNet | 8.87 | 8.87 | 8.87 | 8.87 | 20.31 | 8.87 | 34.99 | 38.17 | 49.73 | 19.64 |
| #67 | Zig-RiR | 8.20 | 29.22 | 8.20 | 21.61 | 12.05 | 35.75 | 27.02 | 29.58 | 9.13 | 19.08 |
| #68 | DoubleUNet | 23.10 | 24.81 | 19.15 | 24.80 | 24.81 | 7.39 | 7.39 | 24.09 | 7.39 | 18.77 |
| #69 | TransAttUnet | 20.75 | 21.21 | 20.57 | 20.69 | 20.93 | 20.38 | 21.75 | 7.00 | 11.85 | 18.65 |
| #70 | DA-TransUNet | 18.84 | 18.58 | 18.19 | 17.39 | 18.08 | 14.99 | 18.46 | 18.46 | 18.84 | 18.10 |
| #71 | U-Net | 19.20 | 18.36 | 19.07 | 18.58 | 19.53 | 19.60 | 19.23 | 19.08 | 6.97 | 17.93 |
| #72 | UNet3+ | 17.68 | 16.66 | 18.16 | 18.30 | 18.54 | 17.29 | 18.96 | 18.69 | 14.32 | 17.78 |
| #73 | TransUnet | 19.63 | 19.34 | 18.31 | 18.43 | 18.18 | 17.67 | 6.86 | 19.72 | 17.62 | 17.53 |
| #74 | CFM-UNet | 15.96 | 23.46 | 22.01 | 21.67 | 7.53 | 24.92 | 7.53 | 23.40 | 21.06 | 17.51 |
| #75 | CPCANet | 18.42 | 18.51 | 17.34 | 18.28 | 17.27 | 18.07 | 17.12 | 13.24 | 17.66 | 17.35 |
| #76 | MedT | 15.50 | 22.46 | 22.84 | 7.32 | 7.32 | 22.86 | 22.69 | 7.32 | 18.43 | 17.00 |
| #77 | UTANet | 18.16 | 18.44 | 17.87 | 18.59 | 17.92 | 18.39 | 18.41 | 17.95 | 6.72 | 16.99 |
| #78 | CaraNet | 18.50 | 18.46 | 18.50 | 18.41 | 18.30 | 18.39 | 13.86 | 18.50 | 16.21 | 16.58 |
| #79 | AttU-Net | 19.41 | 18.17 | 17.43 | 19.50 | 18.74 | 6.97 | 18.78 | 19.07 | 6.97 | 16.50 |
| #80 | H-vmunet | 22.30 | 22.52 | 7.26 | 7.26 | 16.20 | 21.07 | 13.10 | 22.46 | 21.94 | 16.28 |
| #81 | CMU-Net | 16.96 | 17.06 | 16.85 | 16.97 | 16.74 | 15.29 | 12.65 | 16.69 | 16.74 | 16.26 |
| #82 | TransNorm | 17.34 | 17.17 | 17.31 | 15.69 | 16.22 | 14.84 | 6.55 | 17.34 | 6.98 | 15.65 |
| #83 | ScribFormer | 6.98 | 19.45 | 19.11 | 19.88 | 18.33 | 15.30 | 15.32 | 17.00 | 6.98 | 15.64 |
| #84 | ResUNet++ | 7.16 | 15.83 | 20.54 | 18.66 | 20.20 | 7.16 | 21.87 | 7.16 | 15.89 | 15.50 |
| #85 | PraNet | 17.33 | 17.12 | 17.26 | 17.33 | 17.33 | 16.86 | 10.52 | 17.33 | 16.66 | 15.43 |
| #86 | EViT-UNet | 16.03 | 16.16 | 16.18 | 16.18 | 15.75 | 15.31 | 16.18 | 6.38 | 12.62 | 14.70 |
| #87 | UCTransNet | 13.43 | 12.96 | 14.33 | 14.76 | 14.31 | 14.12 | 14.78 | 14.56 | 14.02 | 14.18 |
| #88 | FCBFormer | 14.49 | 14.61 | 14.55 | 14.22 | 14.43 | 13.96 | 11.84 | 14.15 | 14.52 | 14.14 |
| #89 | MSRFNet | 14.46 | 15.05 | 13.35 | 15.31 | 15.93 | 14.23 | 16.11 | 6.73 | 6.37 | 13.36 |
| #90 | D-TrAttUnet | 13.34 | 11.52 | 12.83 | 12.83 | 13.63 | 14.02 | 14.00 | 13.48 | 12.73 | 13.32 |
| #91 | MCA-UNet | 13.39 | 13.30 | 13.37 | 12.75 | 12.86 | 13.39 | 13.30 | 13.14 | 11.88 | 13.09 |
| #92 | Perspective-Unet | 12.12 | 12.32 | 12.06 | 12.23 | 12.15 | 11.69 | 12.26 | 12.17 | 11.76 | 12.07 |
| #93 | UNETR | 7.08 | 7.08 | 7.08 | 7.08 | 7.08 | 7.08 | 19.79 | 14.63 | 17.98 | 11.64 |
| #94 | MERIT | 12.52 | 12.28 | 12.56 | 12.56 | 12.51 | 12.56 | 5.73 | 12.16 | 12.51 | 11.63 |
| #95 | ColonSegNet | 7.28 | 7.28 | 7.28 | 20.52 | 21.00 | 7.28 | 7.28 | 7.28 | 10.79 | 11.40 |
| #96 | DS-TransUNet | 10.72 | 10.74 | 10.60 | 10.80 | 10.80 | 10.53 | 10.80 | 10.61 | 10.79 | 10.72 |
| #97 | UACANet | 12.63 | 12.62 | 12.46 | 12.65 | 12.62 | 12.65 | 7.24 | 12.64 | 5.74 | 10.70 |
| #98 | ConvFormer | 11.76 | 12.10 | 12.34 | 12.29 | 11.65 | 11.65 | 5.70 | 5.70 | 11.85 | 10.08 |
| #99 | CFFormer | 10.13 | 8.21 | 10.00 | 10.09 | 10.13 | 10.13 | 10.13 | 10.09 | 9.41 | 9.84 |
| #100 | MT-UNet | 8.76 | 8.92 | 7.76 | 8.76 | 8.78 | 9.39 | 9.40 | 7.31 | 5.00 | 8.35 |

Table 29: Per-dataset source ranking of 100 u-shape medical image segmentation networks with IoU

Table 30: Per-dataset target ranking of 100 u-shape medical image segmentation networks with IoU. Source → Target.

| Rank | Ultrasound BUSI → BUS | BUSBRA → BUS | TNSCUI → TUCC | Endoscopy Kvasir → CVC300 | Kvasir → CVC-ClinicDB | Dermoscopy ISIC2018 → PH2 | Fundus CHASE → Stare | X-Ray Montgomery → NIH-test | Histopathology Monusac → Tnbcnuclei |
|---|---|---|---|---|---|---|---|---|---|
| #1 | Swin-umamba | MEGANet | MSLAU-Net | PraNet | PraNet | MSLAU-Net | RWKV-UNet | MEGANet | TA-Net |
| #2 | EMCAD | DoubleUNet | MERIT | RWKV-UNet | DS-TransUNet | RWKV-UNet | DS-TransUNet | TransResUNet | CENet |
| #3 | CENet | CENet | EViT-UNet | UACANet | CASCADE | G-CASCADE | SwinUNETR | CaraNet | U-Net++ |
| #4 | G-CASCADE | FCBFormer | Polyp-PVT | MERIT | Swin-umambaD | MERIT | FCBFormer | DA-TransUNet | EMCAD |
| #5 | DA-TransUNet | CASCADE | LGMSNet | MADGNet | EMCAD | MMUNet | MCA-UNet | PraNet | G-CASCADE |
| #6 | PraNet | Polyp-PVT | G-CASCADE | TA-Net | TransResUNet | H2Former | UNETR | MADGNet | UNetV2 |
| #7 | CASCADE | TransResUNet | CaraNet | EViT-UNet | MADGNet | CMUNeXt | DA-TransUNet | Swin-umambaD | CSWin-UNet |
| #8 | TransNorm | U-Net++ | H2Former | UTANet | CFFormer | UACANet | MSRFNet | TransUnet | DAEFormer |
| #9 | MCA-UNet | MCA-UNet | MEGANet | DS-TransUNet | DoubleUNet | MADGNet | DoubleUNet | TransNorm | DA-TransUNet |
| #10 | CaraNet | G-CASCADE | Swin-umamba | CASCADE | MEGANet | MCA-UNet | Swin-umamba | RWKV-UNet | MedVKAN |
| #11 | TransResUNet | Swin-umamba | HiFormer | Swin-umamba | RWKV-UNet | CFFormer | EViT-UNet | FAT-Net | VMUNet |
| #12 | MEGANet | MedFormer | TransNorm | EMCAD | Polyp-PVT | MedVKAN | CENet | Swin-umamba | DS-TransUNet |
| #13 | VMUNet | EMCAD | MCA-UNet | DoubleUNet | MambaUnet | GH-UNet | TransNorm | UACANet | MADGNet |
| #14 | CFFormer | EViT-UNet | DAEFormer | TransFuse | LGMSNet | LGMSNet | ERDUnet | TransFuse | SwinUNet |
| #15 | Polyp-PVT | MSLAU-Net | VMUNetV2 | AURA-Net | UACANet | MEGANet | VMUNet | HiFormer | GH-UNet |
| #16 | TA-Net | CaraNet | LV-UNet | TransResUNet | G-CASCADE | LV-UNet | AC-MambaSeg | AURA-Net | CMUNeXt |
| #17 | MSLAU-Net | FAT-Net | PraNet | CaraNet | MERIT | U-RWKV | DAEFormer | CE-Net | MERIT |
| #18 | UACANet | UACANet | MedFormer | HiFormer | VMUNetV2 | CaraNet | CFFormer | CFFormer | FCBFormer |
| #19 | RWKV-UNet | MissFormer | FCBFormer | LV-UNet | HiFormer | TransFuse | MissFormer | LV-UNet | AC-MambaSeg |
| #20 | Swin-umambaD | RWKV-UNet | CASCADE | VMUNetV2 | VMUNet | CFPNet-M | ULite | MSLAU-Net | MDSA-UNet |
| #21 | MERIT | Perspective-Unet | RWKV-UNet | DDANet | CaraNet | MBSNet | TransUnet | AC-MambaSeg | MissFormer |
| #22 | CE-Net | AURA-Net | CSCANet | MEGANet | ESKNet | VMUNetV2 | FAT-Net | G-CASCADE | AC-MambaSeg |
| #23 | TransUnet | CSCAUNet | DDS-UNet | CENet | Swin-umamba | CASCADE | AURA-Net | VMUNet | CA-Net |
| #24 | TransFuse | CPCANet | ESKNet | CFFormer | AURA-Net | U-Net++ | TransUnet | TA-Net | TransNorm |
| #25 | AC-MambaSeg | UTANet | AC-MambaSeg | Swin-umambaD | MSRFNet | Swin-umamba | HiFormer | DoubleUNet | RWKV-UNet |
| #26 | MADGNet | VMUNetV2 | ConvFormer | MSLAU-Net | FCBFormer | UTANet | TransAttUnet | Perspective-Unet | MEGANet |
| #27 | FCBFormer | MADGNet | TA-Net | MBSNet | CENet | CSCAUNet | UNet3+ | CASCADE | DDS-UNet |
| #28 | CPCANet | DS-TransUNet | ResU-KAN | G-CASCADE | MBSNet | Zig-RiR | CMUNeXt | Polyp-PVT | Swin-umambaD |
| #29 | EViT-UNet | Mobile U-ViT | CA-Net | ResU-KAN | ResU-KAN | D-TrAttUnet | MT-UNet | MCA-UNet | LeViT-UNet |
| #30 | DS-TransUNet | ESKNet | Swin-umamba | Perspective-Unet | TransFuse | ResU-KAN | CA-Net | MUCM-Net | PraNet |
| #31 | LGMSNet | Swin-umambaD | U-KAN | RollingUnet | Perspective-Unet | MT-UNet | MedT | DS-TransUNet | MALUNet |
| #32 | AURA-Net | TA-Net | GH-UNet | ESKNet | TA-Net | AC-MambaSeg | EMCAD | LGMSNet | Swin-umamba |
| #33 | CSCAUNet | TransNorm | UACANet | GH-UNet | FAT-Net | PraNet | LV-UNet | H-vmunet | U-KAN |
| #34 | H2Former | ERDUnet | MBSNet | MambaUnet | EViT-UNet | CPCANet | MADGNet | CE-Net | CMU-Net |
| #35 | FAT-Net | HiFormer | TransAttUnet | CE-Net | MSLAU-Net | VMUNet | AttU-Net | SwinUNETR | CASCADE |
| #36 | Mobile U-ViT | PraNet | CFFormer | CPCANet | TransAttUnet | MDSA-UNet | G-CASCADE | FCBFormer | CPCANet |
| #37 | UTNet | SwinUNETR | RollingUnet | ConvFormer | DDS-UNet | FAT-Net | CFPNet-M | DDS-UNet | H-vmunet |
| #38 | MedFormer | DA-TransUNet | VMUNet | VMUNet | UTANet | TransResUNet | CSCAUNet | VMUNetV2 | MMUNet |
| #39 | DDS-UNet | DCSAU-Net | ERDUnet | CMUNeXt | CSCAUNet | MedFormer | MedVKAN | MERIT | CSCAUNet |
| #40 | GH-UNet | DAEFormer | CMUNeXt | SCUNet++ | CE-Net | TA-Net | MMUNet | UNet3+ | CSCAUNet |
| #41 | U-KAN | U-KAN | CENet | DDS-UNet | AC-MambaSeg | MedT | DDANet | ResU-KAN | ESKNet |
| #42 | UTANet | TransFuse | CE-Net | CMU-Net | DDANet | UTNet | ResU-KAN | ResU-KAN | CFPNet-M |
| #43 | ESKNet | MambaUnet | DCSAU-Net | UCTransNet | DA-TransUNet | CENet | D-TrAttUnet | CSCANet | UltraLight-VM-UNet |
| #44 | U-Net++ | TransUnet | DS-TransUNet | Mobile U-ViT | GH-UNet | CFM-UNet | H2Former | CMU-Net | ConvFormer |
| #45 | ERDUnet | H2Former | UNetV2 | D-TrAttUnet | H2Former | DS-TransUNet | SCUNet++ | EMCAD | Perspective-Unet |
| #46 | MMUNet | UTNet | DA-TransUNet | FCBFormer | CMU-Net | CE-Net | RollingUnet | Mobile U-ViT | SCUNet++ |
| #47 | MBSNet | CE-Net | Perspective-Unet | DCSAU-Net | U-Net | U-Net | CASCADE | MALUNet | TransResUNet |
| #48 | Perspective-Unet | TransAttUnet | TransFuse | LGMSNet | U-KAN | ULite | Mobile U-ViT | ESKNet | UCTransNet |
| #49 | CA-Net | RollingUnet | MMUNet | ScribFormer | Mobile U-ViT | ERDUnet | ULite | AttU-Net | TransFuse |
| #50 | CMU-Net | CMU-Net | MedT | MMUNet | MedFormer | AURA-Net | DC-UNet | U-Net | ERDUnet |
| #51 | H-vmunet | DDS-UNet | UTANet | FAT-Net | UNet3+ | Polyp-PVT | LGMSNet | D-TrAttUnet | MambaUnet |
| #52 | TransAttUnet | LGMSNet | CMU-Net | H2Former | MCA-UNet | TransAttUnet | MSLAU-Net | CENet | Tinyunet |
| #53 | SwinUNETR | H-vmunet | U-RWKV | TransAttUnet | LV-UNet | Mobile U-ViT | U-KAN | MMUNet | CFFormer |
| #54 | LV-UNet | MERIT | AURA-Net | MedFormer | D-TrAttUnet | FCBFormer | CMU-Net | MissFormer | CaraNet |
| #55 | SCUNet++ | GH-UNet | MissFormer | Polyp-PVT | RollingUnet | ESKNet | Perspective-Unet | CFM-UNet | SwinUNETR |
| #56 | AttU-Net | CSWin-UNet | MADGNet | CA-Net | DCSAU-Net | EViT-UNet | UCTransNet | U-KAN | D-TrAttUnet |
| #57 | DoubleUNet | ResU-KAN | TransAttUnet | AttU-Net | MMUNet | SwinUNETR | MUCM-Net | UTNet | UNeXt |
| #58 | MissFormer | MBSNet | EMCAD | U-KAN | UCTransNet | EMCAD | MEGANet | SimpleUNet | TransFuse |
| #59 | DAEFormer | CA-Net | MDSA-UNet | AC-MambaSeg | TransNorm | Perspective-Unet | TA-Net | SCUNet++ | UNETR |
| #60 | ResU-KAN | CFPNet-M | CFPNet-M | UTNet | SwinUNETR | H-vmunet | MBSNet | UNetV2 | Polyp-PVT |
| #61 | VMUNetV2 | ConvFormer | UCTransNet | MSRFNet | UTNet | UCTransNet | CPCANet | Zig-RiR | HiFormer |
| #62 | RollingUnet | UNetV2 | TransUnet | ERDUnet | CPCANet | ConvFormer | UTNet | CSWin-UNet | MCA-UNet |
| #63 | U-Net | SCUNet++ | CPCANet | TransUnet | MissFormer | DDANet | GH-UNet | CMUNeXt | CFM-UNet |
| #64 | HiFormer | LV-UNet | ResUNet++ | MCA-UNet | TransUnet | UTANet | UTANet | Tinyunet | CE-Net |
| #65 | MDSA-UNet | MedT | U-Net | MissFormer | SCUNet++ | MissFormer | DDS-UNet | UNeXt | ULite |
| #66 | DDANet | MDSA-UNet | ScribFormer | UNet3+ | ResUNet++ | Swin-umambaD | U-RWKV | LeViT-UNet | UTNet |
| #67 | CFPNet-M | ScribFormer | ResU-KAN | ResU-KAN | AttU-Net | U-KAN | UNeXt | UltraLight-VM-UNet | MUCM-Net |
| #68 | MedVKAN | MedVKAN | Mobile U-ViT | CFPNet-M | UNetV2 | UNet3+ | Tinyunet | DAEFormer | Mobile U-ViT |
| #69 | ConvFormer | AC-MambaSeg | FAT-Net | CSWin-UNet | CA-Net | CSWin-UNet | ResUNet++ | SwinUnet | ResU-KAN |
| #70 | MALUNet | CMUNeXt | DDANet | U-Net++ | ColonSegNet | MUCM-Net | MedFormer | ScribFormer | MSLAU-Net |
| #71 | D-TrAttUnet | UNeXt | UTNet | BEFUnet | CMUNeXt | CMU-Net | ColonSegNet | MBSNet | MedT |
| #72 | MUCM-Net | MUCM-Net | SwinUNETR | DA-TransUNet | ConvFormer | MSRFNet | DCSAU-Net | RollingUnet | MedFormer |
| #73 | DCSAU-Net | MSRFNet | DC-UNet | MultiResUNet | DAEFormer | SwinUnet | U-Net++ | GH-UNet | AURA-Net |
| #74 | UNet3+ | U-RWKV | MedVKAN | Tinyunet | ScribFormer | TransNorm | CE-Net | CA-Net | LFU-Net |
| #75 | UNeXt | CFM-UNet | Tinyunet | U-Net | CFPNet-M | Tinyunet | LFU-Net | H2Former | LGMSNet |
| #76 | ULite | MMUNet | SCUNet++ | TransNorm | U-Net++ | RollingUnet | MambaUnet | BEFUnet | BEFUnet |
| #77 | MSRFNet | MALUNet | CFM-UNet | MDSA-UNet | MultiResUNet | DA-TransUNet | BEFUnet | DCSAU-Net | SimpleUNet |
| #78 | BEFUnet | U-Net | D-TrAttUnet | UNetV2 | MDSA-UNet | DAEFormer | ScribFormer | MDSA-UNet | EViT-UNet |
| #79 | UCTransNet | AttU-Net | AttU-Net | ColonSegNet | MT-UNet | DCSAU-Net | ESKNet | LFU-Net | ResUNet++ |
| #80 | MT-UNet | MT-UNet | UNeXt | CFM-UNet | BEFUnet | HiFormer | SimpleUNet | CPCANet | UNet3+ |
| #81 | Tinyunet | Zig-RiR | MUCM-Net | DAEFormer | CSWin-UNet | SCUNet++ | SwinUnet | UNETR | FAT-Net |
| #82 | U-RWKV | UNet3+ | DDANet | ResUNet++ | DC-UNet | ScribFormer | Swin-umambaD | U-Net++ | MBSNet |
| #83 | BEFUnet | DDANet | MultiResUNet | MT-UNet | H-vmunet | CA-Net | UltraLight-VM-UNet | DDANet | LV-UNet |
| #84 | UltraLight-VM-UNet | UCTransNet | MALUNet | MSRFNet | MedVKAN | MultiResUNet | MERIT | MultiResUNet | H2Former |
| #85 | CSWin-UNet | D-TrAttUnet | MALUNet | DC-UNet | LeViT-UNet | UNeXt | MDSA-UNet | MT-UNet | TransAttUnet |
| #86 | MedT | ResUNet++ | U-Net++ | Zig-RiR | UNeXt | UNetV2 | VMUNetV2 | ERDUnet | Zig-RiR |
| #87 | CFM-UNet | CFFormer | CSWin-UNet | SwinUNETR | Zig-RiR | SimpleUNet | TransFuse | U-RWKV | DoubleUNet |
| #88 | MultiResUNet | DC-UNet | MT-UNet | UNeXt | CFM-UNet | DDS-UNet | Zig-RiR | CFPNet-M | MT-UNet |
| #89 | DC-UNet | ULite | BEFUnet | ULite | ULite | MambaUnet | H-vmunet | MSRFNet | UACANet |
| #90 | SwinUnet | Tinyunet | ULite | UNETR | Tinyunet | ColonSegNet | LeViT-UNet | ColonSegNet | U-RWKV |
| #91 | ScribFormer | SwinUnet | H-vmunet | LeViT-UNet | U-RWKV | LFU-Net | UNetV2 | ULite | U-Net |
| #92 | UNETR | MultiResUNet | UltraLight-VM-UNet | H-vmunet | ERDUnet | MedT | ConvFormer | ResUNet++ | MultiResUNet |
| #93 | ResUNet++ | UltraLight-VM-UNet | LeViT-UNet | U-RWKV | MedT | UNETR | CFM-UNet | ConvFormer | UTANet |
| #94 | LeViT-UNet | BEFUnet | Zig-RiR | MALUNet | UNETR | LeViT-UNet | Polyp-PVT | MedVKAN | DDANet |
| #95 | Zig-RiR | UNETR | LFU-Net | MedT | MALUNet | ResUNet++ | CaraNet | TransAttUnet | ColonSegNet |
| #96 | ColonSegNet | LFU-Net | SimpleUNet | SimpleUNet | SimpleUNet | DC-UNet | UACANet | MedT | RollingUnet |
| #97 | LFU-Net | SimpleUNet | ColonSegNet | MUCM-Net | MUCM-Net | DoubleUNet | PraNet | MedFormer | MSRFNet |
| #98 | SimpleUNet | LeViT-UNet | MambaUnet | LFU-Net | SwinUnet | AttU-Net | CSWin-UNet | DC-UNet | DC-UNet |
| #99 | MambaUnet | ColonSegNet | UNETR | SwinUnet | LFU-Net | UltraLight-VM-UNet | MultiResUNet | MambaUnet | AttU-Net |
| #100 | CMUNeXt | VMUNet | SwinUnet | UltraLight-VM-UNet | UltraLight-VM-UNet | BEFUnet | MALUNet | EViT-UNet | ScribFormer |

Table 31: Per-dataset source ranking of 100 u-shape medical image segmentation networks with U-Score

Table 32: Per-dataset target ranking of 100 u-shape medical image segmentation networks with U-Score. Source → Target.

| Rank | Ultrasound | | | Endoscopy | | Dermoscopy | Fundus | X-Ray | Histopathology |
|---|---|---|---|---|---|---|---|---|---|
| | BUSI → BUS | BUSBRA → BUS | TNSCUI → TUCC | Kvasir → CVC300 | Kvasir → CVC-ClinicDB | ISIC2018 → PH2 | CHASE → Stare | Montgomery → NIH-test | Monusac → Tnbcnuclei |
| #1 | LGMSNet | LV-UNet | LV-UNet | LV-UNet | LGMSNet | LV-UNet | ULite | LV-UNet | CMUNeXt |
| #2 | LV-UNet | LGMSNet | LGMSNet | MBSNet | MBSNet | LGMSNet | LV-UNet | MUCM-Net | MALUNet |
| #3 | MBSNet | SwinUNETR | MBSNet | CMUNeXt | LV-UNet | CMUNeXt | SwinUNETR | LGMSNet | MDSA-UNet |
| #4 | Mobile U-ViT | Mobile U-ViT | CMUNeXt | LGMSNet | MambaUnet | U-RWKV | CMUNeXt | MALUNet | DCSAU-Net |
| #5 | U-KAN | MBSNet | U-RWKV | MambaUnet | Mobile U-ViT | MBSNet | MUCM-Net | SwinUNETR | UltraLight-VM-UNet |
| #6 | SwinUNETR | UNeXt | U-KAN | Mobile U-ViT | U-KAN | CFPNet-M | CFPNet-M | Mobile U-ViT | U-KAN |
| #7 | MALUNet | U-KAN | DCSAU-Net | Tinyunet | VMUNetV2 | ULite | UNeXt | SimpleUNet | CFPNet-M |
| #8 | MUCM-Net | DCSAU-Net | CFPNet-M | DCSAU-Net | RWKV-UNet | MDSA-UNet | LGMSNet | RWKV-UNet | Tinyunet |
| #9 | UNeXt | MUCM-Net | MDSA-UNet | U-KAN | SwinUNETR | Mobile U-ViT | Swin-umambaD | Tinyunet | TA-Net |
| #10 | CFPNet-M | CFPNet-M | VMUNetV2 | VMUNetV2 | CFPNet-M | Swin-umambaD | CE-Net | CE-Net | UNetV2 |
| #11 | RWKV-UNet | MambaUnet | RWKV-UNet | VMUNetV2 | CMUNeXt | SwinUNETR | U-RWKV | U-KAN | G-CASCADE |
| #12 | ULite | CMUNeXt | Tinyunet | RWKV-UNet | DCSAU-Net | VMUNetV2 | LFU-Net | Tinyunet | RWKV-UNet |
| #13 | Swin-umambaD | U-RWKV | ResU-KAN | DDANet | Polyp-PVT | ResU-KAN | Mobile U-ViT | UNeXt | UNeXt |
| #14 | CE-Net | MDSA-UNet | Swin-umambaD | ResU-KAN | ResU-KAN | MUCM-Net | U-KAN | Polyp-PVT | EMCAD |
| #15 | Polyp-PVT | VMUNetV2 | Mobile U-ViT | TA-Net | G-CASCADE | G-CASCADE | RWKV-UNet | TA-Net | Swin-umambaD |
| #16 | MDSA-UNet | MALUNet | Swin-umambaD | CE-Net | TA-Net | UTNet | DCSAU-Net | VMUNetV2 | VMUNetV2 |
| #17 | TA-Net | RWKV-UNet | G-CASCADE | G-CASCADE | DDANet | CE-Net | ResU-KAN | ResU-KAN | MambaUnet |
| #18 | G-CASCADE | Polyp-PVT | SwinUNETR | EMCAD | EMCAD | TA-Net | SimpleUNet | G-CASCADE | ULite |
| #19 | UTNet | Swin-umambaD | TA-Net | TransFuse | CE-Net | TransFuse | DDANet | TransFuse | LeViT-UNet |
| #20 | DCSAU-Net | TA-Net | CE-Net | MDSA-UNet | CFPNet-M | U-KAN | G-CASCADE | CMUNeXt | SwinUNet |
| #21 | EMCAD | CE-Net | UNetV2 | MultiResUNet | TransResUNet | MEGANet | MambaUnet | TransResUNet | MUCM-Net |
| #22 | TransFuse | G-CASCADE | UNeXt | Polyp-PVT | MEGANet | Polyp-PVT | UTNet | UltraLight-VM-UNet | MEGANet |
| #23 | VMUNetV2 | UTNet | MEGANet | TransResUNet | TransFuse | Tinyunet | EMCAD | MEGANet | MissFormer |
| #24 | ResU-KAN | ResU-KAN | TransResUNet | ResU-KAN | CASCADE | TransResUNet | HiFormer | HiFormer | VMUNet |
| #25 | TransResUNet | EMCAD | TransFuse | UTNet | UTNet | CASCADE | TA-Net | EMCAD | SwinUNet |
| #26 | MEGANet | TransResUNet | HiFormer | MEGANet | HiFormer | DDANet | TransResUNet | CASCADE | LFU-Net |
| #27 | Tinyunet | MEGANet | ERDUnet | CASCADE | MDSA-UNet | MMUNet | CE-Net | UTNet | Mobile U-ViT |
| #28 | DDANet | TransFuse | DDANet | HiFormer | UNetV2 | ERDUnet | DC-UNet | MBSNet | U-Net++ |
| #29 | CASCADE | UNetV2 | EMCAD | ERDUnet | VMUNet | Swin-umambaD | MissFormer | VMUNet | CASCADE |
| #30 | ERDUnet | ERDUnet | MedFormer | MedFormer | MultiResUNet | MultiResUNet | UltraLight-VM-UNet | UNetV2 | TransResUNet |
| #31 | MedFormer | CASCADE | MUCM-Net | UNetV2 | MedFormer | MedFormer | HiFormer | AC-MambaSeg | LGMSNet |
| #32 | VMUNet | MedFormer | CASCADE | VMUNet | MissFormer | VMUNet | MEGANet | MissFormer | MMUNet |
| #33 | MMUNet | MissFormer | UTNet | MMUNet | MMUNet | U-Net++ | VMUNet | LeViT-UNet | DAEFormer |
| #34 | U-RWKV | HiFormer | MALUNet | MissFormer | AC-MambaSeg | AC-MambaSeg | CASCADE | MMUNet | AC-MambaSeg |
| #35 | MissFormer | U-Net++ | MissFormer | CSCAUNet | CSCAUNet | CSCAUNet | CSCAUNet | FAT-Net | ERDUnet |
| #36 | AC-MambaSeg | CSCAUNet | VMUNet | SCUNet++ | H2Former | H2Former | AC-MambaSeg | CSCAUNet | SimpleUNet |
| #37 | UNetV2 | DAEFormer | MMUNet | MSLAU-Net | MSLAU-Net | MissFormer | MedFormer | MSLAU-Net | Polyp-PVT |
| #38 | HiFormer | H2Former | DC-UNet | AC-MambaSeg | FAT-Net | MSLAU-Net | Swin-umambaD | LFU-Net | CE-Net |
| #39 | U-Net++ | DDANet | AC-MambaSeg | MSLAU-Net | ESKNet | DCSAU-Net | DAEFormer | AURA-Net | UTNet |
| #40 | CSCAUNet | MSLAU-Net | CSCAUNet | U-Net++ | SCUNet++ | Zig-RiR | CSCAUNet | SCUNet++ | ResU-KAN |
| #41 | H2Former | MMUNet | DAEFormer | H2Former | U-Net++ | MedVKAN | SCUNet++ | DCSAU-Net | TransFuse |
| #42 | MSLAU-Net | FAT-Net | H2Former | ESKNet | AURA-Net | FAT-Net | U-Net++ | MDSA-UNet | MedVKAN |
| #43 | SCUNet++ | AC-MambaSeg | MSLAU-Net | FAT-Net | DAEFormer | ESKNet | FAT-Net | ESKNet | SCUNet++ |
| #44 | FAT-Net | SCUNet++ | MultiResUNet | DC-UNet | RollingUnet | AURA-Net | FAT-Net | Zig-RiR | HiFormer |
| #45 | UltraLight-VM-UNet | ESKNet | ESKNet | AURA-Net | BEFUnet | BEFUnet | FAT-Net | FAT-Net | CSWin-UNet |
| #46 | DAEFormer | AURA-Net | CA-Net | RollingUnet | DDS-UNet | HiFormer | DAEFormer | DAEFormer | CA-Net |
| #47 | ESKNet | MedVKAN | FAT-Net | CA-Net | DC-UNet | CSWin-UNet | MedVKAN | DDS-UNet | ESKNet |
| #48 | AURA-Net | RollingUnet | SCUNet++ | DAEFormer | CA-Net | DAEFormer | MSLAU-Net | CSWin-UNet | MedFormer |
| #49 | MedVKAN | CA-Net | RollingUnet | DDS-UNet | DoubleUNet | CFM-UNet | SwinUNet | BEFUnet | DDS-UNet |
| #50 | CA-Net | CSWin-UNet | AURA-Net | CSWin-UNet | Swin-umamba | SCUNet++ | AURA-Net | DoubleUNet | DCS-UNet |
| #51 | DDS-UNet | Zig-RiR | MedVKAN | MedVKAN | ColonSegNet | RollingUnet | CA-Net | H2Former | BEFUnet |
| #52 | RollingUnet | DDS-UNet | DDS-UNet | DoubleUNet | UNeXt | MedT | VMUNetV2 | CFM-UNet | MSLAU-Net |
| #53 | BEFUnet | DoubleUNet | U-Net++ | Swin-umamba | TransAttUnet | UNeXt | RollingUnet | Swin-umamba | MBSNet |
| #54 | DoubleUNet | DC-UNet | MADGNet | CFM-UNet | MADGNet | MADGNet | Swin-umamba | RollingUnet | LV-UNet |
| #55 | Swin-umamba | CFM-UNet | Swin-umamba | Zig-RiR | CSWin-UNet | MultiResUNet | DDS-UNet | H-vmunet | AURA-Net |
| #56 | H-vmunet | Swin-umamba | CFM-UNet | GH-UNet | GH-UNet | GH-UNet | TransFuse | CA-Net | H-vmunet |
| #57 | GH-UNet | H-vmunet | TransAttUnet | TransAttUnet | LeViT-UNet | ResUNet++ | DoubleUNet | DDANet | Swin-umamba |
| #58 | TransAttUnet | MedT | TransAttUnet | ColonSegNet | ResUNet++ | MADGNet | MADGNet | TransAttUnet | GH-UNet |
| #59 | MADGNet | TransAttUnet | ResUNet++ | MADGNet | MADGNet | MedVKAN | TransAttUnet | MADGNet | CFM-UNet |
| #60 | CENet | GH-UNet | CENet | SwinUNETR | CENet | CENet | UNETR | U-Net++ | FAT-Net |
| #61 | CSWin-UNet | MADGNet | MADGNet | CENet | U-Net | CENet | ColonSegNet | TransUnet | MADGNet |
| #62 | TransUnet | CENet | DoubleUNet | ScribFormer | AttU-Net | UNetV2 | TransAttUnet | U-Net | CENet |
| #63 | AttU-Net | ScribFormer | ScribFormer | AttU-Net | UNet3+ | CaraNet | ResNet++ | AttU-Net | DA-TransUNet |
| #64 | U-Net | TransUnet | U-Net | ResUNet++ | ScribFormer | UTANet | CENet | DA-TransUNet | MedT |
| #65 | DA-TransUNet | DA-TransUNet | BEFUnet | UTANet | CaraNet | CPCANet | AttU-Net | UNet3+ | UNETR |
| #66 | CaraNet | CPCANet | CSWin-UNet | U-Net | TransUnet | TransUnet | GH-UNet | CENet | CPCANet |
| #67 | CPCANet | UTANet | TransUnet | TransUnet | DA-TransUNet | UTANet | MADGNet | CaraNet | H2Former |
| #68 | UTANet | CaraNet | TransUnet | UTANet | UTANet | PraNet | TransUnet | UTANet | TransUnet |
| #69 | UNeXt | U-Net | DA-TransUNet | UNet3+ | PraNet | CA-Net | TransUnet | TransNorm | CMU-Net |
| #70 | TransNorm | AttU-Net | UNet3+ | CPCANet | CMU-Net | EViT-UNet | PraNet | PraNet | TransNorm |
| #71 | PraNet | TransNorm | UTANet | DA-TransUNet | CMU-Net | ScribFormer | DA-TransUNet | ScribFormer | PraNet |
| #72 | CMU-Net | PraNet | CMU-Net | UNeXt | TransNorm | CMU-Net | ScribFormer | GH-UNet | CaraNet |
| #73 | EViT-UNet | CMU-Net | CPCANet | PraNet | H-vmunet | DA-TransUNet | CPCANet | CMU-Net | ResNet++ |
| #74 | CFM-UNet | UNet3+ | TransNorm | CMU-Net | MSRFNet | TransNorm | UTANet | ERDUnet | FCBFormer |
| #75 | MedT | EViT-UNet | PraNet | EViT-UNet | EViT-UNet | MSRFNet | TransNorm | TransNorm | UNet3+ |
| #76 | MultiResUNet | ResUNet++ | CMU-Net | TransNorm | TransNorm | FCBFormer | CMU-Net | UCTransNet | UCTransNet |
| #77 | FCBFormer | MSRFNet | EViT-UNet | MSRFNet | UCTransNet | UCTransNet | D-TrAttUNet | FCBFormer | D-TrAttUNet |
| #78 | MSRFNet | FCBFormer | FCBFormer | FCBFormer | FCBFormer | D-TrAttUNet | MSRFNet | D-TrAttUNet | EViT-UNet |
| #79 | UCTransNet | MCA-UNet | UCTransNet | FCBFormer | MCA-UNet | MCA-UNet | FCBFormer | FCBFormer | MERIT |
| #80 | MCA-UNet | UCTransNet | MCA-UNet | D-TrAttUNet | UACANet | SimpleUNet | MCA-UNet | MCA-UNet | MCA-UNet |
| #81 | D-TrAttUNet | UACANet | MSRFNet | MCA-UNet | MERIT | UACANet | UACANet | U-RWKV | ConvFormer |
| #82 | UACANet | Perspective-Unet | D-TrAttUNet | UACANet | UACANet | Perspective-Unet | MCA-UNet | MERIT | TransAttUnet |
| #83 | MERIT | MERIT | MERIT | MERIT | Zig-RiR | MERIT | Perspective-Unet | Perspective-Unet | DS-TransUNet |
| #84 | Perspective-Unet | ConvFormer | UACANet | ConvFormer | ConvFormer | ConvFormer | MERIT | MERIT | CFFormer |
| #85 | ConvFormer | D-TrAttUNet | ConvFormer | Perspective-Unet | DS-TransUNet | DS-TransUNet | DS-TransUNet | CFPNet-M | CFFormer |
| #86 | DS-TransUNet | DS-TransUNet | Perspective-Unet | ULite | CFFormer | CFFormer | CFFormer | DS-TransUNet | U-RWKV |
| #87 | CFFormer | CFFormer | DS-TransUNet | DS-TransUNet | LFU-Net | LFU-Net | Zig-RiR | CFFormer | Zig-RiR |
| #88 | SimpleUNet | LFU-Net | DS-TransUNet | CFFormer | SimpleUNet | SimpleUNet | ULite | ULite | MultiResUNet |
| #89 | LFU-Net | SimpleUNet | CFFormer | SimpleUNet | Tinyunet | UltraLight-VM-UNet | MT-UNet | MambaUnet | DDANet |
| #90 | CMUNeXt | Tinyunet | LFU-Net | ULite | ULite | MultiResUNet | MultiResUNet | DC-UNet | DC-UNet |
| #91 | MambaUnet | UltraLight-VM-UNet | SimpleUNet | MALUNet | MALUNet | MambaUnet | Polyp-PVT | MedFormer | RollingUnet |
| #92 | LeViT-UNet | MultiResUNet | UltraLight-VM-UNet | MUCM-Net | DDS-UNet | MedVKAN | MedVKAN | MedT | DoubleUNet |
| #93 | DC-UNet | MambaUnet | MambaUnet | UltraLight-VM-UNet | UltraLight-VM-UNet | LeViT-UNet | LeViT-UNet | MedT | ColonSegNet |
| #94 | MT-UNet | LeViT-UNet | LeViT-UNet | U-RWKV | U-RWKV | DC-UNet | DC-UNet | MT-UNet | ScribFormer |
| #95 | SwinUNet | BEFUnet | SwinUNet | LeViT-UNet | MT-UNet | MT-UNet | BEFUnet | ColonSegNet | U-Net |
| #96 | Zig-RiR | VMUNet | Zig-RiR | MT-UNet | ERDUnet | DoubleUNet | CFM-UNet | ResUNet++ | AttU-Net |
| #97 | ColonSegNet | SwinUNet | MT-UNet | SwinUNet | SwinUNet | ColonSegNet | ColonSegNet | TransAttUnet | UTANet |
| #98 | ResUNet++ | CFFormer | ColonSegNet | MedT | CFM-UNet | ResUNet++ | PraNet | MSRFNet | MSRFNet |
| #99 | UNETR | ColonSegNet | H-vmunet | H-vmunet | MedT | UNETR | UACANet | EViT-UNet | UACANet |
| #100 | ScribFormer | UNETR | UNETR | UNETR | UNETR | AttU-Net | ConvFormer | ConvFormer | MT-UNet |

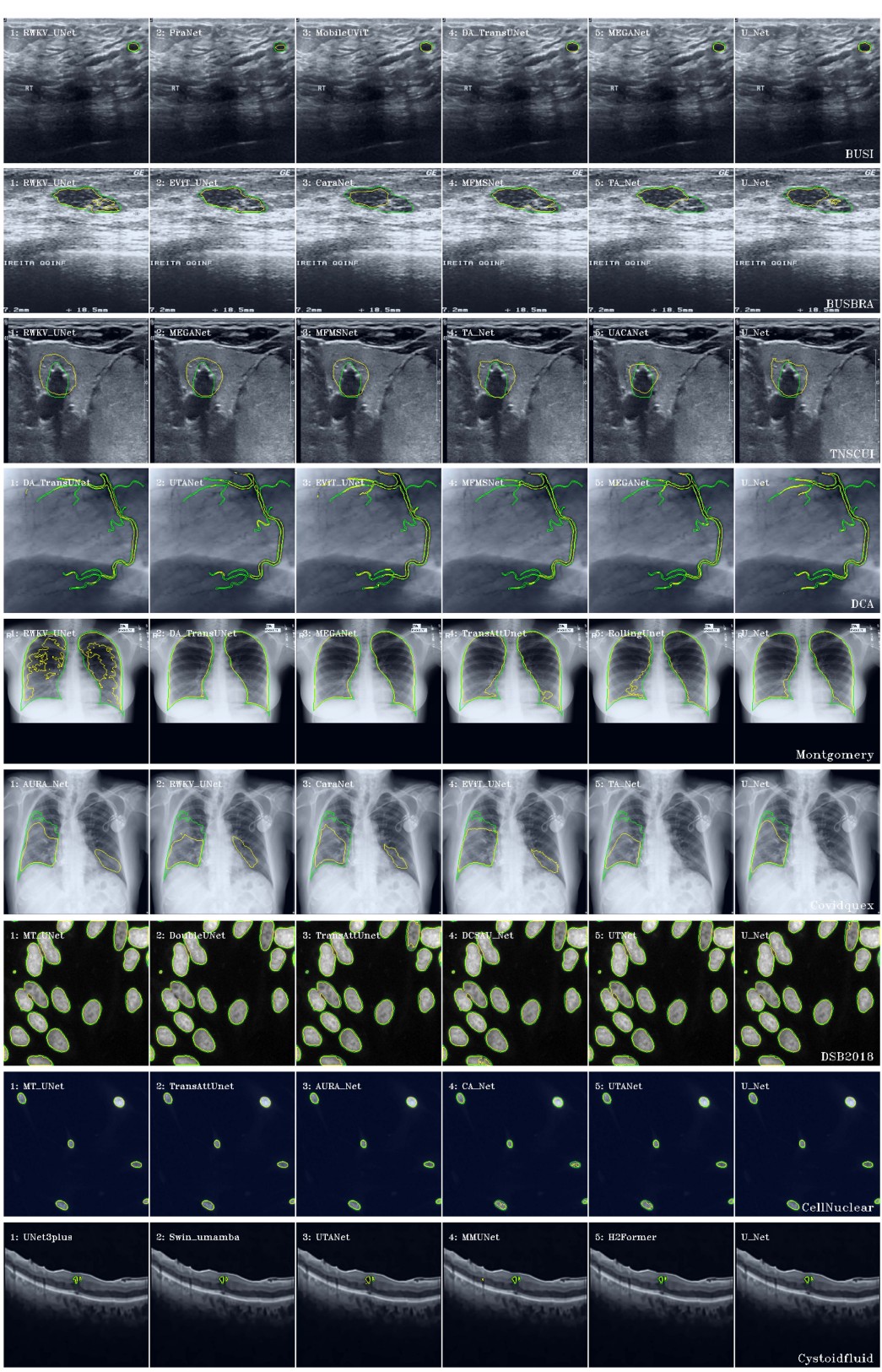

Figure 15: Segmentation results of the Top 5 models and U-Net, where the green curve represents the ground truth and the yellow curve represents the model prediction.

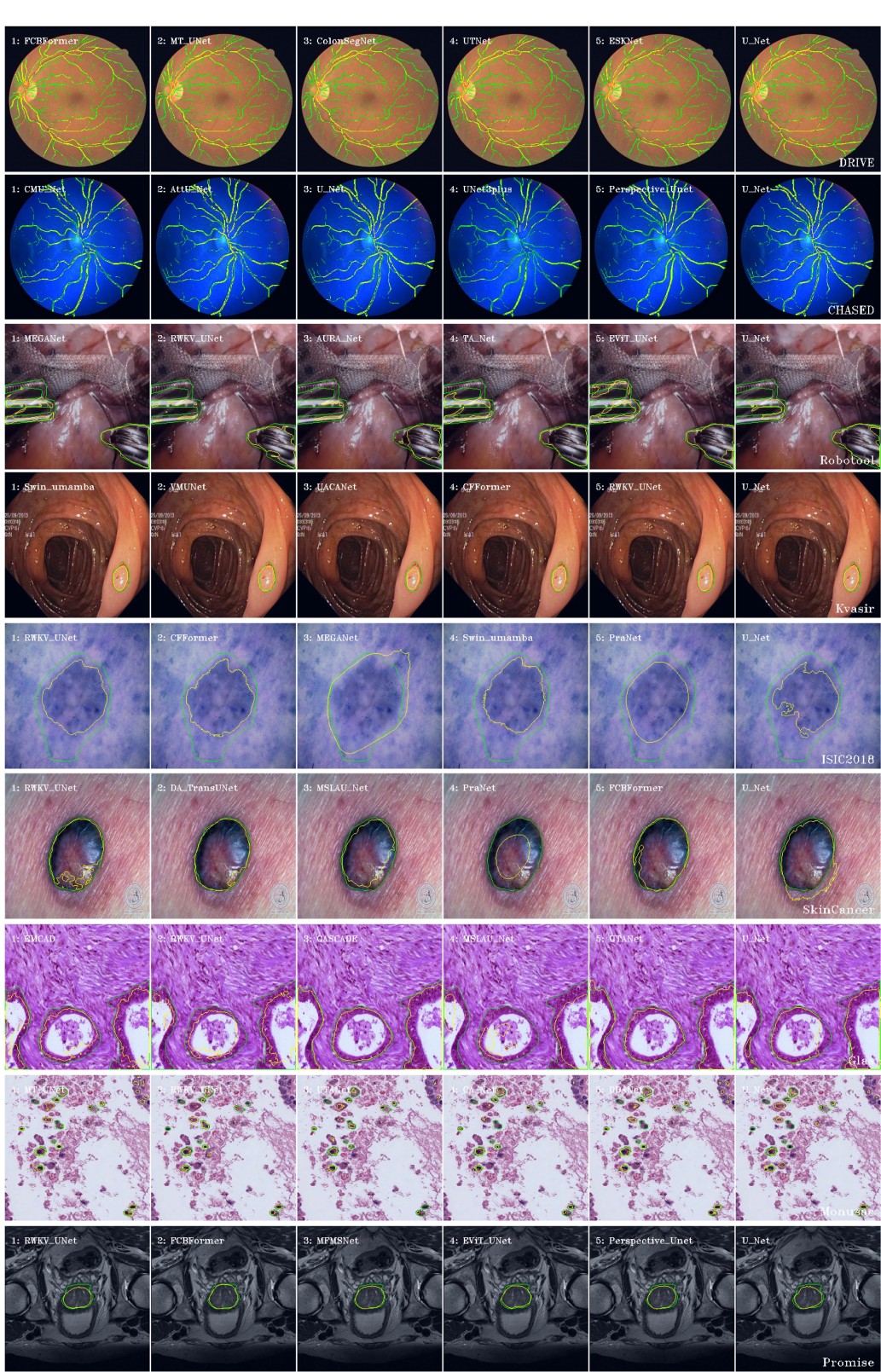

Figure 16: Segmentation results of the Top 5 models and U-Net, where the green curve represents the ground truth and the yellow curve represents the model prediction.

