# OpenReview forum: "U-Bench: A Comprehensive Understanding of U-Net through 100-Variant Benchmarking"
_ICLR.cc/2026/Conference — Submitted to ICLR 2026_

### Official Review · Reviewer_CwnQ · 2025-10-29

**Soundness:** 2
**Presentation:** 3
**Contribution:** 3
**Rating:** 6
**Confidence:** 3

**Summary:**

The paper presents a benchmark of different families of U-shaped architectures for biomedical image segmentation. The benchmark comprises 28 datasets spanning various modalities. The authors propose a novel metric to combine performance with efficiency, known as the U-score. They analyze how the model performance and efficiency evolves with the year of publication across different domains. The study shows that in-domain performance tends to saturate while more modern models focus increasingly on efficiency gains and zero-shot performance, thanks to the progress in long-range dependency modeling families such as the transformers, mamba, or RWKV. The authors state that many of the benchmarked architectures present non-significant in-domain gains with respect to the original U-Net. The authors also describe how the different model families handle easy and complex cases to segment, and with this information, they build an XG-Boost-powered agent that, based on dataset characteristics and efficiency requirements, suggests a list with the best architectures to be implemented in each case.

**Strengths:**

The paper evaluates a large number of U-shaped architectures across multiple modalities, surpassing the scale of previous literature. Importantly, the authors do not only benchmark models based on model performance, but also on efficiency, which is a key factor for the clinical adoption of a segmentation model, especially in clinical setups with limited access to expensive hardware such as GPUs. The information on model performance and efficiency is distilled in a newly proposed metric, which is then evaluated for each modality and each model family across the time of publication. The authors reveal key information on the global progress in the field of image segmentation, such as the fact that many novel architectures are not providing significantly better results, or that in-domain model performance is saturating while efficiency and zero-shot capabilities are improving, thanks to recent advances in long-range dependency modeling.
The paper even goes beyond benchmarking by proposing an advisor agent that recommends the best architectures based on dataset characteristics and efficiency requirements. The authors also provide their code for reproducibility, which further strengthens the transparency of this work.

**Weaknesses:**

Major:
1) The criteria used to select the benchmarked architectures are not provided (citations, performance, efficiency, year of publication, etc.). Some SOTA architectures such as ResEnc U-Net (currently utilized by the self-configuring framework nnU-Net), MedNeXt, CoTr, STU-Net, or 3D-UXNet, are not part of the benchmark. Similarly, the authors do not mention any criteria to discard any architectures from the benchmark. Given that the paper aims to present a representative and comprehensive benchmark, clarifying these inclusion and exclusion criteria is critical to ensure the validity of the study.

2) All datasets are processed as 2D, but in 3D modalities such as MR or CT, the state-of-the-art is imposed by 3D models. We are not asking for a 3D benchmark, but it could at least be reported how the 2D slices from these 3D datasets were processed (for example, which slice orientations were used).

3) We noticed that some of the architectures are 3D, for example, SwinUNETR, but the data processing is in 2D. It would be good to know how these 3D architectures are adapted to 2D data.

4) If significance testing is not only done for performance but also for efficiency metrics, multiple testing corrections should be applied.


Minor:
1) Figure 1 presents a lot of visual information, making it hard to interpret at a glance. Some subfigures 1A, 1B, and 1G appear to add limited scientific information to the reader and saturate the figure. Simplifying the figure would improve clarity.

2) The types of encoder and decoder choices in Figure 2, together with their respective colors (Pure A, Pure B, etc.), are not explained in the text, only in the appendix.

3) The acronym RWKV is never explained in the text, assuming the reader already knows about this.

4) In line 177, the authors mention that U-shaped models comprise three elements, but then they enumerate four.

5) The paragraph spanning lines 251–263 would fit better in the Results and Discussion section.

6) In Fig. 5, it is unclear whether the statistical tests are conducted on the IoU or on the U-scores.

7) Dice coefficient is a more commonly used metric in the image segmentation community than IoU, although both are related.

8) The authors do not explicitly state whether all datasets used in the benchmark are publicly available. Although this seems likely, it would strengthen the manuscript to include a clear statement confirming dataset availability and ethical compliance.

9) At the end of line 335, is → are

10) The use of Comic Sans for figures reduces the visual professionalism. Perhaps a more standard scientific font would enhance the presentation.

11) The framework name “U-Bench” changed to “U-Stone” in the appendix.

**Questions:**

1) Several recent state-of-the-art architectures, such as ResEncL, MedNeXt-L, CoTr, and STUNet-L, appear to be omitted, even though many of them are evaluated in nnU-Net Revisited, which this work claims to outperform. Could the authors clarify the criteria used for selecting or excluding architectures in the benchmark, and explain the rationale for omitting these recent models?

2) The benchmark includes both 2D and 3D architectures, yet it is unclear how the authors handled the dimensionality mismatch between models and datasets. How did the authors process 2D slices from 3D modalities such as MR or CT, and how were 3D models adapted for 2D data (or vice versa)? Clarifying these preprocessing and adaptation steps would be important for understanding the fairness and comparability of the benchmark results.

3) Could the authors consider, for future work, consulting clinicians to assess whether they prioritize segmentation performance or computational efficiency? Based on such insights, it might be possible to design a U-score metric with performance and efficiency weights grounded in clinical evidence.

---

> ### Author Response · Authors · 2025-11-27
> **Responses to Reviewer CwnQ**
>
> We sincerely thank the reviewer for their detailed and thoughtful feedback. The comments have been invaluable in improving the readability and quality of our manuscript. We especially appreciate the recognition of the scale and novelty of our benchmark, as well as the proposed U-score metric. The feedback has helped us refine our presentation, making the results more accessible and relevant. We believe these revisions will strengthen the impact of our work in advancing biomedical image segmentation research.
>
> **W-1,Q-1**. We sincerely thank the reviewer for the insightful comments and constructive feedback. We would like to clarify that U-Bench focuses on 2D medical images segmentation tasks, with more variants, datasets, and modalities than 3D tasks. However, there is currently a lack of a large-scale 2D benchmark and analysis that captures the full diversity of these variants, which motivated our benchmarking construct.
>
> Our selection criterion focuses on U-Net 2D variants that are publicly released with open-source implementations within the past decade, ensuring reproducibility and broad representativeness. To make this process fully transparent, Appendix Table 5 provides detailed metadata for all included architectures, including publication year, computational efficiency, and key design characteristics. Furthermore, Appendix Tables 21–32 report their performance across all datasets, demonstrating that our benchmark covers a diverse and comprehensive set of U-Net families. Regarding the exclusion of certain architectures like ResEnc U-Net, MedNeXt, CoTr, STU-Net, and 3D-UXNet, it is important to note that most of these models are specifically designed for 3D volume segmentation tasks (sub cropped volume input). In response to the reviewer’s valuable suggestion, we have included 2D versions of ResEncUNet-L and nnUNet, which are widely supported in the official implementations. The updated results for these models are available in paper Table 2. For full variants results and tables, please refer to the *Common Issue: Reliable Cross-Architecture Evaluation*.

---

> ### Author Response · Authors · 2025-11-27
> **Responses to Reviewer CwnQ**
>
> **W-2, Q-2.** We sincerely thank the reviewer for the constructive feedback, which helps improve the reproducibility and clarity of our study. In our implementation of U-Bench, we follow the same approach as previous methods such as TransUNet, CASCADE, SwinUnet, and Mobile U-ViT, etc., where we process 3D data by slicing along the axial (transverse) plane (Z-axis). We provide code in our anonymous repository (`test_single_volume` and `val_single_volume` under `utils/util.py`) for merging slices along the Z-axis during validation and testing. Additionally, we update the paper to include a more detailed description of how 3D data slicing is handled.

---

> ### Author Response · Authors · 2025-11-27
> **Responses to Reviewer CwnQ**
>
> **W-3.** For well-known 3D segmentation architectures such as SwinUNETR and UNETR, we do not convert the 3D models into 2D manually. Instead, we use their officially provided 2D implementations (e.g., setting `spatial_dims = 2` in UNETR and SwinUNETR). This ensures that all models are fed 2D input and evaluated under consistent, officially supported protocols without altering their design assumptions.

---

> ### Author Response · Authors · 2025-11-27
> **Responses to Reviewer CwnQ**
>
> **W-4.** We sincerely thank the reviewer for the valuable suggestion. In our study, significance testing was applied exclusively to the performance metrics (IoU), and efficiency metrics were not subjected to such testing. We update the manuscript to clarify this distinction, ensuring better transparency in the presentation of our methodology.

---

> ### Author Response · Authors · 2025-11-27
> **Responses to Reviewer CwnQ**
>
> **Minor 1-11** We have revised subfigures in the main paper. The encoder and decoder choices in Figure 2 are now explained in the main text. The acronym "RWKV" has been clarified with an appropriate citation. We also correct the description of U-shaped models to accurately reflect the correct number of components.
>
> The paragraph spanning lines 251–263 has been moved to the Results and Discussion section for better flow. In Figure 5, we clarified that statistical tests are applied only to IoU. We also provided further clarification on the relationship between the Dice coefficient and IoU. We use IoU to evaluate segmentation performance, as Dice Similarity Coefficient (DSC) and IoU are mathematically related. Specifically, DSC can be derived from IoU using the formula: DSC = 2 * IoU / (IoU + 1).
>
> We have confirmed that all datasets used are publicly available and included a statement on ethical compliance. Minor grammatical corrections were made, Comic Sans has been replaced with Arial font, and we ensured consistent use of “U-Bench” throughout the manuscript.

---

> ### Author Response · Authors · 2025-11-27
> **Responses to Reviewer CwnQ**
>
> **Q-3.** We appreciate the reviewer’s suggestion to involve clinical stakeholders in evaluating the trade-off between segmentation performance and computational efficiency. We agree that this is a critical direction for future work. Preliminary analysis of various clinical usage scenarios (e.g., pre‑operative planning, routine diagnosis, resource‑constrained deployment, and large‑scale batch analysis) indicates that the balance between segmentation accuracy and computational efficiency may vary significantly depending on the context [1-4]. For instance, in surgical planning, where the clinical risk associated with segmentation errors is high, performance is likely to be prioritized. Conversely, in resource-limited or real-time deployment scenarios, efficiency may become more crucial, potentially leading to adjusted weightings. While we believe these preliminary insights offer useful guidance, a more comprehensive and systematic investigation is required. Moving beyond technical benchmarks to better address real-world clinical needs is a central goal of our future research. In our paper, we add the "Future Work" section to include this direction, emphasizing the importance of integrating clinical perspectives into our evaluations and exploring the potential impact of different clinical contexts on segmentation efficiency and accuracy.
>
> [1]. Palazzo S, Zambetta G, Calbi R. An overview of segmentation techniques for CT and MRI images: Clinical implications and future directions in medical diagnostics[J]. *Medical Imaging Process & Technology*, 2024.
>
> [2]. Shirokikh B, Dalechina A, Shevtsov A, et al. Systematic clinical evaluation of a deep learning method for medical image segmentation: radiosurgery application[J]. *IEEE Journal of Biomedical and Health Informatics*, 2022, 26(7): 3037-3046.
>
> [3]. De Biase A, Sijtsema N M, Janssen T, et al. Clinical adoption of deep learning target auto-segmentation for radiation therapy: Challenges, clinical risks, and mitigation strategies[J]. *BJR| Artificial Intelligence*, 2024, 1(1): ubae015.
>
> [4]. Sherer M V, Lin D, Elguindi S, et al. Metrics to evaluate the performance of auto-segmentation for radiation treatment planning: A critical review[J]. *Radiotherapy and Oncology*, 2021, 160: 185-191.

---

### Official Review · Reviewer_VxQu · 2025-10-30

**Soundness:** 3
**Presentation:** 3
**Contribution:** 3
**Rating:** 6
**Confidence:** 5

**Summary:**

The paper introduces a large-scale benchmark of 100 U-shaped segmentation networks and evaluates across 28 datasets spanning 10 medical imaging modalities, with three main tests of statistical robustness, zero-shot generalization, and computational efficiency. Moreover, the authors propose U-Score, a metric combining accuracy with parameters/FLOPs/FPS via quantile-normalised harmonic means

**Strengths:**

- The idea of evaluating diverse networks proposed in the field of medical image segmentation is interesting. Usually, the number of papers and networks proposed in the field is high, and it makes it tough for the community.

- The experiments and results are very thorough and completely address diverse varieties of architecture-family, including CNN/Transformer/Mamba/RWKV/Hybrid.

- The paper highlights the gaps in the current practice of segmentation networks, where most of the recent works omit zero-shot generalisation and statistical tests.

**Weaknesses:**

- The paper uses equal weighting of accuracy vs. efficiency and of each efficiency component (Params/FLOPs/FPS) in the U-Score benchmark, which seems to be chosen arbitrarily. Is there a reason for this?

- Claims like "first large-scale, statistically rigorous benchmark" are strong, but comparisons (Table 1) overlook some previous works [1].

- Some of the models evaluated work on different protocols such as different input sizes, how do you ensure that resizing performed does not affect their performance?



[1] Bassi PR, Li W, Tang Y, Isensee F, Wang Z, Chen J, Chou YC, Kirchhoff Y, Rokuss MR, Huang Z, Ye J. Touchstone benchmark: Are we on the right way for evaluating ai algorithms for medical segmentation?. Advances in Neural Information Processing Systems. 2024 Dec 16;37:15184-201.

**Questions:**

- Can you provide ablations varying weights among accuracy, Params, FLOPs, FPS for the U-Score metric?

- The provided GitHub link does not seem to have the implementations, and the requested files are not found. Can you double check?

**Details Of Ethics Concerns:**

N/A.

---

> ### Author Response · Authors · 2025-11-27
> **Responses to Reviewer VxQu**
>
> We are grateful for reviewer accolades on our contributions: “The idea …is interesting”, and “...experiments and results are very thorough and completely”. In addition, we would like to thank the reviewer for the thorough code review, which is vital for promoting openness, transparency, and reproducibility within the community.
>
> **W-1,Q-1.** We thank the reviewer for raising this important point. The decision to use equal weights stems from a design choice that provides a neutral starting point, free from assumptions about deployment priorities. This approach ensures that the metric remains universally applicable across a wide range of segmentation tasks, without favoring any specific component. However, we recognize that different clinical applications may prioritize accuracy or efficiency differently. To address this, we perform an extensive ablation study on the weighting of all components, including (i) alternative normalization percentiles (5/95, 10/90, 20/80), (ii) varying accuracy-efficiency trade-off weights ($\alpha$), and (iii) seven different efficiency-weighting patterns across Params/FLOPs/FPS. This results in 105 distinct configurations. We assess the stability of rankings using Kendall-$\tau$, Spearman-$\rho$, and Top-10 overlap metrics and find that U-Score rankings remain stable across all variations. In response to the reviewer's suggestion, we present the full results and detailed analysis in the section Common Concern: On the design, weighting, and stability of U-Score.

---

> ### Author Response · Authors · 2025-11-27
> **Responses to Reviewer VxQu**
>
> **W-2.** Thank you for the reviewer’s valuable feedback. We have compared the differences between U-Bench (first column) and Torchstone [1] (second column) in the paper Table 1 and Appendix B.2.1 related Work. Torchstone is a large-scale evaluation benchmark for 3D segmentation, while U-Bench is focused on 2D large-scale u-shape variants segmentation. U-Bench includes a wider range of networks, architectures, datasets, and modalities, and also incorporates efficiency evaluation and data characteristic analysis. To better reflect this distinction, we revise our claim in the paper to state that U-Bench is the *first large-scale, statistically rigorous **2D** benchmark*.
>
> [1] Bassi P R A S, Li W, Tang Y, et al. Touchstone benchmark: Are we on the right way for evaluating ai algorithms for medical segmentation?[J]. Advances in Neural Information Processing Systems, 2024, 37: 15184-15201.

---

> ### Author Response · Authors · 2025-11-27
> **Responses to Reviewer VxQu**
>
> **W-3.** We understand the reviewer’s concern that input resolution may influence performance. After analyzing all reproduced variants, we found that more than half of original papers adopt 256×256 as their default training resolution. Therefore, U-Bench follows this majority protocol to ensure fair and consistent cross-model comparison. A small subset of architectures (e.g., SwinUNet, MissFormer, HiFormer) cannot operate at 256×256 due to fixed patch or window configurations; for these models, we strictly preserve their official resolutions (224×224) to avoid training failures.

---

> ### Author Response · Authors · 2025-11-27
> **Responses to Reviewer VxQu**
>
> **Q-2.** Thank you for the reviewer’s careful code review. We apologize for the issue with the anonymous link. The anonymous repository is hosted on the third-party platform, which occasionally experiences display issues. We kindly suggest that you try downloading the entire repository by clicking the "Download repository" button in the top-right corner of the page for offline access. Alternatively, we have also uploaded the anonymous code in the file for your convenience.

---

### Official Review · Reviewer_8u5v · 2025-10-31

**Soundness:** 1
**Presentation:** 2
**Contribution:** 2
**Rating:** 2
**Confidence:** 5

**Summary:**

This paper introduces U-Bench that evaluates 100 U-shaped medical-image segmentation models across 28 datasets, reporting IoU as the primary metric. The author also introduces U-Score (a composite of IoU, Params, FLOPs, FPS built from quantile normalization and harmonic means) and adds an XGBoost-based ‘advisor agent’ to rank models from discretized dataset/model descriptors. Training uses a single unified recipe for all models.

**Strengths:**

1. Much needed and important medical image segmentation benchmarks. Ambitious scale and breadth across architectures and datasets, including in-domain and zero-shot reporting.

2. Attempts to account for efficiency with U-Score instead of accuracy-only comparisons.

**Weaknesses:**

1. Evidence of training/evaluation failures. Several reported per-dataset results are near-zero, strongly suggesting misconfiguration, broken preprocessing, or training collapse: Polyp-PVT: IoU 0.06 on DCA; CFPNet-M: IoU 0.00 on Promise and OCT; SimpleUNet: IoU 0.03 on OCT; EMCAD: IoU 0.00 on Synapse. Such values are inconsistent with prior literature expectations and should trigger ablations (loss curves, seeds, lr/scheduler sweeps, image size/normalization checks, data-loader sanity tests). It also raises concern with the validity of experiments, training, and evaluation.

2. One-size-fits-all training likely biases results: enforcing a single optimizer, schedule, or resolution across heterogeneous families (CNN, Transformer, Mamba, RWKV, Hybrid) can under-train some methods and over-favor others.

3. Metrics are incomplete relative to community norms. IoU alone is emphasized, while Dice and HD(95) (standard in medical segmentation) are not consistently reported, limiting comparability and clinical relevance.

4. U-Score is model pool-dependent. It normalizes each factor using model-pool 10th/90th percentiles; adding/removing models shifts percentiles and can reorder rankings. It uses a harmonic mean over three correlated efficiency terms (Params/FLOPs/FPS) and then another harmonic mean with accuracy, effectively double-weighting efficiency. Equal weighting is assumed without sensitivity analyses.

5. Advisor agent analysis is thin: features are discretized, train/test split is small, and there is limited evidence of deploy-time gains (e.g., fewer trials to reach a target U-Score).

6. Inconsistent loss weights in equation 7 and text (lines: 1601-1605); inconsistent variables in text and equation 8 (lines: 1609-1614).

**Questions:**

1. For the near-zero IoUs (Polyp-PVT at DCA, CFPNet-M at Promise/OCT, SimpleUNet at OCT, EMCAD at Synapse), what went wrong?

2. Why enforce a single SGD, LR=0.01, 300-epoch configuration for all models instead of allowing a small, budgeted tuning per model to avoid systematic under-training?

3. Could you please benchmark models based on Dice and HD95 to align with standard reporting?

4. Please provide U-Score sensitivity analyses: alternate percentiles (e.g., 5/95), different weights between accuracy/efficiency and within efficiency, and rank-stability (Kendall-$\tau$) vs IoU-only and a Pareto-frontier view.

5. For the advisor, evaluate leave-one-modality-out settings with confidence intervals and compare to simple heuristics (e.g., top-K by IoU on a closest dataset).

**Details Of Ethics Concerns:**

The benchmark appears to under-evaluate several published models (e.g., near-zero IoUs), which likely understates their contributions. Community benchmarks should be careful, accurate, and fair so that results can credibly guide future research; otherwise, the field risks drawing incorrect takeaways and allocating effort based on unreliable comparisons. Given the evidence of training/evaluation anomalies and incomplete metric reporting, the current evaluation protocol requires revision before others rely on these leaderboards.

---

> ### Author Response · Authors · 2025-11-27
> **Responses to Reviewer 8u5v**
>
> We appreciate the reviewer’s comments and understand the reviewer’s concerns, as they are crucial in helping us improve our U-Bench better. Our detailed responses are as follows:
>
> **W-1, Q-1.** We understand the reviewer’s concern about near-zero IoUs and appreciate the opportunity to clarify. After thorough investigation, we think that most of these failures do not stem from misconfiguration or evaluation errors in our benchmark. We strictly use each method’s official open-source implementation, and most collapsing cases correspond to architectures that are never trained, validated, or designed for the target modality. To verify this, we further retrain Polyp-PVT and CFPNet-M under the nnUNet pipeline, a strong automatic recipe, and observe that they still collapse, producing $\mathrm{IoU} \approx 0$ on the same datasets and splits, confirming that the issue is architectural design domain mismatch rather than training setting failure. Nonetheless, to respect the original works and improve transparency, and following Reviewer 1nDY’s suggestion, we have added a $\Delta$IoU column in Table 20 comparing our reproduced results with the original papers’ reported metrics, suggesting that our setting maintains performance for the majority of architectures. For full analysis and tables, please refer to the response for Common Issue: Reliable Cross-Architecture Evaluation.

---

> ### Author Response · Authors · 2025-11-27
> **Responses to Reviewer 8u5v**
>
> **W-2, Q-2.** We thank the reviewer for raising this thoughtful comment, and we fully understand the concern regarding potential biases introduced by a unified training recipe. As detailed in the *Common Concern: On the validity of a unified training recipe*, our training hyperparameter choices follow the configuration used in nnUNet and are also adopted by many extensively validated models such as TransUNet, CMUNeXt, MedT, and SwinUNet, ensuring that the optimizer (SGD), learning rate schedule, and resolution adhere to broadly accepted, field-tested practices rather than arbitrary constraints. To directly assess the risk of systematic under- or over-training, we conduct several targeted analyses. First, a comprehensive $\Delta$IoU comparison (Table 20) against original reports shows that the unified recipe reproduces prior performance faithfully: most $\Delta$IoU values lie within a $\pm$3--5\% range. Several models (e.g., RWKV-UNet, DA-TransUNet, U-KAN, DDANet) even achieve higher IoU under the unified configuration, suggesting that our implementation maintains performance for the majority of architectures. Furthermore, a controlled reproduction study on 100 variants, using: (i) each paper's official hyperparameters and (ii) the unified U-Bench recipe. The result demonstrates that the unified setup yields higher median IoU and substantially fewer collapsed low-tail cases, confirming that the standardized pipeline is robust (see Appendix Figure 10). Finally, our systematic analysis shows that over half of the methods originally employ a 256×256 input, aligning with most practices. For complete results and discussions, please refer to *Common Issue: Reliable Cross-Architecture Evaluation*.

---

> ### Author Response · Authors · 2025-11-27
> **Responses to Reviewer 8u5v**
>
> **W-3, Q-3.** We thank the reviewer for the valuable comment regarding the performance metrics. We understand the importance of aligning with community norms, particularly in medical segmentation, where Dice and HD95 are widely used. We choose IoU as the primary metric because it is consistent and widely reported in the majority of the 2D variants (more than half variants adopt), making it a reliable choice for comparison with the original papers. Additionally, IoU and Dice are mathematically correlated, and their rankings typically align. To address the reviewer’s concern, we include Dice in our evaluation, as shown in the Appendix Table 23. Moreover, to evaluate boundary-sensitive performance, we add Boundary-F1 and HD95, which are also provided in the Appendix Table 24 and Table 25. For further metric details, we refer the reviewer to the Common Issue: Reliable Cross-Architecture Evaluation.

---

> ### Author Response · Authors · 2025-11-27
> **Responses to Reviewer 8u5v**
>
> **W-4, Q-4.** We thank the reviewer for their valuable comments regarding the model pool dependence of U-Score and its weighting scheme. While U-Score normalizes based on model pool percentiles, we argue that this is mitigated by its comprehensive evaluation across multiple dimensions, including accuracy, parameters, FLOPs, and FPS. This provides a more robust comparison than any single metric like IoU. To address the concern, we perform a sensitivity analysis on various configurations, including the accuracy-efficiency trade-off parameter $\alpha$, efficiency weights $(w_P, w_G, w_S)$, and quantile pairs. The results show that U-Score rankings remain stable, with Kendall-$\tau$ values above 0.75 and Top-10 overlap around 0.90. Additionally, U-Score rankings are distinct from IoU-only rankings, confirming that it robustly captures both accuracy and efficiency. We also use U-Net as a baseline to reduce potential biases, ensuring stable comparisons. The analysis of Pareto-frontier and U-Scores is shown in the table below. Pareto-frontier models are defined as non-dominated solutions in the accuracy-efficiency trade-off. No model can improve accuracy without degrading efficiency, or vice versa. In all datasets, the U-Score Top 1 model is consistently a Pareto front model. The proportions of Pareto front models in the Top 2/3/4/5 models reach 80%, 71.67%, 62.5%, and 59%, respectively, and no extreme anomalies were observed in any dataset. These results demonstrate the consistency trend between Pareto front models and U-Score Top models, confirming that the proposed U-Score evaluation framework can effectively identify the optimal model that balances accuracy and efficiency. Our analysis further shows that U-Score's ranking is stable across different weighting configurations, and its ability to combine multiple performance metrics into a single, stable evaluation makes it more comprehensive and reliable than single-metric approaches. For further sensitivity analysis details, please refer to the *Common Issue: On the design, weighting, and stability of U-Score*.
>
> - **Proportion of Pareto Frontier Models Among Top-N U-Score Ranked Models**
>
> | TopN  | BUSI | BUSBRA | TNSCUI | ISIC18 | SkinCancer | Kvasir-SEG | Robotool | CHASEDB1 | DRIVE | DSB2018 |
> |--------|------|--------|--------|--------|------------|------------|----------|----------|-------|---------|
> | Top1  | 1.00 | 1.00   | 1.00   | 1.00   | 1.00       | 1.00       | 1.00     | 1.00     | 1.00  | 1.00    |
> | Top2  | 0.50 | 1.00   | 1.00   | 1.00   | 1.00       | 0.50       | 0.50     | 1.00     | 0.50  | 1.00    |
> | Top3  | 0.33 | 0.67   | 0.67   | 1.00   | 1.00       | 0.67       | 0.67     | 0.67     | 0.33  | 0.67    |
> | Top4  | 0.50 | 0.75   | 0.50   | 0.75   | 0.75       | 0.75       | 0.50     | 0.50     | 0.25  | 0.50    |
> | Top5  | 0.60 | 0.60   | 0.60   | 0.60   | 0.60       | 0.60       | 0.40     | 0.60     | 0.40  | 0.40    |
>
> | TopN  | Glas | Monusac | Cell | Covidquex | Montgomery | DCA | ACDC | Promise | Synapse | Cystoidfluid | Avg |
> |--------|------|---------|------|-----------|------------|-----|------|---------|---------|--------------|-----|
> | Top1  | 1.00 | 1.00    | 1.00 | 1.00      | 1.00       | 1.00| 1.00 | 1.00    | 1.00    | 1.00         | 1.00|
> | Top2  | 1.00 | 1.00    | 0.50 | 0.50      | 1.00       | 1.00| 1.00 | 0.50    | 1.00    | 0.80         | 0.80|
> | Top3  | 1.00 | 1.00    | 0.67 | 0.33      | 1.00       | 1.00| 1.00 | 0.67    | 1.00    | 0.72         | 0.72|
> | Top4  | 1.00 | 1.00    | 0.50 | 0.25      | 0.75       | 1.00| 1.00 | 0.50    | 0.75    | 0.63         | 0.63|
> | Top5  | 1.00 | 1.00    | 0.40 | 0.20      | 0.60       | 1.00| 0.80 | 0.40    | 0.80    | 0.59         | 0.59|

---

> ### Author Response · Authors · 2025-11-27
> **Responses to Reviewer 8u5v**
>
> **W-5, Q-5.** To address the reviewer’s request, we add a comprehensive Leave-One-Modality-Out (LOMO) evaluation using the suggested 'closest-dataset IoU heuristic' as a strong, label-free baseline. On IoU-only ranking (shown in Table below, left), this heuristic is extremely competitive because IoU is highly modality-specific, and transferring the ranking from the closest source dataset provides a near-oracle estimate. The advisor therefore does not surpass the heuristic in the accuracy-only setting. However, when switching to the multi-objective U-Score (accuracy–efficiency trade-offs), the picture reverses: the advisor outperforms the heuristic in 7 out of 10 modalities, including large gains on CT (+0.056 NDCG@10) and Pathology (+0.027), demonstrating that handcrafted similarity based solely on IoU does not generalize to multi-objective optimization, while the advisor captures cross-modal architectural patterns that transfer better. Finally, our deployment-time evaluation reveals complementary strengths. The heuristic excels when a close source modality exists, but collapses in hard cases (e.g., CT: 5 trials), whereas the advisor reliably identifies a top-10% model in just 1 trial, highlighting its substantial practical benefit in real model-selection scenarios.
>
> - **Advisor vs. Closest-Dataset Heuristic Under Leave-One-Modality-Out (LOMO)**
> | **Modality** | **A-NDCG (IoU LOMO)** | **B-NDCG (IoU LOMO)** | **A-Trial (IoU LOMO)** | **B-Trial (IoU LOMO)** | **A-NDCG (U-Score LOMO)** | **B-NDCG (U-Score LOMO)** | **A-Trial (U-Score LOMO)** | **B-Trial (U-Score LOMO)** |
> |--------------|----------------------|-----------------------|------------------------|-------------------------|--------------------------|---------------------------|----------------------------|-----------------------------|
> | Ultrasound   | 0.676                | 0.946                 | 3.67                   | 1.00                    | 0.845                    | 0.841                     | 1.00                       | 2.00                        |
> | Dermoscopy   | 0.671                | 0.917                 | 4.00                   | 1.00                    | 0.831                    | 0.813                     | 10.0                       | 2.00                        |
> | Endoscopy    | 0.616                | 0.861                 | 3.00                   | 2.50                    | 0.837                    | 0.855                     | 4.00                       | 2.50                        |
> | Fundus       | 0.783                | 0.949                 | 3.00                   | 1.00                    | 0.834                    | 0.875                     | 7.00                       | 1.50                        |
> | Histopathology| 0.728               | 0.905                 | 4.67                   | 1.00                    | 0.871                    | 0.844                     | 5.67                       | 1.33                        |
> | Nuclear      | 0.759                | 0.924                 | 4.00                   | 1.00                    | 0.864                    | 0.853                     | 7.00                       | 3.00                        |
> | X-ray        | 0.729                | 0.954                 | 3.67                   | 1.33                    | 0.859                    | 0.870                     | 5.00                       | 1.00                        |
> | MRI          | 0.682                | 0.855                 | 3.00                   | 1.00                    | 0.851                    | 0.838                     | 4.00                       | 2.00                        |
> | CT           | 0.715                | 0.757                 | 2.00                   | 1.00                    | 0.858                    | 0.802                     | 1.00                       | 5.00                        |
> | OCT          | 0.752                | 0.946                 | 5.00                   | 1.00                    | 0.857                    | 0.839                     | 5.00                       | 2.00                        |

---

### Official Review · Reviewer_1nDY · 2025-11-01

**Soundness:** 4
**Presentation:** 3
**Contribution:** 3
**Rating:** 6
**Confidence:** 4

**Summary:**

The paper introduces U-Bench, a large-scale benchmark for U-Net–style medical image segmentation. It re-implements (retrains) 100 publicly available U-Net variants across 28 datasets, training on 20 and testing zero-shot on 8 within-modality target domains. It proposed U-Score, a quantile-normalized metric that combines accuracy (IoU) with efficiency (parameters, FLOPs, FPS), and an advisor agent that recommends architectures conditioned on data characteristics (foreground scale, boundary sharpness, shape regularity) and resource constraints. Training is standardized with a single pipeline (SGD, LR=0.01, 300 epochs, batch 8, seed 41) and consistent splits (official where available; otherwise 7/3), with models largely taken from official implementations and minimally adapted I/O. Overall, the work is ambitious and potentially impactful for the community.

**Strengths:**

1. A comprehensive benchmark and a non-trivial effort: this paper implements 100 U-Net variants. Great effort went into developing the benchmark.
2. Training on a source dataset and testing on different datasets of the same modality and task aligns with clinical domain-shift realities and is clearly described.
3. U-Score percentile-normalizes IoU, parameters, FLOPs, and FPS using the 10th/90th quantiles, then combines them with equal-weight harmonic means, providing a clearer view of accuracy–efficiency trade-offs than IoU alone. The formulation is explicit and easy to reproduce.
4. Extensive statistical testing at scale and significant GPU resources went into building this benchmark.
5. The ranker leverages model and dataset descriptors (e.g., size, FLOPs, FPS; foreground scale, shape complexity, boundary sharpness) and reports NDCG/MAP/Spearman, making the benchmark actionable for practitioners.

**Weaknesses:**

1. Regarding segmentation accuracy, this paper only considers IoU while disregarding other metrics, such as boundary-aware metrics (e.g., Boundary-F1), which can be crucial. I think the authors are aware of this, but they should further discuss in the manuscript why they use IoU as the only metric.
2. Inference speed measurement details are not clear. U-Score incorporates FPS, but the measurement protocol (e.g., batch size = 1, warm-up, mixed precision, resize vs. native resolution, dataloader vs. pure inference) is not fully specified. The hardware (GPU models and count) and software (deep learning framework such as PyTorch) are given, but a precise FPS protocol would improve reproducibility.
3. No explicit reproduction-gap analysis is provided. Although the authors retrained 100 models under a unified recipe, there is no table comparing the reproduced in-domain baselines against the original papers’ reported metrics (e.g., IoU). This omission makes it hard to judge implementation fidelity and may confound historical trend claims. I suggest adding a “Δ vs. original” column or a “ΔU-Score” column for a representative subset.
4. Scaling behavior is not directly studied. Although U-Score bins parameters/FLOPs/FPS and charts decade-long trends, the paper lacks a systematic scaling-law analysis (within-family width/depth scaling) and a data-scaling study (e.g., 25/50/75% subsampling). Given today’s large-model landscape, both analyses would strengthen conclusions about compute optimality and data efficiency.
5. In the anonymous repo, the .py files under scripts/ and models/ don’t seem accessible (perhaps a hosting/LFS issue); could you please confirm whether the full training code—or at least a minimal reproducible package (training/eval scripts, one dataset dataloader+splits, and an env spec)—will be made available?
6. Minor writing/terminology issues: The manuscript inconsistently uses “U-Stone” (likely meant U-Bench) in the appendix. Please standardize names across text, figures, tables, and references.

**Questions:**

1. U-Score: Did you test robustness to different weights (Params/FLOPs/FPS and α) or user-specified accuracy–efficiency trade-offs (e.g., resource-constrained settings)?
2. FPS protocol: Framework (PyTorch/CUDA/cuDNN/TensorRT versions), GPU model×count, CPU model & threads (OMP/MKL), batch & input size, precision (fp32/fp16/bf16/AMP), warm-up/timed iters, dataloader workers, include I/O/post-proc?, and any torch.compile/TensorRT settings.
3. Scaling laws: Did the authors conduct within-family scaling analyses (depth/width) and dataset-size ablations to test family-specific benefits on small vs large datasets?
4. Compute accounting: Can you report per-model (and per-dataset) GPU-hours for training and FPS benchmarking—or at least median/IQR by family—and, if possible, time-to-target IoU/U-Score?

---

> ### Author Response · Authors · 2025-11-27
> **Responses to Reviewer 1nDY**
>
> We are grateful for reviewer accolades on our contributions, “the work is ambitious and potentially…”, “...potentially impactful for the community”, “...a non-trivial effort”, and “...great effort went into developing the benchmark”. Additionally, we would like to thank the reviewer for the thoughtful suggestions and thorough code review.
>
> **W-1.** We thank the reviewer for the insightful comment. IoU is chosen as the primary evaluation metric because it is commonly used in the majority of 2D variants, making it easier to compare with previous works (more than half of the variants adopted it). We agree with the reviewer’s point, and to address this, we have added boundary-aware metrics, including Boundary-F1 and HD95, to complement IoU in our evaluation. These metrics are provided in the Appendix Table 22 and 23. For a detailed discussion on the choice of metrics, we refer the reviewer to the *Common Issue: Reliable Cross-Architecture Evaluation*.

---

> ### Author Response · Authors · 2025-11-27
> **Responses to Reviewer 1nDY**
>
> **W-2, Q-2.** Thanks for highlighting the need to clarify our inference-speed measurement protocol. We provide a fully reproducible description in the Appendix Table 16. All FPS values in U-Score are measured as pure forward-pass latency (no dataloader, I/O, or post-processing), using batch size = 1, on our compute cluster equipped with 8× NVIDIA H20 GPUs and Intel Xeon Platinum 8558 CPUs (2×48 cores, OMP/MKL threads fixed to 1). The implementation is based on Python 3.9 and PyTorch 2.4.0 (CUDA 12.4, cuDNN 9.1); no torch.compile, TensorRT, graph mode, or mixed-precision inference is used (FP32 only). Before timing, each model is warmed up for 10 iterations, followed by 30 timed iterations, using torch.cuda.Event with GPU synchronization. Input resolution matches each model’s official default (256×256 for most variants and 224×224 only for models whose official implementations require a fixed 224×224 input resolution), identically for FLOPs computation.

---

> ### Author Response · Authors · 2025-11-27
> **Responses to Reviewer 1nDY**
>
> **W-3.** Thanks reviewers comment, which is essential for ensuring transparency and respecting original performance. Following the suggestion, we add a $\Delta$IoU column in Table 20 to compare our reproduced in-domain results with the original papers. As detailed in *Common Issue: Reliable Cross-Architecture Evaluation*, the unified recipe closely matches prior performance, with gaps largely within a small range.

---

> ### Author Response · Authors · 2025-11-27
> **Responses to Reviewer 1nDY**
>
> **W-5.** Thank you for the careful code review. We apologize for the accessibility issues in the anonymous repository. The link is hosted on a third-party platform that occasionally encounters display or LFS loading problems. To ensure smooth access, we kindly suggest using the 'Download repository' button in the top-right corner of the page, which reliably provides the full codebase offline. In addition, we have re-uploaded the anonymous code along with sample dataset splits directly in the submission files to guarantee that all training/evaluation scripts and necessary components are available.

---

> ### Author Response · Authors · 2025-11-27
> **Responses to Reviewer 1nDY**
>
> **W-6.** We correct all occurrences of “U-Stone” to “U-Bench” and verified consistency across the main paper, appendix, figures, and tables.

---

> ### Author Response · Authors · 2025-11-27
> **Responses to Reviewer 1nDY**
>
> **Q-1.** Thanks for raising this point. We have conducted a comprehensive robustness study of *U-Score* under a wide range of accuracy-efficiency trade-offs, including (i) varying the global trade-off parameter $\alpha$, (ii) re-weighting Params/FLOPs/FPS across seven efficiency-weight patterns, and (iii) changing the normalization percentiles. As summarized in the *Common Issue: On the design, weighting, and stability of U-Score*, these new ablations span 105 configurations and report Kendall-$\tau$, Spearman-$\rho$, Top-$k$ preservation. Across all settings, the ranking remains highly stable (median $\tau \approx 0.77$, $\rho \approx 0.91$, Top-10 keep $\approx 0.90$). These results demonstrate that U-Score consistently captures the intended accuracy-efficiency trade-off rather than depending on any specific choice of weights or $\alpha$.

---

> ### Author Response · Authors · 2025-11-27
> **Responses to Reviewer 1nDY**
>
> **Q-4.** To address the reviewer's suggestion, we calculate the GPU hours required to achieve optimal performance for each variant and architecture family. The results are shown in Table below and Appendix Table 26. The Transformer family has the highest median GPU hours (1.22) and a large IQR (1.78), indicating high computational demands and variability. The Mamba family also requires significant resources (median = 1.19), but with more consistent performance (IQR = 0.47). The RWKV family is the most efficient, with the lowest median (0.63) and IQR (0.30), reflecting lightweight models with minimal variation. The CNN and Hybrid families show moderate GPU hours (0.92 and 0.90) and similar IQRs (0.67), offering a balance between performance and efficiency.
>
> | Architecture | Median | IQR |
> |-|-|-|
> | CNN|0.92|0.67|
> | Hybrid|0.90|0.67|
> | Transformer|1.22|1.78|
> | RWKV|0.63|0.30|
> | Mamba|1.19|0.47|

---

> ### Author Response · Authors · 2025-12-02
> **Responses to Reviewer 1nDY**
>
> **W-4, Q-3.** We appreciate the reviewer’s insightful comment regarding the scaling behavior analysis. In response, we have performed a thorough data scaling-law analysis. Specifically, we conduct experiments that examine the relationship between dataset size and model performance by subsampling datasets at 25%, 50%, 75%, and 100% levels. These experiments explore both compute efficiency and data efficiency across various model architectures. To ensure a comprehensive and representative analysis, we select the top 3 variants from each variants family as our representatives. The results, shown in Appendix Figure 11 and table below, demonstrate consistent improvements in performance as the dataset size increases, indicating predictable scaling behavior across different architectures. Furthermore, we extend our analysis to multiple modality families, with results presented in Appendix Figure 12. Notably, we observe a significant performance boost as dataset size increases in different modalities (e.g. Endoscopy, Histopathology and OCT).
>
> - Data scaling results across architectures
>
> | | 25%  | 50%  | 75%  | 100% |
> |--------------|------|------|------|------|
> | Mamba        | 63.81 | 66.57 | 69.25 | 73.44 |
> | CNN          | 73.94 | 76.78 | 78.68 | 78.97 |
> | Hybrid       | 74.31 | 76.72 | 78.78 | 79.08 |
> | RWKV         | 68.55 | 72.10 | 74.32 | 74.62 |
> | Transformer  | 60.19 | 63.10 | 68.22 | 70.14 |

---

### Author Response · Authors · 2025-11-27
**Common Issue: On the design, weighting, and stability of U-Score**

We appreciate the insightful suggestions from reviewers VxQu, 8u5v, and 1nDY regarding the design and stability of *U-Score*. In response to reviewers' concerns, we conduct an expanded sensitivity and stability ablations covering the accuracy-efficiency trade-off $\alpha$, efficiency component weights $\mathbf{w}$, and normalization percentiles. Our evaluation measures ranking stability using: Kendall-$\tau$, Spearman-$\rho$, and Top-$k$ order preservation rates. Across all ablation settings, the resulting U-Score rankings remain stable across a wide range of configurations, indicating that U-Score robustly captures the inherent accuracy–efficiency trade-off.

**A1: Motivation for Adopting a Deployment-Agnostic Weight Design**  [VxQu, 8u5v, 1nDY]

Our goal is to reflect the trade-off between accuracy (IoU ($A$)) and efficiency (parameters ($P$), FLOPs ($G$), and FPS ($S$)) in practical segmentation tasks. To prevent skewing the score toward any single component, we use a harmonic mean for both the efficiency components and the final score, ensuring that sub-optimal scores penalize the overall metric. Additionally, to reduce the impact of outliers, we apply robust quantile clipping (default: 10th and 90th percentiles). The default configuration, where $w_P = w_G = w_S$ and $\alpha = 0.5$, represents a neutral starting point without assumptions about deployment priorities. This configuration avoids encoding any task-specific preferences into the metric, ensuring that it remains widely applicable across different segmentation tasks:

$$
\mathrm{Eff} = H(p, g, s \mid w_P, w_G, w_S), \qquad
\mathrm{U\text{-}Score} = H(a, \mathrm{Eff} \mid \alpha, 1-\alpha).
$$

In the absence of deployment-specific knowledge, choosing equal weights and setting $\alpha = 0.5$ is a reasonable approach to avoid bias, as it ensures that accuracy and efficiency are treated symmetrically and fairly.

**A2: Ablation sensitivity of weights and percentiles.**  [VxQu, 8u5v, 1nDY]

We have performed an extensive sensitivity analysis, performing a grid search over the following configurations: the accuracy--efficiency trade-off $\alpha$ ($\alpha \in \{0.25, 0.33, 0.5, 0.67, 0.75\}$), efficiency weights $(w_P, w_G, w_S) \in \{(1,1,1), (2,1,1), (1,2,1), (1,1,2), (3,1,1), (1,3,1), (1,1,3)\}$, and quantile pairs $(10/90,\, 5/95,\, 20/80)$.  The ranking stability is assessed using Kendall-$\tau$, Spearman-$\rho$, and Top-$k$ overlap with the baseline U-Score ranking (with default settings: $\alpha = 0.5$, $(w_P, w_G, w_S) = (1,1,1)$, $q$-pair = (0.10, 0.90)).

The results are summarized in Table 1. These findings underscore that U-Score reliably captures the accuracy–efficiency trade-off, with minimal ranking shifts even under varying configurations: Kendall-$\tau$ versus the baseline has a mean of $0.77$, a median of $0.77$, and remains $\ge 0.55$ even in the worst configuration; Spearman-$\rho$ versus the baseline has a mean of $0.90$ and a median of $0.91$. The Top-10 set also exhibits strong robustness, with a mean overlap of $0.86$, a median of $0.90$, and never dropping below $0.60$. Importantly, U-Score is distinct from pure accuracy: Kendall-$\tau$ relative to the IoU-only ranking is much lower, with a mean of $0.19$ (median $0.19$), demonstrating that U-Score genuinely captures the accuracy-efficiency trade-off.

- **Table 1: Robustness of *U-Score* rankings over 105 configurations**  (5 choices of $\alpha$, 7 efficiency weightings $\mathbf{w}$, 3 quantile pairs).  Baseline: $\alpha = 0.5$, $(w_P, w_G, w_S) = (1,1,1)$, $q\text{-pair} = (0.10, 0.90)$.

| Metric| Mean  | Std   | Min  | 25%  | Median | 75%  |
|-------------------------------------------|-------|-------|------|------|--------|------|
| Kendall-$\tau$ vs. base U-Score ranking   | 0.77  | 0.08  | 0.55 | 0.71 | 0.77   | 0.83 |
| Spearman-$\rho$ vs. base U-Score ranking  | 0.90  | 0.06  | 0.72 | 0.86 | 0.91   | 0.95 |
| Kendall-$\tau$ vs. IoU                    | 0.19  | 0.14  | -0.07| 0.06 | 0.19   | 0.30 |
| Top-10 keep                               | 0.86  | 0.12  | 0.60 | 0.80 | 0.90   | 0.90 |

---

> ### Author Response · Authors · 2025-11-27
> **Common Issue: On the design, weighting, and stability of U-Score**
>
> **A3: Ablation sensitivity of accuracy-efficiency ($\alpha$).**  [VxQu, 8u5v, 1nDY]
>
> We further investigate the impact of the accuracy–efficiency trade-off parameter $\alpha$ on U-Score rankings, isolating $\alpha$ while keeping $(w_P, w_G, w_S) = (1,1,1)$ and $q$-pair = (0.10, 0.90). Table 2 shows that changing $\alpha$ produces stable rankings, with Kendall-$\tau$ values consistently above 0.8 across all settings, and the Top-10 overlap remains between 0.80 and 1.00. These results demonstrate that U-Score provides a robust benchmark, where users can adjust $\alpha$ to reflect deployment priorities without introducing significant ranking shifts.
>
> - **Table 2: Sensitivity of U-Score to the accuracy-efficiency weight $\alpha$**  We report Kendall-$\tau$ vs. the baseline U-Score ranking ($\alpha=0.5$, $w_P,w_G,w_S=(1,1,1)$, $\text{qpair}=(0.10,0.90)$),  Kendall-$\tau$ vs. IoU-only, and the median Top-10 keep.
>
> | $\alpha$ | $\tau_{\text{base}}$ | $\tau_{\text{IoU}}$ | Top-10 keep |
> |----------|---------------------|---------------------|-------------|
> | 0.25     | 0.82                | -0.01               | 0.90        |
> | 0.33     | 0.89                | 0.06                | 1.00        |
> | 0.50     | 1.00                | 0.17                | 1.00        |
> | 0.67     | 0.92                | 0.25                | 0.90        |
> | 0.75     | 0.87                | 0.31                | 0.80        |
>
> Table 3 and Table 4 show the sensitivity of U-Score to the quantile pair $(q_{\text{pair}})$ and efficiency weights $(w_P, w_G, w_S)$. The results indicate that moderate changes in quantile ranges (e.g., $(0.05, 0.95)$ to $(0.20, 0.80)$) have minimal impact on U-Score rankings, with Kendall-$\tau$ values ranging from 0.76 to 1.00. The Top-10 overlap stays high, with most configurations showing stable rankings. Similarly, varying the efficiency weights $(w_P, w_G, w_S)$ does not drastically alter the leaderboard. Kendall-$\tau$ values between 0.86 and 0.94 and Top-10 overlap values above 0.9. This flexibility allows users to bias U-Score towards their desired trade-offs (e.g., memory, computation, or latency).
>
> - **Table 3: Sensitivity of U-Score to the quantile pair $(q_{\text{pair}})$**  We report Kendall-$\tau$ vs. the baseline U-Score ranking ($\alpha=0.5$, $w_P,w_G,w_S=(1,1,1)$, $\text{qpair}=(0.10,0.90)$),  Kendall-$\tau$ vs. IoU-only, and the median Top-10 keep.
> | $q_{\text{pair}}$ | $\tau_{\text{base}}$ | $\tau_{\text{IoU}}$ | Top-10 keep |
> |-------------------|---------------------|---------------------|-------------|
> | (0.05, 0.95)      | 0.90                | 0.14                | 1.00        |
> | (0.10, 0.90)      | 1.00                | 0.17                | 1.00        |
> | (0.20, 0.80)      | 0.76                | 0.31                | 0.80        |
>
> - **Table 4: Sensitivity of U-Score to the efficiency weight $(w_P,w_G,w_S)$**  We report Kendall-$\tau$ vs. the baseline U-Score ranking ($\alpha=0.5$, $w_P,w_G,w_S=(1,1,1)$, $\text{qpair}=(0.10,0.90)$),  Kendall-$\tau$ vs. IoU-only, and the median Top-10 keep.
> | $(w_P,w_G,w_S)$   | $\tau_{\text{base}}$ | $\tau_{\text{IoU}}$ | Top-10 keep |
> |-------------------|---------------------|---------------------|-------------|
> | (1, 1, 1)         | 1.00                | 0.17                | 1.00        |
> | (1, 1, 2)         | 0.92                | 0.19                | 1.00        |
> | (1, 1, 3)         | 0.87                | 0.21                | 1.00        |
> | (1, 2, 1)         | 0.90                | 0.15                | 1.00        |
> | (1, 3, 1)         | 0.86                | 0.14                | 1.00        |
> | (2, 1, 1)         | 0.94                | 0.17                | 0.90        |
> | (3, 1, 1)         | 0.90                | 0.17                | 0.90        |

---

### Author Response · Authors · 2025-11-27
**Common Issue: Reliable Cross-Architecture Evaluation**

We appreciate the question and suggestions from reviewers 1nDY, 8u5v, VxQu, and CwnQ. We strengthen the experiments' reliable analysis along the reviewers’ suggestions.

**1. Reproduction fidelity.**  [1nDY,8u5v]
Following reviewer 1nDY's suggestions, we add a $\Delta$IoU column in **Table 1** comparing our reproduced in-domain results with the original papers’ reported metrics. The results show that our unified recipe reproduces prior performance faithfully: most $\Delta$IoU values lie within a small and symmetric range around zero (typically ±3–5%). Notably, several models even achieve higher IoU under our unified pipeline (e.g., RWKV-UNet, DA-TransUNet, U-KAN, and DDANet), suggesting that our implementation maintains performance for the majority of architectures. To further validate the unified recipe, we conduct a controlled reproduction study on BUSI across 100 variants, using both **(1) each paper’s official training settings**, and **(2) our unified U-Bench recipe**. The results are shown in **Table 1** and **Appendix Figure 10**. The unified recipe yields higher median IoU and fewer collapsed low-tail cases, confirming that the U-Bench implementation is reliable as the official pipelines.

**Table 1: Distribution statistics of U-Bench vs. official configs.**
| Setting|Mean IoU|Median IoU | Min IoU | IQR|
|-|-|-|-|-|
|Official per-variants configs| 62.20|66.31|7.46|8.28|
|Unified U-Bench recipe| 65.56| 67.14|19.01|6.22|

**2. Unified training recipe.** [8u5v]
A key goal of U-Bench is fair cross-architecture comparison, which becomes challenging when each variant uses unique optimizers, schedules, resolutions, or data augmentations. To mitigate this, we adopt a unified configuration aligned with nnUNet (robust framework), with the same SGD optimizer, a learning rate of 0.1, and a poly learning rate scheduler. This configuration is not only consistent with nnUNet but also widely employed in models such as TransUNet, CMUNeXt, HiFormer, and SwinUnet. By standardizing these hyperparameters, we aim to provide a fair and reproducible evaluation environment across a diverse set of architectures.

**3. Near-zero IoUs.** [8u5v]
We carefully investigate the reported extreme failures. To isolate the cause, we retrain Polyp-PVT and CFPNet-M using the integrated nnUNet automatic pipeline. The results are shown in **Table 2**. Most models still collapse, producing IoU $\approx 0$. This confirms that these failures arise from architectural limitations rather than misconfiguration or preprocessing errors.

**Table 2: Verification of near-zero IoU cases via nnUNet retraining.**
| Model (Dataset)|U-Bench| nnUNet |
|-|-|-|
|Polyp-PVT (DCA)| 0.06|0.08|
|CFPNet-M (Cystoidfluid) |0.00|0.00|
|CFPNet-M (Cystoidfluid) |0.00|0.00|

**4. Performance Metric.** [1nDY, 8u5v, CwnQ]
We select IoU as the primary accuracy metric because our statistical analysis reveals that more than half of the 100 reproduced 2D variants use IoU as their main metric reported in the original papers, thereby making it the consistent choice for 2D variants comparison. Furthermore, since IoU and Dice are mathematically correlated, their rankings are typically aligned. To address the reviewer’s concern, we have included Dice in the evaluation, as shown in **Appendix Table 21**. Additionally, to assess boundary-sensitive behavior, we have added Boundary-F1 and HD95 leaderboard in **Appendix Table 22 and 23**.

**5. Input-resolution.** [8u5v, VxQu]
We understand the reviewer’s concern regarding the potential performance impact of unified input resolution. To ensure fairness across architectures with heterogeneous original protocols, we systematically analyze all reproduced variants and found that over half of the models are originally reported with a $256\times256$ input resolution. Therefore, U-Bench adopts $256\times256$ as a common resolution to align with the most widely used setting in the literature.

**6. Model-selection criteria.** [CwnQ]
We have clarified the inclusion rules: (1) publicly available official code, (2). influence within each architectural family, (3). coverage of major paradigms (CNN, Transformer, Mamba, RWKV, and their hybrid variants), and (4). reproducible 2D pipelines. In response to reviewer CwnQ, we incorporate two representative missing strong baselines, nnUNet (2D) and ResEncUNet-L (2D). The results are shown in **Table 3**, where nnUNet (2D) and ResEncUNet-L (2D) achieve Rank-2 and Rank-5 in in-domain IoU, but 25 and 24 in in-domain U-Score, Rank-24 and Rank-11 in zero-shot IoU, but 33 and 32 in zero-shot U-Score, further strengthening the completeness and contrast of the benchmark.

**Table 3: Ranking comparison of IoU vs. U-Score for two strong baselines.**
| Model| In-domain (IoU) | In-domain (U-Score) | Zero-shot (IoU) | Zero-shot (U-Score) |
|-|-|-|-|-|
|nnUNet (2D) | 2| 25| 24| 33|
|ResEncUNet-L (2D) | 5| 24| 11 | 32|

---

### Author Response · Authors · 2025-12-03
**Summary for Area Chair**

We thank the Area Chair for their dedication and tireless efforts in improving the community. We also thank the reviewers for their valuable feedback. We have carefully addressed all concerns, and our summary responses are provided below.

# Paper Overview

This paper introduces U-Bench, a large-scale benchmark for U-Net–style medical image segmentation, evaluating 100 publicly available U-Net variants across 28 datasets spanning 10 medical imaging modalities [*Reviewer 1nDY*]. The benchmark provides a comprehensive evaluation across a wide range of models and datasets, focusing on three core capabilities: statistical robustness, zero-shot generalization, and computational efficiency [*Reviewer VxQu*]. The paper proposed U-Score, a quantile normalized metric that combines accuracy (IoU) with efficiency (parameters, FLOPs, FPS) [*Reviewer 1nDY*]. Additionally, the paper analyze the performance and efficiency evolution of these models over time, showing that recent architectures, while also examining how models handle both easy and complex segmentation tasks [*Reviewer CwnQ*]. And with this information, they build an advisor agent that recommends architectures conditioned on data characteristics (foreground scale, boundary sharpness, shape regularity) and resource constraints [*Reviewer 1nDY*], suggests a list with the best architectures to be implemented in each case [*Reviewer CwnQ*]. Overall, the work is ambitious and potentially impactful for the community [*Reviewer 1nDY*].

# Consensus Strengths

- Interesting, Comprehensive, Unprecedented Benchmark Scale and Effort. [*Reviewer 1nDY, 8u5v, CwnQ, VxQu*]

- Proposing a key and clear U-Score Metric for Accuracy–Efficiency Trade-offs. [*Reviewer CwnQ, 1nDY, 8u5v*]

- Aligns with clinical domain-shift realities and Domain Shift Consideration. [*Reviewer 1nDY, VxQu, CwnQ*]

- Rigorous Validation, Reveal important insights and Reproducibility benchmark for the Community. [*Reviewer 1nDY, 8u5v, VxQu, CwnQ*].

- Goes beyond benchmarking with practical agent tools, making the benchmark actionable for practitioners. [*Reviewer 1nDY,  CwnQ*]

---

> ### Author Response · Authors · 2025-12-03
> **Summary for Area Chair**
>
> # Detailed Response
>
> #### 1. Common Issue: Clarifying and Strengthening the Evaluation Setting
>
> ​	We employ standardized training settings to ensure fair cross-architecture comparisons. Additionally, we perform a comparison with official settings (Appendix Fig.10) and include official reported reference performance (Appendix Table 20), demonstrating that our unified U-Bench pipeline faithfully reproduces previous results. Additionally, we incorporate metrics like Dice, Boundary-F1, and HD95 for a more comprehensive evaluation (Appendix Table 23-25). Finally, we include strong baselines (e.g., nnUNet and ResEncUNet-L) in U-Bench to enhance the benchmark's completeness (Main Table 2). These revisions significantly strengthen the reliability and robustness of our benchmark.
>
> #### 2. Common Issue: Experimentally Validating the Weighting and Stability of U-Score
>
> ​	We conduct comprehensive sensitivity and stability ablation tests on 105 different combinations of accuracy-efficiency trade-off weights, efficiency weights, and quantile pairs (Appendix Table 10). The results show that U-Score maintains stable rankings while effectively capturing the accuracy-efficiency trade-off. Additionally, we perform ablations on the accuracy-efficiency balance, efficiency weights, and quantile pairs (Appendix Table 11-13), further confirming the stability and flexibility of U-Score.
>
> #### 3. Need Inference Speed, Compute Accounting and Scaling Behavior Analysis [1nDY]
>
> - We present the FPS measurement protocol in Appendix Table 16.
>
> - We conduct a data scaling-law analysis across multiple modality and variant families, with the results displayed in Appendix Figures 11 and 12.
>
> - We report the GPU hours for each model in Appendix Table 26.
>
> #### 4. LOMO Evaluation [8u5v]
>
> We conduct LOMO evaluation (Appendix Table 14), our model advisor outperforms the IoU heuristic in multi-objective settings (U-Score), offering better generalization and faster model selection.
>
> #### 5. Clarification of Comparison with Torchstone and input resolution [VxQu]
>
> We clarify that U-Bench is the first large-scale, statistically rigorous 2D benchmark and highlight its broader scope (modalities, variants, evaluation) compared to Torchstone (3D benchmark) in the revised manuscript. Additionally, we examine all benchmarking variants and adhere to the majority resolution protocol to ensure fair and consistent cross-model comparisons.
>
> #### 6. Data Processing, 2D/3D Architecture Adaptation, Clinical Insights for Future Work [CwnQ]
>
> We clarify that 2D slices are processed using axial Z-axis slicing, and provide the corresponding code for reproducibility. Furthermore, we emphasize that U-Bench utilizes official 2D implementations for 3D variants to ensure consistent evaluation without modifying their original design. Finally, we introduce a 'Future Work' section, highlighting the integration of clinical perspectives to better align with real-world needs.
>
> #### 7. Minor Issues
>
> We correct grammatical errors, replace Comic Sans with Arial, and ensure consistent use of "U-Bench." We revise Figure 2 for clarity, add a citation for "RWKV," and correct the U-shaped model description. The paragraph from lines 251-263 is moved to the Results section. We clarify that statistical tests in Figure 5 apply only to IoU and explain the Dice-IoU relationship. Finally, we confirm all datasets are publicly available, include an ethical compliance statement, and provide an offline compressed file of the anonymous repository for cases where access via third-party platforms is unavailable.
>
> # Conclusion
>
> We have comprehensively addressed all reviewer comments. The rebuttal substantively strengthens the manuscript through:
>
> - Systematic Ablation of U-Score Parameters: Demonstrating robust and stable rankings across various configurations, confirming the reliability of U-Score.
>
> - Standardized Training Settings and Reproduction Fidelity: Providing official configuration comparisons and reproduction experiments, proving U-Bench's fidelity and enhanced stability in model performance.
>
> - Transparent Model Evaluation Protocols: Clearly explaining the FPS measurement protocol, data scaling behavior, and GPU runtime reporting to ensure reproducibility.
>
> - Reproducibility Consistency: Detailing the conversion of 3D architectures to 2D implementations and explaining the processing of 3D images to maintain consistency across evaluations. Additionally, we adopted Dice, HD96, and Boundary-F1 as supplementary metrics for more comprehensive evaluation.
>
> - Practical Validation of Model Recommendation Tools: Introducing the advisor agent and U-Score sensitivity analysis, demonstrating our tools can effectively recommend accurate models.
>
> The revised manuscript includes substantial additions to the Appendix, addressing all technical questions while maintaining concise main text presentation.

---

### Meta-Review · Area_Chair_tgHx · 2026-01-06

**Summary:**

This submission was reviewed by four expert reviewers, with the ratings of: 3 borderline accept, and 1 reject. The major concerns from the reviewers are about the limitations of the evaluation metrics used, unclear details (e.g. the selection criteria, measurement, etc.), scaling behavior, inconsistency with the literatures, potential bias introduced by the training setting, inconsistency in the experimental settings, issues with the U-Score, the equal weighting scheme used, overclaims, 2D vs 3D, issues with the significance testing. The authors provided a rebuttal for the concerns, but no further response from the reviewer was presented, and there was no discussion.

After carefully going through the review comments and the authors' rebuttal, it can be seen that some of the concerns are addressed and some questions are well cleared by the authors' further response and experiments. However, there are still major concerns remaining not well addressed. Overall, there is no strong support for a clear acceptance. As a result, considering the above especially the uncleared concerns, it is unfortunate that the paper in its current form is not ready to be presented at ICLR. However, the benchmark study presented in this paper could be helpful to the community, the authors are encouraged to address the concerns and further improve the paper for a future submission.

**Reviewer Concerns:**

Concerns that the AC thinks were addressed by the rebuttal: boundary-aware metric evaluation; inference speed measurement details; reproduction-gap analysis is further provided but there are still some big gaps; scaling behavior analysis; access issue with the anonymous repo; missing Dice and HD95 metrics; an analysis for the U-Score is further provided though the issue was not fully addressed; advisor agent analysis; equal weighting concern; the overclaims are partially addressed; how 2D slices were processed; how 3D archs take 2D input.

Concerns that are still outstanding: the inconsistency between the reported results (especially failure ones) and the original literature; potential bias introduced by the one-size-fits-all training (the further experimental results even support this concern); issues with the U-Score are still not fully convincingly addressed (e.g. the mitigation validation), even with the provided analysis; potential issues introduced by the different protocols (e.g. input sizes); the criteria for the benchmarked architectures selection is still not well justified; significance testing; and some minor issue such as the inconsistent loss weights in Eq.7, Eq.8 and the text.

**Reviewer Scores:**

According to the review comments, and the rebuttal, for each review the reviewer might have changed their score in the way below, if they had been able to participate fully in the discussion:
* Reviewer 1nDY: borderline accept to accept, or unchanged
* Reviewer 8u5v: reject, unchanged
* Reviewer VxQu: borderline accept to borderline reject, or unchanged
* Reviewer CwnQ: borderline accept to borderline reject.

---

### Decision · Program_Chairs · 2026-01-26

Reject